# Dynamic cytoskeletal regulation of cell shape supports resilience of lymphatic endothelium

Hans Schoofs[1,14], Nina Daubel[1,14], Sarah Schnabellehner[1], Max L. B. Grönloh[2], Sebastián Palacios Martínez[3], Aleksi Halme[4], Amanda M. Marks[1], Marie Jeansson[1], Sara Barcos[5], Cord Brakebusch[6], Rui Benedito[7], Britta Engelhardt[5], Dietmar Vestweber[8], Konstantin Gaengel[1], Fabian Linsenmeier[9], Sebastian Schürmann[9], Pipsa Saharinen[4,10], Jaap D. van Buul[2,3,11], Oliver Friedrich[9], Richard S. Smith[12], Mateusz Majda[13] & Taija Mäkinen[1,4,10]✉

Lymphatic capillaries continuously take up interstitial fluid and adapt to resulting changes in vessel calibre[1–3]. The mechanisms by which the permeable monolayer of loosely connected lymphatic endothelial cells (LECs)[4] maintains mechanical stability remain elusive. Here we identify dynamic cytoskeletal regulation of LEC shape, induced by isotropic stretch, as crucial for the integrity and function of dermal lymphatic capillaries. We found that the oak leaf-shaped LECs showed a spectrum of VE-cadherin-based junctional configurations at the lobular intercellular interface and a unique cytoskeletal organization, with microtubules at concave regions and F-actin at convex lobes. Multispectral and longitudinal intravital imaging of capillary LEC shape and actin revealed dynamic remodelling of cellular overlaps in vivo during homeostasis and in response to interstitial fluid volume increase. Akin to puzzle cells of the plant epidermis[5,6], LEC shape was controlled by Rho GTPase CDC42-regulated cytoskeletal dynamics, enhancing monolayer stability. Moreover, cyclic isotropic stretch increased cellular overlaps and junction curvature in primary LECs. Our findings indicate that capillary LEC shape results from continuous remodelling of cellular overlaps that maintain vessel integrity while preserving permeable cell–cell contacts compatible with vessel expansion and fluid uptake. We propose a bellows-like fluid propulsion mechanism, in which fluid-induced lumen expansion and shrinkage of LEC overlaps are countered by actin-based lamellipodia-like overlap extension to aid vessel constriction.

The lymphatic vasculature collects interstitial fluid and transports it through interposed lymph nodes to the systemic circulation[1–3]. Endothelial cells lining the initial blind-ended, fluid absorbing lymphatic capillaries have a unique oak leaf shape and are equipped with specialized discontinuous cell–cell junctions called buttons[4]. The junction-free regions in between buttons have been described as functioning as primary flap valves that permit fluid and macromolecule uptake, as well as immune cell entry, from the interstitium into the vessel lumen[4,7]. The fluid-draining collecting lymphatic vessels instead have continuous zipper-like junctions to prevent lymph leakage[4]. Dynamic transitions have been shown to occur between the two junction types in capillary lymphatic vessels during development and in disease states, with several molecular players identified[8–11]. For example, lymphangiogenic embryonic vessels have zipper junctions that remodel into buttons during late developmental stages[8]. Conversely, new lymphatic sprouts arising from mature lymphatic capillaries in adult tissues undergo a button-to-zipper transition, which is associated with cell elongation[4]. Abnormal lymphatic endothelial cell (LEC) junction 'zippering' has been linked to reduced uptake of dietary fats in the intestine[9] and viral dissemination from the skin[12], and impaired lymphatic drainage in chronic inflammation[4,13]. The specialized junctional organization of LECs is thus critical for efficient lymphatic function. However, given the discontinuity of their junctions, the absence of mural cells and only a thin basement membrane providing structural support[1,2], it remains unknown how capillary LEC monolayers withstand changes in vessel calibre in response to interstitial fluid volume alterations without rupturing.

Here we investigated the specialization of lymphatic capillaries in the dermal vasculature of the mouse ear, which develops postnatally and is accessible to intravital imaging. This vasculature undergoes remodelling into a hierarchical vessel network after 2 weeks of age[14], with developmental maturation of capillary LEC junctions from zippers into buttons occurring by 3 weeks of age[9–12,15]. In the current study, we

[1]Department of Immunology, Genetics and Pathology, Uppsala University, Uppsala, Sweden. [2]Department of Medical Biochemistry at the Amsterdam UMC, location AMC, Amsterdam, The Netherlands. [3]Department of Molecular Cytology, Leeuwenhoek Centre for Advanced Microscopy at Swammerdam Institute for Life Sciences at the University of Amsterdam, Amsterdam, The Netherlands. [4]Translational Cancer Medicine Program and Department of Biochemistry and Developmental Biology, University of Helsinki, Helsinki, Finland. [5]Theodor Kocher Institute, University of Bern, Bern, Switzerland. [6]Biotech Research and Innovation Center, University of Copenhagen, Copenhagen, Denmark. [7]Centro Nacional de Investigaciones Cardiovasculares, Madrid, Spain. [8]Max Planck Institute for Molecular Biomedicine, Münster, Germany. [9]Institute of Medical Biotechnology, Department of Chemical and Biological Engineering, Friedrich-Alexander University Erlangen-Nürnberg, Erlangen, Germany. [10]Wihuri Research Institute, Helsinki, Finland. [11]Amsterdam UMC, Sanquin Research and Landsteiner Laboratory, Amsterdam, The Netherlands. [12]John Innes Centre, Norwich Research Park, Norwich, UK. [13]Department of Plant Molecular Biology, University of Lausanne, Lausanne, Switzerland. [14]These authors contributed equally: Hans Schoofs, Nina Daubel. ✉e-mail: taija.makinen@igp.uu.se

used mouse models and experimental approaches to investigate the mechanisms and drivers underlying the attributes of capillary LECs and their significance for lymphatic vessel physiology. We identified new features of LEC cytoskeleton and dynamic properties of the overlapping cell borders, and uncovered a previously unappreciated spectrum of VE-cadherin-based junctional configurations in capillary LECs across juvenile and adult stages. Our results point to the role of dynamic cytoskeletal regulation of LEC shape in controlling the maintenance of vessel integrity and function.

## Junction heterogeneity in capillary LECs

Analysis of ear skin from 25-week-old VE-cadherin-GFP (green fluorescent protein) mice[16] revealed the previously demonstrated distinctive cell–cell junctions in different lymphatic vessel types[4] (Fig. 1a and Extended Data Fig. 1a). In lymphatic capillaries and precollecting vessels, discontinuous VE-cadherin-GFP distribution, which colocalized with VE-cadherin immunostaining, was interspersed with LYVE1+ lobate segments, whereas collecting vessels showed continuous VE-cadherin+ zipper junctions without LYVE1 (Fig. 1a and Extended Data Fig. 1a). In addition to button junctions oriented parallel along the sides of junction-free regions[4], we observed linear, unsegmented or segmented VE-cadherin+ adherens junctions extending into lobe tips (Fig. 1a and Extended Data Fig. 1a). Confocal imaging of wild-type juvenile (3- and 5-week-old) and adult (25-week-old) mice confirmed diverse VE-cadherin-based junctional configurations in capillary LECs (Fig. 1b,c), with the endothelial tight junction protein claudin 5 (CLDN5) often coinciding with VE-cadherin (Fig. 1b). As previously reported[17], at 3 weeks of age roughly 20% of lymphatic capillary ends showed sprouting (Fig. 1c,d), which was associated with an elongated cell shape, low LYVE1 and the presence of zipper junctions surrounding the entire cell (Fig. 1c). Flow cytometry using Ki67 as a marker of cycling cells revealed LEC proliferation, indicative of active vessel growth, at this stage (Fig. 1e), supporting a link between vascular growth state and junction morphology[8].

To exclude the influence of vascular growth on junction morphology, we focused our analysis on mature LYVE1+ capillary tips with rounded morphology across all age groups. This revealed a spectrum of VE-cadherin-based junctional configurations independent of developmental stage (Fig. 1c). Silver nitrate staining[18] additionally revealed predominantly single or double granular lines at capillary LEC borders, including the tips of the lobes (Fig. 1f and Extended Data Fig. 1b). Silver precipitation is reported to occur due to an unspecified component at the EC intercellular interface or by basement membrane components[18] (see Supplementary Information for discussion). However, the similarity of thin double lines of silver deposits to VE-cadherin distribution suggest that a junctional component at the LEC overlap initiates deposition.

We quantified LEC junction morphology within a region from capillary tip to the first valve. This initial region of lymphatic capillaries lacked LYVE1− zipper junctions (Fig. 1g), defined as continuous VE-cadherin+ junctions around entire LECs. Buttons, defined as punctate VE-cadherin+ deposits at the neck of LYVE1+ regions, were observed in roughly 20% of lobes within an individual capillary vessel end, with no significant increase in their frequency with age (Fig. 1g). Most frequently across all stages, we detected variations of unsegmented or segmented linear distribution of VE-cadherin lining one or both borders of LYVE1+ regions, which we termed curvilinear and double junctions, respectively (Fig. 1g and Extended Data Fig. 1c). Few curvilinear junctions lacked LYVE1 staining (Fig. 1g). These junction types were present along the entire initial vessel segment, including the capillary tip (Extended Data Fig. 1d), and were also observed in the diaphragm lymphatics (Extended Data Fig. 1e and Supplementary Fig. 2a,b), regardless of whether the tissue was fixed by vascular perfusion or immersion (Supplementary Fig. 2c). Acute tissue swelling, induced by PBS injection into the ear to mimic oedema[19], did not result in significant changes in junction type distribution after 10 min (Fig. 1h).

## Dynamic capillary LEC overlaps

To study the other distinctive features of capillary LECs, the lobate shape and cellular overlaps, we performed multicolour mosaic labelling using the *iMb2-Mosaic* reporter[20], which permits stochastic expression of a single membrane-localized fluorescent protein (EYFP, TdTomato or mTFP1) upon Cre recombination (Fig. 2a). Reporter expression in LECs using the *Vegfr3-creER[T2]* transgene[21] confirmed the lobate morphology of capillary LECs and elongated shape of collecting vessel LECs in adult ear skin (Fig. 2b). By contrast, LYVE1[low] LECs of sprouting capillary tips were elongated, lacked lobes and had continuous zipper junctions (Extended Data Fig. 2a). The lobate shape emerged early during dermal vascular development (Fig. 2c), with cell size (Fig. 2d) and lobe number (Fig. 2e) increasing after 3 weeks of age when cell proliferation seizes (Fig. 1d). This was associated with an increase in the diameter of LYVE1+ capillaries (Fig. 2f), suggesting that in the absence of cell proliferation LECs accommodate vessel growth by increasing their size and cell shape complexity.

Mosaic labelling revealed large overlaps between neighbouring LECs expressing different fluorescent proteins (Extended Data Fig. 2b), with LYVE1 localized precisely to these overlap regions (Fig. 2g). Immunolabelling LYVE1 on the cell surface in unpermeabilized tissue, followed by permeabilization and staining of total LYVE1 using a different antibody showed distinctive patterns (Fig. 2h). Cell surface staining predominantly outlined abluminal and intraluminal rims of overlaps (Fig. 2h), suggesting close proximity of plasma membranes at these regions, which become inaccessible for staining after chemical crosslinking. Staining of the intraluminal border presumably occurs where the antibody accesses the lumen through endothelial disruptions created during tissue processing. By contrast, LYVE1 antibody injected into a live mouse ear skin readily accessed intercellular overlaps (Fig. 2h), supporting their previously established role as fluid and macromolecule passage routes[22].

Quantitative analysis of overlaps in *iMb2-Mosaic* mice revealed variability in their dimensions (Fig. 2i and Extended Data Fig. 2c). The average perpendicular overlap width was 2.3 ± 0.8 μm, with an area of 123.0 ± 52.6 μm$^2$ (mean ± s.d., $n = 26$) (Fig. 2j), similar to the previous width measurements of 0.5–2 μm in tracheal lymphatic capillaries[4]. After intradermal fluid injection, we observed a decrease in overlap width (1.7 ± 0.7 μm) and area (83.4 ± 52.0 μm$^2$, mean ± s.d., $n = 35$) (Fig. 2i,j), suggesting responsiveness to interstitial fluid changes. Consistent with previous studies[23,24], transmission electron microscopy (TEM) also revealed variable overlap morphologies, including simple linear and complex convoluted arrangements (Extended Data Fig. 2d and Supplementary Figs. 3 and 4). By applying a 4 μm upper threshold for perpendicular overlap width from confocal data (Extended Data Fig. 2c), TEM measurements revealed an average overlap width of 1.8 ± 0.9 μm (s.d., $n = 34$) (Extended Data Fig. 2e).

To visualize whether LEC contacts remodel during homeostasis, we performed longitudinal intravital imaging of ear skin in *iMb2-Mosaic; Vegfr3-creER[T2]* mice, crossed to a C57BL/6-albino background, using two-photon microscopy (Fig. 2k). Tracking of individual capillary LECs over the course of several weeks (Fig. 2k and Supplementary Fig. 5) and months (Extended Data Fig. 3a) revealed dynamic changes in cell morphology, including lobe remodelling, regression and emergence of new lobes, with no observed migration, proliferation or cell death during the same timeframe. Concave regions of capillary LECs remained mostly stable (Fig. 2k), and no morphological changes were observed in collecting vessel LECs (Extended Data Fig. 3b). Real-time imaging of individual capillary LECs for 4 hours further revealed rapid and continuous remodelling of cell–cell borders (Supplementary Video 1). These findings suggest that capillary LEC lobes represent lamellipodia-like contact sites undergoing dynamic remodelling during homeostasis and respond to acute changes in interstitial fluid volume.

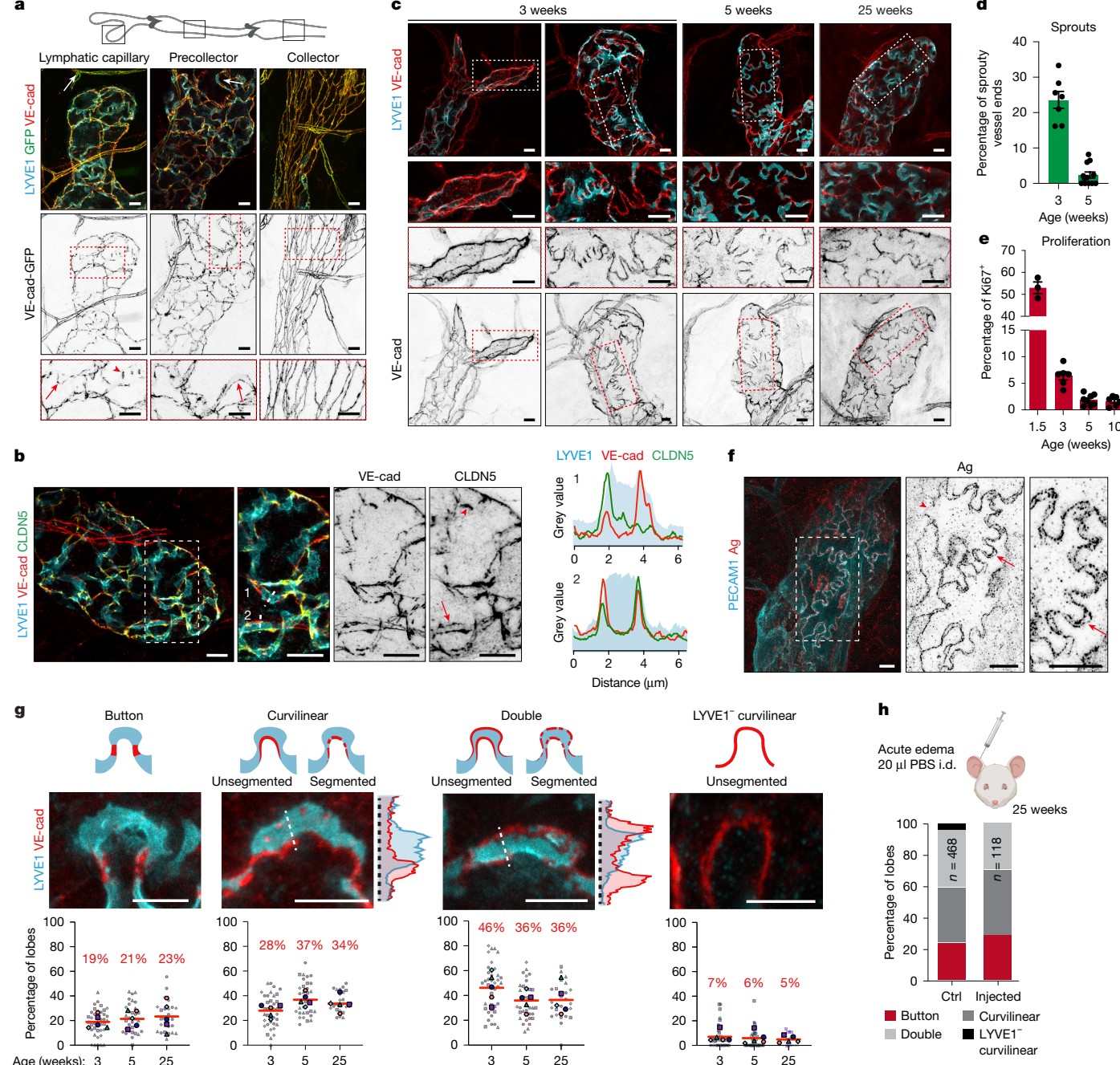

**Fig. 1 | Junctional heterogeneity in capillary LECs. a,** Whole-mount immunofluorescence of ear skin from a 25-week-old *Cdh5-GFP* mouse expressing VE-cadherin-GFP fusion protein (VE-cad). Boxed areas magnified below show unsegmented (arrows) and focal (arrowheads) VE-cadherin[+] junctions in lymphatic capillary (left) and precollecting vessel (middle), and continuous zipper junctions in LYVE1[−] collecting vessel (right). **b,** Immunofluorescence in 12-week-old mouse ear skin showing VE-cadherin colocalization with CLDN5 at junctions. Line intensity profiles through lines 1 and 2 of respective stainings are depicted. **c,** Immunofluorescence at the indicated ages depicting junctional heterogeneity. Boxed areas are magnified. **d,e,** Quantification of lymphatic vessel sprouting (percentage of spiky ends of all lymphatic capillary ends, $n = 7$, 12 per respective stage, mean ± s.e.m.; **d**) and LEC proliferation (percentage Ki67[+] of all LECs by flow cytometry, $n = 3, 6, 8, 6$ mice per respective stage, mean ± s.e.m.; **e**). **f,** Whole-mount silver nitrate (Ag) staining of ear dermis showing deposits around cell perimeter, including the lobe tips (arrow),

with discontinuities (arrowhead). **g,** Immunofluorescence images of dermal capillary LEC lobes and schematics depicting idealized junctional categories (top), with frequencies within a terminal capillary end (bottom) represented as SuperPlot ($n = 4$–5 vessels each mouse, five mice per stage, in total $n = 1,785$ junctions, Supplementary Information). Mean (red line and percentage) of all measurements, large boxed colour-coded shapes represent weighted average for individual mice, smaller shapes individual frequencies at capillary ends from the respective animal. No significant differences between stages, analysed by one-way analysis of variance (ANOVA). Intensity plots of curvilinear and double junctions corresponding to lines across cell–cell contacts are depicted on the right. **h,** Frequency of junction types in 25-week-old mouse ear skin with or without intradermal (i.d.) injection of 20 μl of PBS. Data represent the percentage of all junctions (Ctrl, control, $n = 468$ from five mice; fluid injected: $n = 118$ from four mice). Scale bars, 10 μm (**a**–**c**,**f**), 5 μm (**g**). Illustration in **h** created using BioRender (https://biorender.com).

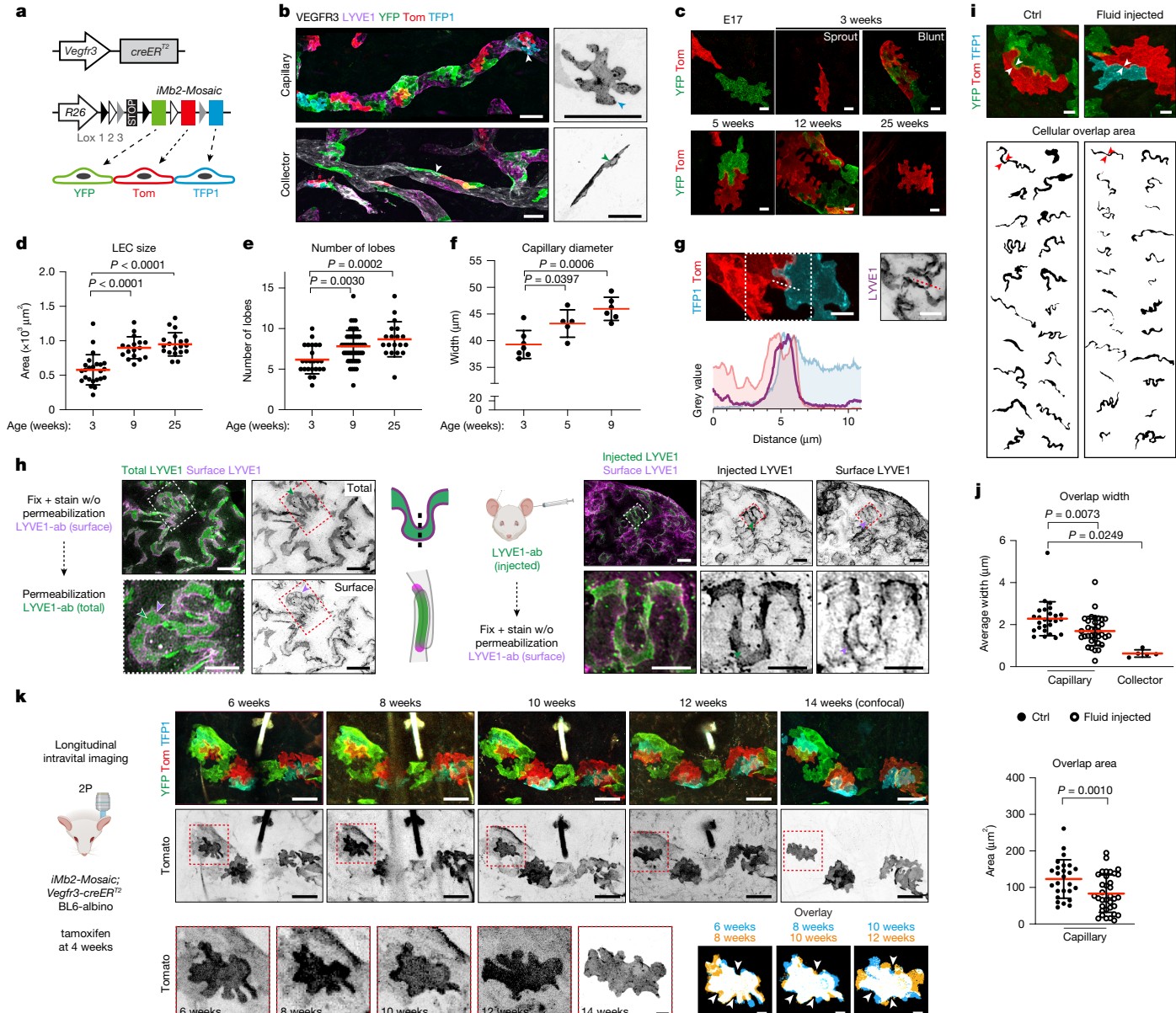

**Fig. 2 | Morphology and remodelling of intercellular overlaps between capillary LECs. a**, Constructs for mosaic multicolour labelling of LECs using membrane-localized fluorescent proteins. **b**, Whole-mount immunofluorescence of mosaically labelled dermal LECs in a 6-week-old *iMb2-Mosaic;Vegfr3-creER[T2]* mouse after 4-OHT treatment at 3 weeks, showing lobate shape in LYVE1+ capillaries and elongated shape in LYVE1− collectors (arrowheads). **c**, Whole-mount immunofluorescence of embryonic back skin (E17) or ear skin at indicated postnatal stages in *iMb2-Mosaic;Vegfr3-creER[T2]* mice. **d**–**f**, Dermal LEC and vessel parameters, represented as mean ± s.d.: cell size (**d**, *n* = 24, 17 and 20 cells per respective stage), lobe number (**e**, *n* = 24, 48 and 20 cells per respective stage) and average lymphatic capillary width (**f**, *n* = 7, 5 and 9 mice per respective stage). Ordinary one-way ANOVA. **g**, Immunofluorescence of ear skin of a 12-week-old *iMb2-Mosaic;Vegfr3-creER[T2]* mouse showing LYVE1 at LEC overlaps, with corresponding intensity plot. **h**, Double staining for cell surface and total LYVE1 (left), or with intradermally injected LYVE1 antibody (right) to visualize intercellular overlaps. w/o, without. **i**,**j**, Visualization (**i**) and quantification (**j**) of cellular overlap width (top, **j**) and area (bottom, **j**) in control and PBS-injected ears in *iMb2-Mosaic;Vegfr3-creER[T2]* mice. Two individual cells and their overlap (arrowheads) are shown as binary images. In **j**, *n* = 26 overlaps from four mice (Ctrl capillary), 35 overlaps from four mice (Fluid injected capillary), five overlaps from two mice (collector), represented as mean ± s.d. Ordinary one-way ANOVA (width), two-sided Mann–Whitney *U*-test (area). **k**, Intravital imaging of capillary LECs in adult *iMb2-Mosaic; Vegfr3-creER[T2]* BL6-albino mice showing remodelling of LEC lobes over time. Cell magnified below (left, red boxes) shown by two-colour overlay of indicated time points (right) show changes in lobes whereas concave areas (arrowheads) show minimal changes. Scale bars, 50 μm (**b**,**k**(top)), 10 μm (**c**,**g**,**h**,**i**,**k**(bottom)). Credits: schematic in **a** adapted from ref. 20 under a CC BY 4.0 licence; illustrations in **h** and **k** created using BioRender (https://biorender.com).

## Capillary LEC cytoskeleton

To understand whether the dynamic regulation of capillary LEC shape is influenced by their junctional or cytoskeletal composition, we analysed single-cell RNA sequencing (scRNA-seq) data from mouse ear skin LECs[25]. Transcript levels of major junctional adhesion molecules (including *Cdh5* (encoding VE-cadherin), *Cldn5*, *F11r* (encoding JAM1), *Tjp1* (encoding ZO1)) were similar across LEC subtypes (Supplementary Fig. 6a). However, capillary LECs showed enrichment of genes encoding regulators of the actin, spectrin and microtubule cytoskeleton compared to collecting vessel and/or valve LECs (Fig. 3a). Similar gene expression patterns were found in the two capillary LEC populations

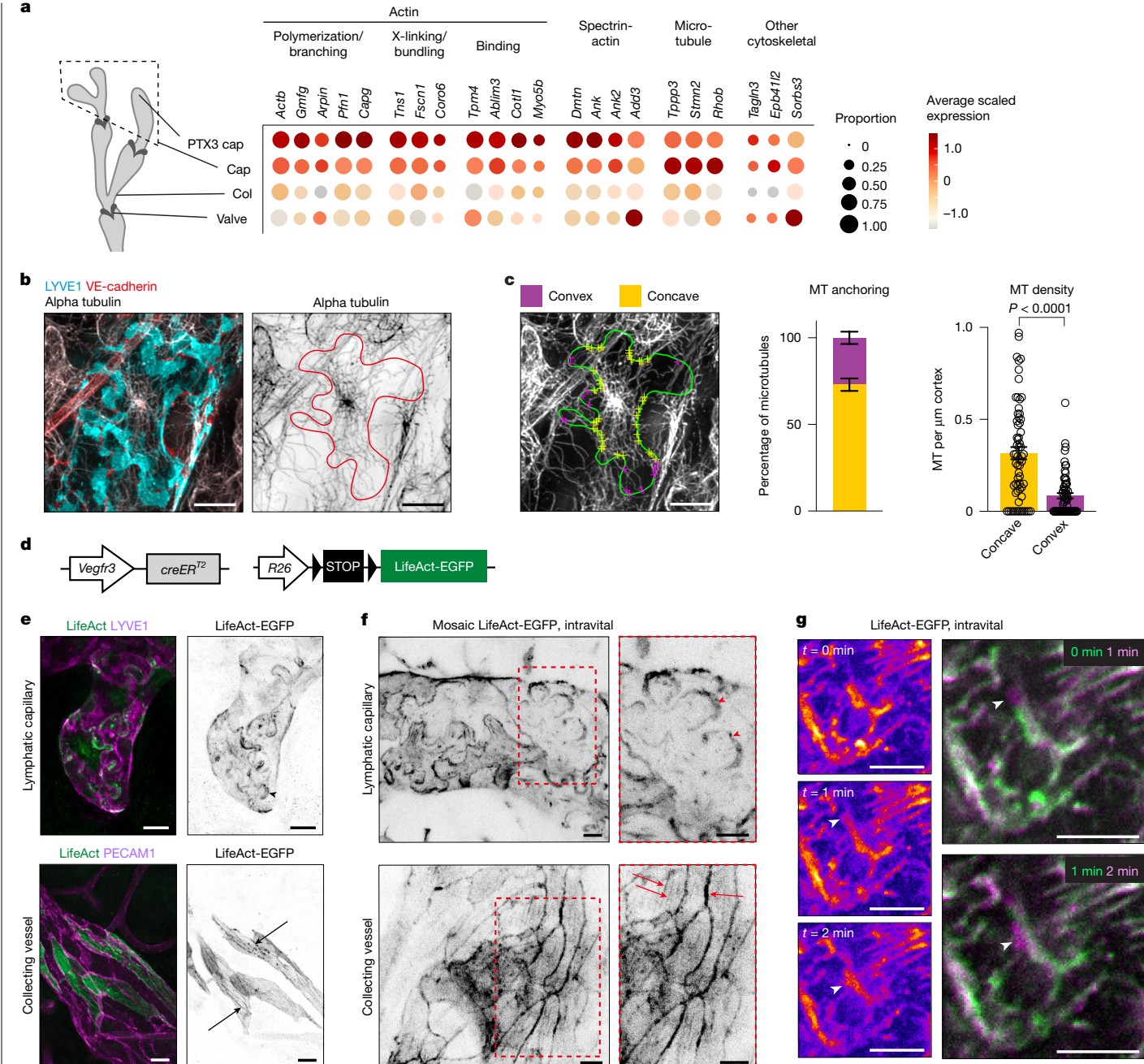

**Fig. 3 | Cytoskeletal organization in lobate capillary LECs. a**, Dot plot showing differential expression of cytoskeletal genes between capillary and collecting vessel LECs. Dot size illustrates percentage of cells with transcript counts, colour illustrates average expression (log$_2$-fold difference). Cap, lymphatic capillary; Col, collecting vessel. **b**, Whole-mount immunofluorescence of adult ear skin showing microtubule network in dermal capillary LECs. Cell outline, based on VE-cadherin and LYVE1 staining, in red. **c**, Quantification of microtubule (MT) anchoring and density in capillary LECs in 9–12-week-old mice. Cell outline from **c** in green, with MT endpoints shown by yellow (concave) and purple (convex) dots. Data represent the percentage of MT anchoring (left; $n = 5$ LECs from five mice, 20–53 MT per cell), or MTs per µm of cortex in concave (right; $n = 156$ MTs, 5 LECs from five mice) versus convex ($n = 56$ MTs, 5 LECs from five mice) regions (mean ± s.e.m.). Two-sided Mann–Whitney $U$-test. **d**, Constructs for LEC-specific visualization of F-actin using LifeAct-EGFP. **e**,**f**, Actin cytoskeleton in LECs from tamoxifen-treated adult *LifeAct-EGFP*; *Vegfr3-creER$^{T2}$* mice after tissue fixation (**e**) and intravital imaging (**f**). Tamoxifen was administered at 6 weeks and ears analysed at 8 weeks of age. Note the enrichment of LifeAct-EGFP in capillary LEC lobes (arrowheads), and cortical actin and stress fibres in collecting vessel LECs (arrows). Boxed areas in **f** are magnified. **g**, Intravital imaging of actin dynamics in *Lifeact-EGFP*;*Vegfr3-creER$^{T2}$* BL6-albino mice. Individual stills (left) and two-colour overlay of stills (right) from Supplementary Videos 3 and 4 show actin remodelling in LEC lobe borders (arrowheads) at the indicated time points (min). Scale bars, 10 µm (**b**,**c**,**e**–**g**).

defined by *Ptx3* expression, with *Ptx3*$^{high}$ LECs localizing within the initial lymphatics[25].

Staining of microtubules in adult mouse skin revealed frequent terminations at concave regions of capillary LECs, while being mostly absent from the convex lobes (Fig. 3b,c and Supplementary Video 2). Phalloidin staining of filamentous actin (F-actin) showed prominent networks in non-ECs of the surrounding tissue, thereby obscuring the analysis of LECs with lower F-actin content (Supplementary Fig. 6b). For cell lineage-specific visualization of F-actin, we generated a Cre-inducible *R26-LifeAct-EGFP* mouse line (Fig. 3d and Supplementary Fig. 6c).

Validation in blood ECs, using pan-endothelial *Tie2-cre* mice, confirmed the expected localization of LifeAct-EGFP in cytoplasmic actin bundles (Supplementary Fig. 6d). LEC-specific expression of LifeAct-EGFP, driven by the *Vegfr3-creER[T2]* transgene, showed a distinct pattern of F-actin in lymphatic capillaries, with a lack of radial actin bundles and an accumulation at the cell borders, observed in both fixed tissue (Fig. 3e) and by intravital two-photon imaging (Fig. 3f). Mosaic expression induced by low tamoxifen dose further revealed enrichment of F-actin on the convex lobes of capillary LECs (Fig. 3f). Line intensity profiles across a capillary LEC contact showed LifeAct-EGFP distribution across LYVE1[+] overlaps (Extended Data Fig. 4a), supported by SPY-555-actin staining of F-actin (Supplementary Fig. 6e). Real-time intravital imaging of dermal capillaries revealed dynamic actin remodelling in subminute time frames in the LEC lobes (Fig. 3g and Supplementary Videos 3 and 4). Collecting vessel LECs instead showed continuous, stable cortical actin rims (Supplementary Videos 5 and 6), with narrow peaks of LifeAct-EGFP signal colocalizing with VE-cadherin (Extended Data Fig. 4a). Furthermore, they showed radial actin bundles oriented along the vessel axis (Fig. 3e,f), resembling the arrangement observed in blood ECs (Supplementary Fig. 6d).

These findings highlight the unique cytoskeletal organization of capillary LECs, with microtubules terminating at concave regions and F-actin enrichment at dynamically remodelling convex lobes. The lobate cell shape and cytoskeletal organization parallels that of puzzle cells in plant epidermis[5,6] (Extended Data Fig. 4b), where microtubules stabilize concave necks and actin-rich convex lobes enable dynamic growth, regulated by plant Rho GTPases[26,27].

## Cytoskeletal regulation of LEC shape

To explore the role of Rho GTPase signalling in capillary LECs, we used *Prox1-creER[T2]* mice[28] to delete *Cdc42*, a key regulator of the actin and microtubule networks, in mature lymphatic endothelium at 6 weeks of age (Fig. 4a). The mice were further crossed with *LifeAct-EGFP* or *iMb2-Mosaic* mice. Immunostaining of the ear skin 3 weeks after tamoxifen administration revealed reduced LifeAct-EGFP at LEC lobes and disrupted microtubule organization (Fig. 4b). Furthermore, there was an increased number of microtubules and loss of their predominant termination at the concave regions (Fig. 4c,d). *Cdc42*-deficient LECs further showed altered morphology, characterized by the formation of thin membrane protrusion and change from a uniform lobate shape to more irregular shapes (Fig. 4e and Extended Data Fig. 5a), reduced LYVE1 expression (Fig. 4e) and cellular overlap width (Fig. 4f), as well as an increase in LYVE1[−] junctions (Fig. 4g,h). Disperse localization of cell surface LYVE1 (Fig. 4i) and the presence of LYVE1[−] areas within the cellular overlaps in *Cdc42*-deficient LECs (Extended Data Fig. 5b) further suggested a loss of integrity within these regions. Despite *Prox1-creER[T2]*-mediated deletion in cardiomyocytes, cardiac function and morphology remained unaffected 3 weeks after tamoxifen administration in *Cdc42* mutant mice (Supplementary Fig. 7a–d), excluding secondary effects from heart failure.

By 12 weeks of age (6 weeks post-tamoxifen), cytoskeletal and cell shape alterations in *Cdc42*-deficient LECs were accompanied by disrupted cell–cell junctions, and formation of intercellular separations (Fig. 4b,g (arrowheads)). TEM analysis of dermal LECs of control mice revealed minimal variation in the width of intercellular clefts, whereas *Cdc42*-deficient LECs showed highly variable cleft width with focal regions of cell separation within cellular overlaps (Extended Data Fig. 5c–e). In addition, clearance of intradermally injected fluorescent tracer was reduced in *Cdc42*-deficient mice (Extended Data Fig. 5f), indicating impaired lymphatic function. Although collecting vessel LECs in mutant mice also showed reduced LifeAct-EGFP, they retained apparently normal organization of adherens junctions (Extended Data Fig. 6a,b), microtubule networks (Extended Data Fig. 6c) and shape (Extended Data Fig. 6d).

Alterations in the cell cytoskeleton and shape, before disruption of monolayer integrity, suggest that cytoskeletal regulation has a primary role in maintaining junctional integrity. By contrast, genetic deletion of major junctional proteins CLDN5 (Extended Data Fig. 7a,b) or VE-cadherin[29] in the mature lymphatic vasculature did not disrupt dermal LEC junctions. Similarly, genetic deletion of the cell-extracellular matrix adhesion receptor integrin β1 in adult mice did not compromise dermal LEC integrity, shape or overlaps, even after an extended period of 10 weeks of gene deletion, despite efficient *Itgb1* deletion in LECs (Extended Data Fig. 7c–e).

## LEC shape and monolayer stability

In plants, puzzle cell shape reduces turgor pressure-induced stress on the cell wall[5,27], preventing excessive bulging during fluid uptake and growth. Mammalian cells have lower turgor pressure due to the absence of a rigid cell wall, whereas their flexible plasma membrane allows cells to change shape. As LECs are relatively large, with only on average three cells forming the vessel circumference, we proposed that simple-shaped cells might deform under stress, similar to plant cells. To investigate the relationship between cell shape and pressure-induced cellular stress, we used finite element method (FEM) modelling, previously used to study plant puzzle cells[27]. The vessel and cell parameters were obtained from in vivo measurements of mature quiescent vessels (Fig. 2d–f), and in silico cellular outlines were fitted to those idealized from 3D confocal stacks of lymphatic capillaries of 9-week-old mouse ear skins (Fig. 5a). The vessel with puzzle-shaped cells was compared to one with linear shapes generated by connecting the tricellular junctions of puzzle cells, using a simplified model that assumes identical connectivity and uniform mechanical properties at cell interfaces. This produced a template with simple-shaped cells with the same average size and neighbour connectivity (Fig. 5b and Extended Data Fig. 8a).

Simulations were first performed without external luminal or abluminal interstitial pressure, using an internal cell pressure of 0.015 kPa, which was chosen at the higher end of the reported range for ECs[30–32]. In the puzzle cell template, notable bulging occurred, particularly on the luminal side, but the vessel remained open (Extended Data Fig. 8a). By contrast, the simple cell template showed bulging that almost completely closed the lumen (Extended Data Fig. 8a). When a pressure gradient with higher abluminal (interstitial) pressure was applied, the vessel formed of puzzle-shaped cells tolerated the increase, whereas the vessel with simple shapes showed exacerbated lumen collapse (Fig. 5c). Furthermore, lower overall stress at both the cell and tissue levels were observed in monolayers composed of puzzle-shaped cells compared to simple-shaped cells (Fig. 5c). These in silico results indicate that the lobate shape enhances LEC monolayer resilience to mechanical strain.

## Effect of isotropic stretch on LECs

The puzzle cell shape in plants emerge from equiaxial tissue growth, resulting in isotropic stretching[6]. Given the need for lymphatic capillaries to accommodate changes in vessel calibre in response to interstitial fluid volume alterations, we proposed that capillary LECs are subjected to intermittent isotropic stretching. Using a custom-engineered 'Multi-Stretcher' device (F.L., S.S. and O.F., unpublished), based on ref. 33, with four parallelly actuated polydimethylsiloxane (PDMS) chambers, we applied cyclic isotropic stretch of 0.01 Hz (100 s per cycle) in-plane to primary human LEC monolayers in vitro (Fig. 5d). After 14 h of stretching, we observed increased cellular overlaps, visualized by PECAM1 staining (Fig. 5e), and presence of F-actin at their borders (Fig. 5f). Some overlaps lacked VE-cadherin, whereas others showed weak or punctate VE-cadherin staining, or double lines of VE-cadherin[+] junctions at the overlap borders (Extended Data Fig. 8b), mimicking the in vivo heterogeneity. An extended stretch period of 22 h further led to

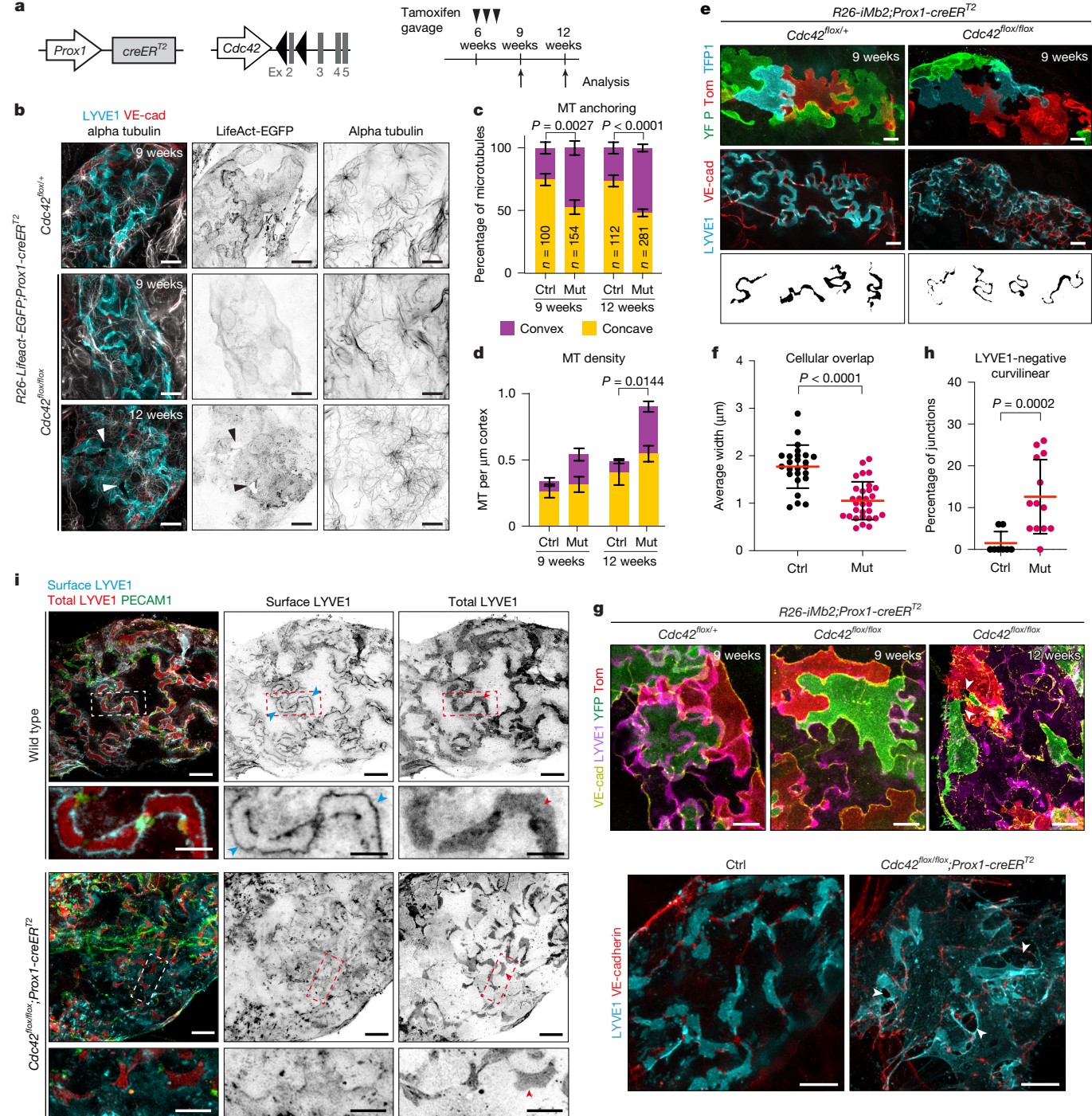

**Fig. 4 | CDC42 in the homeostatic maintenance of LEC cytoskeleton, cell shape and vessel integrity. a**, Scheme for LEC-specific *Cdc42* deletion in mature vasculature using *Prox1-creER^T2* mice. **b**, Actin (LifeAct-EGFP) and microtubule (alpha-tubulin staining) networks in ear skin whole-mounts from *Cdc42^flox^;LifeAct-EGFP;Prox1-creER^T2* mice. **c,d**, Quantification of microtubule (MT) anchoring (**c**) and density (**d**) in capillary LECs in control (Ctrl) and *Cdc42*-deficient (Mut) mice, mean ± s.e.m. (Ctrl, *n* = 100 MTs (9 weeks) or *n* = 112 (12 weeks); Mut, *n* = 154 MTs (9 weeks) or *n* = 281 (12 weeks) from 5–7 LECs/2–3 mice each (Supplementary Information). Two-tailed Fisher's exact test (**c**); two-tailed unpaired Student's *t*-test (**d**). **e–h**, Visualization (**e,g**) and quantification (**f,h**) of cell morphology, cellular overlaps and junctions in control and *Cdc42*-deficient mice, showing intercellular separations in the latter (arrowhead). Images in **g** are from mice carrying the *iMb2-Mosaic* reporter and *Cdc42^flox/+^* control (top), or without a reporter and wild-type control (bottom). In **f**, *n* = 25 overlaps from four mice (Ctrl), *n* = 30 overlaps from four mice (Mut); in **h** *n* = 8 (139) (Ctrl) and *n* = 13 (231) (Mut) vessels (total junctions). Two-sided Mann–Whitney *U*-test. **i**, LYVE1 staining in lymphatic capillaries of control and *Cdc42*-deficient mice without permeabilization (cell surface LYVE1, cyan arrowheads) and with permeabilization (total LYVE1, red arrowheads). Boxed areas are magnified below. In **f**, **h**, data represent mean ± s.d. Two-sided Mann–Whitney *U*-test. Scale bars, 10 μm (**b,e,g,i**(overviews)), 5 μm (**i**(magnifications)).

increased curvature at cell–cell borders and formation of VE-cadherin aggregations (Fig. 5g,h). Notably, there were no apparent changes in the actin cytoskeleton (Fig. 5g).

Inhibition of CDC42 using the selective inhibitor ML141 reduced stretch-induced overlap formation (Fig. 5e,f,i), and resulted in ruptured monolayer integrity, evidenced by intercellular separations

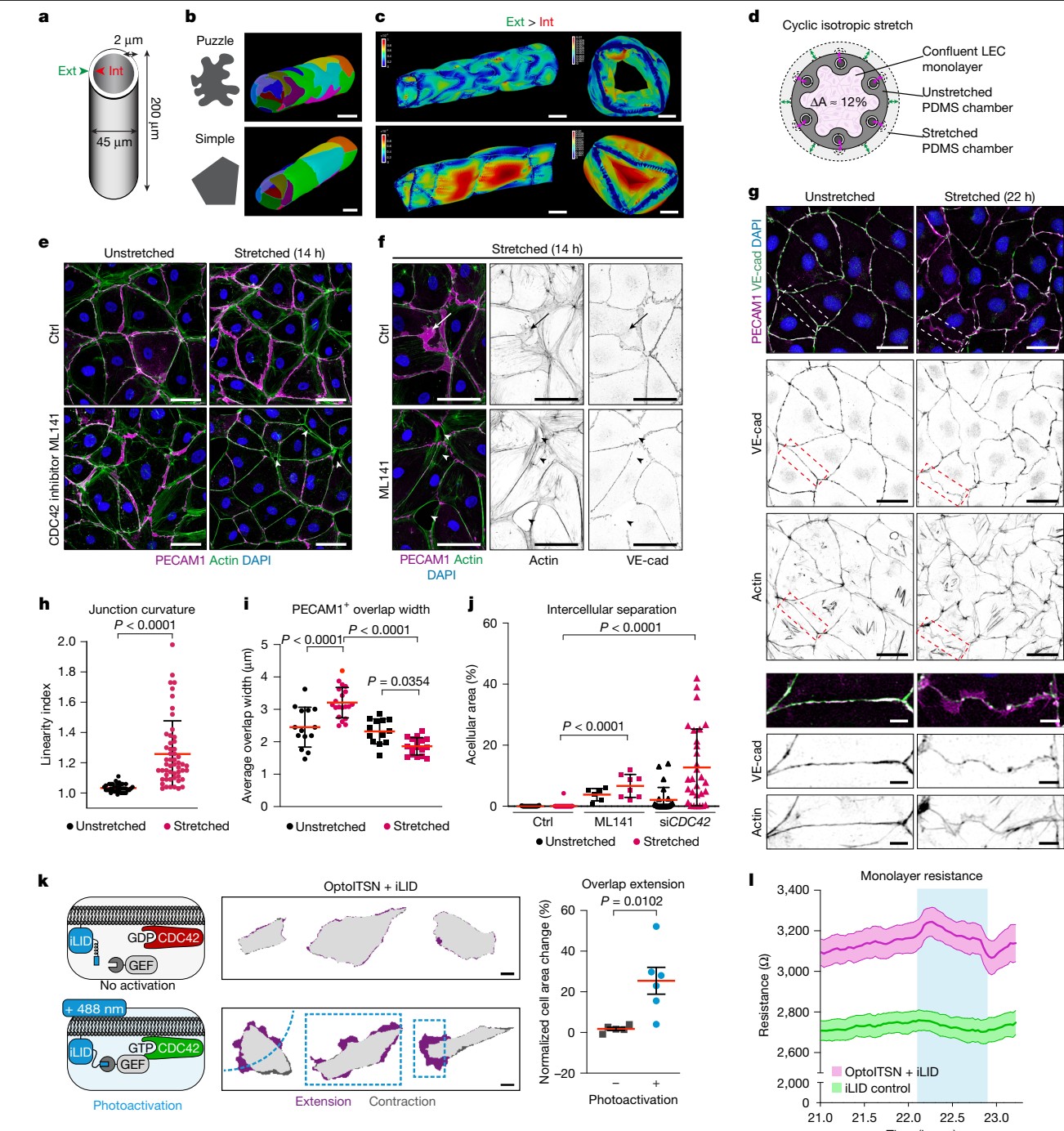

**Fig. 5 | Modelling the effects of mechanical stress on LEC monolayers.**
**a**, Morphological dimensions of a vessel used for FEM simulations. Ext, external wall pressure; int, internal wall pressure. **b**, FEM simulations of cellular stresses on a puzzle-shaped cell template from confocal data (top), or simple-shaped template with equivalent cells size and connectivity (bottom). **c**, Cellular stress patterns from FEM simulations under external pressure. Colour scale shows stress levels (kPa). **d**, Schematic of the isotropic stretching device with elastic PDMS chambers and area change (ΔA, dotted line). **e**,**f**, Immunofluorescence of human dermal LECs showing stretch-induced increase in PECAM1+ overlaps (arrow) compared to unstretched control (Ctrl), blocked by CDC42 inhibitor ML141. Actin (SPY-555), nuclear (DAPI) (**e**,**f**) and VE-cadherin (**f**) stainings are shown. Arrowheads indicate intercellular separations. **g**, Prolonged (22 h) stretch-induced changes in LEC shape with magnified views below. **h**–**j**, Quantification of junction linearity index (ratio of junction contour

length to straight-line junction length; *n* = 38, 43 images; 1 experiment) (**h**), cellular overlaps (*n* = 13, 19, 12, 16 images; 2–3 experiments) (**i**) and intercellular separations (*n* = 32, 34, 6, 8, 24, 29 images; 2–3 experiments) (**j**). Data represent mean ± s.d. Ordinary one-way ANOVA. **k**, Optogenetic CDC42 activation using improved light-induced dimer (iLID) for membrane recruitment of catalytically active RhoGEF ITSN1 (left) and cell area changes in LECs with and without photoactivation (10 min, dashed blue box), quantified on the right. Data represent mean ± s.e.m. (*n* = 5 cells, no activation; *n* = 6 cells, photoactivation). Two-tailed unpaired Student's *t*-test. **l**, Electrical resistance measurements of LEC monolayers with (pink) or without (green) OptoITSN1 photoactivation (cyan bar), using ECIS. Thick lines represent mean (*n* = 4 wells), with 95% CI. Scale bars, 20 μm (**b**,**c** (vessels)), 10 μm (**c** (cross sections),**k**), 50 μm (**e**,**f**), 25 μm (**g** (overviews)), 5 μm (**g** (magnifications)). Schematic in **k** adapted from ref. 34 under a CC BY 4.0 licence.

(Fig. 5f,j). This effect was more pronounced on *CDC42* silencing (Fig. 5j), whereas the integrity of unstretched *CDC42* small-interfering RNA (siRNA)-treated monolayers remained unaffected (Extended Data Fig. 9a–d). Conversely, activation of CDC42 using an optogenetically recruitable RhoGEF OptoISTN1 (ref. 34) induced dynamic cellular protrusions in LECs (Fig. 5k), resembling junction-based lamellipodia[35]. Time-lapse imaging of OptoITSN1-expressing LECs (yellow) revealed protrusions extending beyond VE-cadherin[+] junctions following local optogenetic activation (blue boxed region) (Supplementary Videos 7 and 8). Within 10 min of photoactivation, OptoITSN1-expressing LECs showed a 20–30% increase in cell area (Fig. 5k), whereas cells expressing the plasma membrane-localized optogenetic recruitment tool, the improved light-induced dimer (iLID), alone (data not shown and ref. 34) as well as OptoITSN-expressing LECs not exposed to photoactivation (Fig. 5k) did not show a response. Real-time impedance measurements revealed a high baseline transendothelial resistance of $2,207 \pm 589\ \Omega$ (s.d., $n = 4$ wells) in LECs (Fig. 5l), compared to that reported in other EC types (roughly $1,000–1,500\ \Omega$)[34,36]. OptoISTN-expressing LECs showed a higher resistance compared to control cells, suggesting baseline activity of iLID, which was further increased instantly after global photoactivation of CDC42 within LEC monolayer, and rapidly returned to baseline levels on deactivation (Fig. 5l). These results show that junction-based lamellipodia are associated with LEC monolayer integrity and barrier strength, in line with recent observations in human umbilical vein ECs (HUVECs)[34]. In support of the in vivo findings, integrin β1 inhibition using the function-blocking antibody mAB13 did not influence stretch-induced increase in PECAM1[+] overlaps (Extended Data Fig. 10a–c). Successful inhibition was confirmed at mAB13 concentrations of $0.1–0.2\ \mu g\ ml^{-1}$ (Extended Data Fig. 10a), which were previously shown to block integrin activity without compromising EC attachment[37].

The presence of cellular overlaps is also a feature of certain blood endothelia[38]. To assess if isotropic stretch induces overlaps in other ECs, we exposed HUVECs to stretch. Compared to LECs, HUVECs showed larger irregular cellular overlaps, formed by a reticular VE-cadherin network[39], and numerous stress fibres under baseline conditions (Extended Data Fig. 11a,b). The total cellular overlap area remained unaltered in HUVECs on stretching, but they showed stretch-induced VE-cadherin[+] finger-like protrusions (Extended Data Fig. 11a,b). These results show that, although isotropic stretch induces cellular overlaps and cell–cell contact curvature in LECs, the response differs between LECs and blood ECs.

## Discussion

The lobate oak leaf shape, shared between puzzle-shaped plant epidermal cells and mammalian lymphatic capillary LECs, stands out as a distinctive feature among the diversity of cell shapes observed in nature. Both cell types experience pressure-induced strain related to fluid fluxes—turgor pressure in plants and interstitial fluid pressure in lymphatic capillaries—linking their unique shapes to specialized functions. Similarities in cytoskeletal architecture and Rho GTPase-dependent regulation between plant puzzle cells and LECs, as found in our study, suggest parallel adaptations to withstand mechanical forces while maintaining tissue integrity. Moreover, our study uncovered dynamic, actin-based remodelling of cellular overlaps between capillary LECs in vivo during homeostasis and in response to increased interstitial fluid volume. We interpret our evidence as indicating that the dynamic remodelling is required for maintaining the cellular overlaps associated with the lobate cell shape, which in turn ensures integrity of the LEC monolayer under strain (Extended Data Fig. 12).

Fluid and immune cell entry into lymphatic capillaries is facilitated by discontinuous button junctions, which form from zippers during development[4,8]. In dermal lymphatic capillaries of the mouse ear pinna, zippers dominating at early postnatal stages remodel into buttons as vessel sprouting seizes[9–11,15]. In conditions associated with neo-lymphangiogenesis, including inflammation, the reverse button-to-zipper conversion may thus reflect and serve as a readout of vessel sprouting. In mature non-sprouting capillaries, we found that roughly 20% of capillary LEC lobes showed classical button junctions at 3 weeks of age, with no significant increase in frequency observed in older mice, suggesting limited further maturation. We identified two new capillary LEC junction types, curvilinear and double junctions, characterized by unsegmented or segmented linear distributions of VE-cadherin extending to the tips of LYVE1[+] lobe borders. Previous studies have recognized junctional heterogeneity, classifying 'intermediate' or 'transforming' junctions[8,11], which probably correspond to junction types identified here. However, these intermediate junctions have been primarily regarded as transient states during button-to-zipper or zipper-to-button transformation[8,10,11]. Notably, previous studies reported that button junctions constitute about 50% of junctions in initial lymphatics in mouse ear skin[10], but variations in classification criteria and methods preclude direct comparison of junction type frequencies across studies.

Pioneering work by Leak[22] demonstrated that the intercellular cleft between overlapping capillary LECs serves as the primary passage route for fluid and large molecules. EM analysis showed tracer passage stopping at tight junction barriers but freely passing through spatially separated junction-free intercellular clefts[22]. Consistent with Leak's studies showing both adherens and tight junctions in overlapping regions of adjacent LECs[22,40] and as reported previously[4,8], our immunofluorescence analysis revealed colocalization of VE-cadherin (adherens junctions) and CLDN5 (tight junctions) in LEC overlaps. Because confocal microscopy lacks the resolution of TEM, colocalization of immunofluorescence is described with the understanding that the two junction types are adjacent, not superimposed. Notably, in other vessel types solute and immune cell passage occurs without the presence of buttons and is regulated by phosphorylation of junctional proteins to induce reorganization of the junctions[38]. Also, leukocyte entry into collecting lymphatic vessels can occur through zipper junctions during inflammation[41–43], and alternative transcellular entry routes may contribute to fluid and solute uptake[22,44,45]. Mosaic analysis in our study revealed variability in LYVE1[+] cellular overlaps, their early emergence during development, and shortening on intradermal injection of fluid. This dynamic nature was further highlighted by intravital imaging and studies of genetic mouse models, which revealed continuous remodelling of F-actin-rich LEC overlaps under homeostasis and their dependency on CDC42-mediated regulation. By contrast, junctional proteins VE-cadherin[29] and CLDN5, as well as β1 integrins mediating cell-matrix adhesion, appear to have a more redundant role in homeostasis. This suggests that many tight junction proteins, such as ESAM or JAM-A[4], and cell-matrix adhesion receptors work cooperatively and compensate for each other's functions to maintain lymphatic capillary integrity.

The lobate shape of capillary LECs, recognized since the mid-nineteenth century[46], reflects their specialized function but has remained challenging to replicate in vitro. Cultured LEC monolayers show cobblestone morphology with continuous zipper junctions and mainly cortical actin[23,47] (this study), resembling collecting vessel LECs. In plant puzzle cells, isotropic stretch induces a lobate cell shape[6] by means of a Rho GTPase-dependent mechanism[26,27], increasing structural integrity of the cell wall[48]. In mammals, changes in interstitial fluid volume cause intermittent alterations in the diameter of lymphatic vessel lumens, subjecting the endothelium to repeated isotropic stretching. Using an in-house engineered device to apply isotropic stretch, we observed increased cellular overlaps and junction curvature in LEC monolayers. Inhibition of actin dynamics through CDC42 inhibition prevented the formation of stretch-induced overlaps and compromised monolayer resilience. Although the parameters for these experiments were not grounded in real measurements (see Supplementary Information for discussion on study limitations), and button junctions or

oak leaf shape were not observed under these conditions, our findings highlight isotropic stretch as a new regulatory force in the functional specialization of lymphatic capillaries, warranting further investigation. Notably, other vessel types, including different types of blood vessels and collecting lymphatic vessels, also undergo intermittent dilation and constriction and show prominent cellular overlaps. However, HUVECs responded differently to isotropic stretch, indicating EC type-specific responses, which may be further influenced by other types of mechanical forces in vivo. For example, blood vessels and collecting lymphatic vessels have thicker basement membrane, mural cell coverage, and are subjected to laminar shear stress, whereas capillary LECs uniquely experience isotropic stretch without substantial vessel wall- or flow-regulated forces.

Our data extend on the established key concept of discontinuity of capillary LEC junctions[4] and the role of their overlaps as passage routes for fluid and solutes[22]. The relatively low frequency of LEC lobes with button junctions observed by us in the dermal vasculature suggests the presence of a pool of LEC contacts specialized for fluid entry through a flap valve mechanism. The dynamic nature of LEC overlaps, with abundant presence of VE-cadherin at their borders, resembles junction-based lamellipodia that facilitate blood vessel morphogenesis in zebrafish[35] and increase endothelial barrier strength in vitro[34,39] (this study). We propose that capillary LEC lobes similarly use dynamic VE-cadherin positioning together with actin polymerization to adjust overlap area, enabling lumen shrinkage or expansion in response to fluid volume changes (Extended Data Fig. 12), reminiscent of a 'sliding valve' mechanism predicted by mathematical modelling[49]. Our findings further suggest that the lobate cell shape, along with its associated cytoskeletal and junctional organization, is critical for maintaining vessel integrity and dimension under mechanical stress while preserving a dynamic state of adherens junctions to support a permeable barrier for fluid passage. A thought-provoking implication of this model is that capillary LECs may actively contribute to fluid drainage. In this scenario, passive shortening of LEC overlaps and lumen expansion during oedema is countered by actin-based lobe remodelling to increase cellular overlap and vessel constriction aiding fluid propulsion, a process reminiscent of a bellows-like mechanism. The increase in cellular overlap would, in turn, promote tightening of the endothelial barrier to prevent fluid from re-entering the interstitium[50,51]. This mechanism would work together with the suction effect generated by the downstream collecting vessel contractions, controlled by lymphatic smooth muscle and skeletal muscle movements, to propel lymph[52]. Although a role for anchoring filaments in opening the junctions has also been suggested, such a function has yet to be experimentally demonstrated.

In summary, our study provides new insight into the regulation of lymphatic capillary function, in which cellular overlaps, linked to the lobate LEC shape, act as dynamic lamellipodia-like contact sites that enhance monolayer resilience to changes in interstitial fluid volume.

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

## Methods

### Mouse lines and treatments

*Tie2-cre* (*Tg(Tek-cre)[12Flv]*)[53], *Vegfr3-creER[T2]* (*Flt4[tm2.1(cre/ERT2)Sgo]*)[21] *Prox1-creER[T2]* (*Tg(Prox1-cre/ERT2)[1Tmak]*)[28], *R26-iMb2-Mosaic (Gt(ROSA)26Sor[tm1(CAG-EYFP*,-mEYFP*,-tdTomato*,-mTFP1*)Ben]*)[20], *Cldn5[flox]* (*Cldn5[tm1c(EUCOMM)Wtsi]*)[23], *Cdh5-GFP* (encoding VE-cadherin-GFP fusion protein)[16], *Cdc42[flox]* (*Cdc42[tm1Brak]*)[54] and *Itgb1[flox]* (*Itgb1[tm1Efu]*, The Jackson Laboratory, stock number 004605)[55] mice were previously described, and analysed on a C57BL/6J background, with the exception of the *R26-iMb2-Mosaic; Vegfr3-creER[T2]* mice used for intravital imaging experiments that were crossed to a C57BL/6-albino (*B6(Cg)-Tyrc-2J/J*) background. *R26-LifeAct-EGFP* (*Gt(ROSA)26Sor[tm1(CAG-EGFP)Tmak]*) mice were generated as described in the Supplementary Information. Cre-mediated recombination in *R26-LifeAct-EGFP* mice was induced by topical application of 50 µg of 4-hydroxytamoxifen (4-OHT, H7904, Sigma-Aldrich) dissolved in acetone (10 mg ml[−1]) to the dorsal side of each ear. Cre-mediated recombination in *R26-iMb2* mice, and gene deletion in mice carrying floxed alleles were induced by three (*Cldn5* or *Cdc42*) or five (*Itgb1*) consecutive administrations of 1 mg of tamoxifen (T5648, Sigma-Aldrich) dissolved in peanut oil (10 mg ml[−1], P2144, Sigma-Aldrich), by oral gavage. Littermate controls were included in each experiment, which were tamoxifen-treated Cre[−] mice or Cre[+] mice carrying a heterozygous floxed allele. Increased interstitial fluid volume was induced by intradermal injection of 20 µl of sterile PBS into the ear of sedated mice, and ears were processed after 10 min for further analysis. Experimental procedures on mice were approved by the Uppsala Animal Experiment Ethics Board (permit numbers 130/15, 5.8.18-06383/2020 and 5.8.18-0336/2021) or the National Animal Experiment Board in Finland (licence number ESAVI/15852/2022) and performed in compliance with all relevant national regulations.

### Antibodies

The details of primary antibodies used for immunofluorescence of whole-mount tissues and cells are provided in Supplementary Table 1. Secondary antibodies conjugated to Dylight405, AF405, AF488, AF555, AF594, AF647, AF680, Cy3 or horseradish peroxidase were obtained from Jackson ImmunoResearch Secondary antibodies conjugated to AF405+, AF488+, AF555+, AF594+ or AF647+, were obtained from Thermo Fisher Scientific. Actin was visualized using Phalloidin conjugated to AF647 (A22287, Invitrogen) or SPY-555 actin (SC202, Spirochrome). All were used in 1:200–1:1,000 dilution. Nuclei were visualized using 4,6-diamidino-2-phenylindole (DAPI) (1:1,000 in PBS). Fluorescent protein expression in the *iMb2-Mosaic* line was visualized using antibodies against HA epitope tag, dsRed or GFP.

### Culture and isotropic stretch of primary ECs

Primary human dermal LECs (HDLECs) from juvenile foreskin (C-12216, PromoCell) and primary HUVECs (C-12200, PromoCell) were maintained on fibronectin-coated (2 µg ml[−1]; F1141-2MG, Sigma) cell culture dishes at 37 °C and 5% $CO_2$. The cell lines were authenticated based on morphology and immunohistochemistry profile and tested negative for mycoplasma contamination. HDLECs were supplied with complete Endothelial Cell Growth Medium 2 (ECGMV2; C-22022, PromoCell) and HUVECs with complete ECGMV (C-22010, PromoCell). Cells were passaged up to six times using Trypsin-EDTA (25300054, Thermo Fisher Scientific), diluted to 0.025% with Dulbecco's PBS (DPBS), before being used for the stretching experiments. The autoclaved PDMS chambers were functionalized for cell culture usage with 0.5 mg ml[−1] Sulpho-SANPAH, diluted in sterile MilliQ water, under ultraviolet light for 10 min. The functionalized chambers were extensively washed in DPBS (14190-094, Gibco) and coated with fibronectin before seeding of cells (3 × 10[4] cells per mm[2]). 72 h after cell seeding, the chambers were transferred into 2% prestretch holders and cells were supplied with the complete ECGMV or ECGMV2 supplemented with 0.1% v/v

Pen Strep (15140-122, Gibco) and 1 mM HEPES Buffer Solution (15630-056, Gibco). After 12 h, the chambers were transferred into the Multi-Stretcher device, developed as an extension of the IsoStretcher system[22] to enable a simultaneous in-plane isotropic stretch of four custom-moulded PDMS chambers in parallel as described in the Supplementary Information. Details of the MultiStretcher device will be presented elsewhere (F.L., S.S. and O.F., in preparation). Cyclic isotropic stretching corresponding to roughly 12% change in membrane area was applied at 0.01 Hz (100 s per stretch cycle) for 14 h or 22 h at 37 °C and 5% $CO_2$. Unstretched chambers, kept in prestretch holders, were used as a control. Young's modulus of the PDMS substrate was measured as 2,391 ± 95 kPa (ref. 56).

For integrin β1 inhibition, HDLECs were treated at the onset of stretching with 0.1 or 0.2 µg ml[−1] rat anti-human CD29 (mAb13) (552828, BD Pharmingen). For pharmacological inhibition of CDC42, HDLECs were treated at the onset of stretching with 15 µM ML141 (SML0407-5mg, Sigma) or vehicle (DMSO, 276855-100ML, Sigma-Aldrich). For silencing of *CDC42* expression, HDLECs were first grown to confluence in fibronectin-coated PDMS chambers. Then 34 h before the start of the 14 h long stretching experiment, cells were transfected with either negative control (462001, Invitrogen) or CDC42 Stealth (HSS190761, Thermo Scientific) siRNA at the final concentration of 20 nM using Lipofectamine RNAiMAX Transfection Reagent (56531, Invitrogen) according to the manufacturer's instructions.

### Immunofluorescence

HDLECs and HUVECs were fixed inside the stretch chambers (in stretched state) with ice-cold 4% paraformaldehyde (PFA) in PBS for 10 min, followed by washing in PBS and permeabilization and blocking in 0.2% IGEPAL CA-630 (18896-50ML, Sigma) plus 3% bovine serum albumin (A3295, Sigma-Aldrich) in PBS for 1 h at room temperature. Primary antibodies dissolved in cell blocking buffer were added on cells and incubated overnight at 4 °C. After washing in PBS, cells were incubated with secondary antibodies and SPY-555 actin for 3 h at room temperature. Nuclear staining was done using DAPI (MBD0015-1ML, Sigma-Aldrich) in PBS for 10 min at room temperature, followed by further washing and imaging. To maintain the active conformation of integrin β1, HDLECs stained for active integrin β1 were washed with 1 mM $MgCl_2$ in ice-cold DPBS and then fixed inside the chambers with ice-cold 4% PFA plus 1 mM $MgCl_2$ in PBS for 10 min.

### Optogenetic activation of CDC42

Lentiviral particles for Lck-mTurquoise2-iLID and SspB-HaloTag-ITSN1(DHPH) were produced in human embryonic kidney 293T cells (CRL-3216, American Tissue Culture Collection) as described in ref. 34. The cell line was authenticated based on morphology and tested negative for mycoplasma contamination. The HDLECs were transduced with both lentiviruses 6 days before imaging, and puromycin was added 3 days before imaging. Transduced HDLECs were subsequently seeded onto glass-bottom 12-well culture plates (ø14 mm, MatTek Corporation), coated with fibronectin and grown into confluent monolayers in complete ECGM2-MV medium. During imaging, HDLECs were grown in microscopy medium (20 mM HEPES (pH 7.4), 137 mM NaCl, 5.4 mM KCl, 1.8 mM $CaCl_2$, 0.8 mM $MgCl_2$ and 20 mM glucose) at 37 °C and 5% $CO_2$. The HaloTag was stained 3 h before imaging, with a concentration of 150 nM of Janelia Fluor Dye (JF) JF552nm (red) (Janelia Materials). The medium was replaced before imaging. VE-cadherin was stained by adding Alexa Fluor 647 mouse anti-human CD144 (BD Pharmingen, 561567, 1:40) to the medium 2 min before the start of imaging. Details of the imaging parameters are provided in Supplementary Information. Before photoactivation, Lck-mTurquoise2-iLID was detected with a photomultiplier tube detector (gain 800 V), 447–523 nm emission detection range using a 442-nm diode laser line at 1% intensity. The 442-nm laser line was turned off during photoactivation experiments. Photoactivation was achieved by scanning the defined region

of interest with a 488-nm laser line, the argon laser power set to 15% and intensity 5%, every 2.58 s for a total of 2.5 min. During photoactivation, SspB-HaloTag-ITSN1(DHPH) stained with JF552nm was detected with a photomultiplier tube detector (gain 750 V), 566–629 nm emission detection range with a DPSS 561-nm laser line at 1% intensity. Sequentially, VE-cadherin-Alexa Fluor 647 signal was detected using a HyD detector (gain 50%), 652–775 nm emission detection range in combination with a 647-nm HeNe laser line at 3% intensity.

Resistance measurements were performed using electrical cell-substrate impedance sensing (ECIS) ZTheta (Applied BioPhysics) at 4,000 Hz, representing paracellular permeability, every 10 s at 37 °C and 5% $CO_2$. ECIS arrays (eight-well, ten-electrode, 8W10E PET) were pretreated with 10 mM cysteine (Sigma) for 15 min at 37 °C, washed twice with 0.9% NaCl solution and coated with 10 µg ml$^{-1}$ fibronectin in 0.9% NaCl solution (Sigma) for at least 1 h at 37 °C. Lentivirally transduced HDLECs were seeded at 50,000 cells per well density to grow into a monolayer, and measurements were started immediately after seeding ($n$ = 4 wells per condition). For global photoactivation, started roughly 22 h after seeding, an RGB LED safety strip (Combo 12 V/24 V SMD 3528/50505, Fuegobird) was taped to the lid of the cell culture dish with a rough distance of 1 cm from the cell monolayer and set to blue light (peak 470 nm, highest intensity setting 9) for 45 min. The corresponding blue LED light spectrum was measured as described in ref. 34.

## Whole-mount immunofluorescence

Most tissues (juvenile and adult ear skin, adult diaphragm or embryonic back skin) were harvested from mice that were euthanized by cervical dislocation or $CO_2$ asphyxiation, and immediately dissected and placed for fixation in 4% PFA for 2 h at room temperature or 4 h at 4 °C. Tissue from *Cdh5-GFP* (Fig. 1a,b and Extended Data Fig. 1a) and *Itgb1$^{flox}$* (Extended Data Fig. 6d,e) mice was harvested after transcardial perfusion following three different protocols, yielding a similar diversity of LEC junctions compared to immersion fixation (Supplementary Fig. 2c): (1) perfusion with 10 ml of PBS, followed by 10 ml of 4% PFA (room temperature), ear collection and postfixation by immersion in 4% PFA overnight at 4 °C, (2) perfusion with Hanks balanced salt solution, followed by 4% PFA, ear collection and postfixation by immersion in 4% PFA for 4 h at 4 °C and (3) direct perfusion with 1% formaldehyde for 2–3 min, ear collection and postfixation by immersion in 2% PFA for 4 h at 4 °C. Skin on the dorsal side of the ear pinna was dissected from the underlying cartilage layer before (immersion) or after (perfusion) fixation. Tissues were permeabilized in 0.3% Triton X-100 in PBS (PBST) for 10 min. After blocking in PBST with 2% bovine serum albumin and 1% FBS for 2 h, tissues were incubated with primary antibodies in blocking buffer overnight, followed by PBST washing and incubation with fluorescent dye-conjugated secondary antibodies for 2 h. All incubation steps were carried out at room temperature. Before mounting in Mowiol, samples were repeatedly washed in PBST and water. Details of consecutive staining of surface and total LYVE1, and staining using in vivo injected LYVE1 antibody and total LYVE1 as described in Supplementary Information.

## Silver nitrate staining

Whole-mount AgNO$_3$ staining was performed on immersion-fixed ear skin. Tissue was first blocked in blocking buffer (3% bovine serum albumin, 1% fetal bovine serum in TBS (Tris-buffered saline)) and subsequently stained using primary antibodies overnight at 4 °C dissolved in blocking buffer. The next day, tissue was washed in TBS and incubated with secondary antibodies dissolved in blocking buffer for 2 h at room temperature. Tissue was washed again in TBS for 2 h and was further processed for silver nitrate staining. In brief, tissue was washed twice for 1 min in D-glucose solution (280 mM), followed by immersion for 1 min in freshly filtered AgNO$_3$ solution (15 mM) protected from light. After washing for a further 2 min in fresh glucose solution, the tissue

was mounted and immediately underwent imaging. Silver stain was slowly developed under the microscope by shining white light using the microscope's brightfield function until the desired contrast was reached. Sequential channel imaging was performed, whereby silver particles were imaged using far-red (685 nm) reflected light followed by antibody fluorescent signals (AF594 and AF555, respectively). See Supplementary Information for details about the specificity of silver staining.

## Confocal microscopy and image processing

Confocal images were obtained using a Leica SP8 or Leica Stellaris 5 confocal microscope. Details of lasers and objectives are described in Supplementary Information. Images were deconvolved with Huygens Essential software (v.19.04) (Scientific Volume Imaging) (Figs. 3b,e, 4b and 5f–h, Extended Data Figs. 9d, 10a,b and 11a) or Leica Lightning (Figs. 2h and 4g,i, Extended Data Figs. 2b, 4a and 6b and Supplementary Fig. 6e), which is part of LasX software. Huygens deconvolution was used by using a theoretical point spread function and automatic background estimation. Stopping criteria were set to 40 iterations and a signal-to-noise ratio of 10. Lighting deconvolution was used using an adaptive approach, using the fitting optical parameters in terms of objective lens, corresponding wavelength. Maximum iterations were set to 20 with smoothing. Further processing was done using Fiji and ImageJ. Images represent maximum intensity projections of individual *Z*-stacks unless indicated otherwise. Processing of time-lapse intravital imaging videos was performed in Fiji and ImageJ. Individual frames were stabilized and 3D drift corrected using the image registration plug-in. To remove noise, a rolling average algorithm was used, part of the Multi Kymograph plug-in, which averages three subsequent frames.

## Intravital multiphoton microscopy

Animals undergoing intravital imaging were sedated with an intraperitoneal injection of 100 mg kg$^{-1}$ ketamine and 12.5 mg kg$^{-1}$ xylazine dissolved in sterile saline. The dorsal ear skin was immobilized on a custom-made 3D printed stage for imaging. Animals received eye cream and thermal support during the entire imaging session, and those undergoing longitudinal imaging, were rehydrated using an intraperitoneal injection of saline after imaging. Imaging was performed using a LEICA SP8 DIVE platform equipped with a Ti:Sapphire multiphoton laser emitting a 680–1,300 nm tunable and 1,045 nm fixed laser line. All imaging was done using a HC IRAPO ×25/1.0 numerical aperture (NA) objective.

## Image quantification

Details of image quantification are provided in Supplementary Information. Images of annotated lymphatic capillary junctions in mouse ear skin used for Fig. 1g are available at Zenodo (https://doi.org/10.5281/zenodo.13880404)[57]. Four categories were defined: (1) button junction, a punctate VE-cadherin$^+$ deposit at the neck of LYVE1$^+$ lobe/overlap, with no detectable VE-cadherin at the borders of the overlap; (2) curvilinear junction, unsegmented or segmented distribution of VE-cadherin within one border of LYVE1$^+$ lobe/cellular overlap; (3) double junction, unsegmented or segmented distribution of VE-cadherin within both borders of LYVE1$^+$ lobe–cellular overlap; (4) LYVE1$^-$ curvilinear junction, unsegmented linear VE-cadherin distribution at cell–cell contacts in the absence of LYVE1 and (5) zipper junction, continuous linear VE-cadherin distribution surrounding the entire cell in the absence of LYVE1.

## Dextran clearance assay

Animals undergoing dextran clearance assay were sedated with an intraperitoneal injection of 100 mg kg$^{-1}$ ketamine and 12.5 mg kg$^{-1}$ xylazine dissolved in sterile saline. Animals received eye cream and thermal support during the entire imaging session. The dorsal ear skin was immobilized on a custom-made 3D printed stage for imaging and 1 µl of tracer solution containing 5 mg ml$^{-1}$ TRITC-conjugated dextran

(150 kDa, FD150, Sigma-Aldrich) was injected intradermally using a 32 G needle size Hamilton syringe. The entire ear was imaged using a HCX PL FLUOTAR ×5/0.15 NA objective within 3 min and animals were allowed to recover. After 4 h, animals were resedated and underwent a second round of imaging using the same imaging parameters to determine the clearance of injected tracer.

## TEM

Mice were euthanized by cervical dislocation or $CO_2$ asphyxiation. Skin on the dorsal side of the ear pinna was immediately dissected from the underlying cartilage layer, transferred to 2.5% glutaraldehyde (Ted Pella) + 1% PFA (Merck) in 0.1 M phosphate buffer pH 7.4 and incubated in the fixative at 4 °C overnight. Fixed ears were cut longitudinally along the proximal–distal axis into 2–3 mm strips and further incubated in fresh fixative, which was replenished once more before dehydration. The strips from the central region of the ear were further divided into three parts in the proximal–distal axis. Only the distal part close to the tips of the ear that lacks larger collecting vessels and contains a plexus of lymphatic capillaries and precollecting vessels composed of oak leaf-shaped LECs was used for analysis. Before sectioning, samples were washed in 0.1 M phosphate buffer, incubated in 1% osmium tetroxide in 0.1 M phosphate buffer for 1 h, rinsed again, dehydrated in increasing concentrations of ethanol (50, 70, 95 and 99.9%) for 10 min, and incubated in propylene oxide for 5 min. Samples were then placed in Epon Resin (Ted Pella) and propylene oxide (1:1) for 1 h, followed by 100% resin overnight. Subsequently, samples were embedded in capsules in newly prepared Epon Resin and after 1–2 h finally polymerized at 60 °C for 48 h. Sectioning was carried out using an EM UC7 Ultramicrotome (Leica). Then 60–70-nm-thin sections were placed on a grid and contrasted with 5% uranyl acetate and Reynold's lead citrate. Imaging was carried out on a Tecnai G2 Spirit BioTwin transmission electron microscope (Thermo Fisher/FEI) at 80 kV with an ORIUS SC200 CCD camera and Gatan Digital Micrograph software (both Gatan Inc.).

## scRNA-seq data analysis

Differentially expressed genes between the different LEC subtypes were visualized using a Shiny-based web application for dermal mouse LECs[25] available at https://makinenlab.shinyapps.io/DermaLymphaticEndothelialCells/.

## FEM simulations

The FEM simulations were performed with MorphoMechanX using available models adapted from ref. 27. A regular cylindrical grid of 45 μm wide and 200 μm long was created and outlines from the cells of a lymphatic vessel obtained from confocal imaging were projected onto it and smoothed. These cells were then extruded inward to generate 3D volumetric cells with a depth of 2 μm and triangulated using a threshold area of 4 μm. The template was used as the reference configuration for triangular three-node membrane elements that were given a thickness of 0.1 μm. An isotropic St. Venant material model (linear, large deformation) was used with the Young's modulus set to 100 kPa to match a 10-kPa cell level Young's modulus estimated from the literature for ECs[30–32] (ignoring the cell ends, the 2 × 0.1 μm membrane thickness occupied roughly one-tenth the cross-sectional area of the cells that were 2 μm deep). A uniform internal pressure was applied normal to the inside faces of the elements, which cancels out on the shared walls between cells. For simulations with a lower pressure inside the vessel, the inside faces were assigned a higher pressure. Stresses were visualized as the trace of the stress tensor.

## Statistics

Graphpad Prism (v.9) was used for graphic representation of the data. No statistical methods were used to predetermine sample size. For in vivo experiments, a minimum of three mice per condition was used,

except for Figs. 2j and 4c,d, $n = 2$ for 9-week-old mice. For in vitro experiments a minimum of three biological replicates were used, except for Extended Data Fig. 10a (validation of integrin beta1 inhibition), $n = 1–2$ stretch holders. The sample size of three was chosen as the minimum required to perform statistical tests. Allocation of mice into experimental groups was based on genotype. Data were collected from different litters on different days and experiments were performed for different batches at different time points. For in vitro experiments, allocation into experimental groups was performed randomly. No blinding was done in the analysis and quantifications. Data between two groups were compared using an unpaired two-tailed Student's $t$-test assuming equal variance. When the data were not normally distributed Mann–Whitney $U$-test was used instead. Data between multiple groups were compared using an ordinary one-way analysis of variance (ANOVA) or Brown–Forsythe and Welch ANOVA followed with multiple testing, and categorical variables were compared using Fisher's exact test. Differences were considered statistically significant when $P < 0.05$.

## Reporting summary

Further information on research design is available in the Nature Portfolio Reporting Summary linked to this article.

## Data availability

Images of annotated lymphatic capillary junctions in mouse ear skin used for Fig. 1g are available at Zenodo (https://doi.org/10.5281/zenodo.13880404)[57]. All other data supporting the findings are available within the paper and its Supplementary Information. Source data are provided with this paper.

## Code availability

Code for FEM simulations is based on available models adapted from ref. 27. FEM modelling software and the models are available at Zenodo (https://doi.org/10.5281/zenodo.13880404)[57].

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

**Acknowledgements** We thank S. Ortega (CNIO, Madrid) for the *Vegfr3-creER*[T2] mice and U. Deutsch (Theodor Kocher Institute, University of Bern, Bern) for the *Cdh5-GFP* tissue preparation. We also thank M. Kraft and F. Hofmann for advice on scRNA-seq data analysis; C. Betsholtz and I. Ferby for discussion and insightful comments; K. Staxäng and M. Hodik (The BioVis platform of Uppsala University) for TEM analysis; as well as the BioVis platform for the two-photon microscope usage and support; and S. Lunell Segerqvist, H. Ortsäter, C. Rorsman and A. M. Camara for technical assistance. 3D printing of the microscope stage for the MultiStretcher chambers and 2PM imaging was performed at U-PRINT: Uppsala University's 3D printing facility at the Disciplinary Domain of Medicine and Pharmacy and SciLifeLab Uppsala. This work was supported by grants from Knut and Alice Wallenberg Foundation (grant nos. 2018.0218 to T.M. and 2020.0057 to T.M. and K.G.), the Swedish Research Council (grant nos. 2020-02692 to T.M. and 2021-04896 to K.G.), Göran Gustafsson foundation (to T.M.), the Swedish Cancer Society (grant nos. 9 0220 Pj, 22 2025 Pj to T.M.), the European Union's Horizon 2020 research and innovation programme under the Marie Skłodowska-Curie grant agreement no. 814316 (to H.S. and T.M.), the Swiss National Science Foundation (grant no. 310030_189080 to B.E.), and the Research Council of Finland's Centre of Excellence Program (grant no. 307366 to P.S.). O.F. received funding through an R&D grant by the Bavarian Ministry of Economy & Energy (grant no. 41-6618c/587/2-LSM-2303-0015). R.S.S. was supported by a Biotechnological and Biological Sciences Research Council (BBSRC) Institute Strategic Programme Grant to the John Innes Centre (grant no. BB/X01102X/1).

**Author contributions** H.S. conceived and designed the study; designed, performed, analysed and interpreted data from the analysis of LEC junctions and overlaps and intravital imaging; wrote the manuscript. N.D. conceived and designed the study; designed, performed, analysed and interpreted data from the analysis of LEC cytoskeleton and TEM experiments; contributed with

initial conceptual ideas and to the writing of the first draft of the manuscript. S. Schnabellehner designed, performed, analysed and interpreted data from in vitro stretch experiments. M.L.B.G. and S.P.M. designed, performed, analysed and, with J.D.v.B., interpreted data from optogenetics experiments. A.H. and S.B. performed experiments on *Itgb1*<sup>flox</sup> and *Cdh5-GFP* mice, respectively. A.M.M. and M.J. performed and analysed echocardiography experiments. F.L., S. Schürmann and O.F. designed and manufactured the MultiStrecher device. R.S.S. and M.M. performed, analysed and interpreted data from computer simulations. C.B., R.B., B.E., D.V., K.G. and P.S. provided mouse lines. T.M. conceived and designed the study, analysed and interpreted data, supervised the project and wrote the manuscript. All authors discussed the results and commented on the manuscript.

**Funding** Open access funding provided by Uppsala University.

**Competing interests** F.L., S. Schurmann and O.F. have filed a patent application related to the MultiStretcher device as inventors (DPMA 10 2023 205 399.3). The other authors declare no competing interests.

**Additional information**

**Correspondence and requests for materials** should be addressed to Taija Mäkinen.

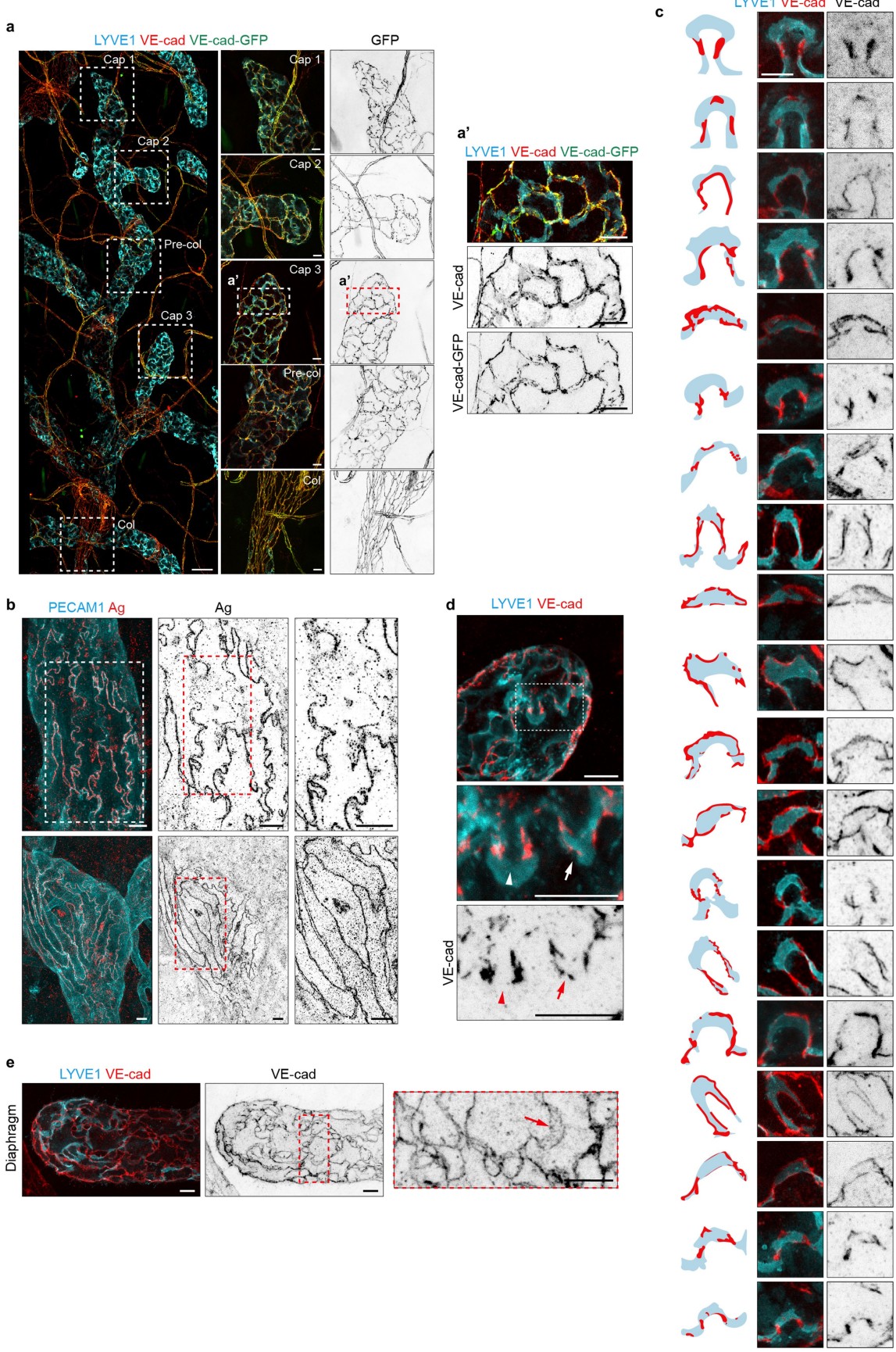

**Extended Data Fig. 1** | See next page for caption.

**Extended Data Fig. 1 | Junctional heterogeneity in capillary LECs.** (**a**) Whole-mount immunofluorescence of lymphatic vessels in the ear skin of a 25-week-old *Cdh5-GFP* mouse expressing VE-cadherin-GFP fusion protein. Boxed areas are magnified on the right (Cap 2, Pre-col and Col also in Fig. 1a) and in **a'** as indicated. Note the overlap of VE-cadherin-GFP signal with VE-cadherin immunostaining. Similar results were obtained from 5 mice in two independent experiments. (**b**) Whole-mount silver nitrate (Ag) staining of mouse ear dermis showing depositis around cell perimeter in lymphatic capillaries (top) and collecting vessels (bottom). Boxed areas are magnified on the right. Similar results were obtained from 4 mice in two independent experiments.

(**c**) Whole-mount immunofluorescence of individual dermal capillary LEC lobes with corresponding schematic drawings, showing a spectrum of VE-cadherin+ junctional arrangements within initial regions of adult lymphatic capillaries. Similar results were obtained from 15 mice in three independent experiments. (**d**, **e**) Whole-mount immunofluorescence of ear skin (3 wk, **d**) and diaphragm (25 wk, **e**) of a wild-type mouse showing junctional heterogeneity, with buttons (arrowheads) as well as curvilinear and double junctions (arrows). Similar results were obtained from 15 mice in three independent experiments. Boxed areas are magnified below (**d**) and on the right (**e**). Scale bar: 50 µm (a, overview), 10 µm (a, a', b, c, d, e).

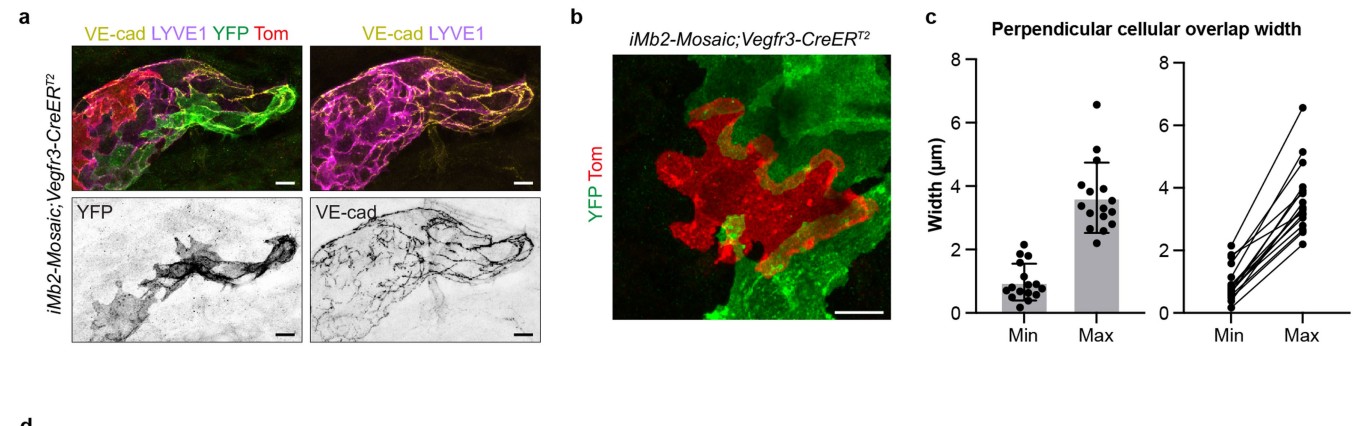

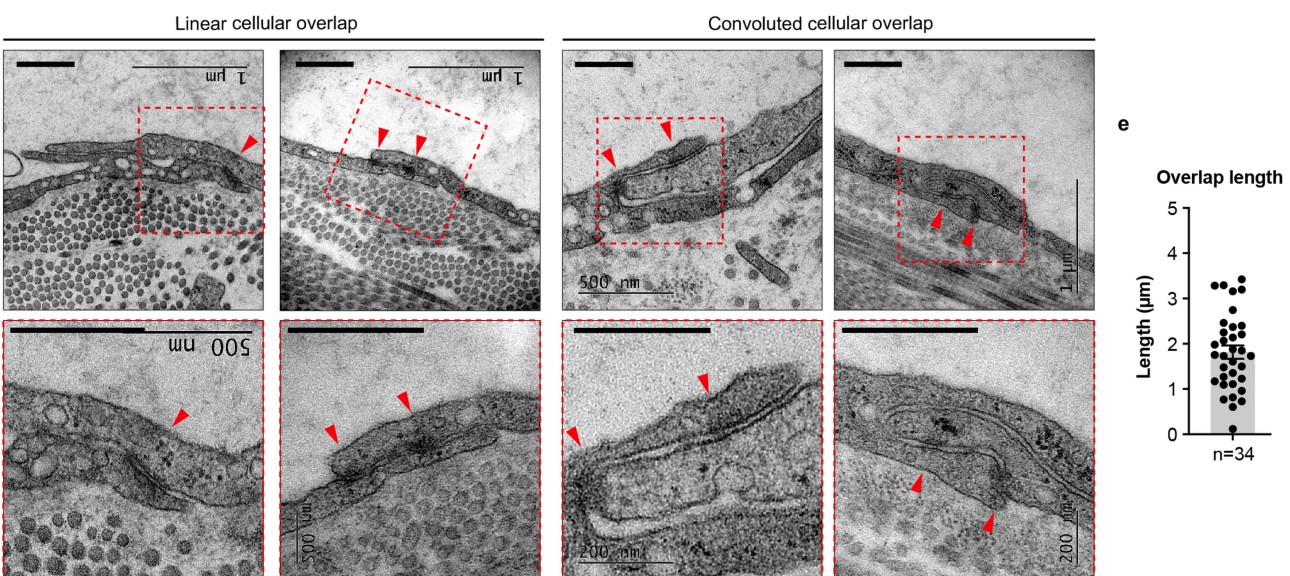

**Extended Data Fig. 2 | Analysis of LEC overlaps.** (**a, b**) Whole-mount immunofluorescence of lymphatic capillaries in *iMb2-Mosaic;Vegfr3-CreER^{T2}* mice, showing a transition in LYVE1 expression, cell shape and junction morphology within a sprout tip (**a**, 3-week-old), and large intercellular overlaps between neighboring capillary LECs (**b**, 9-week-old). Similar results were obtained from 4 mice in two independent experiments (a) or 5 mice in three independent experiments (b). (**c**) Minimum and maximum perpendicular width of extracted overlaps from control mice (from Fig. 2i), represented as mean ± s.d. (left) or as minimum/maximum range within individual overlaps (right) (*n* = 16 overlaps from 4 mice in two independent experiments). No statistical testing was performed. (**d**) TEM analysis of LEC contacts showing variable morphology and junctional organization. Red arrowheads point to electron dense junctions. Boxed areas are magnified below. Similar results were obtained from 3 mice. (**e**) Quantification of the length of overlaps from TEM data, below a 4 µm cut off, represented as mean ± s.d. (*n* = 34 overlaps from 3 mice). Scale bar: 10 µm (a, b), 500 nm (d).

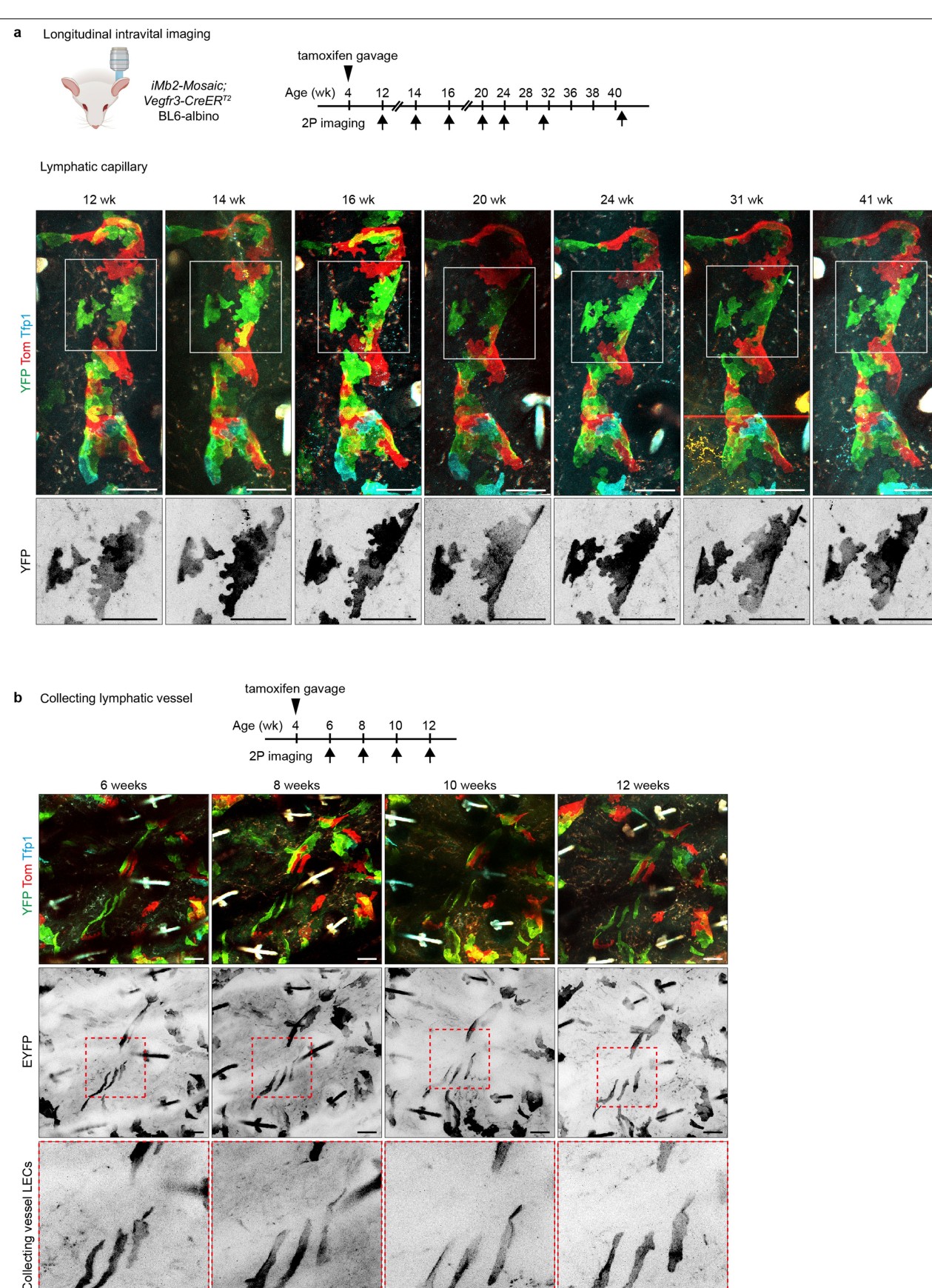

**Extended Data Fig. 3 |** See next page for caption.

Extended Data Fig. 3 | Extended longitudinal intravital imaging of capillary LECs. (a) Longitudinal intravital imaging of an *iMb2-Mosaic;Vegfr3-CreER*[T2] BL6-albino mouse over an extended observational period of 29 weeks. Boxed areas are magnified below. Similar results were obtained from 5 mice over a follow-up period of 8 weeks in two independent experiments, and one mouse over a follow-up period of 35 weeks. (b) Intravital imaging of collecting vessel LECs in an *iMb2-Mosaic;Vegfr3-CreER*[T2] BL6-albino mouse showing no noticeable cell shape changes. Similar findings were obtained from 5 mice in two independent experiments. Scale bar: 50 μm (a, b). Illustration in **a** created using BioRender (https://biorender.com).

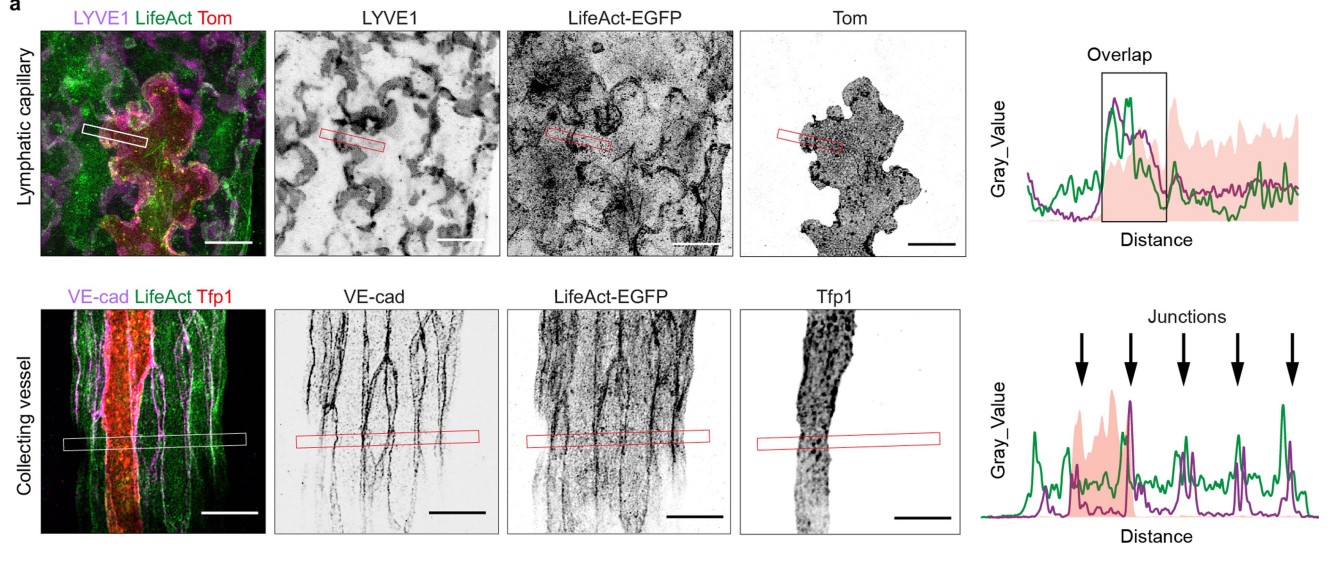

**a**

Lymphatic capillary

LYVE1 LifeAct Tom | LYVE1 | LifeAct-EGFP | Tom

Overlap

Collecting vessel

VE-cad LifeAct Tfp1 | VE-cad | LifeAct-EGFP | Tfp1

Junctions

**b**

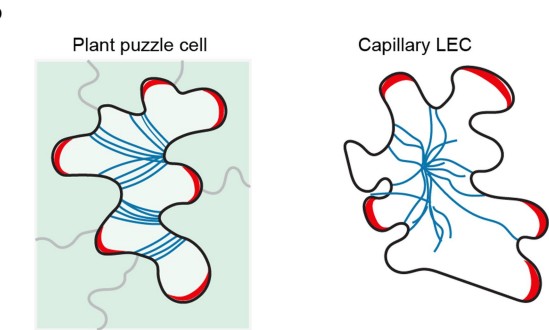

Plant puzzle cell          Capillary LEC

microtubules actin

**Extended Data Fig. 4 | F-actin localisation in LECs of lymphatic capillaries and collecting vessels.** (**a**) Visualization of F-actin in LECs from adult *iMb2-Mosaic;LifeAct-EGFP;Vegfr3-CreER^T2* mice with intensity plots corresponding to the boxed areas. Note F-actin enrichment in LYVE1⁺ lobe of capillary LEC and narrow actin peaks colocalised with VE-cadherin in collecting vessel LECs. Similar results were obtained from 3 mice in two independent experiments. (**b**) Comparison of actin and microtubule organization in lobate puzzle cells of plant epidermis[6] and capillary LECs. Scale bar: 10 μm (a).

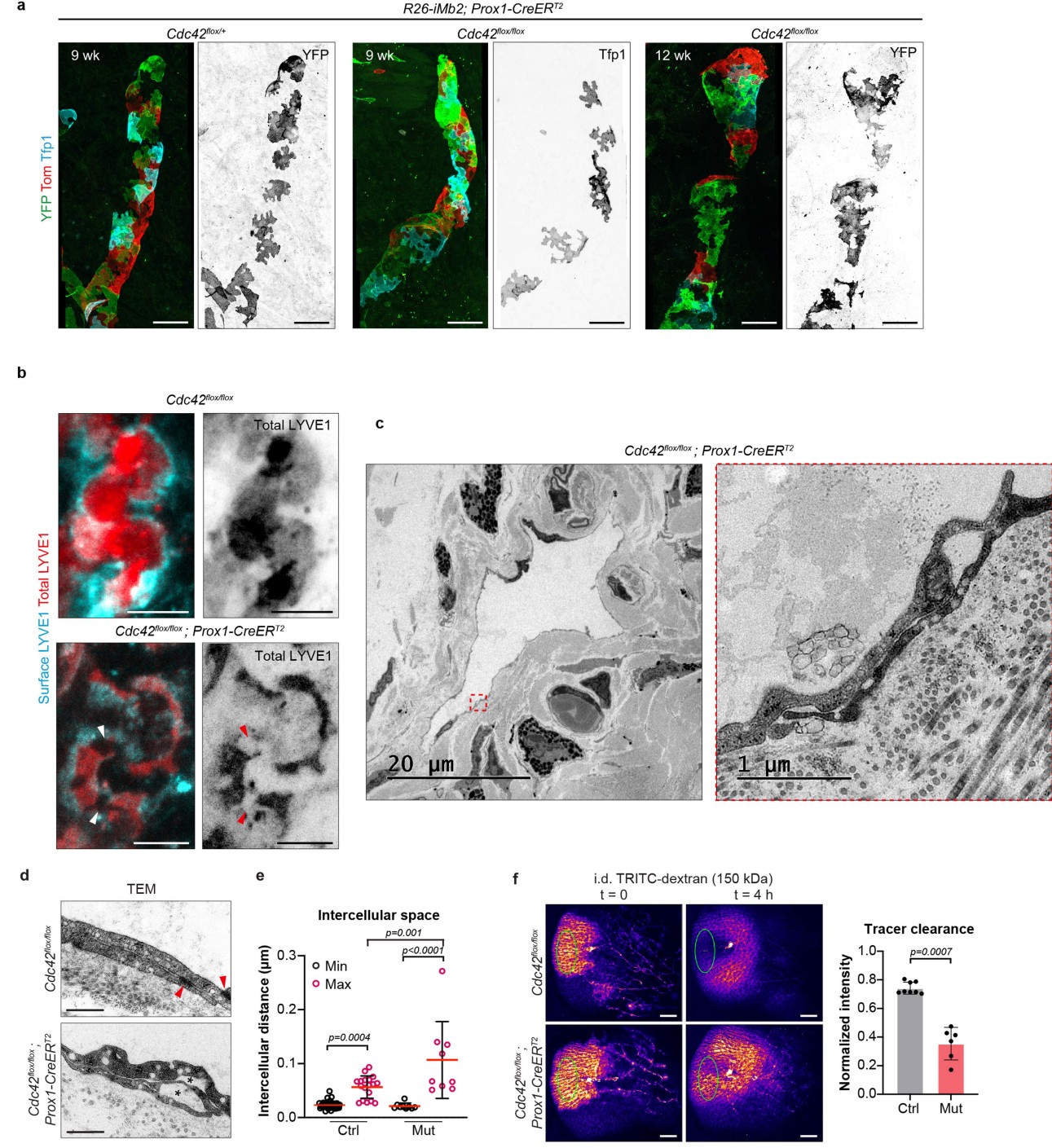

**Extended Data Fig. 5 | Characterisation of cellular defects in *Cdc42*-deficient lymphatic capillaries.** (**a**) Whole-mount immunofluorescence of lymphatic capillaries in control (flox/+) and *Cdc42*-deficient (flox/flox) mice expressing the *iMb2-Mosaic* reporter. Tamoxifen was administered at 6 weeks and ears analyzed at 9 or 12 weeks of age as indicated. Similar findings were obtained from *n* = 8 mice (flox/+), *n* = 4 mice (flox/flox 9 weeks) and *n* = 4 mice (flox/flox 12 weeks). (**b**) Whole-mount immunofluorescence of LEC overlaps and junctions, showing LYVE1⁻ areas within the overlap regions (arrowheads) in *Cdc42*-deficient vessels. Similar results were obtained from 3 mice in two independent experiments. (**c-e**) TEM analysis of LEC overlaps in *Cdc42*-deficient mice two weeks after tamoxifen administration. Whole vessel

overview (**c**, left) and high magnification of a cell-cell contact in boxed area, showing intercellular separation at the overlap region (**c**, right; **d**, asterisks), quantified in (**e**). Red arrowheads in (**d**) point to electron dense junctions. Mean ± s.d., *n* = 18 junctions from 4 mice (Ctrl) and *n* = 10 junctions from 4 mice (Mut) in two independent experiments. Ordinary one-way ANOVA. (**f**) Clearance of intradermally injected tracer (1 μl, 150 kDa TRITC-dextran) in 11-week-old control (Ctrl) and *Cdc42* deficient (Mut) mice 5 weeks post-tamoxifen, normalized to initial intensity at timepoint 0 (mean ± s.d., *n* = 8 mice (Ctrl) and *n* = 6 mice (Mut) in two independent experiments. Two-sided Mann Whitney U test. Scale bar: 50 μm (a), 1 μm (b, c, magnification), 20 μm (c, overview), 0.5 μm (d), 1 mm (f).

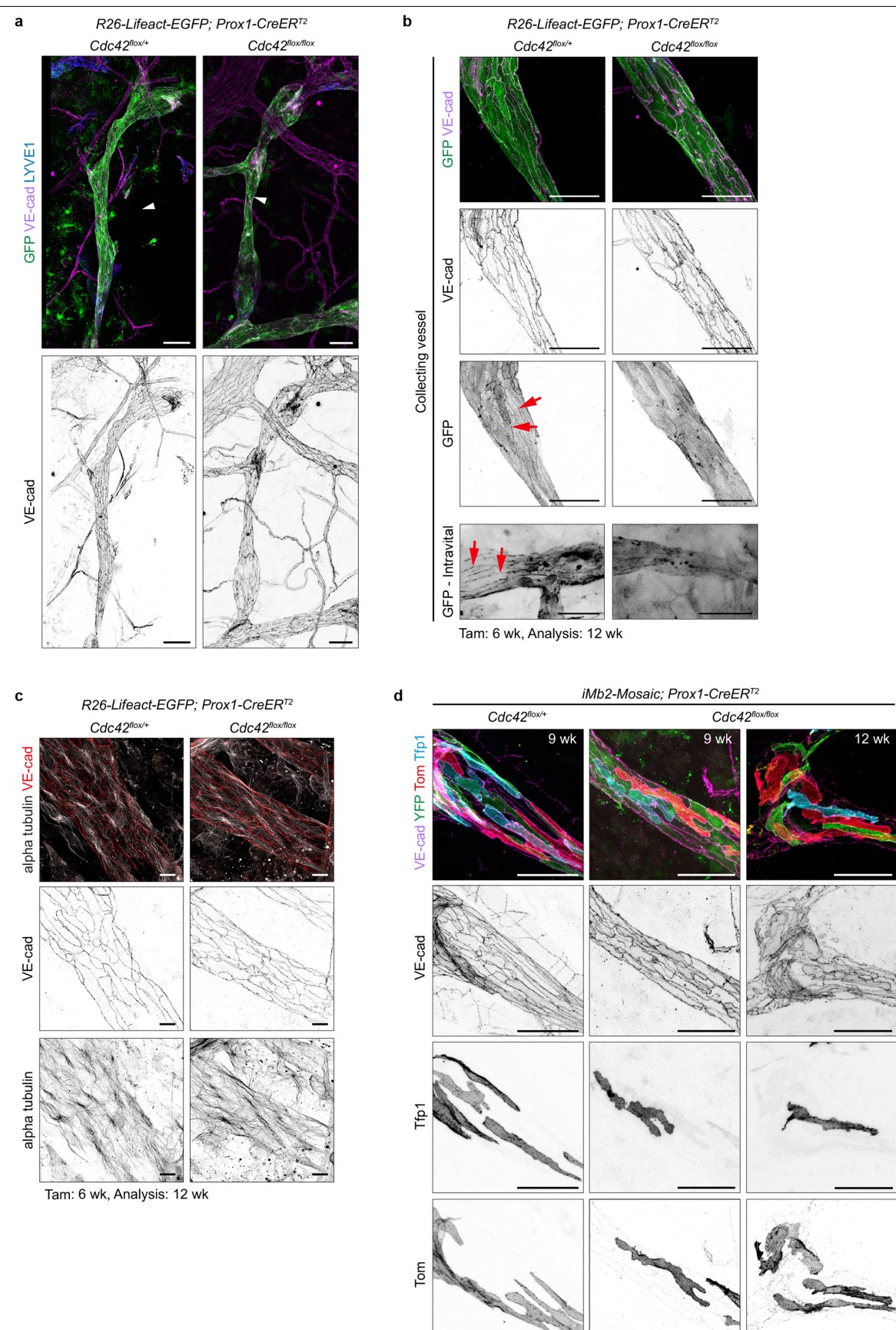

**Extended Data Fig. 6 |** See next page for caption.

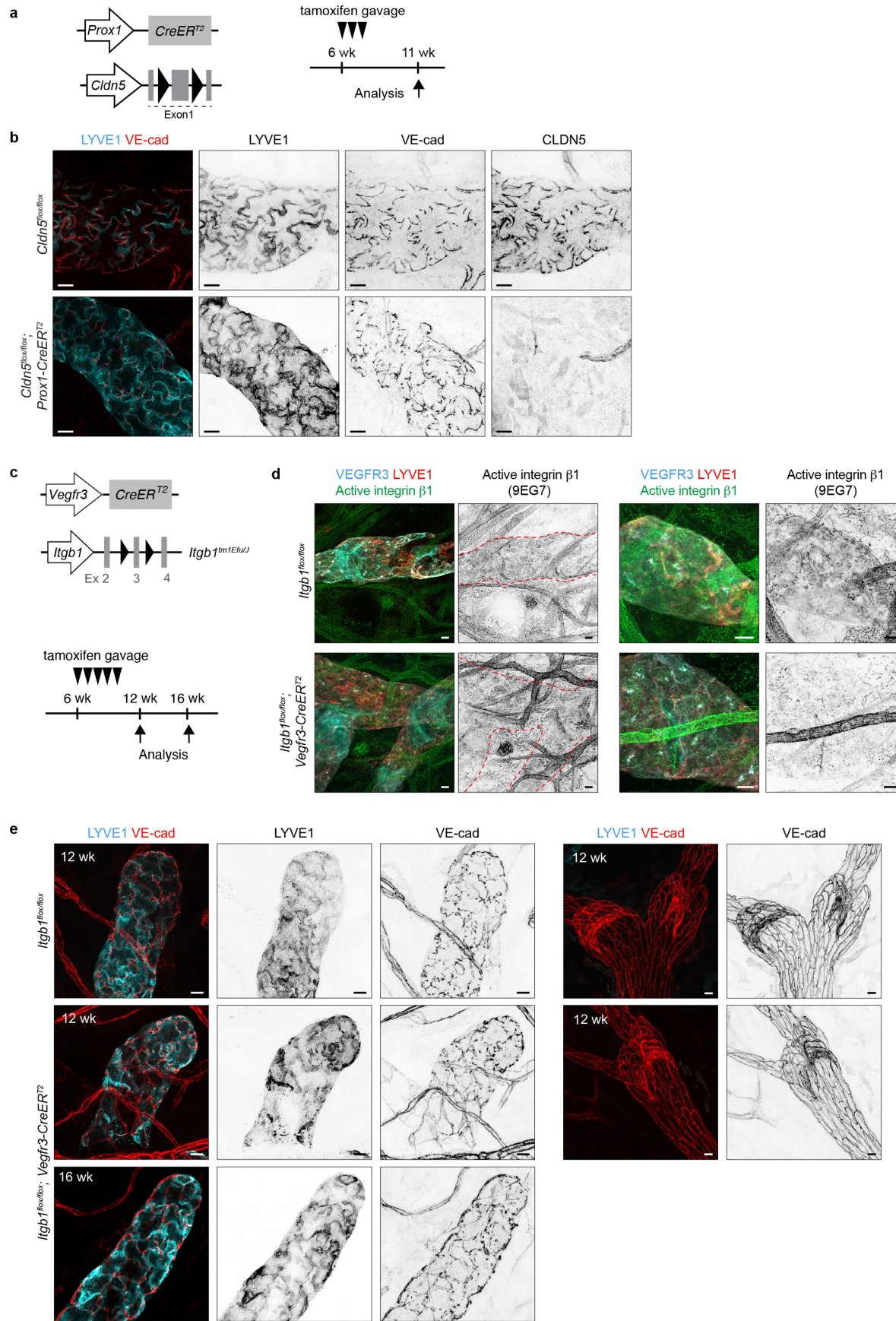

**Extended Data Fig. 7** | See next page for caption.

**Extended Data Fig. 7 | LEC-specific deletion of *Cldn5* or *Itgb1* in adult mice does not disrupt the integrity of lymphatic capillaries.** (**a**) Experimental scheme for LEC-specific deletion of *Cldn5* in mature vasculature using the *Prox1-CreER^T2* mice. (**b**) Whole-mount immunofluorescence of ear skin from a 11-week-old *Cldn5^flox;Prox1-CreER^T2* mice (5 weeks after tamoxifen treatment). *Cldn5* mutant mice show efficient depletion of CLDN5 in LECs, but no effects on their lobate shape (LYVE1 staining) or organization of cell-cell junctions (VE-cadherin staining) compared to a control. Similar results were obtained from 3 mice per genotype in two independent experiments. (**c**) Experimental scheme for LEC-specific deletion of *Itgb1* in mature vasculature using the *Vegfr3-CreER^T2* mice. (**d**) Whole-mount immunofluorescence of ear skin from 12- and 16-week-old *Itgb1^flox;Vegfr3-CreER^T2* mice (6 or 10 weeks after tamoxifen treatment). *Itgb1* mutant mice (*n* = 3 per stage) show efficient depletion of active integrin β1 in LECs, but no effects on their lobate shape (LYVE1 staining) or organization of cell-cell junctions (VE-cadherin staining) compared to a Cre- littermate control (*n* = 3 per stage) (**e**). Scale bar: 10 μm (b, d, e).

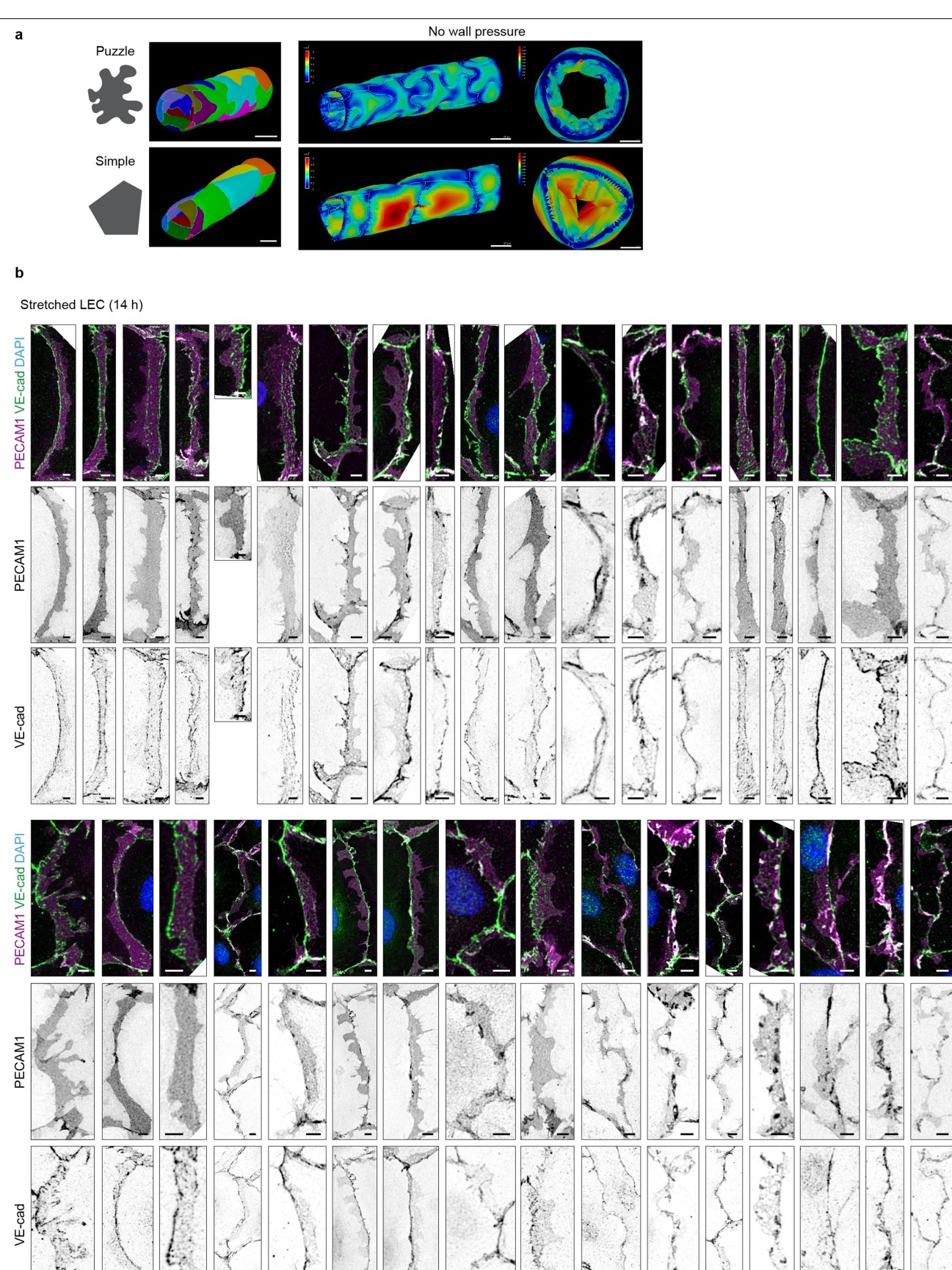

**Extended Data Fig. 8 |** See next page for caption.

**Extended Data Fig. 8 | Modelling of cellular stresses on LECs.** (**a**) Model of lymphatic vessel used for finite element method (FEM) simulations of cellular stresses on a puzzle-shaped cell template extracted from confocal data (upper panel), or simple shaped template with the same average cells size and the same connectivity of the neighbours with cell area matching in vivo measurements (lower panel). Cellular stress patterns from FEM simulations in the absence of wall pressure are shown on the right. Color scale indicated mechanical stress in kPa. (**b**) Immunofluorescence of primary LECs showing a spectrum of VE-cadherin+ junctional arrangements after isotropic stretching for 14 h. Similar results were obtained in 11 independent experiments. Scale bar: 20 μm (a, vessels), 10 μm (a, cross sections), 5 μm (b).

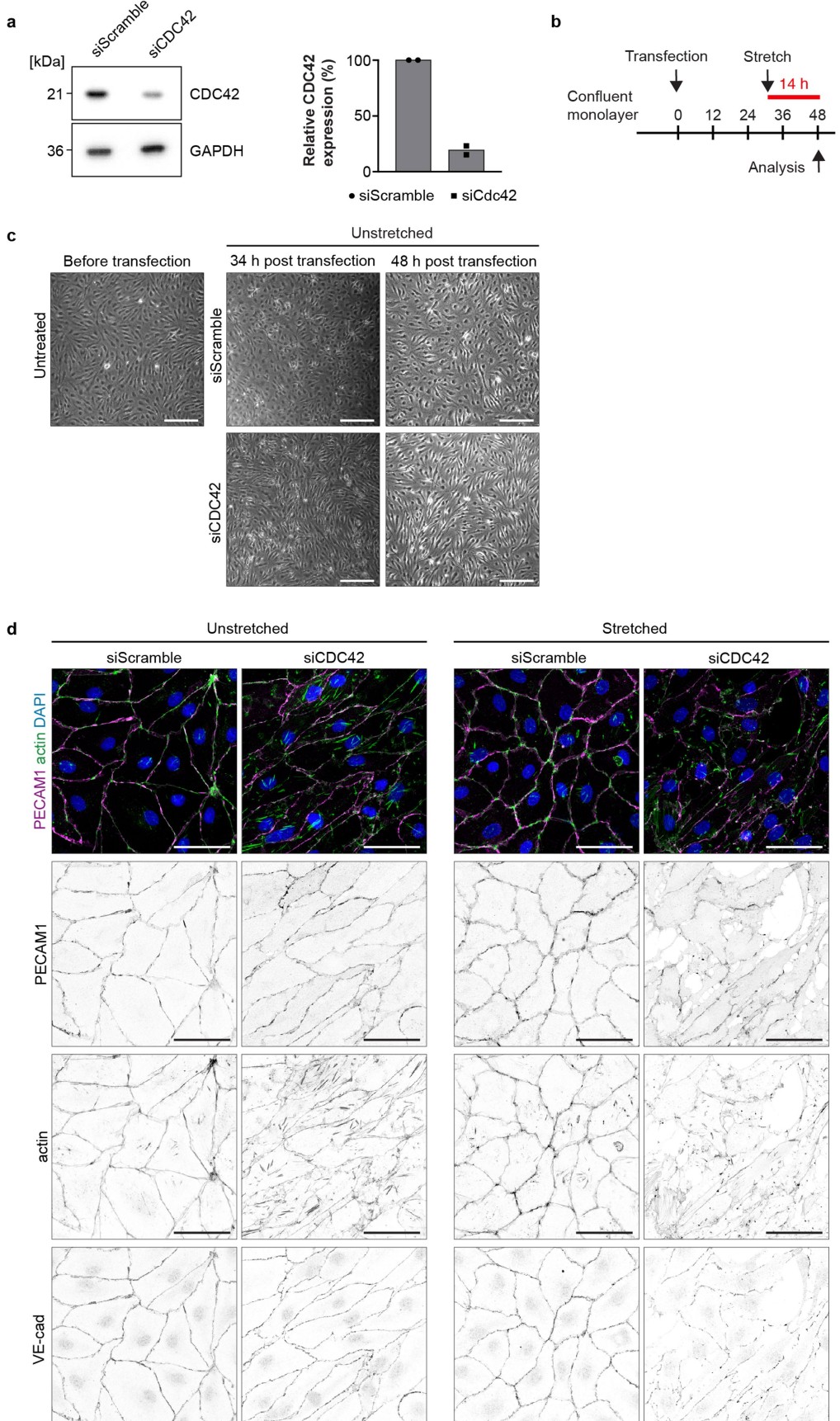

**Extended Data Fig. 9** | See next page for caption.

**Extended Data Fig. 9 | Effect of *CDC42* silencing on isotropic stretch-induced cellular changes in LECs in vitro.** (**a**) Immunoblotting of protein lysates from unstretched LECs cultured in PDMS stretch chambers, showing reduced CDC42 levels 48 h after siRNA-mediated silencing compared to control siRNA (siScramble). Graph on the right depicts quantification of CDC42 levels relative to GAPDH (*n* = 2 independent experiments). For gel source data, see Supplementary Fig. 1. No statistical test was performed. (**b**) Experimental scheme for assessing the effect of isotropic stretch on *CDC42*-silenced LECs. (**c**) Representative brightfield images of LECs cultured in PDMS chambers without stretching before siRNA transfection, and 34 h and 48 h post transfection, showing maintenance of monolayer integrity in unstretched *CDC42*-silenced LECs during the experimental time frame. (**d**) Immunofluorescence of primary LECs showing isotropic stretch-induced monolayer disruption selectively in *CDC42*-silenced LECs, quantified in Fig. 5j. Similar results were obtained in 4 independent experiments. Scale bar: 100 μm (c), 50 μm (d).

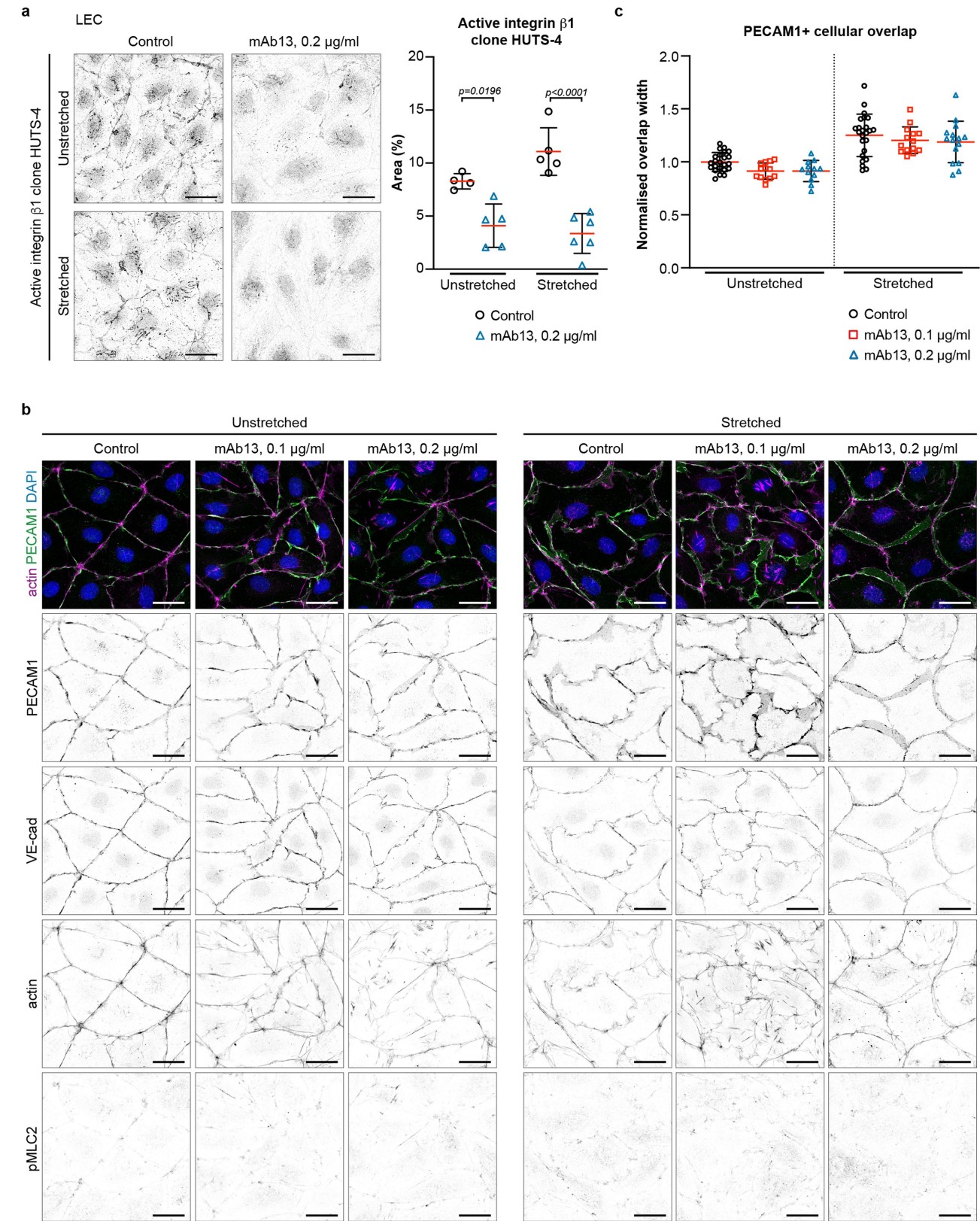

**Extended Data Fig. 10** | See next page for caption.

**Extended Data Fig. 10 | Effect of integrin β1 inhibition on isotropic stretch-induced cellular changes in LECs in vitro.** (**a**) Immunofluorescence staining (left) and quantification (right) of active integrin β1 (visualized using antibody clone HUTS-4) in unstretched and stretched (14 h) primary LECs, showing successful integrin β1 inhibition using a function-blocking antibody (mAb13, 0.2 µg/ml). Data on the right represent area of positive signal, shown as mean ± s.d. (Unstretched ctrl: $n = 4$ [2], Unstrectched mAb13: $n = 5$ [2], Stretched ctrl: $n = 5$ [1], Stretched mAb13: $n = 6$ [2] images [independent stretch holders] from one experiment). Ordinary one-way ANOVA. (**b**) Immunofluorescence of primary LECs showing the effect of isotropic stretch (14 h) on PECAM1$^+$ cellular overlaps and the curvature of VE-cadherin$^+$ cell-cell junctions compared to unstretched controls, not inhibited by integrin β1 blockade. Actin (SPY-555) and pMLC2 staining reveal no apparent stretch-induced changes. (**c**) Quantification of PECAM1$^+$ overlap width after functional blockage of integrin β1 (0.1 µg/ml or 0.2 µg/ml mAb13) in unstretched and stretched HDLECs. Data represent mean ± s.d. (Unstretched ctrl: $n = 23$ [3], mAb13 0.1 µg/ml: $n = 13$ [3], mAb13 0.2 µg/ml: $n = 12$ [3]; Stretched ctrl: $n = 24$ [3] mAb13 0.1 µg/ml: $n = 14$ [3], mAb13 0.2 µg/ml: $n = 15$ [3], images [independent stretch holders] from three independent experiments), Brown-Forsythe and Welch ANOVA. Scale bar: 25 µm (a, b).

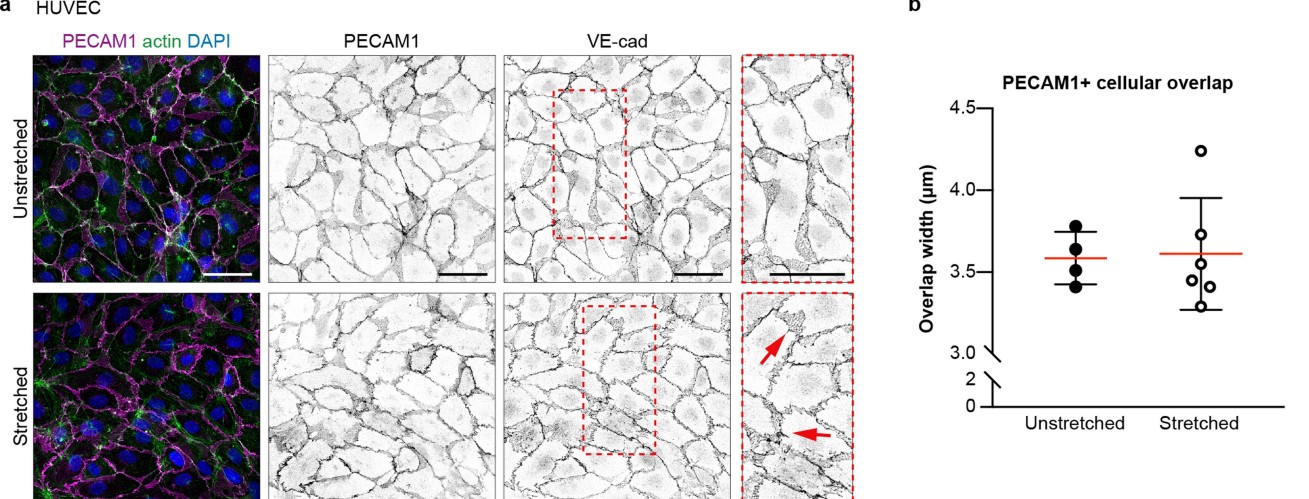

**a** HUVEC

Unstretched / Stretched columns: PECAM1 actin DAPI | PECAM1 | VE-cad

**b** PECAM1+ cellular overlap

Overlap width (µm)

Unstretched    Stretched

**Extended Data Fig. 11 | Effect of isotropic stretch on HUVECs in vitro.** (**a**) Immunofluorescence of primary HUVECs showing isotropic stretch (14 h)-induced VE-cadherin⁺ finger-like protrusions (red arrows) compared to unstretched control. Note that the VE-cadherin staining in HUVECs extends across the entire overlap region under both unstretched and stretched conditions, in contrast to the narrow line of VE-cadherin observed in LECs (Extended Data Fig. 10). (**b**) Quantification of PECAM1⁺ cellular overlap width. Dots represent quantified images (Unstretched $n$ = 4, Stretched $n$ = 6) from one experiment, including two independent stretch chambers per condition, shown as mean ± s.d., two-tailed Unpaired t-test. Scale bar: 50 µm (a).

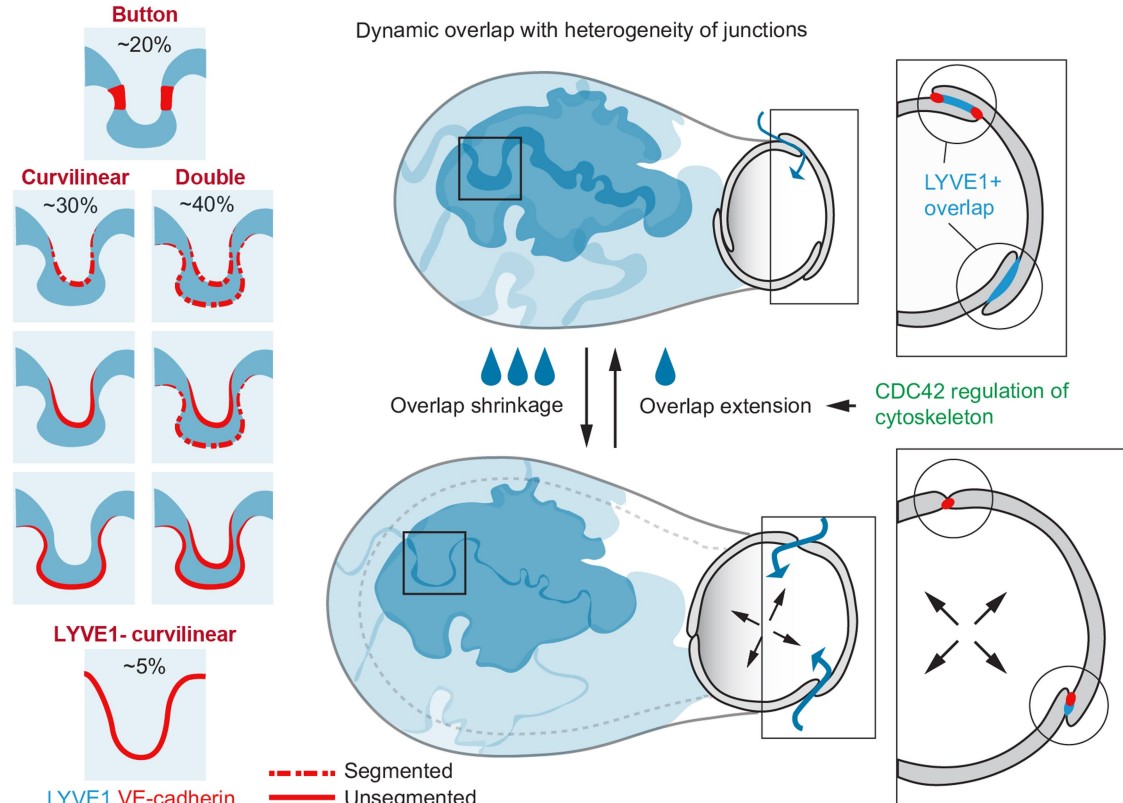

**Extended Data Fig. 12 | Proposed mechanism of lymphatic capillary function driven by dynamic remodelling of LEC overlaps.** Puzzle-shaped capillary LECs have dynamic LYVE1[+] cellular overlaps associated with a spectrum of junctional configurations organized as punctate buttons, as well as segmented or unsegmented double junctions and curvilinear junctions. Increase in interstitial fluid results in shortening of LEC overlaps to facilitate the expansion of vessel lumen upon fluid uptake. This is countered by CDC42-mediated cytoskeletal remodelling that drives the extension of lamellipodia-like cellular overlaps and consequent increase in monolayer integrity and barrier strength. Further potential implication of this model is a bellows-like mechanism, where an increase in capillary LEC overlaps aids in compressing the vessel to facilitate fluid propulsion.

# Reporting Summary

## Statistics

For all statistical analyses, confirm that the following items are present in the figure legend, table legend, main text, or Methods section.

| n/a | Confirmed | |
|---|---|---|
| ☐ | ☒ | The exact sample size (*n*) for each experimental group/condition, given as a discrete number and unit of measurement |
| ☐ | ☒ | A statement on whether measurements were taken from distinct samples or whether the same sample was measured repeatedly |
| ☐ | ☒ | The statistical test(s) used AND whether they are one- or two-sided *Only common tests should be described solely by name; describe more complex techniques in the Methods section.* |
| ☒ | ☐ | A description of all covariates tested |
| ☐ | ☒ | A description of any assumptions or corrections, such as tests of normality and adjustment for multiple comparisons |
| ☐ | ☒ | A full description of the statistical parameters including central tendency (e.g. means) or other basic estimates (e.g. regression coefficient) AND variation (e.g. standard deviation) or associated estimates of uncertainty (e.g. confidence intervals) |
| ☐ | ☒ | For null hypothesis testing, the test statistic (e.g. *F*, *t*, *r*) with confidence intervals, effect sizes, degrees of freedom and *P* value noted *Give P values as exact values whenever suitable.* |
| ☒ | ☐ | For Bayesian analysis, information on the choice of priors and Markov chain Monte Carlo settings |
| ☒ | ☐ | For hierarchical and complex designs, identification of the appropriate level for tests and full reporting of outcomes |
| ☒ | ☐ | Estimates of effect sizes (e.g. Cohen's *d*, Pearson's *r*), indicating how they were calculated |

*Our web collection on statistics for biologists contains articles on many of the points above.*

## Software and code

Policy information about availability of computer code

| Data collection | Leica Application Suite (Version 4.5.0 25531 and earlier) [image aquisition] BD FACSDiva Software (Version 8.0) (BD Biosciences) [flow cytometry] |
|---|---|
| Data analysis | Image J (Version 2.9.0/1.53t or earlier) Adobe Photoshop (Version 27.3) Huygens Essential software (v 19.04) (Scientific Volume Imaging) [image deconvolution] Graphpad Prism 9.0 MATLAB R2020a FlowJo 10.5.0-10.5.3 (TreeStar) [flow cytometry] MorphoDynamX/MorphoMechanX version 2.0 revision: 2-1459 Code for Finite element method (FEM) simulations is available in Zenodo with the identifier doi: 10.5281/zenodo.13880404 |

For manuscripts utilizing custom algorithms or software that are central to the research but not yet described in published literature, software must be made available to editors and reviewers. We strongly encourage code deposition in a community repository (e.g. GitHub). See the Nature Portfolio guidelines for submitting code & software for further information.

## Data

Policy information about availability of data

All manuscripts must include a data availability statement. This statement should provide the following information, where applicable:
- Accession codes, unique identifiers, or web links for publicly available datasets
- A description of any restrictions on data availability
- For clinical datasets or third party data, please ensure that the statement adheres to our policy

All source data supporting the quantitative findings of this study are provided as a Source Data file. Images of annotated lymphatic capillary junctions in mouse ear skin used for Fig. 1g are publicly available in Zenodo with the identifier doi: 10.5281/zenodo.13880404. All other data supporting the findings are available within the paper and its supplementary information files.

## Research involving human participants, their data, or biological material

Policy information about studies with human participants or human data. See also policy information about sex, gender (identity/presentation), and sexual orientation and race, ethnicity and racism.

| | |
|---|---|
| Reporting on sex and gender | n/a |
| Reporting on race, ethnicity, or other socially relevant groupings | n/a |
| Population characteristics | n/a |
| Recruitment | n/a |
| Ethics oversight | n/a |

Note that full information on the approval of the study protocol must also be provided in the manuscript.

# Field-specific reporting

Please select the one below that is the best fit for your research. If you are not sure, read the appropriate sections before making your selection.

☒ Life sciences  ☐ Behavioural & social sciences  ☐ Ecological, evolutionary & environmental sciences

For a reference copy of the document with all sections, see nature.com/documents/nr-reporting-summary-flat.pdf

# Life sciences study design

All studies must disclose on these points even when the disclosure is negative.

| | |
|---|---|
| Sample size | No statistical methods were used to pre-determine sample size.<br>For in vivo experiments, a minimum of 3 mice per condition was used, except for Fig. 4c and d, n=2 for 9 week-old mice.<br>For in vitro experiments a minimum of 3 biological replicates were used, except for ED Fig. 10a (validation of integrin beta1 inhibition), n=1-2 stretch holders.<br>The sample size of 3 was chosen as the minimum required to perform statistical tests. |
| Data exclusions | No data were excluded. |
| Replication | All data has been successfully replicated in at least two independent experiments. |
| Randomization | Allocation of mice into experimental groups was based on genotype. Littermate controls were included. Both female and male mice were included in analyses. Data were collected from different litters on different days and experiments were performed for different batches at different time points.<br>For in vitro experiments, allocation into experimental groups was performed randomly. Most biological replicates were analysed independently in different batches at different time points. |
| Blinding | No blinding was done in the data collection, analysis and quantifications. Quantification of LEC and vessel parameters (cell size, lymphatic vessel diameter, LEC overlap) was done in an unbiased automated fashion using ImageJ or the MATLAB script "REAVER" (Corliss, B. A. et al. Microcirc. N. Y. N 1994 27, e12618 (2020). |

# Reporting for specific materials, systems and methods

We require information from authors about some types of materials, experimental systems and methods used in many studies. Here, indicate whether each material, system or method listed is relevant to your study. If you are not sure if a list item applies to your research, read the appropriate section before selecting a response.

## Materials & experimental systems

| n/a | Involved in the study |
|---|---|
| ☐ | ☒ Antibodies |
| ☐ | ☒ Eukaryotic cell lines |
| ☒ | ☐ Palaeontology and archaeology |
| ☐ | ☒ Animals and other organisms |
| ☒ | ☐ Clinical data |
| ☒ | ☐ Dual use research of concern |
| ☒ | ☐ Plants |

## Methods

| n/a | Involved in the study |
|---|---|
| ☒ | ☐ ChIP-seq |
| ☐ | ☒ Flow cytometry |
| ☒ | ☐ MRI-based neuroimaging |

## Antibodies

**Antibodies used**

The following antibodies were used for whole mount immunofluorescence (dilution 1:100-1:500): chicken anti-GFP (ab13970, Abcam), goat anti-mouse VEGFR3 (AF743, R&D Systems), goat anti-mouse VE-cadherin (R&D Systems, AF1002), goat anti-mouse PECAM1 (R&D, Systems AF3628), mouse anti-HA tag, Alexa Fluor 647 (Cell Signalling Technology, 6E2), rabbit anti-alpha tubulin (Abcam, ab52866), rabbit anti-GFP (A11122, Thermo Fisher Scientific), rabbit anti-DsRed (Takara Bio, 632496), rabbit anti-mouse LYVE1 (Reliatech, 103-PA50AG), rabbit anti-mouse CLDN5 (Invitrogen, 34-1600), rat anti-mouse PECAM1 (553370, BD Pharmingen), rat anti-mouse LYVE1-Alexa Fluor™ 488, Clone ALY7 (Invitrogen, 53-0443-82), rat anti-mouse LYVE1 (R&D Systems (MAB2125), rat Anti-Mouse CD29 Clone 9EG7 (BD Pharmingen, 553715) All secondary antibodies were conjugated to Cy3(JIR, 712-165-153),(JIR, 711-166-152), Dylight 405 (JIR, 712-475-153), Alexa Fluor 488 (JIR, 703-545-155), (JIR, 712-545-153), (JIR, 711-545-152), Alexa Fluor 594 (JIR, 705-585-147), Alexa Fluor 647(JIR, 712-605-153),(JIR, 705-605-147),(JIR, 711-605-152) or Alexa Fluor 680 (JIR, 705-625-147) were raised in donkey and obtained from Jackson ImmunoResearch(JIR) . Secondary antibodies conjugated to AF405+ (# A48268), AF488+(# A48269), AF555+ (# A32794) or AF647+ (# A32849)were obtained from ThermoFisher Scientific.

The following antibodies or reagents were used for immunostaining of cells: DAPI (MBD0015-1ML, Sigma Aldrich), goat anti-mouse VE-cadherin (AF1002, R&D Systems), mouse anti-human PECAM1 (clone JC70A, M0823, Dako), mouse anti-Integrin β1 Antibody, activated, clone HUTS-4 (EMD Millipore, MAB2079Z), mouse Integrin β1/ITGB1 Antibody (TS2/16) Alexa Fluor 488 (Santa Cruz, sc-53711 AF488), rabbit Phospho-Myosin Light Chain 2 (Cell Signalling Technology, 3671), rat anti-Human CD29, Clone Mab 13 (BD Pharmingen, 552828) SPY-555 actin (Spirochrome). All secondary antibodies were conjugated to Cy3(JIR, 712-165-153),(JIR, 711-166-152), Dylight 405 (JIR, 712-475-153), Alexa Fluor 488 (JIR, 715-545-151), (JIR, 703-545-155), (JIR, 712-545-153), (JIR, 711-545-152), Alexa Fluor 594 (JIR, 705-585-147), Alexa Fluor 647(JIR, 712-605-153),(JIR, 705-605-147),(JIR, 711-605-152) were raised in donkey and obtained from Jackson ImmunoResearch(JIR) .

The following antibodies or reagents were used for western blot analysis of cell lysates: rabbit anti-CDC42, Clone 11A11 (Cell Signalling Technology, 2466), rabbit anti-human GAPDH, Clone 14C10 (Cell Signalling Technology, 2118). Secondary antibodies conjugated to HRP were obtained from Jackson ImmunoResearch.

For following antibodies were used for FACS: rat anti-mouse CD16/CD32 (eBioscience 14-0161-85), PDPN (8.1.1, PE, eBioscience 12-5381-81), CD31/PECAM1 (390, PE-Cyanine7, eBioscience 25-0311-82), CD45 (30-F11, eFluor 450, eBioscience 48-0451-82), CD11b (M1/70, eFluor 450, eBioscience 48-0112-82), Ki67 (SolA15, eFluor 660, eBioscience 50-5698-80).

**Validation**

The antibodies used in this study were validated for the species and applications by the manufacturers. They have all been used in previous publications by us and/or others.

Antibodies used for immunostaining:
chicken anti-GFP: Abcam provides several references for validation. https://www.abcam.com/gfp-antibody-ab13970.html
goat anti-mouse VEGFR3: R&D Systems provides several references for validation. https://www.rndsystems.com/products/mouse-vegfr3-flt-4-antibody_af743
goat anti-mouse VE-cadherin: R&D Systems provides several references for validation, and additionally this antibody was validated in this study by using of VE-cadherin-GFP reporter mice. https://www.rndsystems.com/products/mouse-ve-cadherin-antibody_af1002
goat anti-mouse PECAM1: R&D Systems provides several references for validation. https://www.rndsystems.com/products/mouse-rat-cd31-pecam-1-antibody_af3628
mouse anti-HA tag, Alexa Fluor 647: Cell Signalling Technology provides several references for validation. https://www.cellsignal.com/products/antibody-conjugates/ha-tag-6e2-mouse-mab-alexa-fluor-647-conjugate/3444
rabbit anti-alpha tubulin: Abcam provides provides several references for validation. https://www.abcam.com/products/primary-antibodies/alpha-tubulin-antibody-ep1332y-microtubule-marker-ab52866.html
rabbit anti-GFP: Thermo Fisher Scientific provides several references for validation. https://www.thermofisher.com/antibody/product/GFP-Antibody-Polyclonal/A-11122
rabbit anti-mouse CLDN5: Thermo Fisher Scientific provides several refernces for validation. https://www.thermofisher.com/antibody/product/Claudin-5-Antibody-Polyclonal/34-1600
rabbit anti-DsRed: TRakara Bio provides several references for validation. https://www.takarabio.com/products/antibodies-and-elisa/fluorescent-protein-antibodies/red-fluorescent-protein-antibodies
rabbit anti-mouse LYVE1: Reliatech provides several references for validation. https://www.reliatech.de/products/antibodies/polyclonal-antibodies/product/103-pa50ag/
rat anti-mouse PECAM1: BectonDickinson provides several references for validation. https://www.bdbiosciences.com/eu/applications/research/stem-cell-research/cancer-research/mouse/purified-rat-anti-mouse-cd31-mec-133/p/553370

rat anti-mouse LYVE1:  R&D Systems provides several references for validation.https://www.rndsystems.com/products/mouse-lyve-1-antibody-223322_mab2125
rat anti-mouse LYVE1-Alexa Fluor™ 488, Clone ALY7: Invitrogen provides several references for validation. https://www.thermofisher.com/antibody/product/LYVE1-Antibody-clone-ALY7-Monoclonal/53-0443-82
rat anti-mouse CD29, Clone 9EG7 (RUO): BD Pharmingen provides several references for validation. https://www.bdbiosciences.com/en-eu/products/reagents/flow-cytometry-reagents/research-reagents/single-color-antibodies-ruo/purified-rat-anti-mouse-cd29.553715

Antibodies used for immunostaining of cells:
goat anti-mouse VE-cadherin: R&D Systems provides several references for validation. https://www.rndsystems.com/products/mouse-ve-cadherin-antibody_af1002?gclid=Cj0KCQjw2qKmBhCfARIsAFy8buJHxptyt01pulJtf9L724wOf2ropFHp8poOiSS3BphyRhTa2fTmvi8aAt2KEALw_wcB&gclsrc=aw.ds
mouse anti-human PECAM1: Agilent Dako provides several references for validation. https://www.agilent.com/en/product/immunohistochemistry/antibodies-controls/primary-antibodies/cd31-endothelial-cell-%28dako-omnis%29-76224
mouse anti-Integrin β1 Antibody, activated, clone HUTS-4 EMD: Millipore provides several references for validation. https://www.merckmillipore.com/SE/en/product/Anti-Integrin-1-Antibody-activated-clone-HUTS-4-Azide-Free,MM_NF-MAB2079Z
mouse Integrin β1/ITGB1 Antibody (TS2/16), Alexa Fluor 488: Santa Cruz provides several references for validation. https://www.scbt.com/p/integrin-beta1-antibody-ts2-16
rabbit Phospho-Myosin Light Chain 2: Cell Signalling Technology provides several references for validation. https://www.cellsignal.com/products/primary-antibodies/phospho-myosin-light-chain-2-ser19-antibody/3671
rat anti-Human CD29, Clone Mab 13: BD Pharmingen provides several references for validation. https://www.bdbiosciences.com/en-se/products/reagents/flow-cytometry-reagents/research-reagents/single-color-antibodies-ruo/purified-rat-anti-human-cd29.552828

Antibodies used for western blot analysis of cell lysates:
rabbit anti-CDC42, Clone 11A11: Cell Signalling Technology provides several references for validation. https://www.cellsignal.com/products/primary-antibodies/cdc42-11a11-rabbit-mab/2466
rabbit anti-human GAPDH, Clone 14C10: Cell Signalling Technology provides several references for validation. https://www.cellsignal.com/products/primary-antibodies/gapdh-14c10-rabbit-mab/2118

Antibodies used for flow cytometry:
rat anti-mouse CD16/CD32: eBioscience provides several references for validation. https://www.thermofisher.com/antibody/product/CD16-CD32-Antibody-clone-93-Monoclonal/14-0161-82

# Eukaryotic cell lines

Policy information about cell lines and Sex and Gender in Research

| | |
|---|---|
| Cell line source(s) | Primary human dermal lymphatic endothelial cells (HDLEC) from juvenile foreskin (C-12216, PromoCell), human umbilical vein endothelial cells (C-12200, PromoCell), HEK293T cells (CRL-3216, American Tissue Culture Collection; Manassas, VA, USA) |
| Authentication | Authenticated by supplier and verified based on morphology (HEK293T) or immunohistochemistry profile (HDLEC) |
| Mycoplasma contamination | Cells were tested negative for mycoplasma contamination |
| Commonly misidentified lines (See ICLAC register) | No commonly misidentified lines were used in this study |

# Animals and other research organisms

Policy information about studies involving animals; ARRIVE guidelines recommended for reporting animal research, and Sex and Gender in Research

| | |
|---|---|
| Laboratory animals | Tie2-Cre (Koni et al, 2001), Vegfr3-CreERT2 (Martinez-Corral et al, 2016), Prox1-CreERT2 (Bazigou et al, 2011), iMb2-Mosaic (Pontes-Quero et al, 2017), Cdh5-GFP (encoding VE-cadherin-GFP fusion protein) (Winderlich et al, 2009), Cldn5flox (Frye et al, 2020), Cdc42flox (Wu et al, 2006) and Itgb1flox (Raghavan et al, 2000) were analyzed on a C57BL/6J background, with the exception of the iMb2-Mosaic;Vegfr3-CreERT2 mice used for intravital imaging experiments that were crossed to a C57BL/6-albino (B6(Cg)-Tyrc-2J/J) background. R26-LifeAct-EGFP mice were generated as described in the manuscript. Both female and male mice were used for analysis and no differences in the phenotype between the sexes were observed. Both embryonic (E17) and postnatal (up to 41 weeks of age) were used for experiments. The stage/age is stated in the figures and/or legends. Mice were housed in individually ventilated cages (GM500, Tecniplast) under a 12:12-h dark–light cycle (light from 07:00 to 19:00) at 22 ± 1°C under 40-60% humidity with ad libitum access to food and water. |
| Wild animals | The study did not involve wild animals. |
| Reporting on sex | Both female and male mice were included in analyses. No differences in the phenotype between the sexes were observed. |
| Field-collected samples | The study did not involve samples collected from the field. |
| Ethics oversight | All experimental procedures were approved by the Uppsala Laboratory Animal Ethical Committee, Sweden, or the National Animal Experiment Board in Finland. |

Note that full information on the approval of the study protocol must also be provided in the manuscript.

# Flow Cytometry

## Plots

Confirm that:

☐ The axis labels state the marker and fluorochrome used (e.g. CD4-FITC).

☐ The axis scales are clearly visible. Include numbers along axes only for bottom left plot of group (a 'group' is an analysis of identical markers).

☐ All plots are contour plots with outliers or pseudocolor plots.

☐ A numerical value for number of cells or percentage (with statistics) is provided.

## Methodology

| | |
|---|---|
| Sample preparation | For FACS analysis of proliferating cells ear skins were dissected, cut into small pieces and digested in Collagenase IV (Life Technologies) 10 mg/ml, DNase1 (Roche) 0.1 mg/ml and FBS 0.5 % (Life Technologies) in PBS at 37 °C for 30 min. Collagenase activity was quenched by dilution with FACS buffer (PBS, 0.5 % FBS, 2 mM EDTA) and digestion products were filtered twice through 70 μm nylon filters (BD Biosciences). Cells were washed with FACS buffer and immediately processed for immunostaining first by blocking Fc receptor binding with rat anti-mouse CD16/CD32 followed by incubation with antibodies targeting PDPN, CD31/PECAM1, CD45 and CD11b. After staining, cells were washed with PBS and then stained for dead cells using the blue LIVE/DEAD® fixable dead cell stain kit (Life Technologies), followed by fixation and permeabilization using the Foxp3/Transcription factor staining kit according to the manufacturer's instructions. Finally cells were incubated with rat serum and KI67 antibody. |
| Instrument | Cells were analyzed on a BD LSR Fortessa cell analyzer equipped with 5 lasers (355, 405, 488, 561 and 643 nm), or CytoFLEX Flow Cytometer (Beckman Coulter) with 4 lasers (405, 488, 561 and 633 nm). |
| Software | FlowJo software version 10.5.0-10.5.3 (TreeStar) |
| Cell population abundance | Analysis of proliferating LECs: ECs (PECAM1+) of total cells 1.5-2%; LECs (PDPN+) of ECs 20-30%, KI67+ LECs of all LECs 1-60%. |
| Gating strategy | Single viable cells were gated from FSC-A/SSC-A, FSC-H/FSC-W and SSC-H/SSC-W plots followed by exclusion of dead cells in the UV dump channel. FMO controls were used to set up the subsequent gating scheme to obtain cell populations and quantification of proliferating cells. Gating was done as described and exemplified in Martinez-Corral et al, Nat Commun 2020. |

☐ Tick this box to confirm that a figure exemplifying the gating strategy is provided in the Supplementary Information.

