## [Peer Review File · Nature]

Dynamic cytoskeletal regulation of cell shape supports resilience of lymphatic endothelium

Corresponding Author: Professor Taija Makinen

This file contains all reviewer reports in order by version, followed by all author rebuttals in order by version. Parts of this Peer Review File have been redacted as indicated to remove third-party material.

Version 0:

Reviewer comments:

Referee #1

(Remarks to the Author)
General Comments

1. In this clearly written and well-illustrated manuscript, with 6 main figures and 6 supplementary figures, the authors report many novel findings on the presence and functional significance of the overlap of the borders of oak-leaf shaped endothelial cells of initial lymphatics in mouse ear skin. The authors also report similarities between the shape of these endothelial cells and jigsaw puzzle piece-shaped pavement cells of plant epidermis, and conclude that both reflect adaptations to stretch.
2. The novel approaches and findings in the manuscript are benefitted by the contrasting and complementary expertise of Dr. Mäkinen in her studies of lymphatics and Dr. Majda in his studies of shape changes in plant cells.
3. Main: In the introductory section, the authors effectively set out the background of the scalloped shape and distinctive discontinuous (button) junctions in initial lymphatics. They also acknowledge the poor understanding of the control and maintenance of cell shape and button formation and introduce the similarities to plant puzzle cells. However, a more complete description of the specific aims of the project is needed to help readers anticipate the diverse findings in the studies presented. This could easily be achieved by adding a sentence similar to, "The specific aims of the project were to determine ____, ____, ____, and ____" (integrate with lines 72-77).
4. Results- General: The findings address two distinctive features of endothelial cells of initial lymphatics: (a) the cells have an oak-leaf shape with overlapping scalloped borders, and (b) the cells are interconnected by button and zipper junctions. These topics are considered separately in the following comments.
5. Results - Oak-leaf shape of lymphatic endothelial cells
 - a. Mosaic mice: Multicolor labeling of lymphatic endothelial cells in mosaic mice effectively revealed the unusual cell shape and overlap of adjacent cells. This novel feature of the manuscript is documented by striking images of the shape and distribution of these cells in ear skin lymphatics. In regions of lymphatics where adjacent cells have different colors, the extent of overlap of the scalloped borders is beautifully revealed. Use of mosaic mice also revealed differences in appearance of the cells between E17 (back skin) and 3, 5, 9, and 25 weeks postnatally (ear skin).
 - b. Pentraxin: To better understand the results of the scRNA-seq studies (lines 118-121), more detailed justification - including references - is needed for using pentraxin-3 (Ptx3) expression to distinguish endothelial cells of initial lymphatics from those of collecting lymphatics. Please also comment on whether a gradient in Ptx3 expression was found in initial lymphatics from tip to collecting lymphatic (see General Comment 6: Problem 7).
 - c. Cytoskeleton: Another novel feature is the demonstration of contrasting distributions of microtubules and F-actin in oak-leaf lymphatic endothelial cells. Although F-actin is clearly shown to be greater at convex regions of some scalloped borders, the association of microtubules with concave regions is less convincing. Many microtubules in Fig. 2b, c appear to continue beyond the border of the outlined endothelial cell, suggesting that some microtubules are in separate but overlapping cells. Replacement of the images in Fig. 2b-c with images similar to Fig. 3b (top row) is recommended to solve this problem and ensure that the images match the drawing in Fig. 2f and the interpretation (lines 144-146).

d. In vitro stretch: The cyclic isotropic stretch data for human dermal lymphatic endothelial cells in culture (lines 222-231, Fig. 4a-d) are a provocative addition to the story. The increase in overlap after stretching fits the authors' hypothesis, but the absence of both oak-leaf shape and discontinuous junctions raises questions about the relevance to initial lymphatics. As the borders of many cultured endothelial cells overlap, the authors should address the question of specificity by comparing LEC to blood vessel endothelial cells (e.g., HUVEC, BAECs) in confluent monolayer culture with and without isotropic stretch. Greater stretch-induced overlap in LEC than in non-LEC would increase the relevance of these data.

e. Intravital imaging: 2-photon microscopic images showing changes in the borders and overlap of endothelial cells of initial lymphatics - but not collecting lymphatics - at baseline and after saline injection are not only novel findings but also a positive addition to the story (lines 281-293, Fig. 5, Supplementary Figs. 5, 6). Although changes in actin fluorescence are clearly visible in Movies 1 and 2, little change is evident in the shape of cell borders during the 7:31-min:sec movies. This feature of the movies should be described and discussed.

6. Results- Concerns over claims about buttons and zipper junctions: Multiple concerns are raised by the authors' interpretations and conclusions drawn from data for discontinuous (button) junctions and continuous (zipper) junctions in initial lymphatics that are inconsistent with published data on ear skin and other organs. Despite solid evidence to the contrary, the authors argue that discontinuous junctions are rare (10-15%) in ear skin lymphatics. These inconsistencies appear to stem from problems with characterization and categorization of lymphatic junctions, with rigor of the quantitative data, with regions of the lymphatics analyzed, with reconciling confocal and transmission electron microscopic (TEM) images, and with conclusions drawn from the data. The validity and significance of the junction data in relation to published work cannot be interpreted meaningfully until these problems are resolved.

- Problem 1: The authors' claim that 85-90% of junctions in ear skin initial lymphatics are zippers (lines 299-301, 400-403) is inconsistent with data cited in at least 7 publications that document the predominance of button junctions in mouse ear skin lymphatics (Baluk et al., JEM 2007; Zheng et al., Genes Dev 2014; Zhang et al., Science 2018; Hägerling et al., EMBO J 2018; Arasa et al., JEM 2021; Churchill et al., JEM 2022; Jannaway et al., Cell Reports 2023). All of these papers not only show the predominance of buttons but also illustrate the variable appearance of these junctions, both around individual lymphatic endothelial cells and from cell to cell. Despite their structural variability, discontinuous junctions in initial lymphatics (Supplementary Fig. 3b column 3) are conspicuously different from continuous junctions in collecting lymphatics (Supplementary Fig. 2b row 2). The authors' claims to the contrary are also inconsistent with evidence that junction "zippering" (replacement of buttons by zippers) clearly occurs under experimental or pathological conditions (Yao et al. Am J Pathol 2012; Zheng et al., Genes Dev 2014; Zhang et al., Science 2018; Churchill et al., JEM 2022; Jannaway et al., Cell Reports 2023) but could not occur if 85-90% of the junctions were already zippers under normal conditions.

Remedy: The authors should first correct Problems 3-9 described below. Then, they should describe in the revised Results what is known from previous studies of button and zipper junctions in mouse ear skin lymphatics, before presenting their revised junction data. Differences of their data to those published previously on ear skin and other organs should be considered critically. In the Discussion, the strengths and limitations of the authors' approach should be described and any inconsistencies with prior work discussed and given plausible explanations.

- Problem 2: The authors also claim that the oak-leaf phenotype develops before button junctions appear in initial lymphatics of ear skin (lines 83-100). However, their data indicating that buttons constitute only 2% of junctions (Fig. 1e, o) at age 5 weeks are inconsistent with published values of 60% button junctions in ear skin at age 3 weeks (Jannaway et al., Cell Reports 2023). The authors' claim is also inconsistent with published data for initial lymphatics in the trachea and diaphragm, where buttons and oak-leaf shaped endothelial cells appear together around birth and evolve together postnatally (Yao et al., Am J Pathol 2012).

Remedy: The authors should correct Problems 3-9, as for the remedy for Problem 1, and then revisit this issue with the new data.

- Problem 3: The inconsistencies appear to stem in part from the criteria used to apportion lymphatic junctions into "buttons only", "buttons + zipper", and other categories (lines 295-303, Fig. 5c, d). Here, the category names and drawings do NOT accurately reflect the confocal microscopic images in these figures, which in turn leads to other problems. The drawings of "buttons + zipper", "zipper lobe base", and "zipper lobe tip" categories show a uniform red line symbolic of zippers, but no buttons, whereas the black/white confocal images show discontinuous foci of strong VE-cadherin staining, some of which are connected by faint VE-cadherin staining that is very different from zipper junctions (Fig. 1d inset). Indeed, the red/blue confocal images of these categories in Fig. 5c resemble discontinuous buttons more than continuous zippers.

Remedy: A more representative method of categorization is needed. Apportioning junctions into three categories (buttons, intermediate, and zippers) would solve the problem and enable comparison of the new data to previous and future work. The intermediate group would accommodate the known variability of button junction morphology. As an example, previous use of this approach to quantify junctions in initial lymphatics in mouse ear skin at age 3 weeks revealed 60% buttons, 30% intermediate, and 10% zippers, with corresponding values of 50%, 40%, and 10% in the diaphragm (Jannaway et al., Cell Reports 2023).

- Problem 4: Equating discontinuous faint VE-cadherin staining to the continuous strong staining of zippers led to another problem, where 4 of the 5 categories were grouped with zippers to support the authors' claims that "classical button junctions alone accounting for only ~15% of total junctions" (lines 300-301, Fig. 5d) and that button junctions constitute only 10-15% of endothelial cell junctions in mouse ear skin (lines 400-403). This grouping led to interpretations that are inconsistent with the

published data on ear skin lymphatics cited with Problems 1 and 2.

Remedy: The authors should investigate possible explanations for the discontinuous faint VE-cadherin staining that accompanied button junctions. Is the pattern of VE-cadherin similar to staining for tight junction proteins claudin-5, JAM1, ESAM, and occludin? Is faint staining also found between buttons in initial lymphatics of VE-cadherin-GFP mice? Is some of the staining non-specific? Does the faint staining result from cross-reactivity of the VE-cadherin antibody to another protein, tested by preabsorption of their R&D VE-cadherin antibody AF1002 with VE-cadherin protein and by comparison to staining with other antibodies, such as Thermo Fisher rat monoclonal VE-cadherin antibody eBioBV13 (BV13) or BD Pharmingen rat anti-mouse antibody AB_2244733 used in previous studies? Does the faint staining come from superimposition of staining in the opposite membrane of lymphatics that collapse during fixation by immersion (lines 549-550) rather than by conventional vascular perfusion? Answers to these questions should guide the characterization and categorization of junction data included in the manuscript.

- Problem 5: The “double zipper” category presents yet another concern, where the category name and drawing do not match the corresponding images, which show neither two continuous lines of VE-cadherin staining nor an oak-leaf shaped cell. In addition, the $< 10\text{-}\mu\text{m}$ distance between the lines (Fig. 5c, “double zipper” images, scale bar $10\ \mu\text{m}$) exceeds by an order of magnitude the distance between junctions in overlapping regions in TEM images. Unlike the confocal image of the presumptive “double zipper” (Fig. 5c), TEM images of overlapping regions show junction separations of $0.5\ \mu\text{m}$ (Fig. 3j) or $1\ \mu\text{m}$ (Supplementary Fig. 4b). Similarly, the average overlap width was about $2\ \mu\text{m}$ in confocal images of mosaic mouse ears (Fig. 5f). Yet, the authors imply that the TEM images support their interpretation of “double zippers”. No meaningful interpretation of the claim of “double zippers” can be made without quantitative data and more compelling confocal and TEM images of this feature - if it actually exists - in oak-leaf endothelial cells at the tip of initial lymphatics in adult ear skin.

Remedy: Measurements are needed to compare the spacing of double rows of VE-cadherin staining at presumptive “double zippers” to the distance between adherens junctions in TEM images of overlapping regions. Do the values match? If not, other explanations for “double zippers” identified in confocal images should be explored. More convincing confocal and TEM images and robust quantitative data should be added, and the category name, drawing, and interpretation revised to fit the data.

- Problem 6: Related to the “double zipper” problem, all lymphatic endothelial cell junctions in TEM images are described as adherens junctions (lines 305-308; Supplementary Fig. 4b). However, this designation ignores evidence from the authors and many others that lymphatic endothelial cells are connected both by tight junctions (claudin-5, JAM1, ESAM, etc.) and by adherens junctions (VE-cadherin). However, TEM images in Fig. 3j and Supplementary Fig. 4b have insufficient magnification/resolution to distinguish tight junctions from adherens junctions by conventional TEM criteria (Farquhar & Palade, *J Cell Biol* 1963; Brightman & Reese, *J Cell Biol* 1969; Friend & Gilula, *J Cell Biol* 1972).

Remedy: Higher magnification TEM images are needed to distinguish tight junctions from adherens junctions in overlapping regions of lymphatic endothelial cells. In the replacement TEM images used for measurements and illustration, tight junctions should appear as focal sites of fusion of the outer leaflets of adjacent plasma membranes accompanied by electron densities of membrane/cytoplasm, and adherens junctions should appear as focal electron densities of membrane/cytoplasm separated by a narrow - but clearly visible - intercellular space (Farquhar & Palade, *J Cell Biol* 1963; Brightman & Reese, *J Cell Biol* 1969; Friend & Gilula, *J Cell Biol* 1972).

- Problem 7: The authors state that junction measurements were made in ear skin lymphatics from the tip to the first valve (Methods, line 615), apparently based on the assumption that discontinuous junctions are uniformly distributed throughout this segment of lymphatics. A gradient in junction phenotype from the tip downstream toward the collecting lymphatic would explain the finding of sparse button junctions in this region. In the trachea, button junctions are most numerous at the tip and are gradually replaced by zippers downstream, long BEFORE the first valve (see Fig. 1E, Baluk et al., *JEM* 2007). (As described in that paper, valves are restricted to the adventitial surface of the mouse trachea, but the junctions have a steep gradient in mucosal initial lymphatics that have no valves.) If ear skin lymphatics resemble those in the trachea, the region of initial lymphatics downstream from the tip would be expected to have fewer button junctions.

Remedy: Initial lymphatics in ear skin should be characterized to assess the gradient in button junctions from the tip to the first valve. If buttons decrease with distance in the ear as in the trachea, measurements should be restricted to the tip where buttons are most numerous. This approach has been used successfully by others (Fig. 1E, 2G, 3E, S1E, and S1G, Churchill et al., *JEM* 2022).

- Problem 8: Given the inconsistencies described in Problems 1 and 2, concerns about rigor are raised by the use of only 2 mice/group for measurements of % of junctions in various categories at 5, 8, and 25 weeks (Fig. 5d and legend), despite the abundance of junctions measured in each mouse.

Remedy: To increase the robustness of these important data that form the basis of the authors’ conclusions about button junction abundance and to enable assessment of mouse to mouse variability, the group size should be increased from only 2 mice to $N = 5\text{-}6$ mice/group. The approach for making new measurements should reflect the changes made to correct Problems 3-8. The authors should acknowledge and discuss the similarities and differences of the new data in relation to published data and developmental timetables for ear skin and other organs (Yao et al., *Am J Pathol* 2012). The findings should also be discussed in the context of whether ear skin lymphatics are representative of those elsewhere (lines 99-100).

- Problem 9: Fig. 6 does not make sense as a graphic summary of key findings. The schematic showing initial lymphatics

with “double zippers” and “single zippers” but NO button junctions does not reflect the data presented in the manuscript or in previous reports, including their own (Ulvmar & Mäkinen, *Cardiovasc Res* 2016).

Remedy: After the junction problems are corrected, Fig. 6 should be changed to be consistent with the revised text and figures and with the literature. Advances over published evidence should be highlighted and validated in the figure legend.

• Overall integration and reconciliation: After the junction problems are remedied, please ensure that the revised findings and interpretations are described consistently throughout the manuscript, including the Abstract, Results, Discussion, and Figure legends. Readers would also benefit from additional background and discussion that explains: (i) the authors’ rationale for choosing ear skin lymphatics for the present study; and (ii) the unusual postnatal developmental timetable of the mouse ear pinna and accompanying lymphatics (Mäkinen et al., *Genes Dev* 2005), which occurs much later than in the diaphragm, intestinal villi, and trachea.

7. Discussion. Several general aspects of the Discussion deserve attention.

a. Puzzle cells: Because of the importance the authors give to the comparison of oak-leaf lymphatic endothelial cells to jigsaw puzzle-shaped epidermal cells in plants, more detailed background is needed on the similarities and differences in the two cell types. (i) Puzzle cells interdigitate with one another, but do they overlap like oak-leaf endothelial cells? (ii) If so, is the overlap augmented by stretch? (iii) If not, how does the difference bear on the comparison? (iv) Do oak-leaf endothelial cells form their distinctive shape as they enlarge during growth, as reported for puzzle cells? (v) What is the relevance to lymphatic endothelial cells of the reported influence of cell wall composition on the shape of puzzle cells in plants?

b. Limitations and differences from published data: It is understandable that the authors highlight the significance of their findings, but equal attention should be given to the limitations of the approaches and methods and how they could impact the data and interpretations and contribute to differences from published data. Inadequate attention is given to a comparison of the present findings to published data on oak-leaf shaped lymphatic endothelial cells and the junctions between them. Data that are confirmatory should be identified, and findings that depart from the literature described with balanced treatment of the differences and how the discrepancies could be resolved.

c. Are prevailing concepts challenged? As introduced in General Comment 6: Problem 1, the statement, “Our results challenge some of the prevailing concepts of flap valves” (lines 396-403) is not justified by the underlying data and is inconsistent with published data for button junctions in ear skin. Indeed, this claim detracts from the study’s many solid new observations and creates a contradiction that the present data do not resolve. Unless data from additional experiments provide a compelling challenge to prevailing concepts, the authors should replace this sentence with a discussion of the strengths and limitations of their findings from mosaic mice, cytoskeleton staining, and intravital 2-photon imaging in comparison to published data on endothelial cells of initial lymphatics.

d. Starling principle: In the Discussion, the authors should consider the transit of fluid and solutes across lymphatic endothelial cells (lines 436-437) in the context of Starling forces that govern these movements (Michel, Woodcock & Curry, *Acta Anaesthesiol Scand* 2020), to correct the implication that the type of intercellular junctions can explain everything.

Specific Comments

1. “Lymphatic capillaries”: The term “initial lymphatics” is recommended to be used in replace of “lymphatic capillaries” to emphasize that discontinuous junctions are a feature of lymph entry sites at the tip of lymphatics, but not necessarily in other regions of small lymphatics that lack smooth muscle, and to avoid confusion with blood capillaries. The term “distal initial lymphatic” or “pre-collector” could be used for the segment of lymphatics proximal to the first valve that lacks button junctions and smooth muscle.

2. “LEC”: Readers would benefit from the authors’ restricting the use of the abbreviation LEC to isolated lymphatic endothelial cells studied in vitro to avoid confusion with endothelial cells of intact lymphatics.

3. Line 67: Please correct this statement to acknowledge that endothelial cells of initial lymphatics do have a basement membrane (Pflücke & Sixt, *JEM* 2009).

4. Fig. 1m, o: Please describe in the figure legend the meaning of “1.6x” in these figures.

5. Fig. 3h: Please adjust the magnification of images in row 3 to match those in row 1 with which they are compared. Also, please change the color of red arrows that mark structures in the Surface LYVE1 image in row 1, column 2, which are confusing because elsewhere red applies to Total LYVE1. In addition, please explain your interpretation for how the LYVE1 antibody stains the tiny blue region INSIDE the lymphatic without permeabilization (bottom drawing in inset).

6. Fig. 3j: Please specify whether “dermal LECs” (line 195) refers to ear skin lymphatics or another region of skin. Also, please acknowledge that the widened intercellular spaces in TEM images in Fig. 3j (right) could be a technical artifact due to lymphatic collapse during fixation by immersion (lines 596-597) rather than by conventional vascular perfusion. This would not negate the effect of CDC42 deletion on junctional integrity; instead, it could reflect greater susceptibility of junctions to separation during tissue preparation.

7. Lines 198-199: Please revise this text to reflect that the confocal images in Fig. 3i (arrowheads) show staining

“discontinuities” rather than intercellular “gaps”, which would require TEM for confirmation. Also, revise the text to indicate that TEM images in Fig. 3j show focal regions of cell separation but not “large intercellular gaps” because other regions of the same overlapping cell borders are still in contact. Please also change the Y-axis label of Fig. 3k accordingly.

8. Line 202 “...formation of intercellular gaps (Fig. 3b, f, i)”: Please revise as for lines 198-199.

9. Lines 204-206: The finding that deletion of *Cldn5* did not disrupt the discontinuous staining of VE-cadherin at button junctions in ear skin lymphatics is important and deserves to be described in more detail than “did not disrupt LEC junctions in the dermal vasculature”. The description should acknowledge that claudin-5 is only one of several tight junction proteins, and this redundancy is likely to explain why *Cldn5* deletion does not cause vascular junction disruption and rapid death. Also, please specify the antibody used for claudin-5 immunofluorescence (Supplementary Fig. 3b). Because claudin-5 immunoreactivity in the control is weak, perhaps stronger staining could be obtained by using rabbit polyclonal anti-mouse claudin-5 (34-1600, 147 1:1000, Invitrogen). This antibody could also help in resolving Problems 4, 5, and 6.

10. Line 208 “...these results demonstrate that CDC42...”: Please tone down the interpretation. “...these results provide evidence that CDC42...” would be better.

11. Lines 222-231: The strengths and limitations of the in vitro stretch experiments should be described.

12. Lines 232-233 “...formation of intercellular gaps”: Please use a term other than “gaps” (e.g., intercellular separations) for these large (< 25 μm) spaces between cells in endothelial monolayers (Fig. 4b) to avoid confusion with the much smaller gaps (< 1 μm) found in vivo. Please also change the heading and Y-axis label in Fig. 4d accordingly.

13. Lines 256-258: Please explain the relevance of the in silico model of effects of mechanical stress on lymphatic endothelial cell tubes with a cross-sectional wall thickness of > 10 μm (Fig. 4g-h, scale bar 10 μm) to simulate initial lymphatics with a wall thickness of < 1 μm .

14. Lines 264-266 “...lobate shape enhances LEC monolayer resilience to mechanical strain”: Please revise this interpretation to help readers understand the significance of the findings in terms of lymphatic vascular biology. What is the benefit to initial lymphatics of “resilience to mechanical strain”? Why is this feature different from shape changes in endothelial cells of blood vessels that dilate and constrict? Please also acknowledge that overlap is a common feature of vascular endothelial cells that is not restricted to initial lymphatics.

15. Lines 272-274 “...anchored in place by button junctions⁴” and “In contrast to this static structure...LEC lobes might instead be dynamic”: The second of these two statements is misleading because no claim or assumption has been made that the overlapping flaps are static. Instead, they are assumed to change with the amount of lymph flow, chylomicron entry, etc. This issue can be resolved by describing the new findings as additional evidence of the dynamic nature of the valves.

16. Lines 277-280: Cellular “overlaps” and “lamellipodia-like protrusions” look similar in the image in Supplementary Fig. 4a. Evidence for the structures illustrated being lamellipodia is not convincing, and the term is therefore misleading. What criteria were used to distinguish overlapping flaps from lamellipodia? In addition, please correct the spelling of lamellipodia.

17. Legends of Figs. 1, 5, Supplementary Fig. 4 “exemplary”: The context indicates that the intended meaning is “representative” not “exemplary”. Please change accordingly.

18. Line 305 “majority” means 51% or more: Please describe the total number of cell contacts examined by TEM and the percent with junctions. As described by Leak (J Cell Biol 1971), some overlaps have junctions, and others do not. If the authors are trying to modify this concept, quantitative data would be needed for comparison to values predicted for random TEM sections of initial lymphatics with button junctions compared to their proposed junction model.

19. Lines 312-313: Please explain why the absence of change in button junctions in lymphatics after intradermal injection of PBS was surprising, when that is the expected result. The conventional view is that button junctions remain intact when the overlapping borders between the buttons separate during lymph entry. This is an attribute of button junctions that permits lymph entry WITHOUT disassembly of junctions.

20. Lines 322-324 “lamellipodia-like contact sites”: Please see comment on lines 277-280. This interpretation would better fit the data if described as, “Collectively, these data are consistent with rapid changes in the borders of endothelial cells of initial lymphatics under baseline conditions and in response to increases in interstitial fluid pressure.”

21. Line 335 “uncovers”: Should be “uncovered”

22. Line 342 “excess interstitial fluid” (also in first line of Abstract and Main section): “excess” is misleading because lymph flow is a normal process whereby interstitial fluid flows along the hydrostatic pressure gradient from blood vessels to initial lymphatics. “excess” would apply only when interstitial fluid exceeds the capacity of lymphatic drainage. Please delete “excess” in these sentences.

23. Lines 346-349 “In most tissues analyzed^{4,6}, including the ear skin studied here, this transformation commences during late embryonic or early postnatal stage”: This statement is misleading because data for ear skin lymphatics in their study came from mice at age 5 weeks or older, apart from measurements of vessel diameter and endothelial cell size and lobe

number at age 3 weeks (Fig. 1l, m, and n). Please change accordingly and acknowledge published data for age 3 weeks (e.g., Jannaway et al., Cell Reports 2023).

24. Line 354: Please correct the error where claudin-5 is described as an adherens junction protein: "...we found that CLDN5, the second major component of adherens junctions...", to match the correct description as a component of tight junctions (line 204).

25. Lines 358-360 "...loss of the F-actin-enriched LEC lobes with a consequent loss of button junctions and cell junction integrity.": Please change "loss" to "reduction" to fit the data in Fig. 3b, c.

26. Lines 381-386: In the discussion of effects of isotropic stretch on LEC in vitro, please describe the corresponding effects on blood vessel endothelial cells. After performing the recommended experiments, acknowledge that the observed changes were or were not specific to lymphatic endothelial cells. Also mention that, because the cells did not develop an oak-leaf shape or acquire button junctions, the relevance of stretch-induced changes to initial lymphatic structure and function awaits further study.

27. Lines 396-403: As described in General Comment 6: Problem 1, the data presented are inconsistent with published data for button junctions in ear skin and do not justify the statement, "Our results challenge some of the prevailing concepts of flap valves." Indeed, this statement detracts from the study's many solid novel observations and raises an unjustified controversy that the present data do not resolve. Instead, the authors are encouraged to focus on the data from mosaic mice, cytoskeleton staining, and intravital 2-photon imaging that are convincing and novel.

28. Lines 400-403 "...only around 10-15% of cell-cell junctions...represented true buttons": This statement is not only inconsistent with the results of reports from 7 separate laboratories (General Comment 6: Problem 1) but is also inconsistent with evidence that an intradermal injection of VEGFA doubles the number of zippers from 25% to 50% within 30 min (Zhang et al., Science 2018). The current manuscript does not provide convincing evidence to justify this statement, because the data do not invalidate all published work on button junctions in ear skin lymphatics. The necessary changes are described in the remedies for General Comment 6: Problems 1-9.

29. Lines 418-421: Please correct the description of findings reported by Leak (J Cell Biol 1971) to indicate that he found that tracer passage between lymphatic endothelial cells stopped at the barrier created by tight junctions (maculae occludentes) (Fig. 15a, J Cell Biol 1971), whereas adherens junctions (maculae adherentes) maintained structural adhesion of the cells but did not create a barrier. The authors should also note that Leak referred to adherens junctions as focal maculae adherentes, which are now recognized as buttons, not as continuous zonulae adherentes referred to as zippers. Leak's paper was published two years after Brightman & Reese (J Cell Biol 1969) discovered that tight junctions in the form of zonulae occludentes create the endothelial barrier in the brain.

30. Lines 431-433: The proposed mechanism of "zippering" without changes in buttons is not sufficiently clear in relation to what the authors envision or in the context of published data for conditions where button to zipper transformation occurs (Yao et al. Am J Pathol 2012; Zheng et al., Genes Dev 2014; Zhang et al., Science 2018; Churchill et al., JEM 2022; Jannaway et al., Cell Reports 2023). Measurements would be needed to argue that the total length of junctional VE-cadherin is the same before and after zippering. Speculation is fine (if acknowledged as such) but should describe how the authors' speculation differs from what is already known.

31. Lines 434-441: Please broaden the context here to avoid the implication that oak-leaf shaped endothelial cells and button junctions are essential for transit of fluid and cells across the endothelium, given that neither is present in blood vessels where fluid and cells can move in both directions.

32. Lines 437-442: Pflücke & Sixt, JEM 2009 should be cited here to provide perspective.

33. Lines 442-444: Please acknowledge any inconsistencies with the results of previous studies of button junctions in ear skin lymphatics (see General Comment 6: Problem 1).

34. Line 615 "...distal to the first valve.": As button junctions are not expected to be present in lymphatics distal to the first valve, it would seem that the authors intended to describe the region of study as "proximal" to the first valve, which is the segment between the initial tip and the first valve. Please see General Comment 6: Problem 7 and adjust the text accordingly.

35. Line 638 "tresholded": Should be "thresholded"

36. All figure legends: Please report in all figure legends the number of mice used for each measurement AND the statistical significance of differences where N = number of mice per group. Statistics can also be reported for number of lobes, cells, lymphatics, or ears, if desired.

Referee #2

(Remarks to the Author)

The manuscript submitted by Mäkinen and colleagues report on the description and some functional studies on overlapping, oak leaf-shaped lymphatic endothelial cells (LECs) in lymphatic capillaries in the ear skin of mice. Using a mosaic

fluorescent labeling of LECs in lymph vessels, the authors confirm the presence of oak leaf-shaped LECs in lymphatic capillaries (as already reported by Baluk et al. and others) and their absence in the LECs of lymphatic collector vessels. They also confirm previous observations that button-like junctions are found in the lymphatic capillaries, and they show that these buttons develop mainly postnatally. By providing impressive images of lymphatic capillaries, they could demonstrate further that microtubules are located at the concave side of the lobes of the LECs, while actin filaments are exclusively located at the convex side of the lobes. The authors draw parallels between the oak leaf-shaped LECs and the so-called puzzle or epidermal pavement cells of plants. Likely because Rho-related GTPases have been shown to shape these puzzle cells in plants by regulating both actin and tubulin assemblies, Mäkinen and colleagues knocked out the Rho-family kinase *cdc42*, and they report that the typical oak leaf shape of the LECs changed, concomitant with changes in microtubules and junctions. Using mechanical stretching experiments, the authors further show that in human LECs, pharmacologic inhibition of *cdc42* reduces the overlaps of the LECs (that are found in lymphatic capillaries), and they present an *in silico* model that these changes could result in a collapse of the lymphatic capillaries when the interstitial fluid pressure outside these vessels increases (but they do not show this in experiments). Finally, using time-lapse imaging over several weeks, Mäkinen and colleagues indicate that LECs can change their shapes over several weeks and, more notably that actin filaments in these LEC quickly rearrange. Most notably, the authors show that by injecting PBS into the ear skin (as previously reported by Planas Paz and colleagues) increases the interstitial fluid pressure and reduces the overlap of the lobes of the LECs, likely causing the diameter of the lymphatic capillaries to widen. The authors conclude that the overlaps are not mere valves that allow inflow rather than outflow of fluid, but that these overlaps serve as a cellular reservoir for being able to widen the lumen of a lymphatic capillaries. In the opinion of this referee, this finding represents the most novel aspect of the submitted manuscript, as it shifts our view about these overlaps from flap valves for directed flow of fluid into the capillary lumen to cellular reservoirs for lumen expansion.

The following suggestions arose while reading the submitted manuscript, and they are grouped into major and minor ones. Major suggestions:

1. The molecular mechanism of oak leaf-shape and overlap changes remains largely unclear, except that *cdc42* is required for maintenance *in vivo* and resistance to mechanical stretch *in vitro*. Since *cdc42* does many things in a cell, the pathway must be better explored. In particular, beta1 integrins have been widely reported to be absolutely essential for lymphatic development and for mechanosensing in lymphatic and blood vascular endothelial cells. Even though integrins do not exist in plants, integrin like proteins have been identified (such as membrane-bound formins) as well as several integrin-linked kinases, and some of these have been shown to be involved in the shape of puzzle cells. Since manipulation of integrins is well-established, the authors must link their findings on *cdc42* and stretching with beta1 integrins using KO of *Itgb1* and/or RGD peptides and/or integrin blocking or activating antibodies. In this regard, the authors are mistaken to give anchoring filaments a big role in their suggested model, in particular since genetic deletion of filament proteins has minor effects on the lymph vessels compared to deletion of beta1 integrin or integrin-linked kinase. Therefore, the role of integrins, ECM in their model must be urgently incorporated.
2. The authors are asked to not only look at the role of *cdc42* and integrins in the shape of the LECs, but are also asked to analyze how the overlap width is influenced by *cdc42* and (likely its upstream signal) beta1-integrin. The widening of the lymphatic vascular lumen shall also be quantified, if possible.

Minor suggestions:

1. Overview images are needed for many images shown. It is the impression of this referee that a distinction between the capillary part and the collector part of the vessels is largely made by looking at the shape of the LECs. However, when manipulating the LEC shapes *in vivo* using *cdc42* deletion or inhibition as well as beta1 integrin deletion or inhibition, it is necessary to see the parts of the lymphatic vessels connected with each other.
2. Fig. 1e: Please add "weeks of age" to make this panel self-explanatory.
3. Fig. 2a: All data on differential gene expression must be included in the Supplement in form of a Table with all genes analyzed as well as log₂ ratios, p values, etc.
4. Fig. 2d-e: The F-actin localization should also be quantified, and this is an easy task.
5. Fig. 3a: The authors switch from a VEGFR3-CreERT2 line to a Prox1-CreERT2 line. What is the reason?
6. Fig. 3j: It looks as if a *cdc42* floxed mouse is used as a control. However, since effects of Cre lines (due to their promoters capturing transcription factors and thereby influencing cell fates) have been reported, the authors must always use the respective Cre line as a control for comparative analyses.
7. Fig. 4a: The authors give the wrong impression that their stretching device results in stretching of a real lymphatic vessel. They must correct this, and they must show how they stretched LECs in their device.
8. Fig. 4b: These analyses shall be complemented by using siRNA to knockdown *cdc42*, since their *cdc42* inhibitor might also cross-inhibit other molecules.
9. Fig. 5a: The long-term studies on the LECs of the ear skin are difficult to interpret and do not help the study. Mice normally grow and get bigger over time, so that some changes in cell shapes are not unlikely to happen. In addition, the angle of imaging might slightly change, so that the cell shape looks different, even though it stayed the same, but is just shown from a different angle. The authors shall consider deleting this Figure panel.
10. Fig. 5d: Quantification of zipper versus button like junctions is not always as trivial as presented. Again, the imaging angle can make a zipper appear like a button. The authors must discuss this issue.
11. Fig. 6: Integrins and ECM must be placed next to *cdc42*; otherwise, the whole model is not mechanistic. Conversely, the anchoring filaments should be removed. They are present, but their impact on lymphatic development and function is limited.
12. Text line 87 and 157: Please reword that VEGFR3- and Prox1-Cre are LEC specific. There is no strict LEC-specific Cre line.
13. Text line 290: The sub-minute remodeling of actin filaments in the LECs of capillaries must be complemented with imaging and quantification of actin filament remodeling in collectors.

Further points:

- Statistical tests shown in the legends to the Figures are valid choices. "Non-significant" could be removed from some panels though.

(Remarks to the Author)

In this manuscript, the authors took an elegant approach to utilize a mosaic mouse line to image and clearly visualize the morphology of individual lymphatic endothelial cells within the mouse ear with a level of clarity that had not been achieved previously. With this tool they were able to quantify many morphological features of the LEC population in this tissue through various stages of post-natal development. Using a model that allowed for LEC specific post-natal deletion of the Rho GTPase Cdc42, the authors demonstrate its importance in regulation of the actin cytoskeleton and specialized junctions in LECs, and in vitro disruption of Cdc42 in the context of stretch compromises monolayer integrity. They note that the unique lobate shape of capillary LEC is similar in morphology to plant puzzle cells, and utilize a computational FEM model to suggest this unique shape may serve to lower the mechanical stress experienced by the cell, as is the case in plant puzzle cells. Lastly the authors present data on changes in cellular junction arrangement in response to a bolus injection of fluid into the ear interstitium.

Major Concerns

1. The computational model, as it is currently described, does not provide strong support that the underlying benefit of the lobate shape of LEC is to minimize intracellular mechanical stress. There are oversimplifications in the model (or at least my interpretation of how it was implemented based on the description provided) that would play a major role in how force is transmitted across the monolayer of cells. These include:
 - a. The assumption of homogenous mechanical properties of the cell. The authors provide experimental data that both stress fibers and junctional proteins are located at unique locations on the cell membrane. Once cell will preferentially transmit force to another cell at these locations. It seems that the model treats the cell-cell connectivity and mechanical properties identical at all of the interfaces between the cells and only changes the shape.
 - b. One of the most important structural features of initial lymphatics crucial to the unique function of lymphatics, is the unique anchoring filaments that connect certain locations of the cell to the interstitium. While the authors acknowledge these in the discussion, their importance are never considered in the interpretation of the data. The mechanics of the interstitium are also not considered. As interstitial fluid pressure increases, the interstitium will expand because it is a poroelastic material. This expansion transmits force through the anchoring filaments to the initial LEC which helps expose gaps in the junctions that allow for the rapid entry of large molecule along a favorable pressure gradient from the outside of the vessel to the inside of the vessel. It is in the context of this scenario that the cell experiences mechanical loading. It is unclear how the simulations would be relevant to this much more complex context.
 - c. In contrast to plant cells, the mechanical load on the cell is supported by the cytoskeleton, specifically F-actin, which exhibits highly anisotropic non-linear strain stiffening behavior, questioning the relevance of an isotropic St. Venant model, particularly in the context of large deformations.
 - d. Based on previous work published in plant puzzle cells (Sapala, *Elife*, 2018) and the simulations that were run, the results are to be expected. The Young's modulus of the cell wall and the internal cell pressure used for the LEC (10 KPa & .015 KPa) are appropriately orders of magnitude lower than what was used for plant cells (300 MPa & 0.5 MPa), however the ratio between the two parameters is quite similar (667 vs 600). Thus it would be expected that the stress would be lower in the puzzle cell configurations for the simulated LEC as well. This phenomenological behavior would be the same whether the computational LECs were configured in a monolayer or in the shape of a tube as was done here. What is not explained, is why one would not also expect to see a puzzle shape for LEC in a collecting vessel, or for any endothelial cell for that matter. If it could be shown that the benefit of this puzzle shape exists only in the context of some feature unique to initial LEC incorporated within the FEM model (e.g. present of anchoring filaments, discontinuous junctions, unique cytoskeleton arrangement, unique internal cellular pressure, unique Young's modulus of the cell wall, other unique external mechanical loading condition, etc...), then this concern would be minimized.
2. I would caution overinterpreting the importance of the observed cytoskeletal changes in of LEC monolayers under isotropic stretch. Others have reported similar qualitative and quantitative changes in junctional and cytoskeletal morphology, and/or barrier function in LEC when mechanically stimulated by oscillatory shear stress (Norden, *eLife*, 2020) or transmural flow (Miteva, *Circ Res*, 2010). These monolayers are still much closer to collecting LEC than capillaries. Also, LEC on the lumen of a collecting lymphatic will undergo dynamic isotropic stretching with at least a magnitude of 6% during the diastolic filling phase. Also, PDMS typically has a Young's modulus around a range of 0.4 – 1 MPa, which is much stiffer than the stiffness experienced by initial LEC (Frye, *Nat Comm*, 2018). More evidence is needed if the author's want to imply that isotropic stress plays a role in the development of capillary-specific attributes.
3. The authors spend a substantial section of the discussion suggesting that the data here challenges current understanding of how initial lymphatic junctions serve as flap valves. Examples include: "Our results challenge some of the prevailing concepts of flap valves."... "Additionally, through intravital imaging, we observed dynamic remodeling of the actin-rich LEC lobes, contradicting their proposed function as stable flap valves."... "These findings also suggest an active function for capillary LECs in fluid drainage, whereby passive shortening of LEC overlaps and lumen expansion in edema is counteracted by active actin-based lobe remodeling to increase cellular overlap and vessel compression aiding fluid movement, a process reminiscent of a bellow-like mechanism". The valve-like function of initial capillaries was originally proposed primarily through functional data several years before the morphological descriptions of "buttons" and "zippers" took hold in the literature: (Trzewik, J, *FASEB J*, 2001). The data here provides further clarity on how one might interpret observations about buttons and zippers, but the initial understanding of initial lymphatic junctions functioning as flap valves is still unchanged.

4. The approach to measure total Lyve-1 and surface Lyve-1 is very elegant and clever and the images in Fig3 are beautiful. It should be remembered, however, that the labeling was done on fixed, unloaded tissue. Also the mouse ear probably lacks much of the extrinsic factors that are so important in facilitating lymph uptake elsewhere. Have the authors tried to inject a small volume of fluorescently Lyve-1 antibody in the live mouse ear prior to tissue fixation? My hypothesis would be that there would be substantially more Lyve-1 staining without permeabilization. Numerous studies have shown that intradermally injected antibodies rapidly enter lymphatics because of their specialized junctions. You could also inject unlabeled Lyve-1 antibodies in a live mouse ear, fix, and then label with secondary antibodies. In this case I would expect similar images to the surface Lyve-1 stain in Fig3h, as the secondary antibodies may have difficulty accessing the junctions of collapsed, unloaded vessels.

5. I think care should be taken when describing junctions and actin as dynamic. The authors use this description to discuss adaptations that they observe over the course of minutes (Fig 5b) and over the course of weeks. It is not which time scale the overall mechanism proposed by the authors in Fig 6 is meant to function. In his original work Leak noted "It is evident that the rapid removal of connective tissue fluids by the lymphatic capillary is not entirely a passive filtration process but also involves the participation of specific morphological structures located at the connective tissue-lymph interface.." and in the decades since then numerous labs have demonstrated that initial lymphatics take up fluid and larger proteins/tracers (from 20 kDa – 200 nm) within seconds of being administered, making it to downstream lymph nodes within a minute. It is hard to imagine the active actin remodeling of initial lymphatics is crucial to this rapid capability in the context of other mechanisms such as the extrinsic pump (muscle contractions, respiration, gut peristalsis, cardiac contractions, etc...) and suction generated by intrinsic lymphatic contractility (Jamalian, *Sci Rep*, 2017).

6. The manuscript would be significantly strengthened if some of the observed morphological differences in *cdc42* KO mice, resulted in deficiencies in initial lymphatic function (e.g. clearance rate of large MW tracers).

Minor

Might differential growth rates along the membrane driven by difference in mechanical loading be something driving the formation of lobes? This has been shown to be important in plant cells (Belletton, *Nat Plants*, 2021) and interestingly the authors show the preferential location of microtubules (Fig. 2b) on the concave regions, as was reported in plant cells.

Injection of 20 μ l of fluid in a tissue as small as the mouse ear skin is orders of magnitude above what would occur in the context of lymph formation. Even in severe burn models of edema in the ear, the measured accumulation of fluid is usually on the order of at most hundreds of nanoliters. While this data certainly shows the mechanical resilience of the initial lymphatics even under the most extreme behaviors, be careful about extrapolating this to make inferences about the normal function fluid and protein uptake.

SFig1b does not provide convincing evidence that F-actin is low in LECs. Also phalloidin does not seem to co-localize with the most prominent signal from the actin reporter in R26 mice (SFig1e).

Line 174 – "revealed unaltered morphology" should read "revealed altered morphology"

Referee #4

(Remarks to the Author)

The manuscript "Resilience of lymphatic endothelium through dynamic cytoskeletal regulation of puzzle cell shape" explores in lymphatic endothelial cells the relationship between cellular shape, cytoskeleton and cell-cell junctions, and the greater biomechanical forces that act upon these cells. Authors first characterize both the cell shapes and cell-cell interfaces in both terminal capillary vessels, and in collecting vessels. They show that most LECs associate via classical "zipper" junctions, while those at vessel tips display more open button junctions. With both mosaic marking of cells (*iMB2-Mosaic;vegfr3-CreERT2*) and a *LifeAct-GFP* reporter, they find that junctions in capillary LECs exhibit a high level of lobulation and variability that they show represents their dynamic behaviors. They go on to show that these cellular features, including the overlap of protrusions (cleverly measured by comparison of surface versus total LYVE1 levels), depends on the classical cytoskeletal regulator *Cdc42*. They then show that mechanical stress induces responses at the cell-cell interface (such as levels of PECAM), and they model both *in vitro* and *in silico* the effects of mechanical stress on overall vessel morphology. In their last data figure, the authors show that fluid pressure *in vivo* directly impacts the extent of lobule/lamellipodia overlap and hence (most likely) the capacity of lymphatic fluid uptake.

Overall, this study provides novel and timely insights on the critical role of the cytoskeleton on fundamentals like cell shape and adhesion, as well as higher level processes such as tissue morphology and likely function. It is an elegant study that borrows from what is known in plant epidermal cells, and applies findings to lymphatics. Strengths of the study are certainly the novelty of the findings in terms of a mammalian tissue such as the lymphatics and the high quality data on aspects of lymphatic cell biology previously unknown. Weaknesses include an incomplete mechanistic understanding of what exactly the *CDC42* and cytoskeleton are doing within the observed cellular overlaps, and the consequent functional relevance. I would recommend some points for the authors to consider going forward:

Bigger points:

-The authors present compelling data that junctional anatomy changes during lymphatic development and upon changes in hydrostatic fluid pressure. However, do they have data showing that this impacts the lymphatics functionally? Can they show

increased uptake of interstitial fluid associated with changes in cellular overlaps (fig.5d,e,f) with a dye or other tracers? What are the functional outcomes correlated to the cellular changes?

-The model presented in fig.5 is an outstanding demonstration of the findings of the paper. What do the authors think that CDC42 is doing to widen the cellular overlaps? Does it involve polymerization of actin and extension of lamellipodia? Or perhaps does it involve motor molecules helping one "flap" crawl over the other, in a ratchet like mechanism (see Belting and Affolter works in blood endothelial cells)? Have the authors examined phosphorylation of myosin, for example, at the overlap area?

-It is sometimes difficult to distinguish lobes (convex, lamellopodia) versus intervening areas (zippers, concave), as when one considers the multiple cells within a vessel, the lobe of cell A will be nestled within the concave region of cell B. Authors should demonstrate with a schematic exactly how these are defined exactly, relative to buttons, and how this works in a tissue with multiple cells, not just a single isolated cell.

Smaller points:

-How are buttons and zippers quantified, such as in Fig.1? There is only one sentence describing the parameters in the methods. Additional explanation is needed. Sometimes the images are not so clear in terms of buttons versus zippers.

-perhaps shade or false color cells in Fig.1c,f,g. Hard to appreciate cells at times in the VE-cadherin or LYVE1 views. While the YFP is reference in the left panels, it is still hard to see cell borders.

-Fig.1i needs higher magnification panels as in c,f,g to show what authors are talking about.

-Define "Isotropic" in reference to stretch somewhere in manuscript for a wide readership.

-authors should comment on LYVE1 being more prominently expressed at the cell borders. This comes as a surprise in Fig. 2b. Why is it not evenly distributed? Cell outline is useful in 2b.

-There are typos here and there that should be fixed: page 6, line 167 "covex", line 174 "unaltered" should be "altered" I think. Page 10, line 272 "extracellular" should be "extravascular" I think. P.36, line 911 "disperse" should be "dispersed" I think.

-panels in Fig.3b are hard to interpret. Need cell outlines. Hard to appreciate the tubulin connections to concave areas.

-Fig.3h,i,j - the authors quantify cleft width, but they should also quantify the overlap area (especially in i).

-What are the time intervals shown on left side of panels in b?

Version 1:

Reviewer comments:

Referee #1

(Remarks to the Author)

Review for Nature of manuscript #2023-08-13841A entitled: Resilience of lymphatic endothelium through isotropic stretch-induced cytoskeletal regulation of puzzle cell shape

General Comments

1. Improvements: The revised manuscript has many improvements over the original version. The authors have clearly made an effort to address many issues identified by the reviewers. Among the solid findings are the age-related decrease in LEC proliferation and sprouts and increase in LEC lobe number, similarity of VE-cadherin and claudin-5 staining in LECs, distinctive features of LEC overlap in initial lymphatics in mosaic mice, uniform staining of LEC overlaps after in vivo intradermal injection of LYVE1 antibody, contrasting distributions of actin and tubulin cytoskeleton in LECs, cytoskeletal similarities of initial lymphatic LECs to plant puzzle cells, and effects of Cdc42 deletion on LEC cytoskeleton, overlap, and tracer clearance.

2. Needing further attention: However, further attention is needed to address ongoing issues identified in the following comments. These include descriptions and interpretations that reflect the authors' preferred view instead of an objective evaluation of the evidence. One of several examples described in detail below is the authors' view that tight junctions are rare in initial lymphatic LECs, despite IHC staining for claudin-5 matching staining for VE-cadherin. The authors are encouraged to review the descriptions and interpretations throughout the manuscript and make the changes necessary to ensure they are objectively consistent with the evidence.

3. Route for fluid entry into lymphatics: Although the fluid and cell entry function of initial lymphatics is not disputed, the

authors' proposal for how this occurs is confusing and is obfuscated by detailed consideration of LEC overlaps, junctions, cytoskeleton, Cdc42, puzzle cells, in silico modeling, and in vitro stretch effects. As a result, the authors' proposed entry route is questionable and not critically assessed in the context of their data and published evidence.

- a. The authors describe the current view of discontinuous button junctions and flap valves (lines 57-61, 443-446), and then raise doubt about this view by reporting that only 20-30% of the junctions were buttons and the remainder were curvilinear or double junctions that were either continuous or discontinuous (lines 116-119).
- b. In reconciling their findings, the authors propose (lines 479-482) that the entry routes are the intercellular clefts described by Leak (JCB 1971), who provided TEM evidence for tracer entry through spaces between LECs.
- c. The authors' proposal becomes confusing and loses plausibility when they claim that entry occurs at intercellular clefts joined by adherens junctions but not by tight junctions (lines 479-482), again citing Leak (JCB 1971).
- d. This claim misrepresents key findings by Leak (JCB 1971) illustrated by TEM images, described in the text, and beautifully summarized in a drawing (his Fig.30) showing entry between LECs through junction-free spaces bordered by maculae (spot) adherentes and maculae occludentes. Leak's images provide evidence for tracers entering lymphatics in junction-free regions of LEC overlaps located between maculae (spot junctions).
- e. Perhaps the misinterpretation stems from Leak's use of the Latin term "maculae" (spot) for discontinuous adherens and tight junctions (now known as buttons). Leak did not describe the continuous junctions (zonulae adherentes and zonulae occludentes, now known as zippers) that form the endothelial barrier of collecting lymphatics and blood vessels.
- f. The concept advanced by Leak (JCB 1971) for tracer entry through junction-free clefts bordered by discontinuous maculae adherentes serves as important background for the current concept of button junctions that border flap valves.
- g. The authors are asked to resolve the confusion by considering this background and the comments below and then explicitly describing their proposal for fluid entry into lymphatics and how their proposal is consistent with or inconsistent with their data and published evidence.
- h. As a starting point for their proposal, the authors should address the question of whether their findings can be integrated with the current view and thereby advance the understanding of primary lymphatic valves by incorporating what they learned about LEC overlaps, cytoskeleton, Cdc42, and in vivo motility.

4. Transparency and balance: Greater attention should be given to the background on which the authors build their story. The authors should also identify more clearly the assumptions underlying their junction-based lamellipodia hypothesis and balance their consideration of evidence for and against this hypothesis.

- a. Although everyone would agree that the mechanisms regulating the formation and maintenance of the specialized junctions of initial lymphatics are not fully understood, the authors should be more transparent about what was known before their study (lines 71-72). At a minimum, references 6, 7, 14, and 15 should be cited for evidence favoring the contributions of VEGFA, VEGFR3, NOTCH1, angiopoietin-2, and glucocorticoid receptor signaling in the formation and remodeling of button junctions.
- b. The discussion of the authors' hypothesis (lines 443-463) should be more balanced. It is not fair to cherry pick evidence that fits their hypothesis and trivialize evidence against it.
- c. Importantly, the implication that their evidence invalidates the LEC flap valve concept is not justified and should be revised.
- d. The authors' do not present convincing evidence to justify the implication (lines 443-463) that their data for LEC junctions are more valid, more meaningful, and should have greater weight than corresponding data in previous reports (see "8. Classification of LEC junctions" and "10. Reasons for discrepancies with published data...").
- e. The authors also do not explain (lines 443-463) why the observed changes in overlap, distribution of the actin cytoskeleton in flaps, and involvement of Cdc42 are inconsistent with the conventional view of LEC primary flap-valves as fluid entry sites. Why can't the flap-valve and lamellipodia concepts be merged into one integrated concept?
- f. To solve these problems the authors should make changes throughout the manuscript to reflect their commitment to transparency, balance, and objectivity as they describe the strengths and limitations of their findings and interpretations and how they build on what was already known.

5. Healthy skepticism: Related to the previous comment, the manuscript in its present form does not exhibit a healthy level of skepticism over the relative strength of the findings and the authors' preferred interpretation over other reasonable interpretations. The manuscript would be strengthened by authors' distinguishing (i) data shown to be solid by reproducibility and statistics and having an unambiguous interpretation from (ii) data and interpretations that could be viewed by readers as less convincing, potentially inconsistent with other evidence, or having interpretations different from the authors' preferred view.

- a. Examples of solid evidence include the similar shape of oak leaf LECs and plant puzzle cells, conspicuous overlap of adjacent LECs viewed in mosaic mice, uniform staining of overlapping regions after in vivo injection of LYVE1 antibody, distribution of actin in oak leaf-shaped LECs, distinctive differences between LEC junctions in initial lymphatics and collecting lymphatics, and effects of Cdc42 deletion on LEC cytoskeleton and overlaps.
- b. Examples where the authors' interpretations would benefit from more skepticism and/or stronger data include the identification and quantitative comparison of tight junctions and adherens junctions in TEM images, relevance of stretch-induced changes in cultured LECs to oak leaf LECs in vivo, comparisons of stretch effects on LECs and HUVEC in culture, and the treatment of differences between their data and published data on junction classification in LECs.
- c. An example where additional data are needed is the claim (lines 125-127 and Fig.6) of changes in % of junction types in LECs (Fig.1g) resulting from increased interstitial fluid pressure after intradermal PBS injection, where the current data came from only n = 2 mice/group, and the significance of differences was not subjected to statistical testing (Fig.1g).

6. Inconsistencies: The revised manuscript has multiple apparent inconsistencies that need to be addressed.

- a. In an innovative and informative experiment, the authors report that injection of LYVE1 antibody into ears resulted in uniform labeling of overlapping flaps of adjacent LECs that was not found after conventional IHC without permeabilization

(Fig.2h). The authors reasonably interpret these findings as showing that regions of overlap permitted entry of antibody in life but not after fixation (lines 154-158). These in vivo LYVE1 antibody experiments seem to undermine and invalidate the authors' junction-based lamellipodia barrier hypothesis, whereby the overlapping flaps are anchored by barrier-forming curvilinear and double junctions (VE-cadherin + claudin-5, lines 450-463). Please revise.

b. The authors' IHC staining clearly shows similar abundance and distribution of VE-cadherin and claudin-5 in LECs of initial lymphatics (Fig.1b). scRNA-seq analysis revealed even greater expression of *Cldn5* than *Cdh5* in the LECs (Suppl Fig.4a). However, the authors interpret their TEM studies as showing that adherens junctions are abundant but tight junctions are rare in LECs (lines 482-483, Extended Data Fig.2g). They then rationalize the finding by proposing that "adherens junctions in LEC...contain...tight junction molecules (lines 484-487)". This interpretation seems far-fetched next to the conventional view that adherens junctions and tight junctions are both abundant and adjacent in LECs. Please see "9. TEM studies of LEC junctions" and revise.

c. The authors' claim that button junctions are too rare (20-30%, line 129) to be sites for significant fluid entry into ear skin initial lymphatics but then argue that intercellular clefts between overlapping LECs serve as the primary route for fluid and solutes entry (lines 479-480). Importantly, only 20% of LEC overlaps examined by TEM had no junctions (Extended Data Fig.2e). To explain the inconsistency, they misrepresent Leak (JCB 1971) as reporting that adherens junctions, unlike tight junctions, permit tracer entry (lines 481-482). However, this interpretation is inconsistent with Leak's description of the entry routes as "intercellular clefts of patent junctions" and "cell junctions devoid of adhesion devices, i.e., maculae occludentes and maculae adherentes (JCB 1971)." In other words, Leak described the entry routes as junction-free clefts located between discontinuous junctions (maculae). Please revise.

d. The authors' report of relatively uncommon button junctions (20-30%, line 129) in ear skin initial lymphatics is inconsistent with values of about 50% reported by Jannaway et al. (reference 14). This discrepancy needs to be addressed more thoroughly and reconciled (see "8. Classification of junctions").

e. The authors claim their finding of infrequent buttons is supported by published data showing that button junctions constitute 25-30% of junctions in adult tracheal lymphatics (lines 431-432 and 528-529). However, this claim is inconsistent with data in that report (Baluk et al. JEM 2007). Please see "Specific Comment 5. Initial lymphatics of trachea and diaphragm" and revise.

f. In explaining the difference in their interpretation of button junctions in ear skin lymphatics from data on intestinal lacteals (lines 582-587), the authors provide the convenient but unconvincing argument that LEC junctions vary in different organs. Please revise.

g. Data showing no differences in abundance of zipper junctions from 3 to 25 weeks of age (Fig.1f) are inconsistent with subsequent statements (lines 420-431) saying the opposite: "...at three weeks of age...lymphatic capillary tips showed...the presence of continuous zipper junctions. The subsequent reduction in the abundance of zippers during further development was associated with the cessation of vessel sprouting and LEC proliferation." Please revise.

h. The manuscript would be strengthened by greater attention to aligning the authors' new data with published data – instead of trying to rationalize the inconsistencies and contradictions.

7. Lymphatic junction heterogeneity: Although improved from the original manuscript, the description of the heterogeneous junctions in initial lymphatics is still subjective, misleading, and detracting from the many attributes of manuscript.

a. Greater care and objectivity in describing the background and interpreting images of junctions in the context of previous publications would strengthen the manuscript without loss of novelty.

b. Characterization of the heterogeneity of discontinuous junctions of initial lymphatics is represented as a novelty of the manuscript, whereas in reality it confirms earlier work.

c. Images in all papers illustrating discontinuous junctions in initial lymphatics, including references 4, 6-8, 13-15, 32, and 45, show junctional heterogeneity. This heterogeneity is described and quantified in Jannaway et al. (reference 14).

d. Statements in the manuscript that disregard this background should be corrected. Examples are: "...we uncovered a spectrum of junctional configurations..." (lines 82-83), and "We observed mainly extended linear VE-cadherin+ adherens junctions...rather than the expected predominance of discontinuous buttons..." (lines 96-98).

e. As the authors naturally build on earlier work, they should be more even-handed in distinguishing what was already known from what is new in their manuscript. A more objective alternative to lines 82-87 is described in Specific Comment 3e.

8. Classification of LEC junctions: The authors' classification of junctions in initial lymphatics continues to be problematic.

a. The authors changed their classification of non-button LEC junctions in initial lymphatics from 5 types of continuous junctions (original Fig.5c) to 2 types of discontinuous junctions (curvilinear and double) and 3 types of continuous junctions (curvilinear, double, and zipper). This change reflects the subjectivity of classifying these junctions, is confusing, and does not make sense.

b. Regardless of what the junctions are called, the most important distinction is whether junctions are discontinuous or continuous. The names "buttons" (discontinuous junctions) and "zippers" (continuous junctions) were introduced to make this distinction (reference 4). The term "intermediate" was subsequently introduced to describe discontinuous junctions that extend around more of the LEC perimeter than buttons. In contrast, "zippers" are continuous junctions that completely surround the cell perimeter, as in collecting lymphatics and blood vessels.

c. Now, the authors' propose a revised classification where some curvilinear and double junctions are discontinuous and others are continuous (lines 796-804, Supplement, Image quantification). This implies that zippers are not the only junctions that completely surround LECs.

d. Apparently the authors' concept of discontinuous or continuous junctions has morphed from entire LECs – as used in previous publications - into individual lobes of LECs (Supplement, Image quantification). Therein lies the source of a problem.

e. A unique feature of initial lymphatic LECs is the presence of discontinuous junctions that permit entry of fluid, solutes, chylomicrons, etc. The presence of "intermediate" junctions indicates that the proportion of LEC perimeter with discontinuous junctions varies from cell to cell, but few LECs of initial lymphatics are completely surrounded by continuous junctions

(zippers).

f. For the authors to solve their classification problem and meaningfully interpret their data for individual lobes (Fig.1f, 1g), additional data are needed for the (i) % of individual LECs with one or more discontinuous junctions; (ii) % of the perimeter of individual LECs with discontinuous junctions; and (iii) frequency of discontinuous button, curvilinear, and double junctions expressed as % of lobes of individual LECs. Please use an LEC sample size to accommodate the cell to cell variability and give the mean \pm SE for each value for adult mice (n = 5 mice).

g. Classification of discontinuous junctions as "button", "curvilinear", or "double" is not a problem, as long as only discontinuous junctions are so classified. However, the authors should explain that they subdivided junctions previously called "intermediate" into two groups, curvilinear and double. The term "zipper" should be restricted to continuous junctions that completely surround individual LECs, for consistency with the original definition.

h. The corresponding text, figures, including summary Fig.6, and figure legends should be revised to accommodate the requested new data and revised descriptions and interpretations.

9. TEM studies of LEC junctions:

a. TEM values of 17% tight junctions and 83% adherens junctions (Extended Data Fig.2f, 2g) appear to reflect technical limitations of the TEM images (Extended Data Fig.2c) more than the actual % of junctions.

b. The authors' TEM data appear to suffer from technical limitations where the identity of junctions cannot be determined with certainty in some panels of Extended Data Fig.2c A1-A5 (top row left to right) and B1-B5 (bottom row left to right) where plasma membranes were cut obliquely or sections had a thickness (60-70 nm, line 830) greater than optimal for resolving the trilaminar structure of unit membranes (<60 nm).

c. The authors do not acknowledge that some regions marked by red arrowheads in panels A1-A3, A5, B3 in Extended Data Fig.2c are uninterpretable for the presence of tight junctions and some regions marked by blue arrowheads in panels B2, B4 are oblique sections uninterpretable for any junctions.

d. In addition to the section plane and thickness issues, the trilaminar structure of unit membranes was difficult to visualize in the absence of en-block staining with uranyl acetate during tissue processing (lines 816-833). See Fig.11 in Leak, JCB 1971 and Fig.21 in Brightman & Reese, JCB 1969, where uranyl acetate en-block staining was used, for comparison.

e. Confocal images (Fig.1b, Extended Data Fig.6b) show that VE-cadherin and claudin-5 have largely identical distributions, consistent with previous work (reference 4). Therefore, tight junctions would be expected to be at least as numerous as adherens junctions. Expression data for *Cldn5* than *Cdh5* mRNA (Suppl Fig.4a) are consistent with the IHC similarity.

f. The authors attempt to reconcile the discrepancies by speculating that adherens junctions contain tight junction proteins like claudin-5 (lines 486-487). They also argue that deletion of claudin-5, "the second major component of LEC junctions" (lines 472-473), did not disrupt LEC junctions (lines 287-288), but fail to acknowledge that button junctions contain multiple tight junction proteins, including ESAM and JAM-A, in addition to claudin-5 (reference 4).

g. From their interpretations and claims, the authors seem to use infrequency of LEC tight junctions to support their lamellipodia hypothesis. However, the evidence argues otherwise and raises questions over the authors' interpretations and whether the TEM studies were technically suitable and sufficiently rigorous to determine the abundance of tight junctions.

h. Although the optimal solution is for the authors to obtain data from new TEM images that meet the standards of their predecessors cited above, an acceptable alternative would be to (i) delete Extended Data Figs.2e and 2g that show values for the number and % of tight junctions and adherens junctions; (ii) delete or relabel Extended Data Fig.2f to show the intercellular distance values do not apply to tight junctions. Unless better documented, values >20 nm are unlikely to fit with adherens junctions; (iii) replace all TEM images in Extended Data Fig.2c with three high mag images of LECs: a tight junction, an adherens junction, and no junction; (iv) delete from the text (lines 482-483 and elsewhere) all descriptions of TEM data for the number or % of junction type; and (v) acknowledge the technical limitations faced in visualizing LEC junctions by TEM.

10. Reasons for discrepancies with published data on lymphatic junctions: The authors' explanation for why their data for lymphatic buttons and other junctions differ from corresponding data in multiple previous papers is incomplete and unconvincing.

a. The Introduction (lines 57-76) should set out more background on what is known from previous work on junctions in initial lymphatics (cite references 4, 7, 8, 13-15, 32, 45).

b. In the context of this background, the specific aims (lines 78-87) should more clearly define the intended purpose for analyzing and classifying LEC junctions in this study in the context of what is already known. See Specific Comment 3e.

c. The authors claim that their IHC and confocal imaging techniques had greater sensitivity and resolution than previous studies (lines 432-436). But this implies their data are more accurate and believable. However, this claim is not supported by evidence of greater sensitivity and resolution. It is unclear how sensitivity and resolution would be compared among studies.

d. Differences in developmental age are mentioned but unlikely to be factors because the authors' own data show no difference between age 3 and 25 weeks (Fig.1f, Extended Data Fig.1b).

e. A factor not considered is the exposure of lymphatics to mechanical trauma by removing and cutting ear skin from cartilage BEFORE fixation, instead of fixing the tissue by vascular perfusion before removal. Manipulation of ear skin before fixation could contribute to morphological changes in lymphatics that would be prevented by vascular perfusion fixation (reference 4). Please address these issues in the Discussion.

f. Differences in classification of lymphatic junctions in this and previous studies should also be considered in more detail, acknowledging the subjectivity and pros and cons of each approach.

g. Regardless of the terms used, the classification should distinguish discontinuous junctions (maculae) from continuous junctions (zonulae) in LECs.

11. LEC and HUVEC stretched in vitro: The in vitro studies of isotropic stretch described in the revised manuscript are strengthened by the addition of HUVEC for comparison to LECs. However, the comparison is limited by several problems:

a. The studies of stretch-induced shape changes in LEC overlap deserve more thorough consideration of the underlying

assumptions. The presumption that studies of LEC stretch in vitro mimicked the process that initial lymphatics undergo during fluid entry is insufficient.

b. What is known about the relative contributions of stretch and expansion without stretch as collapsed lymphatics fill under physiologic conditions? Is the process different from venules that collapse and expand moment by moment during positional changes? What process of lymphatic biology is mimicked by cyclic stretch for 100 seconds per cycle for 14 or 22 hours?

c. The subtlety of shape changes in LECs after 22 hours of stretch shown in the images (Fig.5g) should be acknowledged. The description of stretch-induced changes (lines 507-509) does not faithfully reflect the subtle changes shown by the images. Please revise.

d. The conspicuous broad overlap of lobes in control LEC in vivo (Fig.2i) differs markedly from the narrow overlap of stretched LECs in vitro, except in focal regions (Fig.5g). However, the overlap measurements indicate the opposite ($\sim 2.4 \mu\text{m}$ in vivo, Fig.2j vs $\sim 3.1 \mu\text{m}$ in vitro, Fig.5i). These differences are inconsistent with the statements on lines 507-509. Please revise.

e. Shape changes in the LEC monolayer shown in Supplementary Movie 7 after Cdc42 activation are more subtle than implied by the description (lines 359-362, 457-459). They are also not remotely reminiscent of oak leaf LECs in vivo. These discrepancies raise questions over whether the description and data in Fig.5k are truly representative of cells in the monolayer. Please align the text and Fig.5k more closely with what readers can see in the movie.

f. The statement, "Together, these results show that junction-based lamellipodial protrusions promote LEC monolayer integrity (lines 371-372)" is not justified, as the reported correlation of shape change and monolayer resistance does not prove cause and effect. Please revise.

g. The interpretation that stretch increases overlap of LECs but not HUVEC in vitro (lines 507-509, 515-517) is misleading because HUVEC already had more overlap before stretch ($\sim 3.6 \mu\text{m}$, Extended Data Fig.10b) than LECs had after stretch ($\sim 3.1 \mu\text{m}$, Fig.5i). The conspicuous difference in overlap of HUVEC compared to LECs, with or without stretch, is evident in the images (Fig.5g vs Extended Data Fig.10b). Please revise the descriptions to reflect these differences and address concerns over the meaningfulness of the LEC/HUVEC comparison.

h. Another difference that deserves comment is that most overlaps of "lobes" in stretched LECs lack VE-cadherin staining (Fig.5f, 5g) and do not resemble LEC lobes in vivo (Fig.1a, 1c) or overlaps of HUVEC that have uniform VE-cadherin staining (Extended Data Fig.10a). These differences weigh against the relevance of shape changes of LECs in vitro. Please address.

i. Images and measurements for LECs and HUVEC stretched in vitro should be shown side-by-side in the same figure to make it easier for readers to compare the two cell types.

12. Statistics: Please address the following issues:

a. The authors should specify the statistical test used for graphs with three or more groups that show no P values (Figs.1e, 1f, 2d, 2f, 4k). Please confirm that these comparisons were made by ANOVA, as described in the Methods, and that Student's t-tests were used only when the dataset included two groups. Absence of statistical significance should be marked as such.

b. Violin plots are confusing when added to graphs that show all individual values and should be removed unless clearly justified. As the individual values show the actual distribution, the empirical distribution shown by violin plots is unnecessary. Violin plots also overlap and obscure some data points and have the distraction of extending above and below the actual data. Prism statistical software recommends not using violin plots with plots of individual data points.

c. Showing medians (e.g., Fig.2d, 2e, 2f, 2j, 4f, 4h, 4k) instead of means implies the data are not normally distributed, yet parametric statistical tests were used. Please justify and change the statistical test or change medians to means.

d. In the analysis of junction types, it is unclear why Individual datapoints are presented as weighted averages per mouse instead of means. Please change to conventional means or justify the use of weighted averages and describe the weighting procedure.

e. Where values are normalized, specify the reference used for normalization, e.g., Fig.4l.

f. The number of mice per group should be increased for data reported for only 2 mice/group.

13. Response to reviewers' comments: The current Responses to Reviewers' comments gives much greater attention to explaining the authors' views than to describing the changes made to address reviewers' comments. Please take to heart that the comments are intended to strengthen the manuscript by alerting the authors to issues likely to be problematic for readers. If given the opportunity to prepare another revision, the authors would benefit from focusing their responses on the changes made in the text and figures to address each comment and identifying where in the manuscript these changes were made (line numbers, figure numbers). Lengthy justifications of the authors' preferred views, instead of focusing on the changes made, are unnecessary and work against the opportunity to strengthen the manuscript for a broad readership.

Specific Comments

1. Junctions on entire LECs versus individual lobes: Values in Fig.1f, 1g reflect the percent of each junction type determined by "numbering of individual lobes of...LECs and subsequent categorizing of lobe-associated junctions based on VE-cadherin signal (Supplement, page 2)."

a. Values for junctions on individual lobes do not inform the junctional composition of entire LECs on which barrier function is dependent.

b. The Methods (lines 796-804, Supplement: Image quantification) describe curvilinear and double junctions as continuous or discontinuous, but this distinction has little significance when applied to individual lobes instead of entire LECs.

c. When entire cells are considered, most LECs illustrated have regions of discontinuous VE-cadherin staining, which fits the original description of discontinuous LEC junctions, regardless of what they are called (maculae, buttons, curvilinear, etc.).

- d. Other junctions are continuous (zonulae, zippers) around entire cells, but these were uncommon on oak leaf-shaped LECs.
- e. In Fig.6, the junction labeled “Curvilinear” and “Zipper” is confusing because it is classified as both types of junction and is also misleading because it differs from the conventional meaning of “zipper” as a continuous junction around an entire endothelial cell. Please see “8. Classification of LEC junctions” and revise.
- f. Fig.6 should also be revised to match the authors’ data, including evidence for no significant difference in junction types resulting from PBS injection (Fig.1g).

2. Categories of discontinuous junctions in ear skin: The idealized drawings of junctions (Fig.1f, Fig.6) are misleading because they do not faithfully match the corresponding images of these junctions.

- a. The drawings show uniform bands of VE-cadherin, whereas the images show variable staining. Images throughout the figures show that most junctions around oak leaf shaped LECs are discontinuous, regardless of whether they are designated button, curvilinear, or double junctions (e.g., Fig.1b, 3b, 4b Cdc42flox/+, 4g Cdc42flox/+, 4g Ctrl, Extended Data Fig.1c). Please revise.
- b. As described in “8. Classification of LEC junctions”, these junctions are distinctly different from continuous zonulae (zipper junctions) in collecting lymphatics (Fig. 1a right, 3g lower, etc.).
- c. The drawings in Fig.1f and Fig.6 and the corresponding descriptions should reflect the discontinuity of the junctions. Please see “8. Classification of LEC junctions” and revise.

3. “Spectrum of junctional configurations”: Descriptions throughout the manuscript of initial lymphatics having a “spectrum of junctional configurations” are potentially misleading because they are insufficiently informative.

- a. “Spectrum of junctional configurations” could be interpreted as a mixture of tight junctions, adherens junctions, gap junctions, desmosomes, etc.
- b. This is of particular concern in the Summary and Introduction.
- c. More objective and informative alternatives are shown below in 3d and 3e.
- d. Summary (lines 39-40): “LECs in dermal lymphatic capillaries had a heterogeneous mixture of buttons and other discontinuous junctions and unique...”
- e. Introduction (lines 82-87): “In the current study, we used novel mouse models and approaches to characterize endothelial cells of initial lymphatics in mouse ear skin and mechanisms responsible for the distinctive oak leaf shape of the LECs. We identified new features of LEC overlap in mosaic mice and confirmed the heterogeneous mixture of buttons and other discontinuous junctions between the LECs, as shown previously (references 4, 6-8, 13-15, 32, 45). We then determined the contributions of cortical actin, microtubules, and Cdc42 in regulating the oak leaf shape of LEC.”

4. “Dynamic” remodeling of LECs: The authors’ reference to “dynamic” remodeling of LECs (lines 406-411 and elsewhere) is confusing because they use the same term for diverse changes occurring at very different rates.

- a. Actin remodeling in LEC lobes occurred in seconds (Fig. 3h, Supplementary Movie 3 and 4).
- b. Changes in LEC junctions and lobe overlap after intradermal injection of PBS (lines 130-131, Fig.1g, lines 165-167, Fig.2i, 2j) occurred over 10 minutes.
- c. Changes in the shape of cultured LECs exposed to cyclic stretch (lines 346-351, Fig. 5e-h) occurred over 14 or 22 hours.
- d. Spontaneous changes in LEC lobes observed by intravital imaging (Fig.2k) occurred over 6 to 14 weeks.
- e. Given the very different durations of these processes, the authors should distinguish changes occurring in seconds or minutes from those occurring in hours, days, or weeks. Use of the term “remodeling” for all these events conflates processes that could have different mechanisms and functional implications.
- f. As remodeling is, by definition, a dynamic process, the word “dynamic” in dynamic remodeling is redundant. Could remodeling be static? The same applies to “active” dynamics. Please justify or change.

5. Initial lymphatics of trachea and diaphragm:

- a. The claim that initial lymphatics of the trachea and diaphragm had junctions similar to those in the ear (lines 122, 527) is not convincing based on one image of one lymphatic in each organ (Extended Data Fig.1d) and no quantitative data.
- b. The statement, “This finding is consistent with the original description of buttons in the terminal ends of adult tracheal lymphatic capillaries, where they were observed with a frequency of 25-30% (lines 431-432).” is incorrect. Instead, in reference 4, 24% of buttons in the initial 1500 μ m of initial lymphatics in mouse tracheas were located within 125 μ m of the tip, 50% were within 250 μ m of the tip, and 75% were within 500 μ m of the tip (Fig.1E of reference 4). Only 25% of buttons were located from 500-1500 μ m from the tip, indicative of the heterogeneity of junctions and decreasing number of buttons with increasing distance from the tip. The proportions of buttons and other types of discontinuous junctions were not reported in reference 4.
- c. Furthermore, the authors’ claim applies to tracheas and diaphragms removed from dead mice before fixation, whereas published data were obtained from lymphatics fixed in situ by vascular perfusion. Please see “10. Reasons for discrepancies with published data...”.
- d. It is unclear why the authors try to generalize their findings to apply to the trachea and diaphragm and then treat intestinal lacteals (not examined) as a special case (lines 581-585). Please see “6. Inconsistencies” and revise.
- e. The authors’ generalization of their findings in ear lymphatics to apply to lymphatics in other organs lacks adequate supportive evidence. These claims should be validated by more convincing images and quantitative data obtained after perfusion fixation or revised.

6. Additional issues:

- a. Lines 49 and 550: “bellow-like” should be “bellows-like”: Please explain how the proposed “bellows-like” constriction of initial lymphatics, in the absence of a primary flap valve, would propel lymph toward the secondary valve without also driving it backward into the interstitium.

- b. Lines 732 and 816: For animal welfare purposes, please specify the anesthetic, dose, route of administration and conditions for mice used for IHC whole mount staining or TEM. Also specify the method of euthanasia and describe in more detail the preparation of ear skin whole mounts, from anesthesia to immersion fixation.
- c. Line 756: "euthanized" is recommended in place of "sacrificed".
- d. Lines 890-897: In "Author contributions", please identify those responsible for preparing tissues for TEM, taking electron micrographs, and interpreting TEM images.

Referee #2

(Remarks to the Author)

The revised Schoofs et al. manuscript addresses part of the raised concerns about novelty, molecular mechanism and technical controls in an insufficient manner. The question about human relevance of the reported findings was not even raised by any referee, so that a thorough rework of the manuscript with regard to molecular mechanism and technical controls was expected for publication in any kind of journal. Unfortunately, this expectation was not met by the authors. The specific points of critique about the authors' additions to the manuscript in response to this referee's suggestions are the following.

Major suggestion 1.

This referee criticized that cdc42 does many things in a cell, so that the pathway involving cdc42 must be better explored to obtain specificity and functional relevance. Just changing a major regulator of the F-actin cytoskeleton will induce many defects and alterations in a cell, without providing major insights into the physiologic mechanisms and functional significances of cdc42-regulated dynamic junctional overlaps in capillary LECs.

This referee asked (as major point 1) to integrate the role of integrins and ECM in their model, in particular since several publications showed that (i) integrins are mechanosensors and essential molecules in LECs (see e.g., Planas-Paz et al., EMBO J 2022; Kumaravel et al., Am J Physiol Cell Physiol 2020, 319: C1045) and (ii) integrin beta 1 and cdc42 functionally interact in different cell types (see e.g., Keely et al., Nature 1997, 390: 632; Reymond et al., J Cell Biol 2012, 199: 653; Cerutti et al., Cell Rep 2024, 43: 113989), including LECs (see e.g., Valtcheva et al., JBC 2013, 288: 35736 and Liu et al., Development 2018, 145: dev165092). Further, Liu et al. have previously shown in their Development paper that cdc42 affects VE-cadherin junctions in LECs, but these authors even integrated cdc42 into a Rasip1- and integrin-containing molecular pathway, making their paper mechanistically superior over the work presented here. Schoofs et al. conducted experiments on integrin beta 1 though, but the experiments were technically insufficient for drawing strong conclusions. The negative results obtained are difficult to interpret, since they might be due to many possible technical issues. They also failed to expand on the underlying molecular mechanism of LEC overlaps and their functional relevance that likely involves not just a single molecule (that is, cdc42).

Firstly, the authors deleted Itgb1 using Vegfr3-CreERT2, but they compared these mice with floxed mice harboring no Cre, even though this was specifically requested by this referee in Minor Suggestion 6, and this is important since Cre expression has been recently shown to have vascular defect on its own (see e.g., Rashbrook et al., Nat Cardiovasc Res 2022, 1: 806). So, an important control is entirely missing in this revision.

Secondly, a quantification of junctional overlaps was not conducted (quantification of images is supposed to be a normal procedure published in contemporary papers, even in small journals).

Thirdly, the lymphatic vessels in the ear were not challenged by fluid injection as to investigate whether Itgb1 changes the physiologic behavior of lymphatic vessels to an enhanced interstitial fluid pressure. These kind of experiments are already part of this manuscript and not at all beyond the scope of this report.

Fourthly, the authors tried to inhibit integrin beta 1 by adding 0.1 to 0.2 µg/ml mAB13 to unstretched and stretched LECs, but they noticed only a slight decrease in CD31-positive cellular overlaps, even though it looks as if there might be a concentration-dependent effect of integrin blockade. Even though investigations on isolated integrin molecules are published with such low concentrations of antibodies and even though the authors observe an (again not quantified) reduction in integrin beta 1 activation, for cell culture experiments higher concentrations are often used, such 1 µg/ml mAB13 or a 1:10-dilution of purified antibody (see e.g., Lee et al., Circ Res 1995, 76: 209; Aqino et al., BBRC 2024, 703: 149575). Thus, the authors could miss an effect of integrin beta 1 by using too low concentrations. Other inhibitors, such as the RGD peptides, as well as knockdown experiments for Itgb1 are urgently needed as well in order to draw solid conclusions.

Fifthly, if integrin appears to be not involved based on all experiments suggested above (and contrary to a large body of literature on its role in LEC and cdc42), the authors must at least obtain the Rasip1 KO mice and conduct their studies in these mice with genetic rescue experiments to show that cdc42-dependent LEC overlaps are integrated within this pathway (that also involves integrins though) to follow up on the paper by Liu et al. in Development 2018.

Major suggestion 2:

The referee pointed out that a functional relevance of cdc42-dependent cellular overlaps of LECs could be lumen widening in order to be able to take up more fluid from the interstitial space. The authors now respond that they did not observe any differences in cdc42-deficient versus control lymphatic capillaries (data not shown). They argue that an increase of around 1 µm cannot be quantified due to high variability. This argument is surprising given that they quantified changes in LEC overlaps with an average width of 2 µm that decreases significantly to around 1.5 µm in their hands. So, the finding of no lumen increase is at odds to the authors' most relevant finding, i.e. the existence of overlaps that they suggest play a key role in lymphatic vessels. So, what is (expressed in 1-2 sentences) the key novel message that the authors wish to convey in terms of mechanistic insights into lymphatic biology that is of strong functional relevance? Why is the paper more relevant and why does it present a substantial increase in knowledge compared to the Liu et al., Development 2018 paper that shows cdc42 influencing LEC junctions by regulating VE-cadherin expression that involves Rasip1?

Minor suggestion 5:

The referee asked the question why the authors switched from Vegfr3-CreERT2 to Prox1-CreERT2 to delete *cdc42*. The authors argued that Prox1 is stronger for gene deletion, but not specific for endothelium, while Vegfr3 is weaker, but specific for endothelium. Given this argument, couldn't it well be that deleting *cdc42* via Prox1-Cre introduces cardiac defects that subsequently affect lymphatic vessels and interstitial fluid? Is there any possibility to inject tamoxifen into the ear skin to obtain more selective effects rather than bystander effects? How do the authors otherwise rule out that deletion of *cdc42* in other tissues (including other vascular beds) affects the lymphatics indirectly rather than directly?

Minor suggestion 6:

The referee asked for Cre lines as controls due to a recent publication indicating that Cre lines must be taken as controls (see Rashbrook et al., Nat Cardiovasc Res 2022, 1: 806). The authors answer that they did not observe any changes in TEM of LEC overlaps between Cre- and Cre+ mice, but this referee does not see any quantification of Cre- versus Cre+ LEC overlaps in Frye et al., eLife 2020. The TEM images shown in the Figure referred to did not have any Cre- control either. The referee wishes the authors to include Cre+ Tx injected control mice for all comparisons to avoid artifacts induced by the presence of Cre recombinase, as reported by Ruhrberg and colleagues in Nature Cardiovasc Res 2022

Minor suggestion 8:

The referee asked for *cdc42* knockdown experiments that the authors provided. However, the results obtained are in contrast to Liu et al., Development 2018, who observed major alterations in F-actin intensity and lumen width in LECs after *cdc42* knockdown experiments. How do the authors explain and compare their results with these published and more mechanistic data?

Version 2:

Reviewer comments:

Referee #1

(Remarks to the Author)

General Comments

1. Purpose of review: In the evaluation of Revision 2, the reviewer takes the authors at their word that they "are committed to transparency" in response to General Comment 4f of the previous review. Although the availability of all source data reflects this commitment, the purpose of this comment was to ensure that the text and figures in Revision 2 have the same transparency, accuracy, clarity, and balance as the source data. This was not the case in Revision 1. The reviewer is now charged with reassessing the authors' effort to achieve this transparency, accuracy, clarity, and balance by determining whether the changes made in the revised manuscript (not just in the Rebuttal) adequately address the issues raised previously by the reviewers.

2. Overall assessment of Revision 2:

a. Problem: In Revision 2 the authors have addressed some but not all of the reviewer's comments by making changes in the manuscript text and/or figures. General Comment 13 in the previous review recommended the authors to focus their Rebuttal on the changes they make in the text and figures to address each comment. The authors responded in their Rebuttal that they "hope that our responses are now satisfactory in this regard". However, again, much of their detailed 38-page response to the reviewers' comments describes the authors' views and explanations for why the requested changes were or were not made. By declining to make some of the recommended changes, the authors not only rejected this constructive feedback for strengthening the manuscript, but also ignored the likelihood that some readers would have the same concerns as the reviewers. And it prolonged the review process.

b. As an example of changes described in the Rebuttal but not in the manuscript, the authors were asked in the initial review to address concerns over whether the reported features of lymphatic junctions resulted from using immersion fixation instead of vascular perfusion fixation. The request was repeated in the second review because the comment was not addressed in Revision 1. Now the authors claim the issue was addressed, but the findings were reported only in the Rebuttal, not in the manuscript. In Revision 2, the issue is addressed in the Methods, but still the findings are not described in the Results or illustrated in the figures (see Specific Comment for lines 1000-1007).

c. Another example is reflected by the newly added data for silver nitrate staining, where the methods and findings are included in the manuscript, but the background and limitations of interpreting silver staining are not considered in the manuscript. Instead, the author argue in the Rebuttal – but not in the manuscript - that the findings are evidence against lobes being junction-free flap valves and in favor of the presence of double junctions (see General Comment 6).

d. Solution: The authors are asked to make changes in the text and figures that address: (1) the problems described below as not adequately resolved in the manuscript, and (2) issues considered in the Rebuttal but not in the manuscript. Parts of the Discussion can be trimmed if length constraints become an issue.

3. Junction classification:

a. Problem: The authors' description of junctions between lymphatic endothelial cells has been problematic from the beginning because their classification was confusing in the context of which findings were confirmatory and which were new and how the new findings fit with the literature. Changes made in Revision 1 and Revision 2 reduced but did not eliminate

these problems.

b. Solution: As described in General Comments 4 and 5, the authors are asked to remedy the problems by: (1) changing the description of curvilinear and double junctions from “discontinuous or continuous” to “segmented or unsegmented” (or similar terms), and explicitly acknowledging that these junctions are, like buttons, discontinuous around endothelial cells, because they would otherwise be called zippers; and (2) using the term “zipper” only for continuous junctions that surround entire endothelial cells, as originally defined in Reference 4.

4. Continuous vs. discontinuous junctions:

a. Problem: The authors’ continued description of some curvilinear and double junctions as “continuous” is baffling. As described in previous reviews, the term “buttons” was introduced to distinguish discontinuous junctions in endothelial cells of initial lymphatics from continuous junctions (“zippers”) around endothelial cells of collecting lymphatics and blood vessels: Reference 4, page 2350: “Endothelial cells of initial lymphatics were joined by discontinuous buttons (Fig. 1C), whereas endothelial cells of collecting lymphatics were joined by continuous zippers (Fig. 1D), similar to those in adjacent blood vessels”.

b. To describe some curvilinear and double junctions as “continuous” is inconsistent with the intended meaning of continuous junctions in endothelial cells, confusing, and in conflict with the authors’ own drawings in Fig. 1g and Fig. 6, showing red lines for curvilinear and double junctions that are discontinuous at the base of lobes.

c. If some curvilinear and double junctions are continuous on a lobe but not around the entire endothelial cell, then some button junctions would also be continuous. Describing discontinuous junctions as “continuous” is not only self-contradictory but is also inconsistent with a commitment to transparency, accuracy, clarity, and balance.

d. If some curvilinear and double junctions are unsegmented, then call them “unsegmented”, not “continuous”, and call the others “segmented” and reserve the term “continuous” for zipper junctions that surround entire endothelial cells, as originally defined in Reference 4.

e. Solution: The authors are asked to fix this problem by using the term “continuous” only for zipper junctions that surround entire endothelial cells and use the term “discontinuous” for all non-zipper junctions, regardless of whether they are called buttons, intermediate junctions, curvilinear junctions, or double junctions.

“Segmented” or similar term can be used for the subset of button, curvilinear, and double junctions that are subdivided or fragmented on a lobe. “Unsegmented” or similar term can be used for the subset of these junctions that are not subdivided on a lobe.

The solution will require four changes: (1) correcting the text (lines 99, 123-124, 448-449, 1057-1060, Supplement lines 81-83, etc.); (2) relabeling Fig. 1g to distinguish discontinuous junctions (all but zippers) from continuous junctions (zippers); (3) reorganizing and relabeling the 8-part panel on the left side of Fig. 6 to distinguish discontinuous junctions (all but zippers) from continuous junctions (zippers); and (4) redefining and distinguishing button junctions, curvilinear junctions, and double junctions from one another in the main text (lines 120-125), Methods (lines 1057-1060), Supplemental Methods (lines 81-83), and legends for Fig. 1g (lines 743-750) and Fig. 6 (lines 873-881).

5. Zipper-like junctions:

a. Problem: As just described, “zipper” was introduced to describe continuous junctions that surround entire endothelial cells typical of collecting lymphatics AND blood vessels, neither of which has an oak leaf shape or lobes. Use of the term “zipper” for a junction on an individual lobe is inconsistent with the definition and usage in the literature, does not make sense, and confuses the authors’ message.

b. To remedy this issue, the authors were asked in the previous review to add data that express junctions per endothelial cell to distinguish endothelial cells with continuous junctions (zippers) around their entire perimeter from those with discontinuous junctions, regardless of what they are called. Instead, the authors’ rejected this recommendation and continued to use their redefined term “zipper” to apply to an individual lobe (Figs. 1g, 1h, and 6, Supplement lines 75-90).

c. As a result, (1) endothelial cells with curvilinear or double junctions designated “continuous” are not distinguished from endothelial cells with continuous junctions around the entire cell, (2) readers are left to ponder the authors’ distinction between “zippers”, “continuous” curvilinear junctions, and “continuous” double junctions on lobes, and (3) readers must also puzzle over whether endothelial cells at the entry region of lymphatics have junctions that are discontinuous or continuous in the conventional sense, because the data apply to individual lobes instead of entire endothelial cells.

d. Solution: The authors are asked again to fix these problems by using the term “zipper” only for junctions that surround entire endothelial cells. This will require correcting the text (lines 118-120, 268-269, Supplement lines 83-84, etc.) and labels in Figs. 1g, 1h, 4h, and 6 to reflect the definition of zippers as junctions around entire endothelial cells and eliminating the use of “zippers” for junctions that are not continuous around the entire cell.

Description of the presence or absence of LYVE1 staining is informative, but LYVE1 staining should not be a criterion for the identification of zippers, which are simply defined as junctions that surround entire endothelial cells. LYVE1 staining and zipper junction designation are independent criteria and should be treated as such.

Values for zippers in graphs shown in Figs.1g and 1h should be adjusted to reflect the historical meaning of zippers.

No changes are needed where “zippers” or “zipper-like” or “zippering” is used correctly for developing lymphatics, collecting lymphatics, and blood vessels (lines 63, 66, 68, 69, 81, 96, 106, 150, 432, 433, 440-443, etc.).

6. Silver nitrate staining:

a. Problem: The authors’ addition of new data for silver nitrate staining (lines 114-117, 126, 741-743, 1015-1027) to support their interpretation of junctions on lobes is puzzling. Why add data that raise more questions instead of using the space to strengthen existing data (e.g., expansion of data from staining for claudin-5 and other tight junction proteins) and discuss in more detail the strengths and limitations of assumptions and interpretations?

b. The authors confirm that silver nitrate staining outlines the approximate location of the border of lymphatic endothelial cells. However, silver nitrate also stains multiple other regions and does NOT faithfully mark intercellular junctions. As is well-documented in the literature, silver nitrate staining not only occurs in “intercellular cement” (yet to be identified) between overlapping endothelial cells but also in the basement membrane at gaps between endothelial cells where no junctions are present and some other regions (Majno et al. *Virchows Arch A* 408: 75-91, 1985; McDonald *Am J Physiol* 266:L61-83, 1994; Reference 16). The authors’ colocalization data illustrate this property by showing that silver staining coincides with PECAM (Fig. 1f, Extended Data Fig. 1b). PECAM, which is an adhesion molecule not a junctional protein, does not colocalize with VE-cadherin or claudin-5 at buttons, and deletion of PECAM does not change button structure (Reference 4).

c. Silver nitrate staining at the borders of lymphatic endothelial cell lobes cannot be equated to junctions and does not add unambiguous support for the authors’ view of junctions. In the absence of balanced consideration of what is known and not known about silver staining of endothelial cells accompanied by realistic interpretations of the observed staining in lymphatics, the authors’ new data are misleading and do not strengthen their story.

d. These new data in the Results are an example of the problem described in General Comment 2, where the authors do not explain in the Introduction their rationale for adding the findings and do not consider possible interpretations in the Discussion, leaving readers in the dark over the reason for adding silver nitrate staining. Instead, this is discussed in the Rebuttal seen only by the reviewers.

e. The authors’ prioritizing the addition of these new data to the manuscript also argues against their claims in their Rebuttal that space constraints prevented or abbreviated the changes they made in the manuscript in response to some reviewers’ comments.

f. Solution: The authors are asked to fix this problem by adding to the Introduction, Results, and Discussion the missing information that describes the (1) background and rationale for adding silver nitrate staining, (2) complex nature of silver staining that is not specific for tight junctions or adherens junctions, and (3) assumptions underlying their interpretations of the findings. If space constraints become an issue, the authors could remove these new data and methods to give room for changes made in response to the reviewers’ comments.

7. Future responses to reviewers’ comments:

a. Problem: As described in General Comment 2a, the authors have encumbered the review process by not simply making changes in the manuscript to address issues recommended by reviewers to improve the accuracy, clarity, and rigor of their story, but instead devoting unnecessarily large amounts of the Rebuttal to explaining their reasons for making or not making changes in the manuscript.

b. Solution: If given the opportunity to revise their manuscript, the authors are asked to streamline their Rebuttal by simply (1) confirming the changes they made in response to the reviewer’s requests to correct inaccuracies and issues that could confuse readers, and (2) specifying the manuscript line numbers and figure panel numbers where the changes can be found.

c. As the changes should speak for themselves, it is not necessary to explain them in the Rebuttal or to repeat the revised text or the authors’ views described in the manuscript.

Specific Comments

1. Lines 111-112: “we focused solely on blunt-ended LYVE1+ vessels in all analyzed age groups...”

Problem: As the authors prefer to use “blunt-ended” instead of “initial lymphatics”, readers should be reminded that “blunt-ended” lymphatics are functionally the beginning – not the end – of lymphatics.

Solution: Please change to: ...“we focused on the entry region of initial lymphatics by examining only the blunt origin of LYVE1+ vessels, sometimes referred to as “blunt-ended” lymphatics, in all analyzed age groups...”

2. Lines 129-130: Text describing lymphatics in the trachea and diaphragm (Extended Data Fig.1e).

Problem: The examples of lymphatic junctions in the trachea and diaphragm shown in Extended Data Fig.1e are not representative of those in the literature.

Solution: These examples should be: (1) described as selected to illustrate curvilinear and double junctions, and (2) acknowledged as not intended to be representative of initial lymphatics in these organs. Also see Comment for lines 597-599.

3. Lines 149-150: "LYVE1-low sprouts located at the distal tip of lymphatic capillaries also showed elongated shape without lobes and the associated zipper junctions (Extended Data Fig. 2a)."

Problem: As written, the phrase "without lobes and the associated zipper junctions" could be misinterpreted as not having zipper junctions.

Solution: Please change to: "LYVE1-negative LEC of sprouts at the tip of lymphatic capillaries were elongated, lacked lobes, and had zipper junctions (Extended Data Fig. 2a)."

4. Lines 161-167: Parallel lines of LYVE1 surface staining shown in Fig. 2h of lymphatics fixed and stained without permeabilization.

Problem: It is unclear how the LYVE1 antibody could access the interior edge of overlapping LEC flaps to create the second line of LYVE1 staining in the absence of permeabilization.

Solution: The authors should explain in the Results or Discussion their interpretation of how the antibody gained access to the interior edge - as well as the exterior edge - of LEC flaps to create two parallel lines of LYVE1 staining without permeabilization in the surface staining experiments.

5. Lines 424-427: "This dynamic remodelling is required for maintaining the cellular overlaps that define the lobate cell shape, which in turn ensures integrity of the LEC monolayer under strain that is imposed by interstitial fluid pressure alterations."

Problem: This statement is the authors' interpretation, not an established fact.

Solution: Please rewrite this statement as an interpretation, e.g., "We interpret our evidence as indicating that the remodeling is required for maintaining the cellular overlaps..."

6. Line 429: "Lymphatic vessels collect interstitial fluid from tissues back into the bloodstream..."

Problem: "...collect fluid...back to bloodstream..." does not make sense.

Solution: Please change to: "Lymphatic vessels collect and transport interstitial fluid from tissues back to the bloodstream..."

7. Lines 452-453: "...intermediate junctions were described at developmental stages and were previously considered a transient state undergoing zipper-to-button transformation (Reference 6)."

Problem: This statement is inaccurate because it omits published descriptions of intermediate junctions under normal conditions, after infection, and with treatment.

Solution: Please replace with: "...intermediate junctions have been considered a stable intermediate between buttons and zippers in normal mice or a transient or transforming state during development, after genetic manipulation, or with untreated or treated infection when the junctions undergo button-to-zipper or zipper-to-button transformation (References 6, 8, 9)."

8. Lines 453-455: "...variations in the criteria..."

Problem: This statement is inaccurate because it omits relevant published data.

Solution: Please change to: "Button junctions have been reported to constitute about 50% of junctions in initial lymphatics in mouse ear skin (Reference 8), but variations in the criteria..."

9. Lines 484-485: "...CLDN5, the second major component of LEC junctions..."

Problem: This statement is misleading.

Solution: Please change to: "...claudin-5, a tight junction protein in LEC..."

10. Lines 487-488: "multiple cell-cell and cell-matrix adhesion receptors, such as the tight junction proteins ESAM or JAM-A present in capillary LEC junctions..."

Problem: "cell-matrix adhesion receptors, such as the tight junction proteins" is not correct.

Solution: Please change to: "multiple tight junction proteins, such as ESAM or JAM-A, and cell-matrix adhesion receptors, such as β 1 integrins, in capillary LEC..."

11. Lines 494-496: "Leak's studies also indicate that while adherens junctions are prevalent, ultrastructurally defined tight junctions are rarely observed in LECs (References 21,37)."

Problem: This sentence does not accurately represent Leak's findings reported in References 21 and 37, where no measurements were made of the relative abundance of tight junctions (maculae occludentes) and adherens junctions (called desmosomes in Reference 37). En block staining with uranyl acetate, which is required to convincingly distinguish the two types of junction, was used only in two figures (Fig.11 and 11a, Reference 21). There is no question of the historical value of Leak's beautiful work, but using his non-quantitative TEM images as a reference for comparison to junction frequencies in contemporary immunofluorescence imaging is unjustified and misleading.

Solution: Please replace this sentence with the following: "Leak's ultrastructural studies revealed both adherens junctions (maculae adherentes) and tight junctions (maculae occludentes) in overlapping regions of adjacent LECs (References 21,37)."

12. Lines 496-498: "It is therefore unclear why immunofluorescence analysis revealed that the classical tight junction molecule CLDN5 co-localizes with VE-cadherin at the majority of LEC junctions, as reported previously (References 4,6), including in curvilinear and double junctions."

Problem: This statement is not a valid or meaningful interpretation of coincident VE-cadherin and claudin-5 immunofluorescence in LEC. The obvious and most likely interpretation is that tight junctions and adherens junctions are both present in regions of coincident fluorescence. However, coincident immunofluorescence does not mean the junctions would be coincident when viewed at higher resolution by TEM, where they would be seen as adjacent.

Solution: Please replace this sentence with the following: "Consistent with Leak's findings, our immunofluorescence analysis revealed colocalization of VE-cadherin (adherens junctions) and claudin-5 (tight junctions) in overlapping regions of LECs, as reported previously (References 4,6). However, because confocal microscopy does not have the resolution of TEM, colocalization of immunofluorescence is described with the understanding that the two types of junction are adjacent, not superimposed."

13. Lines 563-566: Concept of tightening of the endothelial barrier to prevent fluid from re-entering the interstitium.

Problem: The mechanism described here resembles the primary/secondary lymphatic valve concept reported more than 20 years ago by Geert Schmid-Schönbein and colleagues.

Solution: Schmid-Schönbein's historically important concept should be acknowledged here as a precedent for the authors' proposal by citing: Trzewik J et al. Evidence for a second valve system in lymphatics: endothelial microvalves. *FASEB J* 15: 1711-7, 2001; Mendoza E, Schmid-Schönbein GW. A model for mechanics of primary lymphatic valves. *J Biomech Eng* 125: 407-4148, 2003.

14. Lines 568-570: "...limited to a few electron microscopic studies (Reference 43)..."

Problem: This statement is inaccurate.

Solution: Please acknowledge at least one of the publications on the fibrillin composition of anchoring filaments, e.g., Gerli R, Solito R, Weber E, Aglianó M. Specific adhesion molecules bind anchoring filaments and endothelial cells in human skin initial lymphatics. *Lymphology*. 2000 Dec;33(4):148-57.

15. Lines 597-599: "Although we found the presence of curvilinear and double junctions in lymphatic capillaries also in the diaphragm and trachea in adult mice, our analysis is not sufficient to generalize our findings from the skin to different organs."

Problem: The images illustrating lymphatic junctions in the diagram and trachea do not justify this statement.

Solution: Please change to: "Although we obtained evidence of curvilinear and double junctions in lymphatic capillaries in the diaphragm and trachea of adult mice, the proportions were not measured and our analysis was insufficient to determine whether our findings from skin apply to other organs."

16. Line 602: "...large molecules such as chylomicrons..."

Problem: This statement is inaccurate because chylomicrons are multi-component lipoprotein protein particles, not large molecules.

Solution: (1) Please correct this error, and (2) correct the misleading implication that there is evidence documenting that lacteals permit the entry of molecules or particles larger than can enter other initial lymphatics.

17. Lines 728-881: Please correct the typos in the figure legends.

18. Line 879: "bellow-like" should be "bellows-like".

19. Lines 1000-1007: Comparison in the Methods of lymphatic junctions after perfusion fixation versus immersion fixation.

Problem: The three words (“...yielding similar results”) in the Methods are not convincing evidence to justify the statement that LEC junction morphology was identical after immersion fixation and perfusion fixation.

Solution: Please correct this problem by (1) describing in the Results (lines 90-140) the findings obtained in the comparison that support the authors’ claim on line 1001 (“yielding similar results”); (2) identifying the figure panels that enable readers to review evidence underlying the claim; and (3) adding “fixation by immersion” or “fixation by vascular perfusion” to the legend of each example figure used for the comparison.

20. Lines 1196-1206: Extended Data Fig.1 legend.

Problem: The legend for Extended Data Fig.1 does not match the figure.

Solution: Please (1) correct this figure legend, and (2) check all other figure legends to ensure the numbering and panel descriptions match the figures.

21. Problematic sequence of some figure panels: Please make the sequence easier for readers to understand in Fig. 1a, c, d, b, f, g, h and other figures that lack of the conventional a, b, c, d, left to right, top to bottom sequence of figure panels.

22. Figure numbers: Please add numbers to all figures to facilitate future reviews.

Referee #2

(Remarks to the Author)

Schoofs et al. resubmitted a revised manuscript for publication in Nature. This referee had asked to put *cdc42* into the context of what has been published on LEC and stretch-dependent mechanisms (already in her/his first review). The authors now argue that they expect multiple pathways to be upstream of *cdc42*, but without wishing to reveal one of them or even part of a pathway involved. They also argue that quantification of LEC junctional overlaps in *Itgb1* deficient mice requires crossing R26-iMb2 mice with *Itgb1* KO mice, which they argue requires too much time. The referee sees this point, even though a more thorough investigation of a well-characterized pathway upstream of *cdc42* would have been useful to put the data into any kind of molecular context (in particular given that the authors wish to publish in Nature). Further, the authors argue that injecting fluid into the ears of their *Itgb1* KO mice is beyond the scope of this manuscript, which irritates this referee even more, given that fluid injections into mouse ears take a day or two. Therefore, the role of integrin-beta1 as a key upstream receptor under experimental conditions that the authors describe and use in their manuscript (since the first submission) has not been investigated. Again, the authors seem to not wish to explore this pathway or any pathway upstream of *cdc42*, even with tools that they have used in this manuscript. The authors also do not wish to investigate *Rasip1* and argue that it only plays a role in the embryo, but not adult, which is too strong a statement given that Liu et al. deleted *Rasip1* only postnatally and analyzed the neonatal mice within a week after birth, so that according to this referee a role cannot be excluded for the situation that Schoofs et al. investigate: the ears of 9-weeks-old adult mice after fluid challenge. The authors were also asked to exclude the possibility that *Prox1*-Cre-driven deletion of *cdc42* affects cardiac tissue and thereby (as a secondary effect of a starting heart failure) affects the lymphatics, since it is well-known that fluid retention and lymphatic congestion can develop after cardiac issues (when the heart muscle is affected that expresses *Prox1*). The authors state that they did not observe any major vascular defects, making this referee wonder (since no data are shown) what that actually means. Did the authors analyze heart pump function in their mice to exclude this issue that might entirely explain their *cdc42*-lymphatic phenotype? Finally, when looking at the raw data sheets, some of these do not align with the respective Figure panels and are not well-explained or sufficiently transparent. For example, in Figure 2j, the authors show the average width of collector vessels, but the last data in the excel table show as Area/length three values: 2.490, 1.246 and 1.269. This referee fails to see these values or their average in Figure 2j – what has happened? Were they or was their average taken out as outlier? In Ext. Data Figure 9c, the authors show 14 dots for stretch control (no-str.), but 23 values are shown in the corresponding excel sheet. So, where are the values? Same Figure: the authors show as “No stretch 0.1 mAb13” 8 or 9 dots in Ext. Data Figure 9c, whereas the corresponding excel sheet shows 13 values.

Version 3:

Reviewer comments:

Referee #1

(Remarks to the Author)

General Comments

1. The authors were very responsive and have adequately addressed the comments on the previous version of their manuscript. The remaining issues in the revised manuscript are minor.

2. The edits requested under Specific Comments are to correct minor errors and potentially misleading text and to improve clarity.

3. The authors are encouraged to make the title shorter and more eye-catching.

Specific Comments

1. Title: Current title "Resilience of lymphatic endothelium through isotropic stretch-induced cytoskeletal regulation of puzzle cell shape" describes in 13 words some of the findings but seems unnecessarily long, complex, and detailed to be sufficiently eye-catching to attract a broad readership.

Suggested alternatives are:

- Puzzle cell shape reinforces lymphatic vessel fluid uptake
- Puzzle cell shape increases resilience of initial lymphatics
- Lymphatic endothelial cell puzzle shape facilitates fluid uptake
- Lymphatic vessel resilience increased by puzzle cell shape
- Lymphatic vessel fluid uptake facilitated by puzzle cell shape
- Lymphatic endothelial cell puzzle shape increases resilience for fluid uptake

2. Line 38: "...spectrum of VE-cadherin-based junctional configurations at the lobular intercellular interface..."

•Needing attention: The discontinuous nature of these diverse junctions should be described in the Abstract.

•Recommended change: "...spectrum of shapes of discontinuous VE-cadherin-containing junctions at cell-cell interfaces..."

3. Line 58: "...intermittent junction-free regions..."

•Needing attention: "intermittent" is confusing (temporal versus spatial?) and unnecessary.

•Recommended change: Delete "intermittent".

4. Line 83: "...new features of LEC cytoskeleton and their dynamic overlaps..."

•Needing attention: Check grammar. In this phrase, "their" refers to "cytoskeleton", whereas it should refer to LEC.

•Recommended change: "...new features of the LEC cytoskeleton and dynamic properties of the overlapping cell borders..."

5. Line 100: "The classical tight junction protein Claudin 5 (CLDN5) was present in both buttons and in most, but not all, linear VE-cadherin+ adherens junctions (Fig. 1b)."

•Needing attention: This sentence is misleading and confusing. Claudin-5 is not "the classical" tight junction protein and is not present in adherens junctions. Occludin was the first tight junction protein identified in 1993 and has a broader distribution among cell types than claudin-5. Claudin-5, first described in 1999, is typically associated with endothelial cell tight junctions.

•Recommended change: "Staining for the endothelial cell tight junction protein claudin 5 (Cldn5) coincided with VE-cadherin staining of most buttons and other junctions in these LEC (Fig. 1b)."

6. Lines 102-103: "...a significant proportion of lymphatic capillary ends displayed sprouting at three weeks of age (Fig. 1c, d)..."

•Needing attention: Significant in comparison to what? No statistics are shown in Fig. 1d.

•Recommended change: Delete "significant" or change to a percent.

7. Lines 113-114: "Similar diversity was observed upon histological staining using silver nitrate..."

•Needing attention: This description of "similar diversity" is inconsistent with the images of silver staining in Fig. 1f and ED Fig. 1b, which show granular but largely continuous single or double silver lines at cell borders, unlike the heterogeneous discontinuous VE-cadherin staining shown in Fig. 1g and ED Fig. 1d.

•Recommended change: "Silver nitrate staining formed largely continuous single or double granular lines at the border of initial lymphatic LECs, unlike the heterogeneous, discontinuous VE-cadherin staining."

8. Lines 116-117: "Silver precipitation occurs due to an unspecified junctional component at the EC intercellular interface."

•Needing attention: The word "junctional" is misleading because it overstates what is known about the location of silver staining in the context of adherens junctions and tight junctions.

•Recommended change: "Silver nitrate marks endothelial cell borders by staining unknown components of the intercellular interface".

9. Lines 124-125: "This revealed the lack of zipper junctions, defined as linear LYVE1- junctions that surround the entire cell (Fig. 1g)."

•Needing attention: This statement is misleading because VE-cadherin is not mentioned and therefore is inconsistent with the definition on Line 94: "...collecting vessels exhibited continuous VE-cadherin+ zipper junctions with no LYVE1 expression". Another issue is that the authors should explicitly state that the absence of zipper junctions in these vessels means that the junctions were discontinuous, albeit variable in morphology. A further issue is that the figure cited (Fig. 1g) does not show zipper junctions. However, zipper junctions are shown in Fig. 1a.

•Recommended changes: "This initial region of lymphatics lacked zipper junctions, defined as continuous VE-cadherin+ junctions around entire LECs (Fig. 1a). Initial lymphatics had discontinuous junctions with variable morphologies on individual lobes and were LYVE1+, unlike collecting lymphatics that had zipper junctions and lacked LYVE1 staining."

10. Lines 127-128: "...no significant increase in their frequency in older mice (Fig. 1g)."

•Needing attention: Significant in comparison to what? No statistics are shown in Fig. 1g.

•Recommended change: Delete "significant" or change to a percent.

11. Lines 130-131: "Curvilinear junctions that were LYVE1- were rare observed (Fig. 1g)."

•Needing attention: Incorrect grammar.

•Recommended change: "Few curvilinear junctions lacked LYVE1 staining (Fig. 1g)."

12. Lines 137-138: "...Analysis of lymphatic vessels after 10 minutes revealed no significant changes in the relative frequency of junction types (Fig. 1h)."

•Needing attention: Significant in comparison to what? No statistics are shown in Fig. 1h.

•Recommended change: Delete "significant" or add results of statistical tests.

13. Lines 173-174: "...which become inaccessible for staining after chemical crosslinking despite the antibody accessing the lumen at sites where the tissue was disrupted during preparation."

•Needing attention: Rewording this sentence is recommended to reflect the speculative nature of the explanation of the staining of both lines.

•Recommended change: "...which become inaccessible for staining after chemical crosslinking. Staining of the intraluminal border presumably occurs where silver nitrate accesses the lumen through endothelial disruptions created during tissue processing."

14. Lines 194-198: "remodelling"

•Needing attention: The use of the term "remodelling" for shape changes that occur over weeks or months (Lines 194-198) AND for shape changes that occur in minutes or hours (Lines 202-204) is confusing in the absence of clear descriptions of the different meanings of remodelling in these contexts.

•Recommended change: Please address this comment along with the comment for Lines 202-204.

15. Lines 202-204: "...revealed continuous remodelling of cell cell borders". This comment also applies to: Line 477: "Intravital imaging further revealed continuous remodelling of LEC overlaps...", Line 574: "Continuous actin-driven remodelling of cellular overlaps...", legend for Supplementary Movie 1: "...showing remodelling of cell-cell borders", and other similar statements in the manuscript.

•Needing attention: The broad range of time-courses of "continuous" or "dynamic" remodelling reported throughout the manuscript and in figure legends as shape changes that occur in minutes to hours or over weeks to months is not adequately described or discussed. Shape changes occurring in minutes (Supplementary Movie 1) clearly differ mechanistically and functionally from the shape changes occurring over weeks or months (Fig. 2k). Another concern is that the changes shown in Movie 1 are subtle and occur in only focal regions of cells. The absence of the authors' definition of the remodelling observed over minutes raises the question of whether "remodelling" in Supplementary Movie 1 refers to the tiny transient filopodial projections that rapidly form and retract on some lobes. Readers should be told the authors' intended meanings of remodelling, where to look for remodelling in images and movies, and what they should expect to see.

•Recommended change: Please describe the types of remodelling in the Results and give more detailed descriptions of

specific examples of remodelling in the corresponding figure legends, e.g., Supplementary Movie 1 (minutes to hours), Fig. 2k (weeks to months), to explain what is meant by “remodelling” in each case the term is used in the manuscript.

16. Lines 220: “terminal capillary”; line 1190: “terminal capillaries”

- Needing attention: These terms, which are used only in these two locations, are confusing because the authors apparently mean the beginning of initial lymphatics rather than the end (terminal) of lymphatic capillaries where they join pre-collector or collecting lymphatics. The term “terminal capillary” was apparently adopted from studies of blood capillaries. The term “initial lymphatics” helps readers think functionally about these vessels, where flow is away from the tip, and reinforces the difference from blood capillaries, where flow is toward the tip of sprouts.

- Recommended change: “initial lymphatics” or “initial region of lymphatic capillaries”.

17. Line 274: “...and loss of a uniform lobate shape (Fig. 4e, Extended Data Fig. 4a).”

- Needing attention: This is misleading. The cell shape change shown in these figures does not include a loss of lobate shape or conversion to the smooth contour of LEC in collecting venules. Instead, lobes are still present but the cell contour is more irregular.

- Recommended change: “...and change from the uniform oak leaf shape of normal LECs in initial lymphatics to more irregular shapes (Fig. 4e, Extended Data Fig. 4a). Importantly, the cells do not convert into the smooth shape of LEC in collecting venules.”

18. Lines 340-341: “...the cells displayed significant bulging, leading to almost complete closure of the lumen (Extended Data Fig. 7a).”

- Needing attention: Significant in comparison to what? No statistics are shown in ED Fig. 7a.

- Recommended change: Delete “significant” or add results of statistical tests.

19. Line 439: “...interstitial fluid pressure alterations (Supplementary Fig. 8).”

- Needing attention: Supplemental Fig. 8 appears to be identical to Fig. 5m.

- Recommended change: Move Supplementary Fig. 8 to become new main Fig. 6, delete Fig. 5m, and change line 439 to “... interstitial fluid pressure alterations (Fig. 6).”

20. Lines 457-458: “...~20% of cellular lobes exhibited classical button junctions, with no significant increase in their frequency in older mice to suggest developmental maturation.”

- Needing attention: Does this mean no significant age-related changes were found in button junctions? If so, add the results of statistical tests to Fig. 1g. If not, delete “significant”.

- Recommended change: Delete “significant” or add results of statistical tests.

21. Lines 467-470: “Button junctions have been reported to constitute about 50% of junctions in initial lymphatics...”

- Needing attention: Reference citations are missing.

- Recommended change: Add references that support this statement.

22. Line 545: “...exhibit significant cellular overlaps...”

- Needing attention: Where is the statistical significance reported?

- Recommended change: Delete “significant” or replace with a non-statistical term.

23. Line 552: “...the absence of significant flow-induced mechanical forces...”

- Needing attention: Where is the statistical significance reported?

- Recommended change: Delete “significant” or replace with a non-statistical term.

24. Lines 757-758: Fig. 1f legend: “...including the lobe tips (arrow) with discontinuities (arrowhead).”

- Needing attention: Please add that the tiny discontinuities in the granular silver staining are smaller than the discontinuities in VE-cadherin staining of initial lymphatic LECs. The legend should acknowledge that silver staining follows cell borders but does not faithfully match VE-cadherin staining in the images shown.

•Recommended change: "...including the lobe tips (arrow) with small discontinuities (arrowhead) accompanying the granular silver staining. These discontinuities are smaller than the discontinuities in VE-cadherin staining in Fig. 1a,b,g, and the silver staining pattern is largely continuous, unlike the staining of VE-cadherin at adherens junctions."

25. Line 797 Fig. 3 legend: "log2 fold change".

•Needing attention: The comparison of capillary and collecting regions of lymphatics showed a difference in the two regions, not a change.

•Recommended change: "log2-fold difference".

26. Line 852 Fig. 5 legend: "(h-j) Quantification of junction linearity (n=38, 43 images; one experiment)..."

•Needing attention: The description in the Supplementary Methods (page 9), where normalized values are expressed as %, does not fit with normalized linearity values ranging from 1.0-2.0 in Fig.5h or with the statement (Line 856) "Junction linearity was normalized to the average of controls."

•Recommended change: Confirm the location in the manuscript where the method used to calculate linearity is explained (Supplementary Methods?). Also, briefly describe the meaning of "junction linearity" in Fig. 5h legend and add the units to the Y-axis label in Fig. 5h.

27. Lines 864-865 Fig. 5m and Supplementary Fig. 8

•Needing attention: These figures appear to be identical.

•Recommended change: Move Supplementary Fig. 8 to become new main Fig. 6 and delete Fig. 5m.

28. Line 881: "4-hydroxytamoxifen"

•Needing attention: Please unify the abbreviations used for this agent throughout the text and figures.

•Recommended change: Use "tamoxifen" throughout and delete "Tam" and "4-OHT" in the text and figures.

29. Lines 919-920: "Details of the MultiStretcher device will be presented elsewhere (Linsenmeier et al. 2024, in preparation)."

•Needing attention: Check the journal's requirements regarding citation of manuscripts "in preparation".

•Recommended change: If such citations are unacceptable, add the essential details of the device to the Methods or Supplementary Methods.

30. Fig. 5m: This figure appears to be the same as Supplementary Fig. 8.

•Needing attention: Duplication of Fig. 5m and Supplemental Fig. 8

•Recommended change: Delete Fig. 5m and make Supplementary Fig. 8 the new main Fig. 6. See comments for Line 439 and Lines 864-865 regarding Fig. 5m and Supplementary Fig. 8.

31. Extended Data Fig. 3a, 3b

•Needing attention: Additional labels would be helpful.

•Recommended change: Please add labels to these figures to make it easier for readers to understand that 3a shows oak leaf endothelial cells of initial lymphatics, whereas 3b shows endothelial cells of collecting lymphatics. This distinction is evident in the legend but not in the figure.

32. Extended Data Figs. 9 and 10:

•Needing attention: Additional labels would be helpful.

•Recommended change: Please add labels to these figures to make it easier for readers to understand that Figure 9 shows cultured LEC and Figure 10 shows cultured HUVEC. This distinction is evident in the legends but not in the figures. There is plenty of room in the figures to add these labels. Also, in the legends, please comment on the width of VE-cadherin staining at junctions, where VE-cadherin forms a narrow line in LECs regardless of the width of overlap, but VE-cadherin staining is as wide as the overlap in HUVEC under unstretched and stretched conditions.

33. Supplementary Fig. 8 legend: "...a spectrum of junctional configurations organized as discontinuous buttons, as well as segmented or unsegmented double junctions and curvilinear junctions."

•Needing attention: Please correct the unintended but misleading implication that double and curvilinear junctions are not discontinuous.

•Recommended change: "...a spectrum of discontinuous junctions, including buttons, segmented or unsegmented double junctions, and curvilinear junctions."

Response to the Reviewers' comments (#2023-08-13841)

We thank the Reviewers for their insightful and constructive comments. We have revised the manuscript accordingly and added a substantial amount of new experimental data. We have additionally reorganized certain sections of the manuscript and discussion, particularly emphasizing the distinction between confirmatory data, novel findings, and proposed implications. Furthermore, we have addressed discrepancies between our findings and those of previous studies regarding LEC junctions in dermal lymphatic vessels, providing potential explanations, and discussed the limitations to our work. We provide a separate Word document with changes referred to in this letter highlighted in red.

All specific points are addressed below. In brief, the key new data included in the revised figures are as follows:

- Extended analysis of LEC junctions and definition of new categorization (completely revised **Fig. 1**) – Our new definition, based on a substantial amount of new data, categorize capillary LEC junctions into four distinct types: buttons, zippers, curvilinear junctions and double junctions, of which we find the last two categories to predominate. The 20-30% frequency of buttons we observed in dermal lymphatic capillaries is consistent with the original description of buttons by Baluk et al¹ in the terminal ends of adult tracheal intial lymphatics/lymphatic capillaries, where they were observed with a frequency of 25-30%, but contrasts the subsequently reported predominance of buttons in dermal capillary LECs²⁻⁶. This discrepancy and plausible explanations are discussed below and in the manuscript. We acknowledge that terms 'intermediate' and 'transforming' junctions used in some studies likely correspond to the curvilinear and double junctions we define here. However, these terms lack precision and have been used to imply a transition between zippers and buttons, with the latter considered a mature phenotype of functional lymphatic capillaries in the current literature. While this may initially seem like a semantic distinction, we believe it holds important biological significance. Our study, with its more precise definition of junctions, instead suggests that a 'continuum of discontinuity' is the primary feature of capillary LEC junctions, driven by the dynamic nature of their cellular overlaps and junctions within. This, along with its implications, is discussed in depth below in our point-by-point answers.
- Characterisation of LYVE1 localization in LEC overlaps (**Fig. 2g, h**)
- Extended analysis of the effect of *Cdc42* deletion on LEC shape and cellular overlaps using *Imb2-Mosaic* mice (**Fig. 4e-g, Extended Data Fig. 4a**), as well as the consequences to lymphatic vessel function (**Fig. 4l**)
- Isotropic stretch for an extended period leading to increased curvature of cell-cell borders (**Fig. 5g, h**) – showing, to our knowledge, this hallmark of capillary LEC cell shape *in vivo* for the first time in an *in vitro* setting and providing direct evidence for the role of isotropic stretch as an upstream regulator of the unique lobate cell shape
- Loss of monolayer integrity in stretched LECs upon siRNA-mediated *CDC42* silencing (**Fig. 5j, Extended Data Fig. 8**) – confirming our previous data obtained using a CDC42 inhibitor
- Optogenetic activation of *CDC42* in cultured LECs leading to increase in cellular overlaps and consequent increase in transendothelial barrier strength (**Fig. 5l, m**) – providing direct evidence for CDC42 in regulating the extension of dynamic lamellipodia-like cellular protrusions and monolayer stability in LECs
- Revised schematic figure more accurately describing the current findings and avoiding speculations that are now instead only included in the discussion (**Fig. 6**)

In addition, new key data in supplementary figures are as follows:

- Extended TEM analysis of cellular overlaps (**Extended Data Fig. 2, Supplementary Fig. 1, 2**)
- Integrin $\beta 1$ deletion in LECs *in vivo* (**Extended Data Fig. 6**) and inhibition in stretched LECs *in vitro* (**Extended Data Fig. 9**)
- Isotropic stretching of HUVECs *in vitro* (**Extended Data Fig. 10**)

Referees' comments:

Referee #1 (Remarks to the Author):

General Comments

1. In this clearly written and well-illustrated manuscript, with 6 main figures and 6 supplementary figures, the authors report many novel findings on the presence and functional significance of the overlap of the borders of oak-leaf shaped endothelial cells of initial lymphatics in mouse ear skin. The authors also report similarities between the shape of these endothelial cells and jigsaw puzzle piece-shaped pavement cells of plant epidermis, and conclude that both reflect adaptations to stretch.

2. The novel approaches and findings in the manuscript are benefitted by the contrasting and complementary expertise of Dr. Mäkinen in her studies of lymphatics and Dr. Majda in his studies of shape changes in plant cells.

3. Main: In the introductory section, the authors effectively set out the background of the scalloped shape and distinctive discontinuous (button) junctions in initial lymphatics. They also acknowledge the poor understanding of the control and maintenance of cell shape and button formation and introduce the similarities to plant puzzle cells. However, a more complete description of the specific aims of the project is needed to help readers anticipate the diverse findings in the studies presented. This could easily be achieved by adding a sentence similar to, "The specific aims of the project were to determine ____, ____, ____, and ____" (integrate with lines 72-77).

Response: We have added a sentence describing the aim of the study (lines 83-87).

4. Results- General: The findings address two distinctive features of endothelial cells of initial lymphatics: (a) the cells have an oak-leaf shape with overlapping scalloped borders, and (b) the cells are interconnected by button and zipper junctions. These topics are considered separately in the following comments.

5. Results - Oak-leaf shape of lymphatic endothelial cells

a. Mosaic mice: Multicolor labeling of lymphatic endothelial cells in mosaic mice effectively revealed the unusual cell shape and overlap of adjacent cells. This novel feature of the manuscript is documented by striking images of the shape and distribution of these cells in ear skin lymphatics. In regions of lymphatics where adjacent cells have different colors, the extent of overlap of the scalloped borders is beautifully revealed. Use of mosaic mice also revealed differences in appearance of the cells between E17 (back skin) and 3, 5, 9, and 25 weeks postnatally (ear skin).

b. Pentraxin: To better understand the results of the scRNA-seq studies (lines 118-121), more detailed justification - including references - is needed for using pentraxin-3 (Ptx3) expression to distinguish endothelial cells of initial lymphatics from those of collecting lymphatics. Please also comment on whether a gradient in Ptx3 expression was found in initial lymphatics from tip to collecting lymphatic (see General Comment 6: Problem 7).

Response: We identified a Ptx3⁺ subtype of LYVE1⁺ LECs by single-cell RNA sequencing of ECs isolated from mouse ear skin⁷. In the same study we also showed by immunofluorescence staining the presence of PTX3⁺ LECs at the capillary ‘terminals’. As PTX3 is secreted and deposited around the vessels, determining a potential gradient in its expression is not straightforward. Therefore, we hesitate to strictly assign Ptx3⁺ LECs as LECs restricted to the very terminal end of the capillaries, but have included clarification for this population and reference to the publication in the text (line 209-210). Notably, the expression levels of actin regulators are very similar between the two populations of capillary LECs, whether Ptx3⁺ or Ptx3⁻ (Fig. 3a (previously Fig. 2a)), suggesting that this signature is a common feature of all capillary LECs, potentially associated with their lobate shape.

c. Cytoskeleton: Another novel feature is the demonstration of contrasting distributions of microtubules and F-actin in oak-leaf lymphatic endothelial cells. Although F-actin is clearly shown to be greater at convex regions of some scalloped borders, the association of microtubules with concave regions is less convincing. Many microtubules in Fig. 2b, c appear to continue beyond the border of the outlined endothelial cell, suggesting that some microtubules are in separate but overlapping cells. Replacement of the images in Fig. 2b-c with images similar to Fig. 3b (top row) is recommended to solve this problem and ensure that the images match the drawing in Fig. 2f and the interpretation (lines 144-146).

Response: We agree with the Reviewer that it is challenging to observe the microtubule endings in the maximum intensity projection image, due to the presence of overlapping signal from neighboring cells. It is essential to note that quantifications that form the basis of our conclusions were done manually from z-stacks, and not from z-projections, to guarantee accurate assignment of microtubule ends. We have included an example of a z-stack with annotated microtubules as **Supplementary Video 2**.

We attempted to improve the visualization by extracting microtubule signal from individual LECs using IMARIS surface mask based on membrane-targeted fluorescence protein expression. However, technical challenges due to insufficient z-resolution prevented excluding microtubules from neighboring cells, especially at overlap regions. As a result, in these images, we frequently observed not only the microtubule network within analyzed cell but also a cluster of disconnected microtubule endings at the lobe tips, originating from microtubule endings at the concave regions of a neighboring cell (Fig. 1 for Reviewers, indicated by red arrow and circles).

Fig. 1 for Reviewers. An attempt to visualize the microtubule network in individual LECs in the *iMb2-Mosaic;Vegfr3-CreER^{T2}* mouse. An IMARIS 3D segmentation mask based on the mosaically expressed Tfp1 channel was applied to a deconvolved confocal microscopy image of alpha tubulin staining to extract microtubule signal. Presence of clusters of disconnected microtubules at the lobe tips, originating from microtubule endings at the concave regions of a neighboring cells are shown by red arrows and circles.

d. In vitro stretch: The cyclic isotropic stretch data for human dermal lymphatic endothelial cells in culture (lines 222-231, Fig. 4a-d) are a provocative addition to the story. The increase in overlap after stretching fits the authors’ hypothesis, but the absence of both oak-leaf shape and discontinuous junctions raises questions about the relevance to initial lymphatics. As the borders of many cultured endothelial cells

overlap, the authors should address the question of specificity by comparing LEC to blood vessel endothelial cells (e.g., HUVEC, BAECs) in confluent monolayer culture with and without isotropic stretch. Greater stretch-induced overlap in LEC than in non-LEC would increase the relevance of these data.

Response: The isotropic stretching assay is an artificial model system that can only mimic certain aspects of the *in vivo* context, as it is the case for every other *in vitro* system. Despite their simplicity, important insight into (lymphatic) endothelial biology have indeed been obtained using e.g., flow devices and matrices of different stiffness^{8,9}, even if the observed result may also be misleading. As an example of the latter, actin stress fiber-mediated pulling of button junctions was proposed to regulate junction opening and fluid transfer based on studies in cultured LECs². As shown in our studies, such actin stress fibers were not observed *in vivo*.

Importantly, our new data now show that an extended 22 h period of cyclic isotropic stretching induces increased curvature of cell-cell borders and formation of focal VE-cadherin concentrations in LEC monolayers in addition to lobular overlaps (**Fig. 5g, h**). Despite the absence of true button-like junctions under these conditions, the phenotype induced by isotropic stretch in LECs closely resembles the capillary LEC attributes, which has not been achieved before in an *in vitro* setting. This novel data provides compelling evidence that, similar to plant puzzle cells, cyclic isotropic stretch induces the lobate cell shape of LECs. A phenomenon that, to our knowledge, has not been previously observed in animal cells.

To test the effect of isotropic stretch on blood EC, we performed the same assay using human umbilical vein ECs (HUVECs). As previously reported¹⁰, HUVECs in confluent monolayers without isotropic stretch exhibit irregular cellular overlaps, formed by a reticular network of VE-cadherin¹¹, and numerous stress fibers, the latter differing from LECs. Interestingly, stretching using the same parameters as applied to LECs led to numerous VE-cadherin⁺ finger-like protrusions in HUVECs, while the total cellular overlap area remained unaltered. We thus conclude that HUVECs also respond to isotropic stress but show a qualitatively different response compared to LECs. These data are now included in **Extended Data Fig. 10a, b**.

e. Intravital imaging: 2-photon microscopic images showing changes in the borders and overlap of endothelial cells of initial lymphatics - but not collecting lymphatics - at baseline and after saline injection are not only novel findings but also a positive addition to the story (lines 281-293, Fig. 5, Supplementary Figs. 5, 6). Although changes in actin fluorescence are clearly visible in Movies 1 and 2, little change is evident in the shape of cell borders during the 7:31-min:sec movies. This feature of the movies should be described and discussed.

Response: Movies showing actin dynamics were recorded at high frame rate over only a short time frame of 7-8 minutes during homeostatic baseline conditions. During this short time, we indeed did not observe visible changes in the cell borders. To extend on these observations, we provide new intravital imaging data showing dynamic changes at cell borders during a longer period of 4 h 20 min (**Supplementary Video 1**). These movies show small yet clearly noticeable remodelling of LEC borders specifically at the lobes. Extensive changes in the lobate structures are evident in a longer time-frame of weeks (**Fig. 2k**), consistent with continuous dynamic changes in cell-cell borders. Notably, certain concave regions of the cells showed no morphological changes as seen in new overlay images provided in **Fig. 2k**, suggesting that these regions are more stable, possibly being stabilized by microtubules.

6. Results- Concerns over claims about buttons and zipper junctions: Multiple concerns are raised by the authors' interpretations and conclusions drawn from data for discontinuous (button) junctions and continuous (zipper) junctions in initial lymphatics that are inconsistent with published data on ear skin and other organs. Despite solid evidence to the contrary, the authors argue that discontinuous junctions are rare (10-15%) in ear skin lymphatics. These inconsistencies appear to stem from problems with characterization and categorization of lymphatic junctions, with rigor of the quantitative data, with regions of the lymphatics analyzed, with reconciling confocal and transmission electron microscopic (TEM) images,

and with conclusions drawn from the data. The validity and significance of the junction data in relation to published work cannot be interpreted meaningfully until these problems are resolved.

Response: We hope that the Reviewer finds our new image and quantification data, responding point-by-point to their questions below, to be compelling. We have significantly strengthened our own data on LEC junctions by incorporating additional quantifications and giving more careful consideration to categorization, with the aim of avoiding misleading nomenclature. Additionally, we provide a summary of previously published data (see **Appendix 1**), which we believe is important for understanding and addressing the inconsistencies.

- **Problem 1:** The authors' claim that 85-90% of junctions in ear skin initial lymphatics are zippers (lines 299-301, 400-403) is inconsistent with data cited in at least 7 publications that document the predominance of button junctions in mouse ear skin lymphatics (Baluk et al., JEM 2007; Zheng et al., Genes Dev 2014; Zhang et al., Science 2018; Hägerling et al., EMBO J 2018; Arasa et al., JEM 2021; Churchill et al., JEM 2022; Jannaway et al., Cell Reports 2023). All of these papers not only show the predominance of buttons but also illustrate the variable appearance of these junctions, both around individual lymphatic endothelial cells and from cell to cell. Despite their structural variability, discontinuous junctions in initial lymphatics (Supplementary Fig. 3b column 3) are conspicuously different from continuous junctions in collecting lymphatics (Supplementary Fig. 2b row 2). The authors' claims to the contrary are also inconsistent with evidence that junction "zippering" (replacement of buttons by zippers) clearly occurs under experimental or pathological conditions (Yao et al. Am J Pathol 2012; Zheng et al., Genes Dev 2014; Zhang et al., Science 2018; Churchill et al., JEM 2022; Jannaway et al., Cell Reports 2023) but could not occur if 85-90% of the junctions were already zippers under normal conditions.

Response: We thank the Reviewer for raising this question, which prompted us to systemically look into the definition of buttons and the image data used as the basis of quantifications in the previous literature. We identify the following three aspects as the main sources of inconsistency:

1. Definition of button junctions. We would like to begin by summarizing in Appendix 1 (redacted) how previous studies have analyzed and defined button junctions. The short conclusion from this summary is that currently, there are no consistent criteria for the classification of buttons. Some studies classify junctions as buttons, zippers, and, in some cases, intermediate based on manual qualitative assessment, while others have defined quantitative cut-offs for the area or length of VE-cadherin⁺ staining. These particular quantitative criteria, however, are not consistent across different studies, or well motivated. For example, Jannaway et al defined buttons as discontinuous VE-cadherin⁺ fragments <4.72 μm, without providing a rationale for this precise but relatively large threshold. For comparison, the average width of a cellular overlap between neighboring LECs is ~2 μm (**Fig. 2j**), and a previous study defined buttons as discontinuous segments of VE-cadherin staining of ~3 μm in length and ~0.5 μm in width¹. Most studies do not include LYVE1 staining and cell shape as criteria. None of the studies provide the original image data used for quantification, and most are based on data points with high spread, presented as a single average value, with some including only a relatively small number of data points, presented as a single average value that does not represent the observed biological variability (see **Appendix 1, Fig. 2 for reviewers**). The notion that there are significant inconsistencies in the definition of buttons among previous publications serves as a background to the presentation and discussion of our data in response to the questions raised by the Reviewer.

Notably, the 20-30% frequency of buttons we observed in dermal lymphatic capillaries is consistent with the original seminal description of buttons in the terminal ends of adult tracheal lymphatic capillaries, where they were observed with a frequency of 25-30%¹, but contrasts the previously reported predominance of buttons in dermal capillary LECs in the ear skin²⁻⁶. This discrepancy and plausible explanations are now discussed in the manuscript (lines 432-439).

-
-
-
-
-
-

:= I F9 Ž @; 9B8 F9857H98

Fig. 3 for Reviewers. Effect of thresholding on the detection of VE-cadherin signal.

3. Influence of the state of the vasculature. As previously reported¹, the lymphangiogenic stage impacts the junctions and cell shape, with LECs in sprouting lymphatic vessels showing continuous junctions accompanied by a more elongated shape. We show in **Fig. 1d** that at three weeks of age approximately 25% of lymphatic capillary ends in the mouse ear skin still show a sprouty morphology, indicating active vascular

growth, which is consistent with significant LEC proliferation (**Fig. 1e**). We therefore separated blunt-ended and 'sprouty' vessels ends, and while images of both types are included in the **Source Data**, only blunt-ended vessel ends were included in the quantification. This is an important consideration when assessing phenotypes; any stimulus or genetic defect leading to active lymphatic vessel sprouting would also lead to junction 'zippering'. Without considering the different parameters – cell shape, LYVE1 expression and junction morphology – these findings suggest that it is problematic to decouple junctional changes from morphogenetic alterations. None of the previous studies provide the original image data, and it is therefore not possible to assess the growth state of the vessels used for quantification.

Remedy: The authors should first correct Problems 3-9 described below. Then, they should describe in the revised Results what is known from previous studies of button and zipper junctions in mouse ear skin lymphatics, before presenting their revised junction data. Differences of their data to those published previously on ear skin and other organs should be considered critically. In the Discussion, the strengths and limitations of the authors' approach should be described and any inconsistencies with prior work discussed and given plausible explanations.

Response: We have addressed problems 3-9 below. We have briefly described the conclusion from the previous work on button junctions in the mouse ear skin, namely that by three weeks of age developmental maturation of junctions into buttons has been described to occur, with the majority of junctions exhibiting this characteristic morphology²⁻⁶. This has been included in the introduction (lines 81-82) and discussion (418-420), before presenting and discussing our own results. We have additionally discussed inconsistencies between our results with prior work, and plausible explanations accordingly (lines 432-439).

- Problem 2: The authors also claim that the oak-leaf phenotype develops before button junctions appear in initial lymphatics of ear skin (lines 83-100). However, their data indicating that buttons constitute only 2% of junctions (Fig. 1e, o) at age 5 weeks are inconsistent with published values of 60% button junctions in ear skin at age 3 weeks (Jannaway et al., Cell Reports 2023). The authors' claim is also inconsistent with published data for initial lymphatics in the trachea and diaphragm, where buttons and oak-leaf shaped endothelial cells appear together around birth and evolve together postnatally (Yao et al., Am J Pathol 2012).

Remedy: The authors should correct Problems 3-9, as for the remedy for Problem 1, and then revisit this issue with the new data.

Response: To facilitate direct comparison of our data to those presented by Jannaway et al, we have now included analysis of 3-week-old mice. Collectively, our analysis of a substantial number of junctions (in total n=1785) from n=5 mice and 5 vessels from each across three timepoints (3, 8 and 25 weeks), is presented transparently with all raw images and their annotation (**Source Data**). Consistent with our earlier data (**Fig. 1d** (previously Fig. 1k)), we observe that at three weeks of age, a significant proportion of lymphatic capillaries still display active sprouts. These sprouts show continuous junctions, low or no LYVE1, and elongated cell shape (**Fig. 1b, Extended Data Fig. 2a**), fulfilling all the criteria for zippers (see response to Problem 3 below). Quantification of junction types was therefore performed exclusively on blunt-ended capillaries, using refined classification that is explained in the next point (Problem 3). According to the refined criteria, we found that ~25% of junctions in the blunt-ended lymphatic capillaries of 3-week-old mice are buttons, with a similar proportion observed at later developmental stages (**Fig. 1f**). This finding supports our earlier conclusion that only a minor proportion of the junctions represent buttons, irrespective of the developmental stages analysed. This finding is also consistent with the original description of buttons in the terminal ends of adult tracheal lymphatic capillaries, where they were observed with a frequency of 25-30%¹. The discrepancy in our data compared to the study by Jannaway et al. may be attributed to the higher sensitivity and resolution of imaging techniques utilized in the current study, which allowed for the detection of even weaker fluorescent signals, revealing linear VE-cadherin junctions at the capillary LEC borders. In addition, the definition of buttons as considerably long VE-

cadherin⁺ fragments of <4.71 μm in length may explain the higher frequency of buttons reported in this study (for comparison, the average width of a cellular overlap is 2.3 μm, Fig. 2j) (see Problem 1).

• Problem 3: The inconsistencies appear to stem in part from the criteria used to apportion lymphatic junctions into “buttons only”, “buttons + zipper”, and other categories (lines 295-303, Fig. 5c, d). Here, the category names and drawings do NOT accurately reflect the confocal microscopic images in these figures, which in turn leads to other problems. The drawings of “buttons + zipper”, “zipper lobe base”, and “zipper lobe tip” categories show a uniform red line symbolic of zippers, but no buttons, whereas the black/white confocal images show discontinuous foci of strong VE-cadherin staining, some of which are connected by faint VE-cadherin staining that is very different from zipper junctions (Fig. 1d inset). Indeed, the red/blue confocal images of these categories in Fig. 5c resemble discontinuous buttons more than continuous zippers.

Remedy: A more representative method of categorization is needed. Apportioning junctions into three categories (buttons, intermediate, and zippers) would solve the problem and enable comparison of the new data to previous and future work. The intermediate group would accommodate the known variability of button junction morphology. As an example, previous use of this approach to quantify junctions in initial lymphatics in mouse ear skin at age 3 weeks revealed 60% buttons, 30% intermediate, and 10% zippers, with corresponding values of 50%, 40%, and 10% in the diaphragm (Jannaway et al., Cell Reports 2023).

Response: We highly value the Reviewer’s comment, and we have carefully reconsidered the categorization. We agree that equating VE-cadherin distribution, whether continuous or discontinuous, within the borders of capillary LECs to continuous VE-cadherin⁺ zippers observed in collecting vessel LECs is misleading.

To establish a representative categorization, we believe that the inclusion of LEC shape and the presence of LYVE1 as additional parameters besides the pattern of VE-cadherin distribution is important. We have therefore defined the following categories to reconcile previous and new data, aiming to accurately describe the spectrum of junction types observed. This categorization should also provide a more accurate assessment of capillary LEC ‘zippering’ in disease contexts:

- 1) **Button junction** – a punctate VE-cadherin⁺ deposit at the neck of LYVE1⁺ lobe/cellular overlap, with no detectable VE-cadherin at the borders of the overlap. Observed in oak leaf-shaped LECs only, in line with the original description of buttons¹
- 2) **Curvilinear junction** – continuous or discontinuous distribution of VE-cadherin within one border of LYVE1⁺ lobe/cellular overlap. Observed in oak leaf-shaped LECs only.
- 3) **Double junction** – continuous or discontinuous distribution of VE-cadherin within both borders of LYVE1⁺ lobe/cellular overlap. Observed in oak leaf-shaped LECs only.
- 4) **Zipper junction** – continuous linear VE-cadherin distribution at cell-cell contacts in the absence of LYVE1. Observed in collecting vessel LECs and capillary LECs of sprouting vessels.

As discussed above, and as pointed out by the Reviewer, terms ‘intermediate’ and ‘transforming’ junctions have been used in some studies and likely correspond to the curvilinear and double junctions we define here. However, these terms lack precision and have been used to imply a transition from zippers to buttons, with the latter considered a mature phenotype of functional lymphatic capillaries in the current literature. While this may initially seem like a semantic distinction, we believe it holds important biological significance. Our study, with its more precise definition of junctions, instead suggests that a ‘continuum of discontinuity’ is the primary feature of capillary LEC junctions, driven by the dynamic nature of their cellular overlaps and junctions within. The absence of previous descriptions of curvilinear and double junctions likely arises from the lower sensitivity and resolution of imaging techniques utilized in earlier studies.

We provide new illustrations of the junction categories, together with more accurate schematic drawings and line intensity profiles of LYVE1 or VE-cadherin signals across representative cell-cell contacts, as well as rigorous quantification (**Fig. 1f**, **Extended Data Fig. 1b**, **Source Data**). As explained above, our revised data indeed does not concur with the findings by Jannaway et al showing 60% of buttons at three weeks of age. Instead, we find that up to ~25% of LEC junctions within the blunt-ended lymphatic capillaries exhibit buttons at this stage, but also at later developmental stages (**Fig. 1f**), which is consistent with the original description of button frequency of 25-30% in the terminal ends of adult tracheal lymphatic capillaries¹. We observe double junctions and curvilinear junctions as being the most frequently observed junction types across all stages in dermal lymphatic capillaries (**Fig. 1f**), and also observed in the trachea and diaphragm (**Extended Data Fig. 1d**).

- **Problem 4:** Equating discontinuous faint VE-cadherin staining to the continuous strong staining of zippers led to another problem, where 4 of the 5 categories were grouped with zippers to support the authors' claims that "classical button junctions alone accounting for only ~15% of total junctions" (lines 300-301, Fig. 5d) and that button junctions constitute only 10-15% of endothelial cell junctions in mouse ear skin (lines 400-403). This grouping led to interpretations that are inconsistent with the published data on ear skin lymphatics cited with Problems 1 and 2.

Remedy: The authors should investigate possible explanations for the discontinuous faint VE-cadherin staining that accompanied button junctions. Is the pattern of VE-cadherin similar to staining for tight junction proteins claudin-5, JAM1, ESAM, and occludin? Is faint staining also found between buttons in initial lymphatics of VE-cadherin-GFP mice? Is some of the staining non-specific? Does the faint staining result from cross-reactivity of the VE-cadherin antibody to another protein, tested by preabsorption of their R&D VE-cadherin antibody AF1002 with VE-cadherin protein and by comparison to staining with other antibodies, such as Thermo Fisher rat monoclonal VE-cadherin antibody eBioBV13 (BV13) or BD Pharmingen rat anti-mouse antibody AB_2244733 used in previous studies? Does the faint staining come from superimposition of staining in the opposite membrane of lymphatics that collapse during fixation by immersion (lines 549-550) rather than by conventional vascular perfusion? Answers to these questions should guide the characterization and categorization of junction data included in the manuscript.

Response: To validate the immunostaining showing VE-cadherin distribution along the borders of LEC lobes, we analyzed ear skin from knock-in mice that express VE-cadherin-GFP fusion protein under the control of the endogenous *Cdh5* (encoding VE-cadherin) gene promoter¹². Tissues were preserved through cardiac perfusion with the fixative, to exclude the possibility of vessel collapse suggested by the Reviewer. As shown in **Fig. 1a** and **Extended Data Fig. 1a**, VE-cadherin-GFP fluorescence precisely co-localized with VE-cadherin immunostaining, also in curvilinear and double junctions, validating the specificity of the staining. These data thus also validate VE-cadherin distribution along the borders of LEC lobes, indicating it as a characteristic feature of capillary LEC junctions.

Data from both immunofluorescence and TEM analyses provide additional evidence against lymphatic vessel collapse caused by immersion fixation, proposed to lead to superimposition of staining within the opposite membranes of two LECs. All quantification of junctional staining was performed from immersion-fixed tissue where we typically quantified both LEC layers of the vessels (**Fig. 4 for Reviewers, Source Data**), frequently showing curvilinear and double junctions (**Fig. 1b, f**) in a single LEC layer. In addition, low magnification images and reconstruction of vessels from TEM images also did not reveal collapse of vessel lumens (**Supplementary Fig. 2, 3, 7c**).

Fig. 4 for Reviewers. Confocal sub-stacks of two lymphatic capillary ends in immersion-fixed ear skin from a 5-week-old mouse. Visualization of the two LEC layers in non-overlapping sub-stacks indicate their separation.

To analyze the localization of tight junction proteins in capillary LECs, we performed additional staining using antibodies against CLDN5, ZO1 and OCLN. CLDN5 showed co-localization with VE-cadherin in the different junction types including curvilinear and double junctions (**Fig. 1b**). In contrast, we observed only weak ZO1 staining, and no OCLN staining, in dermal lymphatic capillaries, despite successful staining of other dermal cell types (**Fig. 5 for Reviewers**). This pattern of expression is consistent with scRNA-seq data from murine dermal LECs⁷, showing low *Tjp1* (encoding ZO1) and no *Ocln* expression (**Fig. 5 for Reviewers**).

Fig. 5 for Reviewers. Expression of tight junction proteins ZO1 and OCLN in the mouse ear skin. Whole-mount immunofluorescence of 8-week-old ear skin showing ZO1 staining in EC junctions in blood vessels (red arrows, a) while no distinct ZO1 staining pattern in LECs (a), and no OCLN staining (b) is observed. Violin plots on the right show low *Tjp1* (encoding ZO1, a) and no *Ocln* (b) transcript expression in the five dermal LEC subtypes defined in Petkova et al⁷. Data extracted from: <https://makinenlab.shinyapps.io/DermaLymphaticEndothelialCells/>

- Problem 5: The “double zipper” category presents yet another concern, where the category name and drawing do not match the corresponding images, which show neither two continuous lines of VE-cadherin staining nor an oak-leaf shaped cell. In addition, the < 10- μ m distance between the lines (Fig. 5c, “double zipper” images, scale bar 10 μ m) exceeds by an order of magnitude the distance between junctions in overlapping regions in TEM images. Unlike the confocal image of the presumptive “double zipper” (Fig. 5c), TEM images of overlapping regions show junction separations of 0.5 μ m (Fig. 3j) or 1 μ m (Supplementary Fig. 4b). Similarly, the average overlap width was about 2 μ m in confocal images of mosaic mouse ears (Fig. 5f). Yet, the authors imply that the TEM images support their interpretation of “double zippers”. No meaningful interpretation of the claim of “double zippers” can be made without quantitative data and more compelling confocal and TEM images of this feature - if it actually exists - in oak-leaf endothelial cells at the tip of initial lymphatics in adult ear skin.

Remedy: Measurements are needed to compare the spacing of double rows of VE-cadherin staining at presumptive “double zippers” to the distance between adherens junctions in TEM images of overlapping regions. Do the values match? If not, other explanations for “double zippers” identified in confocal images

should be explored. More convincing confocal and TEM images and robust quantitative data should be added, and the category name, drawing, and interpretation revised to fit the data.

Response: As explained above (Problem 3), we have revised the nomenclature and instead of ‘double zipper’ we now use ‘double junction’ to avoid equating VE-cadherin distribution within the borders of capillary LECs to continuous VE-cadherin⁺ zippers in LYVE1⁻ collecting or sprouting capillary vessel LECs. The questions are addressed below separately for confocal microscopy and TEM data.

Confocal data: We provide new high-resolution confocal images with line intensity scans and schematic illustrations for a clearer visualization and description of the findings (**Fig. 1f**). Importantly, using the mosaic membrane-bound fluorescent proteins, we show that LYVE1 is precisely localized to overlaps between two cells (**Fig. 2g**), within which we observe VE-cadherin at one or both overlap borders, consistent with curvilinear or double junctions, respectively (**Fig. 1f**). Our new imaging data also reveal that double junctions are frequently present within the borders of the entire LYVE1⁺ cellular overlap, and the abundance of this junction type is now emphasized by classifying it as a separate category (**Fig. 1f**). We further show that the double junction is not a specific feature of ear skin dermal lymphatic vessels but it is abundant also in other organs such as trachea and diaphragm (**Extended Data Fig. 1d**). Of note, when reviewing previously published data, we observed both curvilinear and double junctions when images of sufficient quality and resolution were available (For example, Fig. 2L in Yao *et al.* Am J Pathol 2012 reproduced in **Fig. 6 for Reviewers**).

Fig. 6 for Reviewers. Presence of curvilinear and double junctions in a previous study. Recolored Fig. 2 L and G from Yao et al. Am J Pathol. 2012, showing tracheal lymphatic vessels in P70 mouse. Green arrows point to apparent curvilinear junctions, purple arrows to double junctions.

: ⇨ | F9 F9857H98

TEM data: We would first like to thank the Reviewer for noting our mistake with the annotation of the scale bar in the mentioned image. The correct scale bar is 5 μm (instead of 10 μm).

To address the Reviewer’s concerns, we performed additional quantifications and acquired higher magnification TEM images. First, it is important to acknowledge the major inherent challenge with quantitative analyses from TEM images. As illustrated below (**Fig. 7 for Reviewers**), depending on the sectioning angle, the ‘overlap’ may occur either perpendicular across or longitudinally along the cell-cell borders, with only the former being suitable for meaningful interpretation. Indeed, we observed a significant variability in the overlap length, as now clearly illustrated also in the reconstruction of the entire vessel perimeter (**Supplementary Fig. 2, 3**).

Fig. 7 for Reviewers. Influence of sectioning angle on overlap morphology and length.

Schematic illustration showing the effect of the sectioning angle (dotted line) in TEM, resulting in highly variable appearance and length of cellular overlaps.

With an attempt to provide meaningful quantitative data from TEM imaging, we measured the minimum ($0.97 \pm 0.58 \mu\text{m}$ (mean \pm s.d., $n=16$)) and maximum ($3.63 \pm 1.11 \mu\text{m}$ ($n=16$)) perpendicular width of cell-cell overlaps from confocal images (**Extended Data Fig. 2b**). Based on this, we only included in quantification TEM images showing an overlap of less than $4 \mu\text{m}$ in length, as likely representing perpendicular orientation. Below this threshold, the average overlap width measured in TEM images ($1.8 \pm 0.9 \mu\text{m}$ (mean \pm s.d., $n=34$)) (**Extended Data Fig. 2d**) was similar to that of confocal images ($2.3 \pm 0.8 \mu\text{m}$ ($n=26$)) (**Fig. 2j**). Two regions of electron densities, suggesting the presence of double junctions, were observed in 33% of cellular overlaps, and the majority of overlaps (76%) had at least one electron density (**Extended Data Fig. 2e**).

However, we agree with the Reviewer's concern regarding the interpretation of double junctions from TEM images. The measurement of overlap width in confocal data roughly corresponds to the spacing of double junctions, since these are found at the overlap borders (see e.g. **Fig. 1f**). From TEM images, with a higher spatial resolution, we measured a spacing of $0.93 \pm 1.1 \mu\text{m}$ (mean \pm s.d., $n=7$) between two adherens junctions. However, this data should be taken with caution because of the above-mentioned issues with the sectioning angle. Therefore, we have decided not to include quantitative data on the spacing of junctions. Instead, we provide new imaging data (see **Source Data** for **Fig. 1f**), including line scans across the cell-cell contacts (**Fig. 1f**), and new images illustrating the presence of double junctions in oak-leaf endothelial cells at the tip of initial dermal lymphatic vessels (**Extended Data Fig. 1c**).

- **Problem 6:** Related to the “double zipper” problem, all lymphatic endothelial cell junctions in TEM images are described as adherens junctions (lines 305-308; Supplementary Fig. 4b). However, this designation ignores evidence from the authors and many others that lymphatic endothelial cells are connected both by tight junctions (claudin-5, JAM1, ESAM, etc.) and by adherens junctions (VE-cadherin). However, TEM images in Fig. 3j and Supplementary Fig. 4b have insufficient magnification/resolution to distinguish tight junctions from adherens junctions by conventional TEM criteria (Farquhar & Palade, J Cell Biol 1963; Brightman & Reese, J Cell Biol 1969; Friend & Gilula, J Cell Biol 1972).

Remedy: Higher magnification TEM images are needed to distinguish tight junctions from adherens junctions in overlapping regions of lymphatic endothelial cells. In the replacement TEM images used for measurements and illustration, tight junctions should appear as focal sites of fusion of the outer leaflets of adjacent plasma membranes accompanied by electron densities of membrane/cytoplasm, and adherens junctions should appear as focal electron densities of membrane/cytoplasm separated by a narrow - but

clearly visible - intercellular space (Farquhar & Palade, J Cell Biol 1963; Brightman & Reese, J Cell Biol 1969; Friend & Gilula, J Cell Biol 1972).

Response: We have carefully reviewed the early TEM studies by Leak on intercellular junctions in lymphatic vessels. Leak indeed reported the presence of macula occludentes, i.e. tight junctions in the ear skin of guinea pig, mouse and rat, but these were observed only rarely^{13,14}. The majority of junctions were described as adherens junctions (Fig. 3, 6, 7, 12, 13 in¹⁴). Our extended TEM analysis aligns with these findings. As explained above, we obtained higher magnification TEM images that were used to quantify junction parameters, including the distance between adjacent plasma membranes at the sites of electron densities. In the majority of the junctions (73 of 88, 83%), we observed a narrow intercellular space of approximately 20 nm (**Extended Data Fig. 2f**), consistent with the reported width of an adherens junction. In 15 of 88 junctions (17%) we did not observe a clear separation of plasma membranes, suggesting the presence of junctions that ultrastructurally fulfil the criteria of tight junctions (**Extended Data Fig. 2c, g**). We also measured the minimum (22.8 ± 0.7 nm (mean \pm s.d., n=18)) and maximum (56.4 ± 2.0 nm (n=18)) width of intercellular clefts within the overlaps in control mice, which is incorporated in **Fig. 4k** to provide comparison to *Cdc42* mutant.

It should also be noted that despite the rare occurrence of ultrastructurally defined tight junctions, immunofluorescent detection of the classical tight and adherens junction proteins CLDN5 and VE-cadherin, respectively, showed large overlap in LECs as shown in **Fig. 1b**, and as reported previously^{1,15}. In TEM analysis, all but one junction classified as tight junction (of 15), was accompanied by an adherens junction, within a distance of $0.6 \mu\text{m} \pm 0.5 \mu\text{m}$. The close proximity may explain co-localization of immunostaining. However, the discrepancy between CLDN5 abundance at LEC borders, including in curvilinear and double junctions by immunostaining (**Fig. 1b**), and the rare presence of ultrastructurally defined tight junctions (**Extended Data Fig. 2c, g**) suggests that the adherens junctions in LECs also contain some classical tight junction molecules. Another notable difference between LEC junctions compared to those in BECs is the coordination of the formation and maintenance of adherens junctions and tight junctions in the former, at least *in vitro*. In BECs, adherens junctions are required for tight junction formation and stabilization, at least in part due to the regulation of CLDN5 expression by VE-cadherin¹⁶. In contrast, in cultured LECs, VE-cadherin silencing did not disrupt CLDN5 at the cell-cell junctions¹⁷, suggesting independent actions. This is further supported by observations *in vivo*, where VE-cadherin inactivation using a function-blocking antibody did not alter the distribution of the classical tight junction protein ZO1, despite inducing the dispersion of VE-cadherin in capillary LECs¹.

- **Problem 7:** The authors state that junction measurements were made in ear skin lymphatics from the tip to the first valve (Methods, line 615), apparently based on the assumption that discontinuous junctions are uniformly distributed throughout this segment of lymphatics. A gradient in junction phenotype from the tip downstream toward the collecting lymphatic would explain the finding of sparse button junctions in this region. In the trachea, button junctions are most numerous at the tip and are gradually replaced by zippers downstream, long BEFORE the first valve (see Fig. 1E, Baluk et al., JEM 2007). (As described in that paper, valves are restricted to the adventitial surface of the mouse trachea, but the junctions have a steep gradient in mucosal initial lymphatics that have no valves.) If ear skin lymphatics resemble those in the trachea, the region of initial lymphatics downstream from the tip would be expected to have fewer button junctions.

Remedy: Initial lymphatics in ear skin should be characterized to assess the gradient in button junctions from the tip to the first valve. If buttons decrease with distance in the ear as in the trachea, measurements should be restricted to the tip where buttons are most numerous. This approach has been used successfully by others (Fig.1E, 2G, 3E, S1E, and S1G, Churchill et al., JEM 2022).

Response: Baluk et al¹ indeed show a gradient in the presence of button junctions along tracheal lymphatic vessels, with a steep drop in the presence of button junctions around 500-750 μm from the vessel tip, corresponding to the transition of capillaries to (pre-)collecting vessels. Our data do not support such a

gradient in the presence of button junctions from vessel tip to the first valve in the dermal vasculature, nor do we observe buttons being most numerous at the tip region of the vessels. Instead, we frequently observe that the entire length of the initial vessel segment from the tip to the first valve shows high variability in the junctions, with curvilinear and double junctions being the most prevalent overall (**Extended Data Fig. 1a-c, Source Data**). However, it should be noted that compared to trachea, the lymphatic vessel hierarchy and architecture is differently organized in the skin, where the first valve can be found as close as 150 μm from the tip of a dermal lymphatic vessel (**Extended Data Fig. 1a**).

We have restricted all quantitative analyses to non-sprouting blunt-ended vessels and acquired images encompassing solely the region from vessel tip to the first valve. This has now been described in the results (line 108-111) and methods (Supplementary Information, Image quantification).

- **Problem 8:** Given the inconsistencies described in Problems 1 and 2, concerns about rigor are raised by the use of only 2 mice/group for measurements of % of junctions in various categories at 5, 8, and 25 weeks (Fig. 5d and legend), despite the abundance of junctions measured in each mouse.

Remedy: To increase the robustness of these important data that form the basis of the authors' conclusions about button junction abundance and to enable assessment of mouse to mouse variability, the group size should be increased from only 2 mice to $N = 5-6$ mice/group. The approach for making new measurements should reflect the changes made to correct Problems 3-8. The authors should acknowledge and discuss the similarities and differences of the new data in relation to published data and developmental timetables for ear skin and other organs (Yao et al., Am J Pathol 2012). The findings should also be discussed in the context of whether ear skin lymphatics are representative of those elsewhere (lines 99-100).

Response: As suggested and explained above, we have increased the number of mice in the analysis. Data, now based on 5 mice per timepoint and 5 vessels per each mouse, is presented in SuperPlot format¹⁸ with mean values/mouse represented as weighted averages in order to normalize for the number of datapoints per animal (**Fig. 1f, Extended Data Fig. 1b**). The advantage of SuperPlot is that it allows visualization of all measurements (vessel-level variability) as well as variability between biological replicates i.e. mice (repeatability). The data show high vessel-level variability, which was observed in previous studies (see Problem 1 response 1) and evident in our imaging data (**Source Data**). Presenting only a single average value for each mouse does, therefore, not meaningfully represent this biological variability that holds important information. The other advantage of SuperPlot is that it allows superimposing summary statistics from biological replicates on a graph of the entire junction/vessel-level dataset; statistical analysis in this case is calculated across separate mice, not individual vessels, even when data from each vessel are represented on the plot. All quantification data are included in the source data and images of annotated junctions are available in Zenodo with the identifier doi: 10.5281/zenodo.10927082: https://zenodo.org/records/10927082?token=eyJhbGciOiJIUzUxMiJ9.eyJpZCI6ImZjMjRjODBmLWE2ZTEtNDZhZC1hODgzLTcwYWJkNzRhMWMwNyIsImRhdGEiOnt9LCljYW5kb20iOiIlwYjlmODdkMDlmOGY2OTkyODhiZjVhYWZmZnZiOWI5NCJ9.CiItFY2O9_XKtWCxb5DkNr0buqANxA4D7Pnx8iS3f2WbiV-NzJSpWPj2LDUo4FHuk_r-woOC-tf0YCbtqMoAg, to provide full transparency.

In order to show the broader relevance of our quantitative findings in the dermal ear skin, we have included images of the trachea and diaphragm that similarly show the abundance of curvilinear and double junctions as discussed above (Problem 5). Finally, we have discussed our findings from the ear skin in the context of previous studies (lines 428-439).

- **Problem 9:** Fig. 6 does not make sense as a graphic summary of key findings. The schematic showing initial lymphatics with “double zippers” and “single zippers” but NO button junctions does not reflect the data presented in the manuscript or in previous reports, including their own (Ulvmar & Mäkinen, Cardiovasc Res 2016).

Remedy: After the junction problems are corrected, Fig. 6 should be changed to be consistent with the revised text and figures and with the literature. Advances over published evidence should be highlighted and validated in the figure legend.

Response: We have revised the graphic summary to incorporate button junctions and indicate the relative proportions of the different junction types in homeostasis and changes observed upon increase in interstitial fluid volume. We have also removed speculative elements, such as inferring transitions between junction types and the speculated role of overlap remodelling in generating active constriction of the vessels. Instead, we now only discuss these aspects in the text as our interpretations (lines 544-550).

- Overall integration and reconciliation: After the junction problems are remedied, please ensure that the revised findings and interpretations are described consistently throughout the manuscript, including the Abstract, Results, Discussion, and Figure legends. Readers would also benefit from additional background and discussion that explains: (i) the authors' rationale for choosing ear skin lymphatics for the present study; and (ii) the unusual postnatal developmental timetable of the mouse ear pinna and accompanying lymphatics (Mäkinen et al., Genes Dev 2005), which occurs much later than in the diaphragm, intestinal villi, and trachea.

Response: After incorporating the changes as outlined in the specific comments, we hope that the Reviewer finds our revised text now as consistently describing the findings and interpretations. We have discussed the rationale for choosing ear pinna as an ideal model, where vascular growth and maturation occurs postnatally and is therefore accessible to intravital imaging (line 79).

7. Discussion. Several general aspects of the Discussion deserve attention.

a. Puzzle cells: Because of the importance the authors give to the comparison of oak-leaf lymphatic endothelial cells to jigsaw puzzle-shaped epidermal cells in plants, more detailed background is needed on the similarities and differences in the two cell types.

Response: We have now included background comparing oak leaved LECs and plant puzzle cells, which includes discussion on the questions raised below (lines 557-571). We have also adjusted the order of the figures to discuss the implications of the similarities between LEC and plant puzzle cell shape towards the end of the manuscript. This way it becomes clearer that we utilize insight gained from studies on plant puzzle cells to address questions regarding the potential role of puzzle shape and the key mechanisms underlying its acquisition, rather than implying that the two cell types are directly comparable.

(i) Puzzle cells interdigitate with one another, but do they overlap like oak-leaf endothelial cells

Response: Plant epidermal cells are tightly connected by the rigid anticlinal cell walls and do not overlap.

(ii) If so, is the overlap augmented by stretch? (iii) If not, how does the difference bear on the comparison?

Response (to (iii)): Interestingly, puzzle cells, although lacking overlaps, exhibit convex shapes (bulging out). When subjected to stretching forces, they will flatten without breaking. We hypothesize, and provide evidence supporting that monolayers of flat animal cells without overlaps would be prone to breakage under stretching.

However, the comparison of the two cell types is indeed to be done with some caution. Plant epidermal cells are deeper, typically 10-20 microns, and have rigid cell walls, while mammalian LECs are extremely thin and with large overlaps and flexible membranes, as we now stress in the text (lines 559-560). Despite these differences, there are striking similarities: we provide evidence that the unique puzzle cell shape of LECs, like described in plants, exhibits a similar cytoskeletal architecture (**Fig. 3**), is regulated by CDC42-mediated cytoskeletal dynamics (**Fig. 4**) and by isotropic stretch (**Fig. 5**), and serves to increase monolayer stability (**Fig. 4, 5**).

(iv) Do oak-leaf endothelial cells form their distinctive shape as they enlarge during growth, as reported for puzzle cells?

Response: We indeed observed an increase in the number of lobes with increased cell size (**Fig. 2d, e**) during postnatal development.

(iv) What is the relevance to lymphatic endothelial cells of the reported influence of cell wall composition on the shape of puzzle cells in plants?

Response: In plants, the cell walls are exposed to a high tensile force generated by turgor pressure and cell wall expansion is restricted by MTs and cellulose microfibrils that act as springs restricting the growth locally¹⁹. Growth is a stress relaxation process, where cell wall remodelers release some of the tension by cutting and rearranging connections between cellulose microfibrils and specific matrix polysaccharides in the cell wall²⁰. Areas that are stiffer stretch less, and thus will grow less. In LECs, which have no cell wall, the mechanisms of lobe growth are different. Our new data show that dynamic CDC42-mediated actin remodelling is sufficient to drive extension of lamellipodia-like protrusions (**Fig. 5k, Supplementary Video 7**) that increases barrier strength (**Fig. 5l**). These differences, and the limitation of the modelling, are now noted in the discussion (lines 557-571).

b. Limitations and differences from published data: It is understandable that the authors highlight the significance of their findings, but equal attention should be given to the limitations of the approaches and methods and how they could impact the data and interpretations and contribute to differences from published data. Inadequate attention is given to a comparison of the present findings to published data on oak-leaf shaped lymphatic endothelial cells and the junctions between them. Data that are confirmatory should be identified, and findings that depart from the literature described with balanced treatment of the differences and how the discrepancies could be resolved.

Response: We have included discussion of our own findings in relation to the published literature to provide a clearer distinction between confirmatory data and novel findings. Additionally, we have discussed potential reasons for discrepancies between our data and previous studies on LEC junctions in dermal lymphatic capillaries (lines 432-439), as well as the limitations to our work (lines 557-587).

c. Are prevailing concepts challenged? As introduced in General Comment 6: Problem 1, the statement, "Our results challenge some of the prevailing concepts of flap valves" (lines 396-403) is not justified by the underlying data and is inconsistent with published data for button junctions in ear skin. Indeed, this claim detracts from the study's many solid new observations and creates a contradiction that the present data do not resolve. Unless data from additional experiments provide a compelling challenge to prevailing concepts, the authors should replace this sentence with a discussion of the strengths and limitations of their findings from mosaic mice, cytoskeleton staining, and intravital 2-photon imaging in comparison to published data on endothelial cells of initial lymphatics.

Response: The relatively low frequency of button junctions, as observed both by us and previously in the trachea¹, may suggest the presence of a pool of LEC contacts specialized for fluid and immune cell entry through a flap valve mechanism, which we now acknowledge in the discussion (lines 528-531). However, despite our sincere efforts to reconcile our new observations regarding the dynamic nature of the cellular overlaps between capillary LECs with the prevailing view of flap valves, we must acknowledge that these findings do not support some of the key concepts. Specifically:

1. Our extended quantitative data reveal the abundant presence of curvilinear and double junctions in capillary LECs, also after fluid injection. We thus argue that the prevailing concept of 'flap valves' that open to allow fluid passage is implausible given the observed junctional organization.
2. We provide compelling evidence for the dynamic remodelling of cell-cell contacts and actin cytoskeleton within capillary LECs *in vivo* across different time scales:
 - 1) Dynamic remodelling of actin in sub-minute time frame (**Fig. 3h**, (previously Fig. 5b), **Supplementary Video 3, 4**)

- 2) Dynamic remodelling of cell-cell borders in a time frame of hours (new data, **Supplementary Video 1**)
- 3) Dynamic changes in cell shape over the course of weeks and months (observation time frame extended to 41 weeks, **Fig. 2k** (previously Fig. 5a), **Extended Data Fig. 3a**)

Additionally, we provide new data showing that optogenetic activation of CDC42 in LEC monolayers *in vitro* induces the formation of dynamic lamellipodia-like protrusions, reminiscent of junction-based lamellipodia (**Fig. 5k, Supplementary Video 7**), and consequent increase in monolayer barrier strength (**Fig. 5l**). Conversely, the loss of CDC42-mediated actin remodelling leads to a reduction in cellular overlaps in lymphatic capillaries *in vivo* (new data, **Fig. 4e, f**) and in stretched LEC monolayers *in vitro* (**Fig. 5e-g, l, j**), ultimately leading to reduced lymphatic function (new data, **Fig. 4l**) and the loss of monolayer integrity (**Fig. 4g, Extended Data Fig. 4a, b, Fig. 5j**). Notably, our data do not rule out or even address whether e.g. immune cells could cross the endothelial barrier in between the lobular overlaps. However, we do not find evidence that adherens junctions within the overlapping lobes weaken upon fluid injection as would be expected with opening of flap valves.

Of note, additional observations supporting the flap valve concept include the reported detachment of intercellular junctions and the pulling of LECs by anchoring filaments to facilitate the expansion of the vessel lumen. It is important to note that although these concepts are commonly presented in review articles (²¹⁻²⁴ and others, including our own²⁵), the original data supporting these findings are derived from a few electron microscopy studies. For example, Castenholtz reported that lymphatic capillaries have focal regions of intercellular junction detachment that enlarge when the vessels dilate^{26,27}. However, to our knowledge, such openings have not been reported in recent studies utilising other methods. We also did not observe such openings in dermal lymphatic capillaries of wild type mice, even after injection of 20 µl of PBS into the ear and imaging of cellular overlaps in high resolution using the *iMb2-Mosaic* mice. Castenholtz visualized these openings using SEM after casting by interstitial injection of resin into the connective tissue²⁷ or after exposure of the vessels to high interstitial pressure, heat or histamine²⁶, without consideration of the disruptive impact such a procedure would likely incur. In fact, in one of the studies he also reported an open interface between the pre-lymphatic space and the initial lymphatic vessel²⁷ - a conclusion that contradicts current knowledge and suggests the possibility of artefacts introduced by the method used. Similarly, the exact manner in which anchoring filaments connect to the endothelium remains unclear, and their proposed function to facilitate simultaneous opening of interendothelial junctions²⁶ has yet to be experimentally demonstrated. These reflections highlight the need for further investigation and caution in interpreting even existing data, which we have now tried to achieve: throughout the discussion, we now discuss our findings in relation to the results of previous studies to provide a clearer distinction between confirmatory data, novel findings, and speculations. Additionally, we have discussed potential reasons for discrepancies between our data and previous studies on LEC junctions in dermal lymphatic capillaries (lines 432-439), as well as the limitations to our work (lines 557-587).

d. Starling principle: In the Discussion, the authors should consider the transit of fluid and solutes across lymphatic endothelial cells (lines 436-437) in the context of Starling forces that govern these movements (Michel, Woodcock & Curry, *Acta Anaesthesiol Scand* 2020), to correct the implication that the type of intercellular junctions can explain everything.

Response: While discussion of Starling forces is relevant for understanding the dynamics of fluid transport across the capillary barrier, we believe it may not be necessary for conveying the main message here (also keeping within space limits). We have clarified that in other vessel types solute and cellular passage can occur without the presence of buttons through regulated opening of intercellular contacts (lines 487-489).

Specific Comments

1. “Lymphatic capillaries”: The term “initial lymphatics” is recommended to be used in replace of “lymphatic capillaries” to emphasize that discontinuous junctions are a feature of lymph entry sites at the tip of lymphatics, but not necessarily in other regions of small lymphatics that lack smooth muscle, and to avoid confusion with blood capillaries. The term “distal initial lymphatic” or “pre-collector” could be used for the segment of lymphatics proximal to the first valve that lacks button junctions and smooth muscle.

Response: As discussed above (see response to Problem 7), our high-resolution imaging of VE-cadherin⁺ junctions, both based on immunolabelling or endogenously expressed EGFP-tagged protein, does not provide evidence for a gradient in junctional composition in the segment distal from the first valve. We understand the Reviewer’s suggestion that the term initial lymphatic could cause less confusion with blood capillaries, but would like to retain our terminology of lymphatic capillary because of the widely adopted use of the term in the lymphatic field to describe the LYVE1⁺ lymphatic segment which harbors LECs with the characteristic oak leaved morphology. PubMed search return 736 hits with "lymphatic capillar*" (to include both capillary and capillaries) and 255 hits with "initial lymphatic*" (lymphatic and lymphatics).

2. “LEC”: Readers would benefit from the authors’ restricting the use of the abbreviation LEC to isolated lymphatic endothelial cells studied *in vitro* to avoid confusion with endothelial cells of intact lymphatics.

Response: EC, BEC and LEC are commonly used abbreviations for blood/lymphatic endothelial cells in both *in vitro* and *in vivo* contexts. For clarity, we have opted to consistently use these abbreviations throughout the text.

3. Line 67: Please correct this statement to acknowledge that endothelial cells of initial lymphatics do have a basement membrane (Pflücke & Sixt, JEM 2009).

Response: We have corrected the statement and now refer to a thin basement membrane.

4. Fig. 1m, o: Please describe in the figure legend the meaning of “1.6x” in these figures.

Response: We considered this unnecessary and have removed 1.6x from this panel (now Fig. 2d).

5. Fig. 3h: Please adjust the magnification of images in row 3 to match those in row 1 with which they are compared. Also, please change the color of red arrows that mark structures in the Surface LYVE1 image in row 1, column 2, which are confusing because elsewhere red applies to Total LYVE1. In addition, please explain your interpretation for how the LYVE1 antibody stains the tiny blue region INSIDE the lymphatic without permeabilization (bottom drawing in inset).

Response: We have adjusted the magnification in Fig. 4i (previously Fig. 4h). The observed surface LYVE1 pattern is now introduced earlier, in Fig. 2h, and we have altered arrow colours to be consistent with the illustrations and labelling of antibodies, as suggested.

Regarding LYVE1 positive staining under non-permeabilized conditions, which is indeed detected both at the abluminal and intraluminal borders of the cellular overlap. Due to tissue preparation, lymphatic vessels are no longer intact and antibody can access the lumen where the vessel is cut or damaged, and, therefore, stain the intraluminal border.

We were unsure if the Reviewer was also referring to the strong puncta observed in regions of the cell that are not associated with cellular overlaps. Such puncta could potentially represent abnormal LYVE1 clusters at the cell membrane, however, the negative staining for total LYVE1 argues against this. It is more likely that these puncta represent aggregates and/or non-specific binding of antibodies, phenomena that are typically minimized in immunofluorescence staining by the addition of a detergent.

6. Fig. 3j: Please specify whether “dermal LECs” (line 195) refers to ear skin lymphatics or another region of skin. Also, please acknowledge that the widened intercellular spaces in TEM images in Fig. 3j (right) could be a technical artifact due to lymphatic collapse during fixation by immersion (lines 596-597) rather than by

conventional vascular perfusion. This would not negate the effect of CDC42 deletion on junctional integrity; instead, it could reflect greater susceptibility of junctions to separation during tissue preparation.

Response: All TEM images are taken from the tip-region of the ear and represent dermal lymphatic vessels and LECs. This has been specified in the methods. While we cannot be sure that they represent the initial vessel segment between the tip and the first valve, in this region all vessels are LYVE1⁺ and composed of oak leaf shaped LECs.

We now provide lower magnification images (**Extended Data Fig. 4c**) or reconstruction of vessels (**Supplementary Fig. 2, 3**) to demonstrate that all images, including those of *Cdc42* deficient mice, were taken from vessels with a clear lumen and no signs of ‘collapse’.

7. Lines 198-199: Please revise this text to reflect that the confocal images in Fig. 3i (arrowheads) show staining “discontinuities” rather than intercellular “gaps”, which would require TEM for confirmation. Also, revise the text to indicate that TEM images in Fig. 3j show focal regions of cell separation but not “large intercellular gaps” because other regions of the same overlapping cell borders are still in contact. Please also change the Y-axis label of Fig. 3k accordingly.

Response: We now refer to ‘LYVE1⁺ areas within the overlap regions’ when discussing data in **Extended Data Fig. 4b** (previously Fig. 3i). We have moved these panels to the supplementary data, in order to not imply that cell separations observed within cellular overlaps in TEM analysis (**Fig. 4j**) correspond to staining discontinuities in the immunofluorescence data (**Extended Data Fig. 4b**). As suggested, we have also changed the description of the phenotype (now **Fig. 4j, k**) to ‘focal regions of cell separation within cellular overlaps’. Y-axis label has been corrected to ‘intercellular distance’. We have also strengthened the data on intercellular separations in *Cdc42* deficient vessels using the *iMb2-Mosaic* line (**Fig. 4g**).

8. Line 202 “...formation of intercellular gaps (Fig. 3b, f, i)”: Please revise as for lines 198-199.

Response: This has been revised to ‘intercellular separations’.

9. Lines 204-206: The finding that deletion of *Cldn5* did not disrupt the discontinuous staining of VE-cadherin at button junctions in ear skin lymphatics is important and deserves to be described in more detail than “did not disrupt LEC junctions in the dermal vasculature”. The description should acknowledge that claudin-5 is only one of several tight junction proteins, and this redundancy is likely to explain why *Cldn5* deletion does not cause vascular junction disruption and rapid death. Also, please specify the antibody used for claudin-5 immunofluorescence (Supplementary Fig. 3b). Because claudin-5 immunoreactivity in the control is weak, perhaps stronger staining could be obtained by using rabbit polyclonal anti-mouse claudin-5 (34-1600, 147 1:1000, Invitrogen). This antibody could also help in resolving Problems 4, 5, and 6.

Response: We agree with the Reviewer that the lack of a junctional phenotypes in the dermal lymphatic vasculature upon *Cldn5* deletion is an interesting finding. It is in line with our previous observations in the postnatal mesentery where *Prox1-CreER²*-mediated deletion of *Cldn5* also did not cause collecting LEC junction disruption¹⁷. Our findings on *Cldn5*, and those by Hägerling et al on *Cdh5*,²⁸ suggest that the disruption of these major highly expressed cell-cell junction components of the dermal lymphatic vasculature alone does not lead to rapid disintegration of vessel integrity, suggesting that multiple cell-cell adhesion receptors work cooperatively and compensate for each other’s functions (now stated in the discussion, lines 475-477). This is in contrast to the loss of monolayer integrity upon deletion of *Cdc42*, which suggests a critical role of cytoskeletal regulation for the homeostatic maintenance of lymphatic capillaries. While further studies on the role of CLDN5 are of interest for future research, this is beyond the scope of our current study.

We did use rabbit anti-mouse CLDN5 antibody (34-1600, Invitrogen, 1:1000) and have repeated the staining (now **Extended Data Fig. 6b**, see also **Fig. 1b**). This antibody has indeed generally worked well, and we previously validated its specificity in *Cldn5* knock-out tissue¹⁷.

10. Line 208 "...these results demonstrate that CDC42...": Please tone down the interpretation. "...these results provide evidence that CDC42..." would be better.

Response: We have replaced 'demonstrate' with 'provide evidence' as suggested.

11. Lines 222-231: The strengths and limitations of the in vitro stretch experiments should be described.

Response: Discussion on the strengths and limitations of the stretch assay was added (line 571-579).

12. Lines 232-233 "...formation of intercellular gaps": Please use a term other than "gaps" (e.g., intercellular separations) for these large (< 25 μm) spaces between cells in endothelial monolayers (Fig. 4b) to avoid confusion with the much smaller gaps (< 1 μm) found in vivo. Please also change the heading and Y-axis label in Fig. 4d accordingly.

Response: We have revised the text and refer to 'intercellular separations'. Y-axis in **Fig. 4j** (previously Fig. 4b) is now 'intercellular distance'.

13. Lines 256-258: Please explain the relevance of the in silico model of effects of mechanical stress on lymphatic endothelial cell tubes with a cross-sectional wall thickness of > 10 μm (Fig. 4g-h, scale bar 10 μm) to simulate initial lymphatics with a wall thickness of < 1 μm .

Response: The initial LEC membrane thickness is 2 μm in the model, before pressure is applied (**Fig. 5a**). While large part of the cell has much lower thickness, this value corresponds to the average thickness of LECs we measured at the level of the nucleus from TEM images. This value also corresponds the average length of the cellular interface (overlap) where force transmission occurs (**Fig. 2j**). When the model is run and the forces (membrane stiffness, internal cellular pressure, luminal pressure, etc) are applied, LECs inflate and their thickness increases considerably and indeed reaches values > 10 μm . The exact amount depends on the stiffness of the membranes in the model, and we have tried to match reasonable estimates from the literature. We do not imply that such excessive inflation is physiological; nevertheless, the model serves to illustrate the effect of cell shape on stress and structural rigidity at both the cell and tissue levels. We have now emphasized the limitations of the model in the discussion (lines 557-571).

14. Lines 264-266 "...lobate shape enhances LEC monolayer resilience to mechanical strain": Please revise this interpretation to help readers understand the significance of the findings in terms of lymphatic vascular biology. What is the benefit to initial lymphatics of "resilience to mechanical strain"? Why is this feature different from shape changes in endothelial cells of blood vessels that dilate and constrict? Please also acknowledge that overlap is a common feature of vascular endothelial cells that is not restricted to initial lymphatics.

Response: The reviewer is correct in noting that other vessel types, including different types of blood vessels and collecting lymphatic vessels, also undergo dilation and constriction. However, these vessel types also have external support from the basement membrane and mural cell coverage that provide adhesive substrates and mechanical stability. The low abundance of actin stress fibers in capillary LECs *in vivo* (**Fig. 3e, f**) suggests that the thin and discontinuous basement membrane surrounding lymphatic capillaries does not similarly support strong cell-matrix adhesions. Capillary LECs are additionally uniquely exposed to isotropic stretch, in the absence of significant flow-induced mechanical forces, presenting them with a distinct challenge in terms of mechanical strain. Our new data (**Fig. 5h**) provide direct evidence that this previously overlooked type of mechanical strain critically contributes to the acquisition of their cell shape and resilience.

We have added the above discussion in the text (lines 518-524). In addition, we describe cellular overlap as a common feature of many EC types. For example, collecting vessel LECs also possess cellular overlaps, but these are highly consistent in width and much shorter ($0.6 \pm 0.1 \mu\text{m}$ (s.d., n=5)) compared to capillary LECs ($2.3 \pm 0.8 \mu\text{m}$ (s.d., n=26)) (**Fig. 2j**), and do not undergo dynamic remodelling (**Supplementary Video 5, 6**), as observed in capillary LECs (**Supplementary Video 3, 4**).

15. Lines 272-274 "...anchored in place by button junctions⁴" and "In contrast to this static structure...LEC lobes might instead be dynamic": The second of these two statements is misleading because no claim or assumption has been made that the overlapping flaps are static. Instead, they are assumed to change with the amount of lymph flow, chylomicron entry, etc. This issue can be resolved by describing the new findings as additional evidence of the dynamic nature of the valves.

Response: The Reviewer is correct in noting that no explicit claim has been made about the static nature of overlapping flaps. However, it has been proposed, and schematically drawn in a number of papers and review articles, that flap valves represent physical structures, held in place by the spot weld effect of button junctions, that open and close. When we referred to a 'static structure,' we thus meant the existence of such valve structure (flap) that itself remains unchanged under homeostatic conditions, even if the proposed flap can open and close. This is in contrast to our model, which proposes that the extension and shrinkage of the cellular overlap facilitate dynamic movement of cell borders, which serves to maintain barrier strength and a dynamic state of adherens junctions, thereby preserving a 'loose' vascular barrier for fluid passage. To avoid misinterpretation, we have removed the reference to 'static structure' when referring to the previous model of flap valves.

16. Lines 277-280: Cellular "overlaps" and "lamellipodia-like protrusions" look similar in the image in Supplementary Fig. 4a. Evidence for the structures illustrated being lamellipodia is not convincing, and the term is therefore misleading. What criteria were used to distinguish overlapping flaps from lamellipodia? In addition, please correct the spelling of lamellipodia.

Response: We agree that in this particular image the basis for the distinction between 'cellular overlap' and 'lamellipodia-like protrusion' was not clear and we have therefore removed the latter. However, we refer to overlaps overall as lamellipodia-like structures, based on the dynamic remodeling of actin and cell borders *in vivo*, and their close resemblance to junction-based lamellipodia described in blood ECs during embryonic vessel morphogenesis in zebrafish²⁹. Additionally, we provide new data showing that optogenetic activation of CDC42 in LEC monolayers *in vitro* induces the formation of dynamic lamellipodia-like cell-cell overlaps, reminiscent of junction-based lamellipodia (**Fig. 5k, Supplementary Video 7**), and consequent increase in monolayer barrier strength (**Fig. 5l**). Conversely, the loss of CDC42-mediated actin remodeling leads to a reduction in cellular overlaps in lymphatic capillaries *in vivo* (new data, **Fig. 4e, f**) and in stretched LEC monolayers *in vitro* (**Fig. 5e, f, i**), ultimately leading to reduced lymphatic function (new data, **Fig. 4l**) and the loss of monolayer integrity (**Fig. 4g, Extended Data Fig. 4b, c, Fig. 5j**).

We have corrected the spelling of lamellipodia.

17. Legends of Figs. 1, 5, Supplementary Fig. 4 "exemplary": The context indicates that the intended meaning is "representative" not "exemplary". Please change accordingly.

Response: We have replaced 'exemplary' with 'representative'.

18. Line 305 "majority" means 51% or more: Please describe the total number of cell contacts examined by TEM and the percent with junctions. As described by Leak (J Cell Biol 1971), some overlaps have junctions, and others do not. If the authors are trying to modify this concept, quantitative data would be needed for comparison to values predicted for random TEM sections of initial lymphatics with button junctions compared to their proposed junction model.

Response: Our observations concur with those of Leak in that some overlaps have regions of electron densities and others not. With the criteria of analyzing cellular overlaps of <4 μm in length as likely representing perpendicular views across cell-cell contacts (see Problem 5), we included 21 overlaps in the quantification. Of these, we observed 5 (24%) overlaps with no electron density, 9 (43%) with one electron density, and 7 (33%) with two or more electron densities (**Extended Data Fig. 2e**). Therefore, as stated (now line 174), the majority of cellular overlaps (76%) displayed at least one electron-dense junction at the

luminal and/or abluminal border. This is in agreement with confocal microscopy data showing up to ~25% of cellular overlaps with buttons i.e. without VE-cadherin around the LYVE1⁺ regions (Fig. 1f).

Further high-magnification TEM images of additional overlaps with electron densities were acquired to assess junction characteristics. The majority of the junctions within these overlaps (73 of 88, 83%) were adherens junctions with a narrow intercellular space (Extended Data Fig. 2g). In 15 of 88 junctions (17%) we did not observe a clear separation of plasma membranes, indicating the presence of junctions that ultrastructurally fulfil the criteria of tight junctions.

19. Lines 312-313: Please explain why the absence of change in button junctions in lymphatics after intradermal injection of PBS was surprising, when that is the expected result. The conventional view is that button junctions remain intact when the overlapping borders between the buttons separate during lymph entry. This is an attribute of button junctions that permits lymph entry WITHOUT disassembly of junctions. **Response:** We acknowledge that the original description and current view indeed posits that button junctions (i.e. lack cell-cell junctions within the ‘flaps’) permit lymph entry without a need to disassemble junctions¹. Given the relatively low proportion of button junctions observed in our analysis, we therefore considered the finding that even an acute increase in interstitial fluid volume does not increase their abundance unexpected. Yet, to avoid misinterpretation, we have omitted the statement in the revised discussion.

20. Lines 322-324 “lamellipodia-like contact sites”: Please see comment on lines 277-280. This interpretation would better fit the data if described as, “Collectively, these data are consistent with rapid changes in the borders of endothelial cells of initial lymphatics under baseline conditions and in response to increases in interstitial fluid pressure.”

Response: As discussed above (Specific comments point 16), the dynamic cell-cell contact sites that we describe here closely resemble junction-based lamellipodia described in blood ECs during embryonic vessel morphogenesis in zebrafish²⁹, as also suggested by Reviewer 4. Additionally, we provide new data showing that optogenetic activation of CDC42 in LEC monolayers *in vitro* induces the formation of dynamic lamellipodia-like protrusions, highly reminiscent of junction-based lamellipodia (Fig. 5k, Supplementary Video 7), and consequent increase in monolayer barrier strength (Fig. 5l). Thus, we consider it justified to refer to them as ‘lamellipodia-like’ contact sites.

21. Line 335 “uncovers”: Should be “uncovered”

Response: Corrected.

22. Line 342 “excess interstitial fluid” (also in first line of Abstract and Main section): “excess” is misleading because lymph flow is a normal process whereby interstitial fluid flows along the hydrostatic pressure gradient from blood vessels to initial lymphatics. “excess” would apply only when interstitial fluid exceeds the capacity of lymphatic drainage. Please delete “excess” in these sentences.

Response: Deleted.

23. Lines 346-349 “In most tissues analyzed^{4,6}, including the ear skin studied here, this transformation commences during late embryonic or early postnatal stage”: This statement is misleading because data for ear skin lymphatics in their study came from mice at age 5 weeks or older, apart from measurements of vessel diameter and endothelial cell size and lobe number at age 3 weeks (Fig. 1l, m, and n). Please change accordingly and acknowledge published data for age 3 weeks (e.g., Jannaway et al., Cell Reports 2023).

Response: As explained above (Problem 2), we have now included analysis of 3-week-old mice. We have addressed the discrepancy between our findings and those by Jannaway et al. in Problem 2 and in the revised discussion (lines 420-422).

24. Line 354: Please correct the error where claudin-5 is described as an adherens junction protein: “...we

found that CLDN5, the second major component of adherens junctions...”, to match the correct description as a component of tight junctions (line 204).

Response: Thank you for noting the error. As discussed above (Problem 6), previous TEM data by Leak^{13,14} and our new data (**Extended Data Fig. 2g**) indicate rare occurrence of junctions fulfilling the criteria of tight junctions that are spatially separated from adherens junctions in lymphatic endothelia. The abundance of the typical tight junction component CLDN5 at LEC borders by immunostaining and its colocalization with the adherens junction protein VE-cadherin in both zippers and buttons (**Fig. 1b** and ^{1,15}), as well as in curvilinear and double junctions (**Fig. 1b**) instead suggest that adherens junctions in LECs also contain certain tight junction molecules.

To avoid misinterpretation, we have revised the text as follows:

Line 472-473: ... CLDN5, the second major component of LEC junctions, ...

Line 287-288: ... genetic deletion of the major junctional proteins CLDN5 (Extended Data Fig. 6a, b) or VE-cadherin²² in the mature lymphatic vasculature did not disrupt LEC junctions in the dermal vasculature.

25. Lines 358-360 “...loss of the F-actin-enriched LEC lobes with a consequent loss of button junctions and cell junction integrity.”: Please change “loss” to “reduction” to fit the data in Fig. 3b, c.

Response: Text has been revised according to new data added in this figure. We now do not comment on buttons, but increase in LYVE1⁻ zipper junctions (line 267).

26. Lines 381-386: In the discussion of effects of isotropic stretch on LEC *in vitro*, please describe the corresponding effects on blood vessel endothelial cells. After performing the recommended experiments, acknowledge that the observed changes were or were not specific to lymphatic endothelial cells. Also mention that, because the cells did not develop an oak-leaf shape or acquire button junctions, the relevance of stretch-induced changes to initial lymphatic structure and function awaits further study.

Response: As discussed above in point 5d, we performed the isotropic stretch assay using human umbilical vein ECs (HUVECs). As previously reported¹⁰, HUVECs exhibit irregular cellular overlaps, formed by a reticular network of VE-cadherin¹¹, and numerous stress fibers under confluent unstretched conditions, the latter differing from LECs. Interestingly, stretching under the same parameters as applied to LECs led to numerous VE-cadherin⁺ finger-like protrusions in HUVECs, while the total cellular overlap area remained unaltered. We thus conclude that HUVECs also respond to isotropic stress but show a qualitatively different response compared to LECs. These data are included in **Extended Data Fig. 10a, b**.

We have acknowledged that the isotropic stretching assay mimics only certain aspects of the *in vivo* context, and does not recapitulate the full phenotype of capillary LECs including the formation of true buttons junctions (lines 572-574). Notably, however, our new data show that extending the period of cyclic isotropic stretching to 22 hours not only promoted cellular overlaps but also led to increased curvature of cell-cell borders in LEC monolayers (**Fig. 5h**). Despite the absence of button-like junctions under these conditions, the phenotype induced by isotropic stretch in LECs most closely resembles the capillary LEC attributes described so far in an *in vitro* setting.

27. Lines 396-403: As described in General Comment 6: Problem 1, the data presented are inconsistent with published data for button junctions in ear skin and do not justify the statement, “Our results challenge some of the prevailing concepts of flap valves.” Indeed, this statement detracts from the study’s many solid novel observations and raises an unjustified controversy that the present data do not resolve. Instead, the authors are encouraged to focus on the data from mosaic mice, cytoskeleton staining, and intravital 2-photon imaging that are convincing and novel.

Response: We have addressed the inconsistency of our data with previous data, which is likely attributed to the higher sensitivity and resolution of imaging techniques utilized in the current study, which allowed for the detection of even weaker fluorescent signals. Throughout the discussion, we have discussed our findings in relation to the results of previous studies to provide a clearer distinction between confirmatory

data and novel findings. Additionally, we have discussed potential reasons for discrepancies between our data and previous studies on LEC junctions in dermal lymphatic capillaries (lines 432-439), as well as the limitations of our work (line 557-587).

28. Lines 400-403 “...only around 10-15% of cell-cell junctions...represented true buttons”: This statement is not only inconsistent with the results of reports from 7 separate laboratories (General Comment 6: Problem 1) but is also inconsistent with evidence that an intradermal injection of VEGFA doubles the number of zippers from 25% to 50% within 30 min (Zhang et al., Science 2018). The current manuscript does not provide convincing evidence to justify this statement, because the data do not invalidate all published work on button junctions in ear skin lymphatics. The necessary changes are described in the remedies for General Comment 6: Problems 1-9.

Response: As explained above, we have addressed problems 1-9 and provide extensive image data showing distribution of VE-cadherin along the cell borders of LECs, explaining the lower proportion of buttons observed in our analysis compared to previous findings. In our extended analysis, by using the refined criteria and restricting the analysis to the tip regions of blunt-ended vessels, we find up to ~20-25% of buttons at 3, 5 and 25 weeks of age, which is consistent with the original description of buttons in the terminal ends of adult tracheal lymphatic capillaries, where they were observed with a frequency of 25-30%¹, but contrasts the previously reported predominance of buttons in dermal capillary LECs²⁻⁶. We interpret the lack of previous definition of curvilinear and double junctions as likely stemming from the lower sensitivity and resolution of imaging used compared to ours (see Problem 1), and in some cases additionally unprecise definition of button junctions. We present all images and quantification data transparently, including annotated images of the junctions (in Zenodo with the identifier doi: 10.5281/zenodo.10927082), for the Reviewers’ (and readers’) assessment.

Regarding the effect of VEGFA as reported by Zhang et al (Science 2018), we refer back to data reproduced in **Fig 2 for Reviewers**, showing a large spread of data points, representing one initial lymphatic vessel from n=6 mice, with 5 to 55% zippers in control vs. 10 to 75% in VEGFA injected ears. We would also like to note again in this context that, as shown in our analysis, ~25% of vessel tips at three weeks of age, similar to the age (P18-P21) analyzed by Zhang et al, represent sprouting vessels with zipper junctions (**Fig. 1c, d**). Inclusion of a few such vessel ends in quantification could skew the data. Assessment of this is not possible because raw data are not available.

29. Lines 418-421: Please correct the description of findings reported by Leak (J Cell Biol 1971) to indicate that he found that tracer passage between lymphatic endothelial cells stopped at the barrier created by tight junctions (maculae occludentes) (Fig. 15a, J Cell Biol 1971), whereas adherens junctions (maculae adherentes) maintained structural adhesion of the cells but did not create a barrier. The authors should also note that Leak referred to adherens junctions as focal maculae adherentes, which are now recognized as buttons, not as continuous zonulae adherentes referred to as zippers. Leak’s paper was published two years after Brightman & Reese (J Cell Biol 1969) discovered that tight junctions in the form of zonulae occludentes create the endothelial barrier in the brain.

Response: We have carefully read the pivotal work by Leak. As the Reviewer notes, Leak¹³ shows in Figure 15a that injected lanthanum, used as a tracer, does not pass maculae occludentes. In addition, a serial section in the same figure panel shows that the close apposition of cellular membranes is lost, suggesting a focal adhesion device i.e. macula occludentes or tight junction. Such tight junctional devices are, however, rarely observed also by Leak and Burke¹⁴, and consistent with our extended TEM analysis (**Extended Data Fig. 2, Supplementary Fig. 1, 2**, see Problem 6). In the same publication, Leak and Burke¹⁴ also described the presence of both focal maculae adherentes (buttons) as well as continuous zonulae adherentes (zippers) in multiple figures (Fig. 3, 6, 7, 12, 13), and note that maculae adherentes are most frequently found. However, without serial sectioning it is not possible to exclude that the described focal maculae adherentes do not continue in z-dimension. The frequent observation of adherens junctions within cellular overlaps in TEM images, both in Leak’s studies and ours (**Extended Data Fig. 2, Supplementary Fig. 1, 2**),

also does not support their local confinement to the base of the lobe, since the large overlaps at the lobe tip would be expected to be more frequently represented, compared to the base regions.

We have corrected the description of Leak's findings accordingly (lines 479-483).

30. Lines 431-433: The proposed mechanism of "zippering" without changes in buttons is not sufficiently clear in relation to what the authors envision or in the context of published data for conditions where button to zipper transformation occurs (Yao et al. Am J Pathol 2012; Zheng et al., Genes Dev 2014; Zhang et al., Science 2018; Churchill et al., JEM 2022; Jannaway et al., Cell Reports 2023). Measurements would be needed to argue that the total length of junctional VE-cadherin is the same before and after zippering. Speculation is fine (if acknowledged as such) but should describe how the authors' speculation differs from what is already known.

Response: Previous literature suggests transformation of buttons to zippers, where VE-cadherin is distributed to regions of cell-cell contacts previously devoid of adherens junctions. As we show, curvilinear and double junctions predominate in dermal capillary LECs. We propose that, in the context of increased interstitial fluid, the observed zippering may in fact consist of pulling of the distal and proximal cell-cell borders/junctions together, resulting in reduced cellular overlap width. This closely resembles the model of junction-based lamellipodia recently described in the zebrafish vasculature to facilitate the convergence movement of blood ECs during vessel morphogenesis²⁹, and to increase monolayer integrity and transendothelial barrier strength *in vitro*^{10,11}. We propose that capillary LEC lobes similarly use dynamic VE-cadherin positioning and actin polymerization to actively move within the boundaries of the overlap formed by two neighboring cells. Our model is thus conceptually fundamentally different from the previous view, as also highlighted by Review 2, and hopefully, this novel aspect of our proposed mechanism of action is now better explained in the discussion (lines 544-550).

31. Lines 434-441: Please broaden the context here to avoid the implication that oak-leaf shaped endothelial cells and button junctions are essential for transit of fluid and cells across the endothelium, given that neither is present in blood vessels where fluid and cells can move in both directions.

Response: During the revision of the discussion, this text was deemed redundant and removed.

32. Lines 437-442: Pflücke & Sixt, JEM 2009 should be cited here to provide perspective.

Response: Here (now lines 489-490), we discuss the entry of leukocytes through zipper junctions during inflammation. The study by Pflücke & Sixt, which primarily focuses on the properties of the basement membrane around lymphatic capillaries that enable leukocyte entry, is referenced in the introduction (line 60), however, we do not find it directly relevant in this context.

33. Lines 442-444: Please acknowledge any inconsistencies with the results of previous studies of button junctions in ear skin lymphatics (see General Comment 6: Problem 1).

Response: We have discussed our findings in relation to the results of previous studies as described above.

34. Line 615 "...distal to the first valve.": As button junctions are not expected to be present in lymphatics distal to the first valve, it would seem that the authors intended to describe the region of study as "proximal" to the first valve, which is the segment between the initial tip and the first valve. Please see General Comment 6: Problem 7 and adjust the text accordingly.

Response: In anatomical terms, distal is used to describe parts of the body and anatomical structures that are further away from the trunk or their point of origin/attachment. Thus, we considered the segment between the initial tip and the first valve as being 'distal to the first valve' or the 'distal tip' - in line with the Reviewer's suggestion above to use the term 'distal initial lymphatic' (Specific comment 1). However, we understand that in this context distal can also be interpreted in terms of flow direction, potentially leading to confusion. To avoid misinterpretation, we now describe the region of study as the segment between the initial tip and the first valve.

35. Line 638 “tresholded”: Should be “thresholded”

Response: Corrected.

36. All figure legends: Please report in all figure legends the number of mice used for each measurement AND the statistical significance of differences where N = number of mice per group. Statistics can also be reported for number of lobes, cells, lymphatics, or ears, if desired.

Response: We have included the number of mice and statistical significance based on this in all figure legends. For quantification of junction categories (**Fig. 1f, Extended Data Fig. 1b**), we used SuperPlot that allows superimposing summary statistics from biological replicates (=mice) on a graph of the entire junction/vessel-level dataset; statistical analysis is calculated across separate mice, not individual vessels, even when data from each vessel is represented on the plot.

Referee #2 (Remarks to the Author):

The manuscript submitted by Mäkinen and colleagues report on the description and some functional studies on overlapping, oak leaf-shaped lymphatic endothelial cells (LECs) in lymphatic capillaries in the ear skin of mice. Using a mosaic fluorescent labeling of LECs in lymph vessels, the authors confirm the presence of oak leaf-shaped LECs in lymphatic capillaries (as already reported by Baluk et al. and others) and their absence in the LECs of lymphatic collector vessels. They also confirm previous observations that button-like junctions are found in the lymphatic capillaries, and they show that these buttons develop mainly postnatally. By providing impressive images of lymphatic capillaries, they could demonstrate further that microtubules are located at the concave side of the lobes of the LECs, while actin filaments are exclusively located at the convex side of the lobes. The authors draw parallels between the oak leaf-shaped LECs and the so-called puzzle or epidermal pavement cells of plants. Likely because Rho-related GTPases have been shown to shape these puzzle cells in plants by regulating both actin and tubulin assemblies, Mäkinen and colleagues knocked out the Rho-family kinase *cdc42*, and they report that the typical oak leaf shape of the LECs changed, concomitant with changes in microtubules and junctions. Using mechanical stretching experiments, the authors further show that in human LECs, pharmacologic inhibition of *cdc42* reduces the overlaps of the LECs (that are found in lymphatic capillaries), and they present an *in silico* model that these changes could result in a collapse of the lymphatic capillaries when the interstitial fluid pressure outside these vessels increases (but they do not show this in experiments). Finally, using time-lapse imaging over several weeks, Mäkinen and colleagues indicate that LECs can change their shapes over several weeks and, more notably that actin filaments in these LEC quickly rearrange. Most notably, the authors show that by injecting PBS into the ear skin (as previously reported by Planas Paz and colleagues) increases the interstitial fluid pressure and reduces the overlap of the lobes of the LECs, likely causing the diameter of the lymphatic capillaries to widen. The authors conclude that the overlaps are not mere valves that allow inflow rather than outflow of fluid, but that these overlaps serve as a cellular reservoir for being able to widen the lumen of a lymphatic capillaries. In the opinion of this referee, this finding represents the most novel aspect of the submitted manuscript, as it shifts our view about these overlaps from flap valves for directed flow of fluid into the capillary lumen to cellular reservoirs for lumen expansion.

We thank the Reviewer for their helpful comments, and for highlighting the novel aspect of our study.

The following suggestions arose while reading the submitted manuscript, and they are grouped into major and minor ones.

Major suggestions:

1. The molecular mechanism of oak leaf-shape and overlap changes remains largely unclear, except that *cdc42* is required for maintenance *in vivo* and resistance to mechanical stretch *in vitro*. Since *cdc42* does many things in a cell, the pathway must be better explored. In particular, beta1 integrins have been widely

reported to be absolutely essential for lymphatic development and for mechanosensing in lymphatic and blood vascular endothelial cells. Even though integrins do not exist in plants, integrin like proteins have been identified (such as membrane-bound formins) as well as several integrin-linked kinases, and some of these have been shown to be involved in the shape of puzzle cells. Since manipulation of integrins is well-established, the authors must link their findings on cdc42 and stretching with beta1 integrins using KO of *Itgb1* and/or RGD peptides and/or integrin blocking or activating antibodies. In this regard, the authors are mistaken to give anchoring filaments a big role in their suggested model, in particular since genetic deletion of filament proteins has minor effects on the lymph vessels compared to deletion of beta1 integrin or integrin-linked kinase. Therefore, the role of integrins, ECM in their model must be urgently incorporated.

Response: As suggested, we have addressed the role of integrins both *in vivo* and *in vitro*. First, we induced genetic deletion of *Itgb1* using *Vegfr3-CreER^{T2}* in mice at 6 weeks of age, to circumvent developmental defects (**Extended Data Fig. 6c**). Junction morphology within lymphatic capillaries was assessed 6 and 10 weeks later, at 12 and 16 weeks of age, respectively. Despite the efficient deletion of *Itgb1* in LECs, shown by the absence of active integrin β 1 staining in Cre⁺ mice compared to littermate controls (**Extended Data Fig. 6d**), we did not observe apparent defects in LEC shape, overlaps or cell-cell junctions in mutant mice (**Extended Data Fig. 6e**). *Itgb1* deficient vessels exhibited a normal architecture and lobate LEC morphology, with VE-cadherin deposition along mostly curvilinear or double junctions (**Extended Data Fig. 6e**). While we cannot rule out the possibility that junctional defects emerge upon challenge such as edema or inflammation, further in-depth characterization of the potential role of integrin β 1 in these conditions is beyond the scope of this study. It is possible that multiple cell-matrix adhesion receptors work cooperatively and compensate for the function of integrin β 1 in maintaining adult lymphatic vessels.

We also assessed the impact of integrin β 1 inhibition on LECs under isotropic stretch. A function blocking antibody (rat anti-human CD29, clone: MAB13) was added at the onset of stretching at concentrations (0.1 and 0.2 μ g/ml) that were previously shown to inhibit integrin β 1 activity, without compromising EC attachment and monolayer integrity³⁰. Staining with antibodies detecting the active conformation of integrin β 1 (clone: HUTS-4) confirmed successful inhibition (**Extended Data Fig. 9a**). However, no effect was observed on stretch-induced increase in PECAM1-positive overlap width (**Extended Data Fig. 9b, c**). Notably, the higher concentration (0.2 μ g/ml) of mMAB13 appeared to result in straighter cell-cell junctions (**Extended Data Fig. 9b**), which is an interesting observation, yet this would require further investigations that are beyond the scope of this study.

Together, these *in vivo* and *in vitro* findings strongly argue against a direct role of integrin β 1 signalling as an upstream regulator of CDC42-mediated actin dynamics in LEC overlaps. As noted by the Reviewer, CDC42 has many cellular functions, and, similarly multiple upstream regulators. Additionally, besides acting through direct effectors, disruption of the cytoskeleton upon loss of CDC42 can indirectly cause multiple effects, especially in *in vivo* studies where there is a significant period between gene deletion and phenotype analysis. To provide further mechanistic insight and direct evidence regarding our proposed role of CDC42 in cellular overlaps, we utilized optogenetics to induce CDC42 activation in a temporally controlled manner in human LECs *in vitro*. To achieve this, we transduced LEC monolayers with a lentiviral construct expressing guanine-nucleotide exchange factor (GEF) domain from ITS1 linked to the improved light-induced dimer (iLID), allowing optogenetic CDC42 activation¹⁰. Local activation of CDC42 in LEC monolayers induced the formation of dynamic protrusions (**Fig. 5k, Supplementary Video 7**), reminiscent of junction-based lamellipodia²⁹. Control cells expressing the optogenetic recruitment iLID alone (data not shown and ¹⁰) as well as OptoITSN expressing LECs not exposed to photoactivation (**Fig. 5k**) did not show a response. Real-time electric cell-substrate impedance sensing measurements revealed a high baseline transendothelial resistance of $2207 \pm 589 \Omega$ (s.d., n=4 wells) in LECs, compared to that reported in other EC types (~ 1000 - 1500Ω)^{10,31} (**Fig. 5l**). OptoITSN-expressing LECs showed a higher barrier resistance compared to control cells, which was further increased instantaneously after global photoactivation of CDC42 within the entire confluent LEC monolayer (**Fig. 5l**), providing direct evidence for a role of CDC42 in driving this

process. Together, these results show that junction-based lamellipodial protrusions promote LEC monolayer integrity and, similar to what was recently reported in HUVECs¹⁰, modulate barrier strength.

Notably, the dynamic behaviour of LEC overlaps observed *in vivo* (**Fig. 3h, Supplementary Video 3, 4**) and now also *in vitro* (**Fig. 5k, Supplementary Video 7**) closely resembles junction-based lamellipodia, similarly characterized by variable VE-cadherin deposition and presence of actin at their tips, which were shown to facilitate convergence movement of blood ECs during embryonic vessel morphogenesis in zebrafish²⁹, and to increase monolayer integrity and transendothelial barrier strength *in vitro*^{10,11}. These new findings thus provide further support for our proposed model that capillary LEC lobes utilize dynamic VE-cadherin positioning in conjunction with CDC42-mediated actin polymerization to increase overlap area and thereby facilitate strengthening of the monolayer integrity. We have incorporated these additional mechanistic insights into the discussion.

2. The authors are asked to not only look at the role of cdc42 and integrins in the shape of the LECs, but are also asked to analyze how the overlap width is influenced by cdc42 and (likely its upstream signal) beta1-integrin. The widening of the lymphatic vascular lumen shall also be quantified, if possible.

Response: To answer these questions, we quantified cellular overlaps in *Cdc42* deficient lymphatic capillaries using the *iMb2-Mosaic;Prox1-CreER^{T2}* mice 3 weeks after induction of gene deletion. At this stage, changes in cell cytoskeleton and shape were evident without a major loss of cell junction integrity (**Fig. 4e, g**). We found that in *Cdc42* deficient vessels, the width of cellular overlap between LECs decreased compared to littermate control vessels (**Fig. 4e, f**), which coincided with the increase in LYVE1 negative zippers (**Fig. 4i**). Six weeks after gene deletion, the lymphatic monolayer was severely disrupted preventing the analysis as only a small proportion of LECs retained intercellular overlaps (**Fig. 4e, g, Extended Data Fig. 4a**).

We also quantified capillary width, serving as a readout of lumen width, but did not observe differences between *Cdc42* deficient and control vessels (data not shown). Given the observed reduction in the cellular overlap width from 2 μm to 1 μm (**Fig. 4f**), and with an average of three cells forming the vessel perimeter (*in vivo* parameters used for modelling, **Fig. 5b**), the expected increase in vessel circumference was calculated to be 3 μm . This corresponds to an increase of 0.95 μm in diameter, which is too small to be detected in capillaries that exhibit variability of several μm even in control mice (**Fig. 2f**).

Minor suggestions:

1. Overview images are needed for many images shown. It is the impression of this referee that a distinction between the capillary part and the collector part of the vessels is largely made by looking at the shape of the LECs. However, when manipulating the LEC shapes *in vivo* using *cdc42* deletion or inhibition as well as beta1 integrin deletion or inhibition, it is necessary to see the parts of the lymphatic vessels connected with each other.

Response: We have now provided the images used for quantifications of junction morphology, showing the blunt-ended tip of the vessel (**Source Data**). We have also included overview pictures of cell morphology upon *Cdc42* deletion in **Extended Data Fig. 4a**. For the quantification of microtubules in control (**Fig. 3b**) and *Cdc42* mutant (**Fig. 4c, d**) mice, high resolution and magnification was necessary. Nevertheless, it is important to note that all imaging and quantification were conducted in the segment between the initial tip and the first valve, a detail we have now emphasized in both the results (line 112) and methods (Supplementary Information, Image quantification) sections.

2. Fig. 1e: Please add “weeks of age” to make this panel self-explanatory.

Response: During the revision, this panel became redundant and was replaced with corresponding (extended) data included in **Fig. 1f** where Age (wk) is added as a clarification.

3. Fig. 2a: All data on differential gene expression must be included in the Supplement in form of a Table with all genes analyzed as well as log₂ ratios, p values, etc.

Response: Differentially expressed genes between capillary (merged Ptx3 Cap and Cap clusters) and collecting vessel LECs were identified using our published dataset⁷. Data now included in **Source Data** shows (1) a list of differentially expressed genes between capillary (merged Cap and Ptx3 Cap) and collecting vessel LECs, including log₂ fold change and statistics, and (2) data for Dot Plot presentation of selected capillary LEC-enriched cytoskeletal genes selected from (1) and shown in Fig. 2a, which was visualized using a ShinyApp web application for the same published dataset⁷:
<https://makinenlab.shinyapps.io/DermalLymphaticEndothelialCells/>.

4. Fig. 2d-e: The F-actin localization should also be quantified, and this is an easy task.

Response: We now provide line intensity profiles of Lifeact-EGFP signal across representative cell-cell contacts in both lymphatic capillaries and collecting vessels (**Fig. 3g**). In capillary LECs, Lifeact-EGFP is observed along the entire width of the LYVE1⁺ cellular overlap. In collecting vessels, Lifeact-EGFP signal shows a narrow peak colocalizing with VE-cadherin⁺ zipper junctions, suggesting a thin cortical actin rim at the cellular perimeter in these cells.

5. Fig. 3a: The authors switch from a VEGFR3-CreERT2 line to a Prox1-CreERT2 line. What is the reason?

Response: Based on our experience with multiple floxed alleles, we have observed that the *Vegfr3-CreER^{T2}* line does not provide as efficient gene deletion as *Prox1-CreER^{T2}* line at adult stages. The choice of an efficient Cre line was particularly important in the context of *Cdc42* deletion, to achieve simultaneous recombination of three alleles (two floxed *Cdc42* alleles and one *Rosa* allele (*Lifeact-EGFP* or *iMb2-mosaic*)) in each individual LEC. In contrast, the *Vegfr3-CreER^{T2}* line was an ideal choice for inducing mosaic expression of reporters, due to lack of non-endothelial expression that is observed in the *Prox1-CreER^{T2}* line.

6. Fig. 3j: It looks as if a *cdc42* floxed mouse is used as a control. However, since effects of Cre lines (due to their promoters capturing transcription factors and thereby influencing cell fates) have been reported, the authors must always use the respective Cre line as a control for comparative analyses.

Response: Indeed, in the TEM analysis, a Cre⁺ control was not included. However, for the other analyses Cre⁺ *Cdc42^{fllox/+}* mice were used and no phenotypes were observed in these (**Fig. 4e, g**). In addition, in previous TEM analyses of skin lymphatic vessels of tamoxifen-treated Prox1-CreER^{T2+} control mice, we did not observe phenotypes in the lymphatic vessels or their cell-cell junctions (Frye et al, eLife 2020: <https://iiif.elifesciences.org/lax/57732%2Felife-57732-fig2-v1.tif/full/1500/0/default.jpg>).

7. Fig. 4a: The authors give the wrong impression that their stretching device results in stretching of a real lymphatic vessel. They must correct this, and they must show how they stretched LECs in their device.

Response: We have improved the schematic drawing that hopefully provides a clearer representation of the experimental setup.

8. Fig. 4b: These analyses shall be complemented by using siRNA to knockdown *cdc42*, since their *cdc42* inhibitor might also cross-inhibit other molecules.

Response: We have performed the experiment using siRNA knock-down of CDC42 with the same conclusions as obtained using the inhibitor. Data is incorporated in **Fig. 5j** and **Extended Data Fig. 8a-d**. In the timeframe of the experiment (**Extended Data Fig. 8b**) CDC42 silencing did not affect integrity of unstretched monolayers (**Extended Data Fig. 8c**), but led to the loss of integrity upon isotropic stretching (**Extended Data Fig. 8d**).

9. Fig. 5a: The long-term studies on the LECs of the ear skin are difficult to interpret and do not help the study. Mice normally grow and get bigger over time, so that some changes in cell shapes are not unlikely to happen. In addition, the angle of imaging might slightly change, so that the cell shape looks different, even

though it stayed the same, but is just shown from a different angle. The authors shall consider deleting this Figure panel.

Response: We used young adult mice (> 6 weeks old) to avoid effects of tissue (and vessel) growth on LECs. The presented images are maximum intensity projections of z-stacks, ensuring the capture of the entire cell 'footprint'. Therefore, the suggested change of an imaging angle does not apply, which can be readily appreciated from the multiple examples provided to the Reviewers, including the large tile-scan views (**Supplementary Fig. 3**). To further strengthen the data, we have extended the observation period to 41 weeks. The new data clearly demonstrate that even in 9-month-old mice, in the absence of tissue and vessel growth, as well as the absence of LEC proliferation, migration or cell loss as indicated by the unchanged number and positioning of fluorescently labelled cells, the lobate shape of individual LECs continues to undergo changes (**Extended Data Fig. 3a**).

We consider this panel a highly novel aspect of the study, representing state-of-the-art technology and constituting a critical part of our conclusions, as it is the anticipated long-term outcome of the highly dynamic cell-cell contact properties we observe in a short time-frame. As such, we choose to retain these data.

10. Fig. 5d: Quantification of zipper versus button like junctions is not always as trivial as presented. Again, the imaging angle can make a zipper appear like a button. The authors must discuss this issue.

Response: The classification of the junctions is indeed challenging, and there are no consistent criteria used in the previous literature (see **Appendix 1**). As described in our responses to Reviewer 1's comments, we have significantly improved the quantification and defined a new categorization that more accurately reflects the *in vivo* observations (see response to Reviewer 1, Problem 3). This is presented in **Fig. 1f**, and we have included all raw data together with annotations in the **Source Data** for full transparency.

While other imaging and image processing parameters are critical for this analysis (see response to Reviewer 1, Problem 1, response 2), the imaging angle does not impact the results. The presented images are maximum intensity projections of z-stacks, ensuring the capture of the entire cell layer.

11. Fig. 6: Integrins and ECM must be placed next to cdc42; otherwise, the whole model is not mechanistic. Conversely, the anchoring filaments should be removed. They are present, but their impact on lymphatic development and function is limited.

Response: We agree that experimental evidence showing the importance of anchoring filaments is lacking, and we have therefore removed them from the revised schematic models as suggested. Considering that our new data do not support a role of integrins as upstream regulators of CDC42 in mediating dynamic remodelling of cellular overlaps (**Extended Data Fig. 6c-e, 9**), we have not added integrins in the model. We have also removed speculative elements, such as inferring transitions between junction types and the speculated role of overlap remodelling in generating active constriction of the vessels. Instead, we now only discuss these aspects in the text as our interpretations (lines 544-550). In addition, we emphasize in the text the direct role of CDC42 in regulating the extension of lamellipodia-like cellular protrusions, for which our new experimental data using photo-activatable CDC42 provide direct evidence (**Fig. 5k, I Supplementary Video 7**).

12. Text line 87 and 157: Please reword that VEGFR3- and Prox1-Cre are LEC specific. There is no strict LEC-specific Cre line.

Response: We agree, and have removed 'LEC-specific' in these contexts.

13. Text line 290: The sub-minute remodeling of actin filaments in the LECs of capillaries must be complemented with imaging and quantification of actin filament remodeling in collectors.

Response: We have included these intravital imaging data as **Supplementary Video 5, 6**. No changes in actin, mainly present at the cell cortex, were observed at the cell-cell contacts, despite the presence of typical collecting vessel contractions driven by surrounding smooth muscle cells.

Further points:

- Statistical tests shown in the legends to the Figures are valid choices. "Non-significant" could be removed from some panels though.

Response: We have removed 'non-significant' in the figures and corresponding legends as suggested.

Referee #3 (Remarks to the Author):

In this manuscript, the authors took an elegant approach to utilize a mosaic mouse line to image and clearly visualize the morphology of individual lymphatic endothelial cells within the mouse ear with a level of clarity that had not been achieved previously. With this tool they were able to quantify many morphological features of the LEC population in this tissue through various stages of post-natal development. Using a model that allowed for LEC specific post-natal deletion of the Rho GTPase Cdc42, the authors demonstrate its importance in regulation of the actin cytoskeleton and specialized junctions in LECs, and in vitro disruption of Cdc42 in the context of stretch compromises monolayer integrity. They note that the unique lobate shape of capillary LEC is similar in morphology to plant puzzle cells, and utilize a computational FEM model to suggest this unique shape may serve to lower the mechanical stress experienced by the cell, as is the case in plant puzzle cells. Lastly the authors present data on changes in cellular junction arrangement in response to a bolus injection of fluid into the ear interstitium.

Major Concerns

1. The computational model, as it is currently described, does not provide strong support that the underlying benefit of the lobate shape of LEC is to minimize intracellular mechanical stress. There are oversimplifications in the model (or at least my interpretation of how it was implemented based on the description provided) that would play a major role in how force is transmitted across the monolayer of cells. These include:

a. The assumption of homogenous mechanical properties of the cell. The authors provide experimental data that both stress fibers and junctional proteins are located at unique locations on the cell membrane. Once cell will preferentially transmit force to another cell at these locations. It seems that the model treats the cell-cell connectivity and mechanical properties identical at all of the interfaces between the cells and only changes the shape.

Response: We chose homogenous mechanical properties in order to test the effect of cell shape on vessel mechanics (stress at cell and tissue levels). Considering the heterogeneity in the junctional and cytoskeletal organization in capillary LECs we report here, it would also be difficult to determine how else to connect them. We now acknowledge that the model is simplified by not including the cellular overlaps and by assuming equal properties at the cell interfaces (lines 315-316 and 561-563).

Of note, LECs have cortical actin and very few or no radial stress fibers. Actin is highly enriched in cellular overlaps, and VE-cadherin is distributed along most cell-cell borders, although discontinuous at some of the cellular overlaps. Assuming linear VE-cadherin junction = high force transmission and discontinuous VE-cadherin = low force transmission would also be a simplification. This is highlighted by the finding that increase in cellular overlaps can increase vascular barrier strength in HUVEC monolayers by VE-cadherin-independent mechanism through Rho GTPase-driven extension of cellular protrusions¹⁰, and is further supported by our new data showing a similar transendothelial barrier increase in LEC monolayers upon CDC42 activation (**Fig. 5I**).

b. One of the most important structural features of initial lymphatics crucial to the unique function of lymphatics, is the unique anchoring filaments that connect certain locations of the cell to the interstitium. While the authors acknowledge these in the discussion, their importance are never considered in the interpretation of the data. The mechanics of the interstitium are also not considered. As interstitial fluid pressure increases, the interstitium will expand because it is a porelastic material. This expansion transmits force through the anchoring filaments to the initial LEC which helps expose gaps in the junctions that allow for the rapid entry of large molecule along a favorable pressure gradient from the outside of the vessel to the inside of the vessel. It is in the context of this scenario that the cell experiences mechanical loading. It is unclear how the simulations would be relevant to this much more complex context.

Response: While anchoring filaments have been described and illustrated in review articles and textbooks, direct observations are limited to a few electron microscopy studies. The exact manner in which they would connect to the endothelium remains unclear, and their proposed function to help opening the junctions has yet to be experimentally demonstrated. Integrating anchoring filaments into the model would necessitate making assumptions that may not accurately reflect the physiological reality. Indeed, Reviewer 2 requested removing anchoring filaments even from the schematic model because genetic deletion of filament proteins has shown only minor effects on lymphatic vessels³², and we have followed that Reviewer's suggestion.

c. In contrast to plant cells, the mechanical load on the cell is supported by the cytoskeleton, specifically F-actin, which exhibits highly anisotropic non-linear strain stiffening behavior, questioning the relevance of an isotropic St. Venant model, particularly in the context of large deformations.

Response: Our St. Venant material model generates a linear stress with respect to strain, however it uses Green strain, which is non-linear with respect to displacement (approximately quadratic). Consequently, there is some strain stiffening behavior with large deformations. Since lymphatic capillaries are not covered by a continuous basement membrane, it is not clear that their plasma membrane could tolerate very large stretches. In our model, the strains are less than 10%, which is moderate and unlikely to be very different in a non-linear model. Although there are large displacements in the model as the cells are pressurized, the deformations of the elements themselves are not excessively large.

d. Based on previous work published in plant puzzle cells (Sapala, Elife, 2018) and the simulations that were run, the results are to be expected. The Young's modulus of the cell wall and the internal cell pressure used for the LEC (10 KPa & .015 KPa) are appropriately orders of magnitude lower than what was used for plant cells (300 MPa & 0.5 MPa), however the ratio between the two parameters is quite similar (667 vs 600). Thus it would be expected that the stress would be lower in the puzzle cell configurations for the simulated LEC as well. This phenomenological behavior would be the same whether the computational LECs were configured in a monolayer or in the shape of a tube as was done here. What is not explained, is why one would not also expect to see a puzzle shape for LEC in a collecting vessel, or for any endothelial cell for that matter. If it could be shown that the benefit of this puzzle shape exists only in the context of some feature unique to initial LEC incorporated within the FEM model (e.g. present of anchoring filaments, discontinuous junctions, unique cytoskeleton arrangement, unique internal cellular pressure, unique Young's modulus of the cell wall, other unique external mechanical loading condition, etc...), then this concern would be minimized.

Response: To answer the question of why collecting vessel LECs (and other EC types) do not exhibit a puzzle shape, we note that unlike lymphatic capillaries, the collecting vessels are surrounded by a continuous basement membrane and lymphatic smooth muscle cells, which provide mechanical stability. The puzzle shape is hypothesized to be required to promote monolayer integrity in a tube with a few cells around, and prevent lumen collapse in the absence of a substantial basement membrane or mural cell coverage, against a relatively small pressure gradient from the interstitium to the lumen.

It should also be noted that capillary LECs are exposed to a very low, likely disturbed, flow, and instead experience transmural flow and isotropic stretch. In contrast, collecting vessels (like blood vessels) are exposed to significant shear stress caused by laminar flow of lymph, as evident by cell alignment and elongation along the direction of flow.

We have now discussed these considerations (lines 518-524).

2. I would caution overinterpreting the importance of the observed cytoskeletal changes in of LEC monolayers under isotropic stretch. Others have reported similar qualitative and quantitative changes in junctional and cytoskeletal morphology, and/or barrier function in LEC when mechanically stimulated by oscillatory shear stress (Norden, eLife, 2020) or transmural flow (Miteva, Circ Res, 2010). These monolayers are still much closer to collecting LEC than capillaries. Also, LEC on the lumen of a collecting lymphatic will undergo dynamic isotropic stretching with at least a magnitude of 6% during the diastolic filling phase. Also, PDMS typically has a Young's modulus around a range of 0.4 – 1 MPa, which is much stiffer than the stiffness experienced by initial LEC (Frye, Nat Comm, 2018). More evidence is needed if the author's want to imply that isotropic stress plays a role in the development of capillary-specific attributes.

Response: The cited studies among others have indeed documented changes in junctional and cytoskeletal morphology in cultured LECs when mechanically stimulated by oscillary or transmural flow. Oscillatory flow induces a cuboidal cell shape along with an increase in actin stress fibers forming a thick cortical network, increase in pMLC2 as well as an increase in overlapping contact sites that are assembled into a complex reticular adherens network⁹. Transmural flow induces downregulation of VE-cadherin and PECAM1, as well as discontinuous jagged VE-cadherin⁺ junctions³³ that are typically associated with increased paracellular permeability in both BECs and LECs^{34,35}. We report here that isotropic stretch induces overlaps, but does not induce prominent (oscillatory flow-associated) changes in the actin stress fibers (**Fig. 5g**) or pMLC2 levels (**Extended Data Fig. 9b**). Nor do we observe (transmural flow-associated) downregulation of VE-cadherin or PECAM1, or jagged VE-cadherin junctions. Importantly, our new data show that extended period of isotropic stretching increased the curvature of the cell-cell contacts (**Fig. 5g**). To our knowledge, this is the first instance of observing this hallmark feature of capillary LECs in an *in vitro* system, or in fact more broadly in an animal cell.

To conclude, the cytoskeletal and junctional changes induced by oscillatory or transmural flow in LECs are qualitatively very different from those induced by isotropic stretch. Interestingly, capillary LEC are uniquely exposed to isotropic stretch in the absence of significant fluid shear forces that likely predominate in determining cellular features in the larger collecting lymphatic vessels. Indeed, laminar flow promotes an elongated cell shape, cell alignment in the direction of flow, and the formation of actin stress fibers – attributes observed in collecting vessels *in vivo* by previous studies and by us.

Regarding stiffness, unfortunately no *in vivo* measurements are currently available for lymphatic capillaries. In our previous study by Frye et al⁸, we indeed reported low stiffness value of ~0.1 kPa experienced by the first LEC progenitors that migrate as individual cells within embryonic mesenchyme at E12. However, these values cannot be compared to mature dermal lymphatic capillaries in adult mouse skin, which is expected to be more similar to the value of 4 kPa measured in embryonic vein⁸. Young's modulus of the PDMS substrate used in our experiments is 2391 ± 95 kPa. While this is indeed significantly higher than values *in vivo*, it is significantly lower than the GPa values of glass or plastic that are commonly used in *in vitro* experiments³⁶.

To take into account the Reviewers' valuable comments, we have now clarified the different cellular responses of LECs to isotropic stretch compared to oscillatory and transmural flow (lines 574-579), and the role of laminar flow in collecting vessels (lines 521). We have added the PDMS stiffness value in the methods. To avoid overinterpreting the *in vitro* data, we have also discussed the limitation of the stretching assay (line 574-578). We further discuss dynamic cellular overlaps as an attribute that makes capillary LECs

capable of coping with isotropic stretch, noting that this specific attribute can be induced by isotropic stretching.

3. The authors spend a substantial section of the discussion suggesting that the data here challenges current understanding of how initial lymphatic junctions serve as flap valves. Examples include: “Our results challenge some of the prevailing concepts of flap valves.”...” Additionally, through intravital imaging, we observed dynamic remodeling of the actin-rich LEC lobes, contradicting their proposed function as stable flap valves.”... “These findings also suggest an active function for capillary LECs in fluid drainage, whereby passive shortening of LEC overlaps and lumen expansion in edema is counteracted by active actin-based lobe remodeling to increase cellular overlap and vessel compression aiding fluid movement, a process reminiscent of a bellow-like mechanism”. The valve-like function of initial capillaries was originally proposed primarily through functional data several years before the morphological descriptions of “buttons” and “zippers” took hold in the literature: (Trzewik, J, FASEB J, 2001). The data here provides further clarity on how one might interpret observations about buttons and zippers, but the initial understanding of initial lymphatic junctions functioning as flap valves is still unchanged.

Response: The concept of a ‘flap valve’ implies the presence of immobile structures (valves), anchored by stable button junctions, that open and close. The relatively low frequency of button junctions, as observed both by us and previously in the trachea¹, may suggest the presence of a pool of LEC contacts specialized for fluid and immune cell entry through a flap valve mechanism, which we now acknowledge in the discussion (lines 528-531). However, despite our sincere efforts to reconcile our new observations regarding the dynamic nature of the cellular overlaps between capillary LECs with the prevailing view of flap valves, we must acknowledge that these findings do not support some of the key concepts. Specifically: We would like to argue that our findings do challenge some of the key concepts of flap valves for the following reasons:

1. Our extended quantitative data reveal the predominant presence of curvilinear and double junctions in capillary LECs, also after fluid injection. We thus argue that the prevailing concept of ‘flap valves’ that open to allow fluid passage is implausible given our observed junctional organization. We have addressed the inconsistency of our data with previous data regarding the frequency of button junctions, which we interpret as likely stemming from the lower sensitivity and resolution of imaging used in previous studies.
2. We provide compelling evidence for the dynamic remodelling of cell-cell contacts and actin cytoskeleton within capillary LECs *in vivo* across different time scales:
 - 4) Dynamic remodelling of actin in sub-minute time frame (**Fig. 3h**, (previously Fig. 5b), **Supplementary Video 3, 4**)
 - 5) Dynamic remodelling of cell-cell borders in a time frame of hours (new data, **Supplementary Video 1**)
 - 6) Dynamic changes in cell shape over the course of weeks and months (observation time frame extended to 41 weeks, **Fig. 2k** (previously Fig. 5a), **Extended Data Fig. 3a**)

Additionally, we provide new data showing that optogenetic activation of CDC42 in LEC monolayers *in vitro* induces instantaneous formation of dynamic lamellipodia-like protrusions, reminiscent of junction-based lamellipodia (**Fig. 5k**, **Supplementary Video 7**), and consequent increase in monolayer barrier strength (**Fig. 5l**). Conversely, the loss of CDC42-mediated actin remodelling leads to a reduction in cellular overlaps in lymphatic capillaries *in vivo* (new data, **Fig. 4e, f**) and in stretched LEC monolayers *in vitro* (**Fig. 5e, f, i**), ultimately leading to reduced lymphatic function (new data, **Fig. 4l**) and the loss of monolayer integrity (**Fig. 4g, i**, **Extended Data Fig. 4b, c**, **Fig. 5j**). Based on these data, we propose that the lobate cell shape of capillary LECs, along with its associated cytoskeletal and junctional organization, is a result of continuous actin-driven remodelling of cellular overlaps and cell-cell contacts and serves to maintain barrier strength and a dynamic state of adherens junctions, thereby preserving a ‘loose’ vascular barrier compatible with fluid passage.

Interestingly, the proposed mechanism of flap valves has been challenged in the past based on mathematical modeling of an idealized tissue-lymph interface, which suggested a similar sliding mechanism³⁷ that we propose here.

4. The approach to measure total Lyve-1 and surface Lyve-1 is very elegant and clever and the images in Fig3 are beautiful. It should be remembered, however, that the labeling was done on fixed, unloaded tissue. Also the mouse ear probably lacks much of the extrinsic factors that are so important in facilitating lymph uptake elsewhere. Have the authors tried to inject a small volume of fluorescently Lyve-1 antibody in the live mouse ear prior to tissue fixation? My hypothesis would be that there would be substantially more Lyve-1 staining without permeabilization. Numerous studies have shown that intradermally injected antibodies rapidly enter lymphatics because of their specialized junctions. You could also inject unlabeled Lyve-1 antibodies in a live mouse ear, fix, and then label with secondary antibodies. In this case I would expect similar images to the surface Lyve-1 stain in Fig3h, as the secondary antibodies may have difficulty accessing the junctions of collapsed, unloaded vessels.

Response: We fully agree with the Reviewer that analyzing fixed unchallenged tissue does not reflect the accessibility of antibodies (to serve as a readout of intercellular passage of fluid and macromolecules) to cellular overlaps upon fluid uptake. The purpose of the assay was to highlight the loss of integrity and 'sealing' between neighbouring LECs in *Cdc42* deficient vessels under homeostatic conditions.

Following the Reviewer's suggestion, we performed an experiment where 20 µl of conjugated LYVE1 antibody (antibody A) was injected into a live mouse ear, followed by tissue fixation, staining for unpermeabilized LYVE1 (antibody B) in combination with secondary antibody (anti-A antibody) against the injected LYVE1 antibody. Tissue was subsequently refixed and permeabilized to stain for VE-cadherin (antibody C), followed by secondary antibodies against antibody B and C. As hypothesized by the Reviewer, the injected LYVE1 antibody (antibody A) readily accessed intercellular overlaps but neither staining for unpermeabilized LYVE1 (antibody B) (**Fig. 2h**) nor secondary against injected LYVE1 (anti-A antibody) (not shown) was able to stain the intercellular overlap. The order of consecutive staining is shown in **Fig. 2h**. Our findings are consistent with previous observations of intercellular LEC overlaps serving as a passage route for fluid and macromolecules¹³. The fact that fixation causes sealing and chemical crosslinking of the intercellular overlaps, making them inaccessible for staining, suggests the close proximity of the overlapping LEC plasma membranes despite increase in interstitial pressure upon fluid injection. This argues further against opening of flaps to facilitate fluid uptake.

5. I think care should be taken when describing junctions and actin as dynamic. The authors use this description to discuss adaptations that they observe over the course of minutes (Fig 5b) and over the course of weeks. It is not which time scale the overall mechanism proposed by the authors in Fig 6 is meant to function. In his original work Leak noted "It is evident that the rapid removal of connective tissue fluids by the lymphatic capillary is not entirely a passive filtration process but also involves the participation of specific morphological structures located at the connective tissue-lymph interface.." and in the decades since then numerous labs have demonstrated that initial lymphatics take up fluid and larger proteins/tracers (from 20 kDa – 200 nm) within seconds of being administered, making it to downstream lymph nodes within a minute. It is hard to imagine the active actin remodeling of initial lymphatics is crucial to this rapid capability in the context of other mechanisms such as the extrinsic pump (muscle contractions, respiration, gut peristalsis, cardiac contractions, etc...) and suction generated by intrinsic lymphatic contractility (Jamalian, *Sci Rep*, 2017).

Response: Our earlier and new data provide compelling evidence for the dynamic remodeling of cell-cell contacts and actin cytoskeleton within capillary LECs *in vivo* across different time scales, forming the foundation on describing these as dynamic processes:

- 1) Dynamic remodelling of actin in sub-minute time frame (**Fig. 3h**, (previously Fig. 5b), **Supplementary Video 3, 4**)
- 2) Dynamic remodelling of cell-cell borders in a time frame of hours (new data, **Supplementary Video 2**)
- 3) Dynamic changes in cell shape over the course of weeks and months (observation time frame extended to 41 weeks, **Fig. 2g** (previously Fig. 5a), **Extended Data Fig. 3a**)

Additionally, as also mentioned above, we provide new data showing that optogenetic activation of CDC42 in LEC monolayers *in vitro* induces the formation of dynamic lamellipodia-like protrusions driven by actin remodelling, reminiscent of junction-based lamellipodia within seconds (**Fig. 5k**, **Supplementary Video 7**), and consequent increase in monolayer barrier strength (**Fig. 5l**). Conversely, the loss of CDC42-mediated actin remodelling leads to a reduction in cellular overlaps in lymphatic capillaries *in vivo* (**Fig. 4e, f**) and in stretched LEC monolayers *in vitro* (**Fig. 5e, f, i**), ultimately leading to reduced lymphatic function (**Fig. 4l**) and the loss of monolayer integrity (**Fig. 4g, i, Fig. 5j**).

Thus, we provide several pieces of evidence supporting the dynamic nature of the overlapping cell-cell contacts between capillary LECs that, at least *in vitro*, can be modulated in a time frame of seconds, and we further show their importance for vessel functionality. In particular, we propose that this dynamic remodelling of the overlaps maintains capillary junctions in a state that is permissive for fluid uptake, and ensures monolayer integrity under isotropic stretch.

We fully agree with the Reviewer that the described extrinsic mechanisms of lymph drainage are critical, and this has now been explicitly clarified in the discussion (lines 551-552). We have also discussed the proposed role of anchoring filaments, originally described by Leak. As mentioned above, these filaments have been discussed in review articles and textbooks, but direct observations are limited to a few electron microscopy studies, and their proposed function to help open the junctions has yet to be shown experimentally.

6. The manuscript would be significantly strengthened if some of the observed morphological differences in *cdc42* KO mice, resulted in deficiencies in initial lymphatic function (e.g. clearance rate of large MW tracers). **Response:** As suggested, we have performed tracer injections to assess the consequence of *Cdc42* deletion to lymphatic vessel function 5 weeks after gene deletion when defects were observed in LEC cytoskeleton (**Fig. 4b-d**), cell shape (**Fig. 4e**), and cellular overlaps (new data, **Fig. 4e, f**) and junctions (**Fig. 4g, i**). The clearance of intradermally injected fluorescent tracer by lymphatic vessels was reduced in *Cdc42* deficient mice compared to littermate controls (**Fig. 4l**), indicating compromised lymphatic function. These data indicate a relationship between cellular overlaps and lymphatic drainage, supporting a model whereby cellular overlaps critically facilitate efficient lymphatic drainage and mechanical resilience to maintain vessel integrity upon changes in interstitial fluid volume especially in the context of unaltered collecting vessel integrity (**Extended Data Fig. 5**).

Minor

Might differential growth rates along the membrane driven by difference in mechanical loading be something driving the formation of lobes? This has been shown to be important in plant cells (Belletton, Nat Plants, 2021) and interestingly the authors show the preferential location of microtubules (Fig. 2b) on the concave regions, as was reported in plant cells.

Response: Indeed, we consider the similarities between the cytoskeletal organization of LECs with that of plant puzzle cells intriguing and potentially reflecting similar functions in controlling mechanical loading. Since we have not directly studied the role of microtubules in controlling LEC shape, we prefer not to speculate on this matter. Investigating the instructive role of microtubule anchoring in contributing to the lobate shape will be an interesting topic for future studies.

Injection of 20 ul of fluid in a tissue as small as the mouse ear skin is orders of magnitude above what would occur in the context of lymph formation. Even in severe burn models of edema in the ear, the measured accumulation of fluid is usually on the order of at most hundreds of nanoliters. While this data certainly shows the mechanical resilience of the initial lymphatics even under the most extreme behaviors, be careful about extrapolating this to make inferences about the normal function fluid and protein uptake.

Response: We agree that the amount injected represents an extreme situation. Such an extensive tissue swelling was intentional, to allow assessing a coordinated response required for sustaining mechanical resilience within the endothelium under acute stress. We have now state this in the text (line 122-123).

It should be noted that intravital imaging of actin (**Fig. 3h, Supplementary Video 3, 4**) and cell shape (new data, **Supplementary Video 2**) showing dynamic cell lobe remodelling were done in homeostatic conditions, without fluid injection.

SFig1b does not provide convincing evidence that F-actin is low in LECs. Also phalloidin does not seem to co-localize with the most prominent signal from the actin reporter in R26 mice (SFig1e).

Response: We agree, and now provide a better image to illustrate that F-actin is low in LECs (now **Supplementary Fig. 4b**). We also provide a better image for independent actin staining using another F-actin dye, SPY555-actin (**Supplementary Fig. 4e**). While some signal from the surrounding cells is captured in the image stack, the dye clearly illustrates the presence of F-actin at the lobe borders.

Line 174 – “revealed unaltered morphology” should read “revealed altered morphology”

Response: Corrected (now line 263-264).

Referee #4 (Remarks to the Author):

The manuscript “Resilience of lymphatic endothelium through dynamic cytoskeletal regulation of puzzle cell shape” explores in lymphatic endothelial cells the relationship between cellular shape, cytoskeleton and cell-cell junctions, and the greater biomechanical forces that act upon these cells. Authors first characterize both the cell shapes and cell-cell interfaces in both terminal capillary vessels, and in collecting vessels. They show that most LECs associate via classical “zipper” junctions, while those at vessel tips display more open button junctions. With both mosaic marking of cells (iMB2-Mosaic;vegfr3-CreERT2) and a LifeAct-GFP reporter, they find that junctions in capillary LECs exhibit a high level of lobulation and variability that they show represents their dynamic behaviors. They go onto that these cellular features, including the overlap of protrusions (cleverly measured by comparison of surface versus total LYVE1 levels), depends on the classical cytoskeletal regulator Cdc42. They then show that mechanical stress induces responses at the cell-cell interface (such as levels of PECAM), and they model both in vitro and in silico the effects of mechanical stress on overall vessel morphology. In their last data figure, the authors show that fluid pressure in vivo directly impacts the extent of lobule/lamellipodia overlap and hence (most likely) the capacity of lymphatic fluid uptake.

Overall, this study provides novel and timely insights on the critical role of the cytoskeleton on fundamentals like cell shape and adhesion, as well as higher level processes such as tissue morphology and likely function. It is an elegant study that borrows from what is known in plant epidermal cells, and applies findings to lymphatics. Strengths of the study are certainly the novelty of the findings in terms of a mammalian tissue such as the lymphatics and the high quality data on aspects of lymphatic cell biology previously unknown. Weaknesses include an incomplete mechanistic understanding of what exactly the CDC42 and cytoskeleton re doing within the observed cellular overlaps, and the consequent functional relevance. I would recommend some points for the authors to consider going forward:

We thank the Reviewer for acknowledging the novelty of our findings. In our revised manuscript, we provide new mechanistic insights into the function of CDC42, by showing that optogenetic activation of CDC42 is sufficient to induce immediate and localized extension of lamellipodia-like cellular overlaps in LECs, which, in turn, increase barrier strength of the monolayer. In addition, we provide functional evidence for the requirement of cellular overlaps in maintaining lymphatic drainage function.

Bigger points:

-The authors present compelling data that junctional anatomy changes during lymphatic development and upon changes in hydrostatic fluid pressure. However, do they have data showing that this impacts the lymphatics functionally? Can they show increased uptake of interstitial fluid associated with changes in cellular overlaps (fig.5d,e,f) with a dye or other tracers? What are the functional outcomes correlated to the cellular changes?

Response: This excellent question presents a technical challenge as injection of a tracer leads to an increase in fluid volume within the tissue, consequently affecting cellular overlaps (**Fig. 2e, f**). This prevents decoupling of the remodelling of overlaps from assessing its functional consequences.

To gain further insight into the relationship between cellular overlaps and lymphatic functionality, we utilized the *Cdc42* deficient mice that show loss of lobate cell shape and monolayer integrity of capillary LECs (**Fig. 4**). We now show that prior to the loss of junctional integrity they show shortening of cellular overlaps in vivo (**Fig. 4e, f**), consistent with our findings in vitro (**Fig. 5e, f, i, j**). The clearance of intradermally injected fluorescent tracer by lymphatic vessels was reduced in *Cdc42* deficient mice compared to littermate controls (**Fig. 4l**), indicating compromised lymphatic function. These new data indicate a relationship between cellular overlaps and lymphatic drainage, supporting a model whereby cellular overlaps critically facilitate efficient lymphatic drainage and mechanical resilience to maintain vessel integrity upon changes in interstitial fluid volume.

-The model presented in fig.5 is an outstanding demonstration of the findings of the paper. What do the authors think that CDC42 is doing to widen the cellular overlaps? Does it involve polymerization of actin and extension of lamellipodia? Or perhaps does it involve motor molecules helping one “flap” crawl over the other, in a ratchet like mechanism (see Belting and Affolter works in blood endothelial cells)? Have the authors examined phosphorylation of myosin, for example, at the overlap area?

Response: We fully agree with the Reviewer’s proposition regarding the suggested role of CDC42 in cellular overlaps, acting as a regulator of actin dynamics that facilitates the dynamic extension of lamellipodia-like cellular overlaps. In addition, as discussed (line 533-536), we noted the resemblance of the morphological findings to junction-based lamellipodia described by Belting et al in zebrafish blood endothelia, which indeed suggests that capillary LEC lobes similarly utilize dynamic VE-cadherin positioning in conjunction with actin-driven increase in overlap area.

To experimentally validate this proposition, we utilized optogenetics to induce CDC42 activation in a temporally controlled manner in human LECs in vitro. To achieve this, we transduced LEC monolayers with a lentiviral construct expressing guanine-nucleotide exchange factor (GEF) domain from ITS1N1 linked to the improved light-induced dimer (iLID), allowing optogenetic CDC42 activation¹⁰. Local activation of CDC42 in LEC monolayers induced the formation of dynamic protrusions (**Fig. 5k, Supplementary Video 7**), reminiscent of junction-based lamellipodia²⁹. Control cells expressing the optogenetic recruitment iLID alone (data not shown and ¹⁰) as well as OptoITSN expressing LECs not exposed to photoactivation (**Fig. 5k**) did not show a response. Real-time electric cell-substrate impedance sensing measurements revealed a high baseline transendothelial resistance of $2207 \pm 589 \Omega$ (s.d., n=4 wells) in LECs, compared to that reported in other EC types (approximately 1000-1500 Ω)^{10,31} (**Fig. 5l**). OptoITSN-expressing LECs showed a higher barrier resistance compared to control cells, which was further increased instantly after global photoactivation of CDC42 within the entire confluent LEC monolayer (**Fig. 5l**), providing direct evidence for a role of CDC42 in driving this process. Together, these results show that junction-based lamellipodial

protrusions promote LEC monolayer integrity, and, similar to what was recently reported in HUVECs¹⁰, modulate barrier strength. Interestingly, Mahlandt et al reported that while the state of VE-cadherin influenced the baseline vascular barrier strength in HUVECs, cellular overlaps further modulate the amplitude, inducing instantaneous changes in barrier strength through Rho GTPase-driven formation of junction-based lamellipodia¹⁰.

As suggested by the Reviewer, we have additionally performed pMLC2 staining of LECs both *in vitro* and *in vivo* using a phospho-myosin light chain 2 antibody, which has been previously used in several *in vitro* studies^{9,38,39}. Weak staining has been reported in cultured LECs in baseline conditions, as also observed by us (**Extended Data Fig. 9b**), but increased in response to oscillatory shear⁹. However, we did not observe changes in pMLC2 levels upon isotropic stretch (**Extended Data Fig. 9b**). *In vivo* staining for pMLC2 also showed only very weak staining at LEC junctions and overlaps, despite abundant staining observed in vascular smooth muscle cells within the same tissue. This observation is consistent with the low expression of *MyI9* (encoding smooth muscle myosin light chain 2 isoform) in dermal LECs based on our scRNA-seq **Fig 8 for reviewers**, and *MyI9* being highly selective for smooth muscle cells (<https://www.proteinatlas.org/ENSG00000101335-MYL9/single+cell+type>). We have not been able to evaluate staining for another phosphorylated myosin light chain isoform, *MyI6*, which is abundantly expressed in LECs (**Fig 8 for reviewers**), due to the lack of validated antibodies.

Fig. 8 for Reviewers. pMLC2 immunoreactivity and expression of *MyI* genes in the mouse ear skin. Whole-mount immunofluorescence of 12-week-old ear skin showing weak pMLC2 staining within LEC cell borders, while strong staining can be seen in a discrete spot within LEC cell body (arrows, left panels) and in SMCs (right panels) (a). Scale bar: 10 μ m. Bubble plot in (b) showing low *MyI9* (encoding smooth muscle myosin light chain 2 isoform) transcript expression in the five dermal LEC subtypes defined in Petkova et al⁷. Data extracted from: <https://makinenlab.shinyapps.io/DermaLymphaticEndothelialCells/>

-It is sometimes difficult to distinguish lobes (convex, lamellopodia) versus intervening areas (zippers, concave), as when one considers the multiple cells within a vessel, the lobe of cell A will be nestled within the concave region of cell B. Authors should demonstrate with a schematic exactly how these are defined exactly, relative to buttons, and how this works in a tissue with multiple cells, not just a single isolated cell. **Response:** We have revised the schematic presentations of the different junction types, which are provided with line intensity profiles of LYVE1 or VE-cadherin signals across representative cell-cell contacts (**Fig. 1f**). The summary **Fig. 6** shows overlaps in the context of the monolayer, which are also more clearly illustrated in immunofluorescence images throughout the manuscript (e.g. **Fig. 2c, g, i; Fig. 4e, g**). We hope that the

Reviewer will find that these revisions have improved clarity.

Smaller points:

-How are buttons and zippers quantified, such as in Fig.1? There is only one sentence describing the parameters in the methods. Additional explanation is needed. Sometimes the images are not so clear in terms of buttons versus zippers.

Response: The classification of the junctions is indeed challenging, and there are no consistent criteria used in the previous literature (see **Appendix 1**). As described in our responses to Reviewer 1's comments, we have significantly improved the quantification and defined a new categorization that more accurately reflects the *in vivo* observations (see response to Reviewer 1, Problem 3). The criteria for this categorization has been described in the methods section, and we have included all raw data together with annotations in the **Source Data** for transparency.

-perhaps shade or false color cells in Fig.1c,f,g. Hard to appreciate cells at times in the VE-caderin or LYVE1 views. While the YFP is reference in the left panels, it is still hard to see cell borders.

Response: During revision, these figure panels were replaced. We have paid attention to ensuring that the cell borders are well-defined in the new selected images.

-Fig.1i needs higher magnification panels as in c,f,g to show what authors are talking about.

Response: During revision, this figure panel was also replaced. We have provided higher magnification of the corresponding new image (**Fig. 1a**).

-Define "Isotropic" in reference to stretch somewhere in manuscript for a wide readership.

Response: We have defined isotropic stretching when first described in the context of plant cells as follows: '...tissue growth occurring uniformly in all directions, resulting in isotropic stretching' (line 338-339), and now also provide a schematic drawing of the isotropic stretching device for further clarity (**Fig. 5d**).

-authors should comment on LYVE1 being more prominently expressed at the cell borders. This comes as a surprise in Fig. 2b. Why is it not evenly distributed? Cell outline is useful in 2b.

Response: We have included the description of the subcellular localization of LYVE1 *in vivo* in **Fig. 2c**. Imaging of neighboring LECs expressing a different membrane-targeted fluorescent protein clearly demonstrates that LYVE1 localization is restricted to cell-cell overlaps, as shown in line intensity profiles of LYVE1 and the fluorescent protein signals across a cell-cell contact (**Fig. 2c**). The reason why LYVE1 is localized at areas of overlap between two cells *in vivo*, but not *in vitro*, remains unknown.

-There are typos here and there that should be fixed: page 6, line 167 "covex", line 174 "unaltered" should be "altered" I think. Page 10, line 272 "extracellular" should be "extravascular" I think. P.36, line 911 "disperse" should be "dispersed" I think.

Response: We apologize for these errors that have now been corrected.

-panels in Fig.3b are hard to interpret. Need cell outlines. Hard to appreciate the tubulin connections to concave areas.

Response: We agree with the Reviewer that it is challenging to observe the microtubule endings in the maximum intensity projection image, due to the presence of overlapping signal from neighboring cells. It is essential to note that quantifications that form the basis of our conclusions were done manually from z-stacks, and not from z-projections, to guarantee accurate assignment of microtubule ends. We have included an example of a z-stack with annotated microtubules as **Supplementary Video 2**.

We attempted to improve the visualization by extracting microtubule signal from individual LECs using IMARIS surface mask based on membrane-targeted fluorescence protein expression. However, technical challenges due to insufficient z-resolution prevented excluding microtubules from neighboring cells at

overlap regions. As a result, in these images, illustrated in **Fig. 1 for Reviewers** on page 2, we observed not only the microtubule network of a cell but also a cluster of disconnected microtubule endings at the lobe tips, originating from microtubule endings at the concave regions of a neighboring cell.

-Fig.3h,i,j - the authors quantify cleft width, but they should also quantify the overlap area (especially in i).

Response: Quantifying width or area based on TEM images has inherent problems, as discussed in response to Reviewer 1, Problem 5, as depending on the sectioning angle, the overlap may occur either perpendicular across or longitudinally along the cell-cell contacts, resulting in highly variable dimensions. We further show that in confocal microscopy images overlap area can be extremely variable based on the organization of individual LECs with their neighbours, whereby neighbouring LECs can share a variable number of cellular lobes (**Fig. 2i, 4e**). We have therefore opted to quantify cellular overlap width based on confocal images in *Cdc42* deficient lymphatic capillaries using the *iMb2-Mosaic;Prox1-CreER^{T2}* mice 3 weeks after induction of gene deletion. At this stage, changes in cell cytoskeleton (**Fig. 4b-d**) and shape (**Fig. 4e**) were evident without a major loss of cell junction integrity (**Fig. 4g**). We found that in *Cdc42* deficient vessels, the width of cellular overlap between LECs decreased compared to littermate control vessels (**Fig. 4f**), which coincided with an increase in LYVE1⁺ zippers (**Fig. 4h**).

-What are the time intervals shown on left side of panels in b?

Response: We interpreted that the question referred to Fig. 5b – revised version now in **Fig. 3h**. Time intervals referring to minutes has now been indicated more clearly.

References

1. Baluk, P. *et al.* Functionally specialized junctions between endothelial cells of lymphatic vessels. *J. Exp. Med.* **204**, 2349–2362 (2007).
2. Zarkada, G. *et al.* Chylomicrons Regulate Lacteal Permeability and Intestinal Lipid Absorption. *Circ. Res.* **133**, 333–349 (2023).
3. Zhang, F. *et al.* Lacteal junction zippering protects against diet-induced obesity. *Science* **361**, 599–603 (2018).
4. Jannaway, M. *et al.* VEGFR3 is required for button junction formation in lymphatic vessels. *Cell Rep.* **42**, 112777 (2023).
5. Churchill, M. J. *et al.* Infection-induced lymphatic zippering restricts fluid transport and viral dissemination from skin. *J. Exp. Med.* **219**, e20211830 (2022).
6. Zheng, W. *et al.* Angiopoietin 2 regulates the transformation and integrity of lymphatic endothelial cell junctions. *Genes Dev.* **28**, 1592–1603 (2014).
7. Petkova, M. *et al.* Immune-interacting lymphatic endothelial subtype at capillary terminals drives lymphatic malformation. *J. Exp. Med.* **220**, e20220741 (2023).
8. Frye, M. *et al.* Matrix stiffness controls lymphatic vessel formation through regulation of a GATA2-dependent transcriptional program. *Nat. Commun.* **9**, 1511. DOI: 10.1038/s41467-018-03959-6 (2018).
9. Sabine, A. *et al.* FOXC2 and fluid shear stress stabilize postnatal lymphatic vasculature. *J. Clin. Invest.* **125**, 3861–3877 (2015).
10. Mahlandt, E. K. *et al.* Opto-RhoGEFs, an optimized optogenetic toolbox to reversibly control Rho GTPase activity on a global to subcellular scale, enabling precise control over vascular endothelial barrier strength. *eLife* **12**, RP84364 (2023).
11. Cao, J. & Schnittler, H. Putting VE-cadherin into JAIL for junction remodeling. *J. Cell Sci.* **132**, jcs222893 (2019).
12. Winderlich, M. *et al.* VE-PTP controls blood vessel development by balancing Tie-2 activity. *J. Cell Biol.* **185**, 657–671 (2009).
13. Leak, L. V. Studies on the permeability of lymphatic capillaries. *J. Cell Biol.* **50**, 300–323 (1971).

14. Leak, L. V. & Burke, J. F. Fine structure of the lymphatic capillary and the adjoining connective tissue area. *Am. J. Anat.* **118**, 785–809 (1966).
15. Yao, L.-C., Baluk, P., Srinivasan, R. S., Oliver, G. & McDonald, D. M. Plasticity of button-like junctions in the endothelium of airway lymphatics in development and inflammation. *Am. J. Pathol.* **180**, 2561–2575 (2012).
16. Taddei, A. *et al.* Endothelial adherens junctions control tight junctions by VE-cadherin-mediated upregulation of claudin-5. *Nat. Cell Biol.* **10**, 923–934 (2008).
17. Frye, M. *et al.* EphrinB2-EphB4 signalling provides Rho-mediated homeostatic control of lymphatic endothelial cell junction integrity. *eLife* **9**, e57732 (2020).
18. Lord, S. J., Velle, K. B., Mullins, R. D. & Fritz-Laylin, L. K. SuperPlots: Communicating reproducibility and variability in cell biology. *J. Cell Biol.* **219**, e202001064 (2020).
19. Sampathkumar, A. *et al.* Subcellular and supracellular mechanical stress prescribes cytoskeleton behavior in Arabidopsis cotyledon pavement cells. *eLife* **3**, e01967 (2014).
20. Zhang, Y. *et al.* Molecular insights into the complex mechanics of plant epidermal cell walls. *Science* **372**, 706–711 (2021).
21. Baluk, P. & McDonald, D. M. Buttons and Zippers: Endothelial Junctions in Lymphatic Vessels. *Cold Spring Harb. Perspect. Med.* **12**, a041178 (2022).
22. Zhang, F., Zarkada, G., Yi, S. & Eichmann, A. Lymphatic Endothelial Cell Junctions: Molecular Regulation in Physiology and Diseases. *Front. Physiol.* **11**, 509 (2020).
23. Norden, P. R. & Kume, T. Molecular Mechanisms Controlling Lymphatic Endothelial Junction Integrity. *Front. Cell Dev. Biol.* **8**, 627647 (2020).
24. Petrova, T. V. & Koh, G. Y. Biological functions of lymphatic vessels. *Science* **369**, eaax4063. DOI: 10.1126/science.aax4063 (2020).
25. Stritt, S., Koltowska, K. & Mäkinen, T. Homeostatic maintenance of the lymphatic vasculature. *Trends Mol. Med.* **27**, 955–970 (2021).
26. Castenholz, A. Structural and functional properties of initial lymphatics in the rat tongue: scanning electron microscopic findings. *Lymphology* **20**, 112–125 (1987).
27. Castenholz, A. Interpretation of structural patterns appearing on corrosion casts of small blood and initial lymphatic vessels. *Scanning Microsc.* **3**, 315–325 (1989).
28. Hägerling, R. *et al.* Distinct roles of VE-cadherin for development and maintenance of specific lymph vessel beds. *EMBO J.* **37**, e98271 (2018).
29. Paatero, I. *et al.* Junction-based lamellipodia drive endothelial cell rearrangements in vivo via a VE-cadherin-F-actin based oscillatory cell-cell interaction. *Nat. Commun.* **9**, 3545 (2018).
30. Hakanpaa, L. *et al.* Targeting β 1-integrin inhibits vascular leakage in endotoxemia. *Proc. Natl. Acad. Sci. U. S. A.* **115**, E6467–E6476 (2018).
31. Hilfenhaus, G. *et al.* Vav3-induced cytoskeletal dynamics contribute to heterotypic properties of endothelial barriers. *J. Cell Biol.* **217**, 2813–2830 (2018).
32. Danussi, C. *et al.* Emilin1 deficiency causes structural and functional defects of lymphatic vasculature. *Mol. Cell. Biol.* **28**, 4026–4039 (2008).
33. Miteva, D. O. *et al.* Transmural flow modulates cell and fluid transport functions of lymphatic endothelium. *Circ. Res.* **106**, 920–931 (2010).
34. Claesson-Welsh, L., Dejana, E. & McDonald, D. M. Permeability of the Endothelial Barrier: Identifying and Reconciling Controversies. *Trends Mol. Med.* **27**, 314–331 (2021).
35. Kakei, Y., Akashi, M., Shigeta, T., Hasegawa, T. & Komori, T. Alteration of cell-cell junctions in cultured human lymphatic endothelial cells with inflammatory cytokine stimulation. *Lymphat. Res. Biol.* **12**, 136–143 (2014).
36. Gordon, E., Schimmel, L. & Frye, M. The Importance of Mechanical Forces for in vitro Endothelial Cell Biology. *Front. Physiol.* **11**, 684 (2020).
37. Heppell, C., Roose, T. & Richardson, G. A Model for Interstitial Drainage Through a Sliding Lymphatic Valve. *Bull. Math. Biol.* **77**, 1101–1131 (2015).

38. Norden, P. R. *et al.* Shear stimulation of FOXC1 and FOXC2 differentially regulates cytoskeletal activity during lymphatic valve maturation. *eLife* **9**, e53814 (2020).
39. Hall, J. D. *et al.* Lymphoedema conditions disrupt endothelial barrier function in vitro. *J. R. Soc. Interface* **19**, 20220223 (2022).

APPENDIX REDACTED

APPENDIX REDACTED

Referee #1

Review for Nature of manuscript #2023-08-13841A entitled: Resilience of lymphatic endothelium through isotropic stretch-induced cytoskeletal regulation of puzzle cell shape

General Comments

1. Improvements: The revised manuscript has many improvements over the original version. The authors have clearly made an effort to address many issues identified by the reviewers. Among the solid findings are the age-related decrease in LEC proliferation and sprouts and increase in LEC lobe number, similarity of VE-cadherin and claudin-5 staining in LECs, distinctive features of LEC overlap in initial lymphatics in mosaic mice, uniform staining of LEC overlaps after *in vivo* intradermal injection of antibody, contrasting distributions of actin and tubulin cytoskeleton in LECs, cytoskeletal similarities of initial lymphatic LECs to plant puzzle cells, and effects of Cdc42 deletion on LEC cytoskeleton, overlap, and tracer clearance.

We would like to thank the Reviewer for acknowledging these improvements and for their critical and thorough assessment of the revised manuscript. We have followed the Reviewer's suggestion to remove certain data (TEM data on putative tight junctions) and added new data (additional mice for the analysis of the impact of acute edema on junctions (**Fig. 1h**) and images showing histological visualization (silver nitrate staining) of the junctional morphologies identified in our study (**Fig. 1f**, **Extended Data Fig. 1b**) and highlighting the heterogeneity of capillary LEC junctions *in vivo* (**Extended Data Fig. 1c**) and of stretched LECs *in vitro* (**Extended Data Fig. 7b**). We have also revised the manuscript text with a sincere aim to present our novel findings in the context of current knowledge, without invalidating previous findings.

2. Needing further attention: However, further attention is needed to address ongoing issues identified in the following comments. These include descriptions and interpretations that reflect the authors' preferred view instead of an objective evaluation of the evidence. One of several examples described in detail below is **the authors' view that tight junctions are rare in initial lymphatic LECs, despite IHC staining for claudin-5 matching staining for VE-cadherin**. The authors are encouraged to review the descriptions and interpretations throughout the manuscript and make the changes necessary to ensure they are objectively consistent with the evidence.

We have addressed all comments below point-by-point, with some overlapping questions cross-referenced. Certain topics were discussed during the first revision, and in these cases only a brief summary is provided.

Specifically, the comment on the abundance of tight junctions is addressed in comment 6b.

3. Route for fluid entry into lymphatics: Although the fluid and cell entry function of initial lymphatics is not disputed, the authors' proposal for how this occurs is confusing and is obfuscated by detailed consideration of LEC overlaps, junctions, cytoskeleton, Cdc42, puzzle cells, *in silico* modeling, and *in vitro* stretch effects. As a result, **the authors' proposed entry route is questionable and not critically assessed in the context of their data and published evidence**.

We would like to clarify that we are not proposing a new entry route but our data is consistent with entry of fluid through the overlapping intercellular spaces, as shown in our experiments by LYVE1 antibody injection (**Fig. 2h**). Please see also comment 3g below.

a. The authors describe the current view of discontinuous button junctions and flap valves (lines 57-61, 443-446), and then raise doubt about this view by reporting that only 20-30% of the junctions were

buttons and the remainder were curvilinear or double junctions that were either continuous or discontinuous (lines 116-119).

We have revised the text in the introduction and discussion to avoid presenting our data as invalidating previous findings.

Lines 60-62: The intermittent **junction-free** regions in between buttons ~~are thought to~~ **have been described to** function as primary flap valves that permit fluid and macromolecule uptake, as well as immune cell entry, from the interstitium into the vessel lumen^{4,5}.

Lines 457-458: ~~According to the currently accepted view, The overlapping intercellular contacts between adjacent LECs serve as mechanical flap valves that enable the entry of unidirectional flow of interstitial fluid and entry of immune cells into the vessel lumen^{4,5}. By being anchored in place by stable button junctions, these valves are thought to facilitate lymph entry without the disassembly of junctions.~~

b. In reconciling their findings, the authors propose (lines 479-482) that the entry routes are the intercellular clefts described by Leak (JCB 1971), who provided TEM evidence for tracer entry through spaces between LECs.

Correct.

c. The authors' proposal becomes confusing and loses plausibility when they claim that entry occurs at intercellular clefts joined by adherens junctions but not by tight junctions (lines 479-482), again citing Leak (JCB 1971).

The Reviewer is correct in that our statement about adherens junctions was confusing, and we have now revised the sentence referring to Leak's work as follows:

Line 493-494: EM analysis showed tracer passage stopping at tight junction barriers **but freely passing through spatially separated junction-free intercellular clefts**²¹ ~~but not at adherens junctions.~~

This is in line with Leak's description that tracer passage does not occur at places where there are tight junctions.

Leak: *"In intercellular clefts that are formed by closely apposed endothelial cells, the passage was unrestrained and tracer substance could be followed through most of its length (Figs. 12 and 13). However, in some areas the presence of surface specializations on the apposing endothelial membranes prevented the passage (Fig. 15 a). High magnifications of these sites indicate that they represent tight or occluding junctions, i.e., a fusion of the external leaflets of apposing membranes (Fig. 15c)."*

d. This claim misrepresents key findings by Leak (JCB 1971) illustrated by TEM images, described in the text, and beautifully summarized in a drawing (his Fig.30) showing entry between LECs through junction-free spaces bordered by maculae (spot) adherentes and maculae occludentes. Leak's images provide evidence for tracers entering lymphatics in junction-free regions of LEC overlaps located between maculae (spot junctions).

Addressed together with the next point.

e. Perhaps the misinterpretation stems from **Leak's use of the Latin term "maculae" (spot) for discontinuous adherens and tight junctions (now known as buttons)**. Leak did not describe the continuous junctions (zonulae adherentes and zonulae occludentes, now known as zippers) that form the endothelial barrier of collecting lymphatics and blood vessels.

Leak's description of LEC junction was based on TEM analysis. As explained in the initial rebuttal letter, without analysis of consecutive sections it remains impossible to distinguish whether the observed junctional devices are focal "maculae" junctions (now described as buttons), rather than discontinuities within a linear junction. As also discussed in our previous rebuttal letter, it is not possible from individual TEM images to draw conclusions on the spatial localisation of the observed junctions (i.e. that they are located parallelly at the borders of the junction-free flap valves) since depending on the sectioning angle the overlap may represent either a perpendicular cut across or a longitudinal cut along the cell-cell borders. Leak¹ does provide examples of serial sections (fig. 8, fig. 11), which allowed him to conclude that junctions do not form continuous belts. While discontinuity and focality of junctions can be concluded from fig. 8, it is much less clear in fig. 11, and no information about the frequency of these observations is provided.

We now provide additional confocal images of capillary LEC junctions at the level of individual lobes, together with drawings closely depicting the immunofluorescence signal, in new **Extended Data Fig. 1c**. From these images one may appreciate that a thin cross-section through a discontinuity within a VE-cadherin⁺ junction, as illustrated below, would also misleadingly appear as a junction-free flap (patent junction). Consecutive sections might show continuous VE-cadherin (zonulae adherentes) but also 'spots' of VE-cadherin (maculae adherentes); yet examples such as provided below do not fulfill the definition of buttons and intermittent junction-free flap valves. For additional examples, please see **Extended Data Fig. 1c**. We have revised the schematic drawings (**Fig. 1g, Fig. 6**) to illustrate that curvilinear and double junctions can be discontinuous or continuous.

Interestingly, our new data on the classical histological staining of junctional components using silver nitrate² revealed predominantly continuous deposits around the entire capillary LECs circumference, including at the tips of the lobes (**Fig. 1f, Extended Data Fig. 1b**). The precipitation of silver in this method was previously described in blood endothelium to result from the interaction with junctional components present on the inter-endothelial junctions², although the exact components have remained undefined. This data, in agreement with a previous report³, argues against the majority of lobes constituting junction-free flap valves and provides further evidence for the presence of double junctions.

f. The concept advanced by Leak (JCB 1971) for tracer entry through junction-free clefts bordered by discontinuous maculae adherentes serves as important background for the current concept of button junctions that border flap valves.

We agree that Leak's data show tracer entry through junction-free areas within the intercellular clefts. His data does not, however, provide sufficient evidence for spatial organisation of these regions as junction-free flaps bordered by button junctions.

Please see the previous comment.

g. The authors are asked to resolve the confusion by considering this background and the comments below and then explicitly describing their proposal for fluid entry into lymphatics and how their proposal is consistent with or inconsistent with their data and published evidence.

As explained above, we have revised the sentence referring to Leak's work (lines 493-494). Our data is consistent with entry of fluid through the overlapping intercellular spaces, as shown in our experiments by LYVE1 antibody injection (**Fig. 2h**).

h. As a starting point for their proposal, the authors should address the question of whether their findings can be integrated with the current view and thereby advance the understanding of primary lymphatic valves by incorporating what they learned about LEC overlaps, cytoskeleton, Cdc42, and *in vivo* motility.

The flap valve and continuously remodelling lobes might be integrated in the sense that the same lobes could transition between the two states, especially considering the dynamic formation and regression of individual LEC lobes. We now state this (lines 553-554).

We find it difficult to reconcile how a flap valve completely devoid of VE-cadherin and anchored at its sides by parallelly oriented stable button junctions, could be fully merged with the continuously remodelling cell-cell contacts frequently lined by VE-cadherin⁺ junctional complexes at one or both borders of the cellular overlap that we observe in a larger proportion of LEC lobes.

Importantly, we do not invalidate the LEC flap valve concept. In fact we observed a significant proportion of lobes (~20%) that fulfill the morphological criteria of classical flap valves. We now clarify that our analysis of junctional heterogeneity was done at the level of individual lobes (lines 39, 117-118, 445), and discuss the existence of a pool of LEC contacts specialised for fluid and immune cell entry through a flap valve mechanism (lines 540-543).

However, the observed junctional arrangements in the majority of capillary LEC lobes is similar to junction-based lamellipodia (JBL), which we now introduce earlier in the manuscript (lines 137-139). Such actin-driven JBLs were described by Paatero et al⁴ to be part of an oscillating ratchet-like mechanism, used by ECs to move over each other and thus providing the physical means for cell rearrangements during zebrafish blood vessel morphogenesis (see figure below). They have also been described in cultured ECs, and shown to increase transendothelial barrier resistance *in vitro*^{5,6}, which we now also show in LECs in our study. A key feature of JBLs *in vivo* observed in the earlier work was the presence of double lines of VE-cadherin⁴, reminiscent of double junctions categorised as a novel entity of capillary LEC junctions in our study.

FIGURE REDACTED

From Paatero et al⁴:

Proposed oscillatory mechanism of junction-based lamellipodia (JBL) function. A single cycle is depicted. F-actin (blue) protrusions emanate distally from a stable junction. These protrusions also contain diffuse VE-cadherin (green), but not ZO1 (red). At the distal end of the protrusion, F-actin, ZO1 and VE-cadherin are components of a newly formed junction with VE-cadherin-mediated contact to the underlying cell. Eventually, dynamic F-actin remodeling pulls the proximal junction towards the new junction.

4. Transparency and balance: Greater attention should be given to the background on which the authors build their story. The authors should also identify more clearly the assumptions underlying their junction-based lamellipodia hypothesis and balance their consideration of evidence for and against this hypothesis.

We have revised the text to more clearly define where our findings are confirmatory of previous findings:

- Intercellular overlaps of LECs in initial lymphatics/lymphatic capillaries serve as primary routes for fluid uptake as show by LYVE1 antibody injection. Stated as confirmatory: ... **supporting their previously established role** as a passage route for fluid and macromolecules²¹ (lines 166-167)
- Capillary LECs have a unique shape and large intercellular overlaps with their neighbours. Stated as confirmatory: *Expression of the reporter in LECs using the Vegfr3-CreER^{T2} transgene²⁰ confirmed at single cell level the distinctive shapes of the lobate capillary LECs and elongated collecting vessel LECs...* (lines 146-148)
- Lymphatic capillary junctions are morphologically distinct from other vessel types such as collecting lymphatic vessels and blood vessels. Stated as confirmatory: ...revealed the **previously demonstrated** distinctive organization of cell-cell junctions in different lymphatic vessel types⁴ (lines 92-93)
- We acknowledge the previously described presence of button junctions (i.e. segments of VE-cadherin deposits oriented parallelly along the sides of LYVE1⁺ regions devoid of VE-cadherin) among the junctional spectrum: *In addition to the classical button junctions oriented parallelly along the sides of junction-free regions⁴,...* (lines 97-98)
- We acknowledge that junctional heterogeneity has been described previously in developmental contexts: *Of note, previous studies have recognized junctional heterogeneity and classified 'intermediate' or 'transforming' junctions^{6,9}, which likely represent junction types identified here. However, intermediate junctions were described at developmental*

stages and were previously considered a transient state undergoing zippers-to-button transformation⁶ (lines 450-453)

- We acknowledge that discontinuity of capillary LECs junctions has been described: *Our data extend on the **previously established key concept of discontinuity of capillary LEC junctions**⁴ and **the role of overlaps as passage routes for fluid and solutes**²¹* (lines 539-540)

a. Although everyone would agree that the mechanisms regulating the formation and maintenance of the specialized junctions of initial lymphatics are not fully understood, the authors should be more transparent about what was known before their study (lines 71-72). At a minimum, references 6, 7, 14, and 15 should be cited for evidence favoring the contributions of VEGFA, VEGFR3, NOTCH1, angiopoietin-2, and glucocorticoid receptor signaling in the formation and remodeling of button junctions.

During the revision of text, the sentence was replaced and the suggested revision is no longer actual.

Previous: Hence, the specialised junctional organization of LECs is critical for efficient lymphatic function, yet the mechanisms that control the establishment and maintenance of the unique LEC junctions are not well understood.

Now: The specialised junctional organization of LECs is thus critical for efficient lymphatic function. However, given the discontinuity of their junctions, compounded by the absence of mural cells and only a thin basement membrane providing structural support^{1,2}, it remains unknown how the thin capillary LEC monolayer withstands intermittent changes in vessel calibre in response to alterations in interstitial fluid volume without rupturing.

We have noted to the involvement of several molecular regulators of capillary LEC junctions in an earlier sentence with reference to the suggested studies:

Lines 63-65: Dynamic transitions have been shown to occur between the two junction types in capillary lymphatic vessels during development and in disease states, **with several molecular players identified**⁶⁻⁹.

A deeper discussion delving into the molecular mechanisms of button junction formation is not feasible within the space constraints of the manuscript, and we believe would detract from the key focus of the paper, which is to understand the maintenance of capillary lymphatic endothelium integrity and function during homeostasis.

b. The discussion of the authors' hypothesis (lines 443-463) should be more balanced. It is not fair to cherry pick evidence that fits their hypothesis and trivialize evidence against it.

Lines 443-448 concerned discussion on the discrepancies of our data compared to previous reports on the predominance of buttons in dermal capillary LECs, which was added as requested previously. We do not wish to trivialize earlier work and during the current revision of the discussion, this text was deemed unnecessary and most of it removed. The revised text is as follows:

~~Lines 444-455: The discrepancies in our data compared to previous reports on the predominance of buttons in dermal capillary LECs may be attributed to the higher sensitivity and resolution of imaging techniques utilised in the current study, which allowed for the detection of even weaker fluorescent signals, revealing linear VE-cadherin junctions at the capillary LEC borders.~~ Restricting the analysis to blunt ended lymphatic capillaries at three weeks of age revealed that ~20% of cellular lobes exhibited classical button junctions, with no significant increase in their frequency in older mice to suggest

developmental maturation. We identified two new junction categories, termed curvilinear and double junctions, as the predominant junction types present at the lymphatic capillary terminals at adult stages. These were characterized by variations of continuous or discontinuous linear distribution of VE-cadherin extending to the tip of LYVE1⁺ lobe borders. Of note, previous studies have recognized junctional heterogeneity and classified ‘intermediate’ or ‘transforming’ junctions^{6,9}, which likely represent junction types identified here. However, intermediate junctions were described at developmental stages and were previously considered a transient state undergoing zippers-to-button transformation⁶. Additionally, variations in the criteria and methods used to classify junctions preclude direct comparison of the absolute frequencies of junction types reported in different studies. Variations in the criteria and methods used to classify junctions preclude direct comparison of the absolute frequencies of junction types reported in different studies. We additionally defined two new junction categories, curvilinear and double junctions, as the predominant junction types present at the lymphatic capillary terminals.

Lines 450-463 concerned the introduction of the flap valve concept followed by description of our data. We have revised the first part as follows:

Lines 457-460: ~~According to the currently accepted view,~~ The overlapping intercellular contacts between adjacent LECs ~~serve as mechanical flap valves that enable entry of interstitial fluid and entry of immune cells into the vessel lumen^{4,5}. Previous studies have described junction-free flap valves anchored at their sides by parallelly oriented button junctions, which facilitate lymph entry without the disassembly of these junctions.~~ Our mosaic analysis with membrane-bound fluorescent markers revealed a high degree of variability in the morphology of the LYVE1⁺ cellular overlaps, and their early emergence during development. Interestingly, we observed shortening of cellular overlaps upon intradermal injection of fluid, suggesting their ability to respond to acute changes in interstitial fluid volume.

We hope the Reviewer agrees with us that the precise proportion of different junctional types we and others observe is not of principal importance, but that the steady-state junctional heterogeneity is of biological interest.

c. Importantly, the implication that their evidence invalidates the LEC flap valve concept is not justified and should be revised.

As explained above in comments 3a, h and 4b, we have revised the text concerning flap valves. Indeed, we observed a significant proportion of lobes (~20%) that fulfill the morphological criteria of classical flap valves, and we discuss the existence of a pool of LEC contacts specialised for fluid and immune cell entry through a flap valve mechanism (lines 540-543). In addition, we discuss that the lobes might transition between JBLs and flap valves (lines 553-554).

d. The authors’ do not present convincing evidence to justify the implication (lines 443-463) that their data for LEC junctions are more valid, more meaningful, and should have greater weight than corresponding data in previous reports (see “8. Classification of LEC junctions” and “10. Reasons for discrepancies with published data...”).

We have removed reference to the question of resolution and sensitivity in the revised manuscript. Yet, technological advancements (e.g., detectors) in recent years have greatly improved resolution and sensitivity that is evident (to us) when comparing the quality of (our own) images acquired on our newest microscope system compared to earlier state-of-the-art systems available (to us) 5-10 years ago. Comparing our images to those from previous studies, such as Jannaway et al (see below

comment 6d), we note that some features, such as double junctions, are not clearly resolved in the latter.

e. The authors also do not explain (lines 443-463) why the observed changes in overlap, distribution of the actin cytoskeleton in flaps, and involvement of Cdc42 are inconsistent with the conventional view of LEC primary flap-valves as fluid entry sites. Why can't the flap-valve and lamellipodia concepts be merged into one integrated concept?

The flap valve and continuously remodelling lobes might be merged in the sense that the same lobes could transition between the two states, especially considering the dynamic formation and regression of individual LEC lobes. This is now stated (lines 553-554).

Our analysis of junctional arrangements of individual cellular lobes show a spectrum of junctional confirmations, with the majority not fitting the classical button junction classification. Regardless of the classification, terminology, methodology or absolute frequencies of junctional types reported by us or others, the key novel observations in the context of capillary LEC junctions in our study included:

- The presence of VE-cadherin⁺ regions, as shown by both immunostaining and *Cdh5-EGFP* reporter, at the tip or base of the lobe, ranging from focal junctional deposits to entire lobe perimeters. In addition, the presence of junctional components around the entire capillary LECs circumference, including at the tips of the lobes, shown by silver staining.
- The presence of double junctional deposits at both the tip and base of individual cellular lobes, either in a continuous or discontinuous fashion.
- The persistence of the observed junctional heterogeneity during the course of development and maturation of lymphatic capillaries, which could also be found in other organs such as diaphragm and trachea, both in immersion and *in situ* (vascular perfusion) fixed tissues.
- Actin remodelling at lobe tips, and complete genesis and disappearance of cellular lobes over time revealed by longitudinal intravital imaging, which suggests extensive junctional remodelling.
- Requirement of CDC42-mediated actin dynamics in the homeostatic maintenance of LEC shape and monolayer integrity selectively in lymphatic capillaries.

While we believe these arguments are of importance and a detailed discussion of these topics would provide background in the discussion, due to space constraints we are unable to include them.

f. To solve these problems the authors should make changes throughout the manuscript to reflect their **commitment to transparency, balance, and objectivity** as they describe the strengths and limitations of their findings and interpretations and how they build on what was already known.

We are committed to transparency, and have provided, from the beginning, original annotated image data used for quantification, and now also overviews of statistical testing in the source data, for full transparency. We have revised the results and discussion, sincerely aiming to present our data in the context of previous work, leaving space for other interpretations, and avoiding presenting our data as invalidating previous findings. See comments 3a, h and 4b where these have been specifically addressed and specified.

5. Healthy skepticism: Related to the previous comment, the manuscript in its present form does not exhibit a healthy level of skepticism over the relative strength of the findings and the authors' preferred interpretation over other reasonable interpretations. The manuscript would be strengthened by authors' distinguishing (i) data shown to be solid by reproducibility and statistics and having an unambiguous interpretation from (ii) data and interpretations that could be viewed by readers as less

convincing, potentially inconsistent with other evidence, or having interpretations different from the authors' preferred view.

We hope that the changes we have done throughout the manuscript are now satisfactory in this regard.

a. Examples of solid evidence include the similar shape of oak leaf LECs and plant puzzle cells, conspicuous overlap of adjacent LECs viewed in mosaic mice, uniform staining of overlapping regions after *in vivo* injection of LYVE1 antibody, distribution of actin in oak leaf-shaped LECs, distinctive differences between LEC junctions in initial lymphatics and collecting lymphatics, and effects of Cdc42 deletion on LEC cytoskeleton and overlaps.

We would like to thank the Reviewer for acknowledging the strength of these data.

b. Examples where the authors' interpretations would benefit from more skepticism and/or stronger data include the identification and quantitative comparison of tight junctions and adherens junctions in TEM images,

We have followed the Reviewer's recommendation below (comment 9h) to remove some of these data. We agree that the TEM data had limited value and depth. This data is not central to the study and was added as supplementary data at the revision stage.

relevance of stretch-induced changes in cultured LECs to oak leaf LECs *in vivo*,

We have now mentioned that the exposure of LECs to isotropic stretch *in vivo* is based on a hypothesis (lines 341-344), rather than direct visualisation of such mechanical forces *in vivo*. This hypothesis is supported by compelling data from plant puzzle cells, which identify isotropic stretch as an upstream mechanism influencing their cell shape. We also acknowledge that the parameters used for these experiments are not grounded in real measurements (lines 518-519), and that the model does not accurately replicate *in vivo* conditions and phenotypes (lines 523-524).

Despite the limitations, we consider the novel observations regarding the stretch effects on the curvature of cell-cell junctions and cellular overlaps highly interesting and of high fundamental novelty, in particular given that attempts to recapitulate the key features of *in vivo* capillary LECs in culture conditions have not been successful so far and that the observed isotropic stretch-induced cell shape changes have not been reported for any mammalian cell before. Indeed, despite the inherent differences between *in vitro* and *in vivo* studies (e.g. culturing ECs on unphysiologically stiff matrices (PDMS, plastic or glass)), results obtained from *in vitro* studies can be informative for studying fundamental LEC behaviour and for elucidating *in vivo* mechanisms.

comparisons of stretch effects on LECs and HUVEC in culture,

A comprehensive analysis of the effect of isotropic stretch on HUVECs is out of scope of the current study; the data was added in response to the request by this Reviewer. We fully agree that a more thorough study should be conducted to provide a truly meaningful comparison, but in our opinion this would also require *in vivo* investigations to characterise cellular overlaps in BECs. Although we believe that the current *in vitro* data, as presented, will be of value to the vascular biology community, we are willing to consider its exclusion if requested.

and the treatment of differences between their data and published data on junction classification in LECs.

In the previous revision we were specifically asked to address differences in previously utilised junction classifications, as well as to provide plausible explanations for differences in junction type distribution between our and previous studies, but we recognise that this now detracts from the key focus of the paper and have removed some of the discussion. For the reasons explained in our first revision and further elaborated in comment 6h below, we disagree with the Reviewer's opinion that previous publications (such as Jannaway et al) presents a framework on capillary LEC junction frequency and their classification, according to which our data should be aligned. We also disagree with the view that our findings on junctional heterogeneity are merely confirmatory of previous studies, as will be discussed in detail in comment 7b below. Rather than referring to absolute frequencies that are difficult to compare between different studies due to different quantification methods used, we now focus on the presence of the additional junctional categories identified (curvilinear and double junctions) and their implications to the proposed JBL model.

c. An example where additional data are needed is the claim (lines 125-127 and Fig.6) of changes in % of junction types in LECs (Fig.1g) resulting from increased interstitial fluid pressure after intradermal PBS injection, where the current data came from only n = 2 mice/group, and the significance of differences was not subjected to statistical testing (Fig.1g).

We agree. We have performed analysis on additional fluid-injected mice, and compared the data to the large dataset of 25-week-old uninjected mice generated during the previous revision. Data in **Fig. 1h** are now based on n=5 for the uninjected and n=4 for fluid-injected 25-week-old mice. Statistical analysis of the data (Mann Whitney test) indeed show no significant differences in the frequencies of the junction types in control vs injected animals. We have therefore revised the text and **Fig. 6** accordingly.

6. Inconsistencies: The revised manuscript has multiple apparent inconsistencies that need to be addressed.

a. In an innovative and informative experiment, the authors report that injection of LYVE1 antibody into ears resulted in uniform labeling of overlapping flaps of adjacent LECs that was not found after conventional IHC without permeabilization (Fig.2h). The authors reasonably interpret these findings as showing that regions of overlap permitted entry of antibody in life but not after fixation (lines 154-158). These in vivo LYVE1 antibody experiments seem to undermine and invalidate the authors' junction-based lamellipodia barrier hypothesis, whereby the overlapping flaps are anchored by barrier-forming curvilinear and double junctions (VE-cadherin + claudin-5, lines 450-463). Please revise.

This appears to be a misunderstanding of our data and conclusions, because we have not proposed a 'junction-based lamellipodia barrier hypothesis'. Instead we conclude "...continuous remodelling of cellular overlaps maintain vessel integrity **while at the same time preserving dynamic and permeable cell-cell junctions compatible with vessel expansion and fluid uptake through intercellular overlaps**" (lines 46-48). Entry through cellular overlaps is indeed in line with our model as we also show by the LYVE1 antibody injection method.

b. The authors' IHC staining clearly shows similar abundance and distribution of VE-cadherin and claudin-5 in LECs of initial lymphatics (Fig.1b). scRNA-seq analysis revealed even greater expression of Cldn5 than Cdh5 in the LECs (Suppl Fig.4a). However, the authors interpret their TEM studies as showing that adherens junctions are abundant but tight junctions are rare in LECs (lines 482-483, Extended Data Fig.2g). They then rationalize the finding by proposing that "adherens junctions in LEC...contain...tight junction molecules (lines 484-487)". This interpretation seems far-fetched next to the conventional view that adherens junctions and tight junctions are both abundant and adjacent in LECs. Please see "9. TEM studies of LEC junctions" and revise.

We have not been able to find references for TEM data showing that adherens junctions and tight junctions are both abundant and adjacent in LECs, and we would be grateful for advice from the Reviewer of the published evidence to include in our discussion.

Others⁹, and as confirmed by us, have shown that adherens junction proteins such as VE-cadherin and tight junction proteins such as CLDN5 show largely overlapping signals on an immunohistochemical level. However, despite the abundant presence of both proteins, tight junctional complexes (maculae occludentes) on the ultrastructural level (fusion of plasma membrane) remain rarely observed, whether in a focal maculae or continuous zona conformation, as reported by Leak¹. This inconsistency between immunofluorescence data and ultrastructural observations remains unclear, and we have removed speculative interpretations in the revised text:

Lines 494-496: ~~Both Leak's and our studies indicate that while adherens junctions are prevalent, ultrastructurally defined tight junctions are rarely observed in LECs^{21,37}. It is therefore unclear why Yet, immunofluorescence analysis revealed that the classical tight junction molecule CLDN5 co-localizes with VE-cadherin at the majority of LEC junctions, as reported previously^{4,6}, including in curvilinear and double junctions. This suggests that the adherens junctions in LECs may also contain some classical tight junction molecules.~~

c. The authors' claim that button junctions are too rare (20-30%, line 129) to be sites for significant fluid entry into ear skin initial lymphatics but then argue that intercellular clefts between overlapping LECs serve as the primary route for fluid and solutes entry (lines 479-480). Importantly, only 20% of LEC overlaps examined by TEM had no junctions (Extended Data Fig.2e). To explain the inconsistency, they misrepresent Leak (JCB 1971) as reporting that adherens junctions, unlike tight junctions, permit tracer entry (lines 481-482). However, this interpretation is inconsistent with Leak's description of the entry routes as "intercellular clefts of patent junctions" and "cell junctions devoid of adhesion devices, i.e., maculae occludentes and maculae adherentes (JCB 1971)." In other words, Leak described the entry routes as junction-free clefts located between discontinuous junctions (maculae). Please revise.

As described above in comment 3c, we have revised the sentence referring to Leak's findings.

Line 493-494: EM analysis showed tracer passage stopping at tight junction barriers **but freely passing through spatially separated junction-free intercellular clefts**²¹ ~~but not at adherens junctions.~~

d. The authors' report of relatively uncommon button junctions (20-30%, line 129) in ear skin initial lymphatics is inconsistent with values of about 50% reported by Jannaway et al. (reference 14). **This discrepancy needs to be addressed more thoroughly and reconciled** (see "8. Classification of junctions").

We have provided all source data for full transparency of our analysis and extensively discussed the plausible reasons for the discrepancy in the previous rebuttal letter and additional appendix. Briefly:

- Different methods of quantification: Jannaway et al. defined buttons as discontinuous VE-cadherin⁺ fragments <4.72 μm . No rationale for this precise but relatively large threshold was provided. For comparison, the average width of a cellular overlap between neighboring LECs is $\sim 2 \mu\text{m}$ and a previous study defined buttons as discontinuous segments of VE-cadherin staining of $\sim 3 \mu\text{m}$ in length and $\sim 0.5 \mu\text{m}$ in width⁹. Our quantification was based on manual scoring of individual LEC lobes using high-resolution images.

- Imaging: Jannaway et al. presented seemingly more thresholded images of lower resolution compared to ours. The Reviewer argues that this is our subjective view and we have therefore removed this comment in the revised text, see below. We provide original image data used for quantification for full transparency, which is not available from previous studies.
- Anatomical localization: Our analysis was performed at the blunt ended tips of lymphatic capillaries, earlier publications did not specify where images were acquired
- Data presentation: High spread of datapoints (10-75%), presented as a single average value in Jannaway et al, compared to SuperPlot presentation of all our data, to accurately represent data variability.

We have revised the manuscript text to avoid focus on the precise frequencies of junction types, which we believe is of secondary importance, and as we acknowledge and emphasise that previous studies have quantified these differently (lines 453-455), thus direct comparison is not meaningful. We focus on our observed high heterogeneity with previously unappreciated presence of VE-cadherin⁺ junctions along the cellular overlap borders, reminiscent of JBLs (see comment 3h), and on the characterisation of this heterogeneity at the level of individual lobes, which we now illustrate more clearly in the new **Extended Data Fig. 1c**. Along with these changes, we have also reduced the discussion on controversies and plausible explanations for differences in categorising LEC junctions (see comment 4f), as it detracts from the main scientific question. We believe that elaborating on this issue may be better suited for a review article.

We have also removed references to the question of resolution and sensitivity. Yet, in our humble opinion, the difference in the resolving power required for finer details, such as double junctions, is evident from careful examination of the published data, provided below for comparison. For this reason, we do not believe that thorough discussion on the discrepancies in the frequencies of junction types is meaningful.

FIGURE REDACTED

Whole-mount immuno-fluorescence of dermal lymphatic capillaries from 3-week-old mice (P21) for VE-cadherin. Images on the left are reproduced from the study by Jannaway et al¹⁰. On the right, panels from Fig 1c in our study.

e. The authors claim their finding of infrequent buttons is supported by published data showing that

button junctions constitute 25-30% of junctions in adult tracheal lymphatics (lines 431-432 and 528-529). However, this claim is inconsistent with data in that report (Baluk et al. JEM 2007). Please see “Specific Comment 5. Initial lymphatics of trachea and diaphragm” and revise.

The Reviewer is correct! We had indeed overlooked the details and thank the Reviewer for noting our mistake. These sentences have been removed. Please see also specific comment 5b.

f. In explaining the difference in their interpretation of button junctions in ear skin lymphatics from data on intestinal lacteals (lines 582-587), the authors provide the convenient but unconvincing argument that LEC junctions vary in different organs. Please revise.

We have not performed analysis of lacteal junctions ourselves, recognizing that detailed characterisation of different vascular beds would necessitate an extensive additional study. Rather than generalizing our findings from the skin, we considered it as fair treatment of previous data, rather than an intentional act of convenient ignorance, to acknowledge potential organ-specific differences. To further emphasize the limitation of our analysis, we have revised the sentence as follows:

Lines 597-599: Although we found **the presence of curvilinear and double junctions** similar junctional morphologies in lymphatic capillaries also **in the diaphragm and trachea in adult mice** in other adult organs, our analysis is not sufficient to generalize our findings from the skin to different organs. ~~it is important to acknowledge that the organization and remodelling of junctions may vary in different organs.~~

g. Data showing no differences in abundance of zipper junctions from 3 to 25 weeks of age (Fig. 1f) are inconsistent with subsequent statements (lines 420-431) saying the opposite: “...at three weeks of age...lymphatic capillary tips showed...the presence of continuous zipper junctions. The subsequent reduction in the abundance of zippers during further development was associated with the cessation of vessel sprouting and LEC proliferation.” Please revise.

As described in the results (lines 111-112), discussion (line 444) and methods (Supplementary Information), the quantification shown in **Fig. 1g** only includes vessel ends that showed LYVE1 staining and blunt morphology, i.e. not sprouting tips that are always associated with zippers.

The full sentence indeed reads as follows:

...we found that at three weeks of age approximately 25% of lymphatic capillary tips showed sprouting morphology associated with elongated LEC shape and the presence of continuous zipper junctions. The subsequent reduction in the abundance of zippers during further development was associated with the cessation of vessel sprouting and LEC proliferation.

For improved clarity, we have revised the second sentence (lines 440-441):

The cessation of vessel sprouting and LEC proliferation at later developmental stages was subsequently associated with the loss of zippers

h. The manuscript would be **strengthened by greater attention to aligning the authors’ new data with published data** – instead of trying to rationalize the inconsistencies and contradictions.

As mentioned above, we agree that too much focus was put on rationalizing inconsistencies and much of that discussion has now been removed. Instead we discuss that the flap valve and JBL concepts can be integrated in the sense that cellular overlaps and junctions may transition between the two states.

As explained in detail in comment 6d, direct comparison between the absolute frequencies of junction types between different studies is not meaningful due to different classification and quantification methods used. As also explained in comment 6d, we define additional junction types not described previously. For these reasons, we must respectfully disagree with the suggestion that we should align our results with the previous data in this regard. We hope the Reviewer agrees with us that the precise proportion of different junctional types we and others observe is not of principal importance, but that the steady-state junctional heterogeneity we describe in our study is of biological interest.

7. Lymphatic junction heterogeneity: Although improved from the original manuscript, **the description of the heterogeneous junctions in initial lymphatics is still subjective, misleading, and detracting from the many attributes of manuscript.**

In the interest of transparency and thus to avoid subjectivity we provide all images and quantification source data. We additionally provide new images in **Extended Data Fig. 1c** highlighting the junctional heterogeneity. To classify this enormous heterogeneity and define previously undescribed features such as double junctions as confirmatory of the previous classification of 'intermediate' junctions (as proposed by the Reviewer below, e.g. in comments 8b, 8g, and specific comments 1c, 3e), which were described as a transient state of junctions undergoing zipper-to-button transformation, would, in our opinion, be misleading. See also comments 8a and b below.

- a. Greater care and **objectivity in describing the background and interpreting images of junctions in the context of previous publications** would strengthen the manuscript without loss of novelty.
- b. **Characterization of the heterogeneity of discontinuous junctions of initial lymphatics is represented as a novelty of the manuscript, whereas in reality it confirms earlier work.**
- c. Images in all papers illustrating discontinuous junctions in initial lymphatics, including references 4, 6-8, 13-15, 32, and 45, show junctional heterogeneity. This heterogeneity is described and quantified in Jannaway et al. (reference 14).

We have now clarified that junctional heterogeneity as such has been observed in previous studies, in the developing tissues, and referred to previous publications (lines 450-451).

These questions are addressed in more detail in comment 8a and b below, when discussing the classification of LEC junctions. Briefly, 'intermediate' junction was indeed introduced previously to describe a transient state of junctions transforming from zippers to buttons⁷, with the latter considered a mature phenotype of functional lymphatic capillaries, rather than a junction type that continues to be present in the mature vasculature in adult mice (see image extracted from Yao et al in comment 8b). In addition, as we indeed pointed out in our previous response to the Reviewers, images illustrating discontinuous junctions extending to the tip of the lobe, including double junctions, are clearly visible in the figures of some previous publications, even in the mature vasculature, yet these were not described and discussed in the text in these publications. Jannaway et al. included quantification of 'intermediate' junctions in the ear dermis of juvenile mice, but this was based on arbitrary cut-off values of VE-cadherin⁺ fragment length (>4.72 μm but <7.85 μm), and lacked LYVE1 staining to visualise the relationship of these segments to cell morphology and borders. Their images (see comment 6d) do not resolve the details of junctional morphology described in our study, including double junctions with resemblance to JBLs.

- d. Statements in the manuscript that disregard this background should be corrected. Examples are: "...we uncovered a spectrum of junctional configurations..." (lines 82-83), and "We observed mainly

extended linear VE-cadherin+ adherens junctions...rather than the expected predominance of discontinuous buttons...” (lines 96-98).

We are uncertain about the Reviewer’s proposed revision in this case. As explained in the next point, the description of the different capillary LEC junction morphologies, including double junctions, in mature lymphatic vasculature is a novel and unexpected aspect of our study. We had expected a predominance of buttons based on previous literature. Nevertheless we have now removed this statement and the revised sentence is as follows:

Lines 97-100: In addition to the classical button junctions oriented parallelly along the sides of junction-free regions⁴, we frequently observed linear, either continuous or discontinuous, VE-cadherin+ adherens junctions extending to the tip of the lobe borders (**Fig. 1a, Extended Data Fig. 1a**).

We also acknowledge that junctional heterogeneity was previously described in developing vessels: Lines 450-453: Of note, previous studies have recognized junctional heterogeneity and classified ‘intermediate’ or ‘transforming’ junctions^{6,9}, which likely represent junction types identified here. However, intermediate junctions were described at developmental stages and were previously considered a transient state undergoing zipper-to-button transformation⁶.

e. As the authors naturally build on earlier work, they should be more even-handed in distinguishing what was already known from what is new in their manuscript. A more objective alternative to lines 82-87 is described in Specific Comment 3e.

In Specific Comment 3e, the Reviewer suggests us [**bold below**] to summarise our findings as confirmatory of previous findings, showing a ‘heterogenous mixture of buttons and other discontinuous junctions’ as follows: *“In the current study, we used novel mouse models and approaches to characterize endothelial cells of initial lymphatics in mouse ear skin and mechanisms responsible for the distinctive oak leaf shape of the LECs. We identified new features of LEC overlap in mosaic mice and **confirmed the heterogeneous mixture of buttons and other discontinuous junctions between the LECs, as shown previously** (references 4, 6-8, 13-15, 32, 45). We then determined the contributions of cortical actin, microtubules, and Cdc42 in regulating the oak leaf shape of LEC.”*

Since we also observe continuous VE-cadherin-based junctions in individual LEC lobes (e.g. **Extended Data Fig. 1c**), it would be misleading to state that only ‘buttons and other discontinuous junctions’ are present. In addition, equating our findings on the heterogeneity of the junctions and the identification of new junction types to representing confirmation of previous findings would be misleading, as discussed further in comment 8b below.

To incorporate the Reviewer’s suggestions, but to maintain accuracy and focus on the key aim and novel aspects of the study, we have revised the sentence as follows:

Lines 82-88: In the current study, we used novel mouse models and experimental approaches to characterize endothelial cells of lymphatic capillaries in mouse ear skin, elucidating the mechanisms and drivers underlying their attributes and significance for lymphatic vessel physiology. We identified new features of LEC cytoskeleton and their dynamic overlaps and uncovered a previously unappreciated spectrum of VE-cadherin-based junctional configurations in capillary LECs across juvenile and adult stages. Our results point to the role of dynamic cytoskeletal regulation of LEC shape in controlling the maintenance of vessel integrity and function.

8. Classification of LEC junctions: The authors’ classification of junctions in initial lymphatics

continues to be problematic.

a. The authors changed their classification of non-button LEC junctions in initial lymphatics from 5 types of continuous junctions (original Fig.5c) to 2 types of discontinuous junctions (curvilinear and double) and 3 types of continuous junctions (curvilinear, double, and zipper). This change reflects the subjectivity of classifying these junctions, is confusing, and does not make sense.

We have provided additional images of capillary LEC junctions at the level of a single lobe, together with drawings closely depicting the immunofluorescence signal, in **Extended Data Fig. 1c**. This is to better illustrate the spectrum of VE-cadherin⁺ junctional configurations without imposing subjective classification. We argue that simplifying such heterogeneity into strict classification has limited value. Nevertheless, our categorisation was intended to extract the main distinctive patterns as a means to convey the main new findings. These categories broadly align with the description of the ‘oscillatory stages’ of JBL function in zebrafish (see comment 3h). We have now revised the schematic drawings (**Fig. 1g, Fig. 6**) to illustrate that curvilinear and double junctions can be discontinuous or continuous.

b. Regardless of what the junctions are called, the most important distinction is whether junctions are discontinuous or continuous. The names “buttons” (discontinuous junctions) and “zippers” (continuous junctions) were introduced to make this distinction (reference 4). **The term “intermediate” was subsequently introduced to describe discontinuous junctions that extend around more of the LEC perimeter than buttons. In contrast, “zippers” are continuous junctions that completely surround the cell perimeter**, as in collecting lymphatics and blood vessels.

Our understanding is that the term ‘intermediate’ junction was introduced to describe a transient state of junctions transforming from zippers to buttons⁷, with the latter considered a mature phenotype of functional lymphatic capillaries, rather than a junction type that continues to be present in the mature vasculature in adult mice. See text and figure excerpts from Yao et al (Am J Pathol 2012)⁷ below:

“Intermediate junctions increased rapidly just before birth and then gradually decreased during the next four weeks as they transformed into buttons”.

“Several lines of evidence from studies of developing lymphatics were consistent with the concept that buttons form from zippers via intermediate junctional forms.”

FIGURE REDACTED

From Yao et al, Am J Pathol 2012: ‘Intermediate junctions increased rapidly just before birth and then gradually decreased during the next four weeks as they transformed into buttons.’

Secondly, as we indeed pointed out in our previous response to the Reviewers, images illustrating discontinuous junctions extending to the tip of the lobe, including double junctions, are clearly visible in the figures of some of the previous publications, even in the mature vasculature, yet these morphologies were not described or discussed in the text in these publications. Jannaway et al. included quantification of ‘intermediate’ junctions in the ear dermis of juvenile mice, but this was based on arbitrary cut-off values of VE-cadherin⁺ fragment length (>4.72 μm but <7.85 μm), and lacked LYVE1 staining to visualise the relationship of these segments to cell morphology and borders. Their

images (see comment 6d) do not resolve the details of junctional morphology described in our study, including double junctions with resemblance to JBLs.

We have now clarified that junctional heterogeneity has been observed in the developing tissues in previous studies (lines 450-453). However, considering the available data, it would be incorrect and misleading to describe our findings on junctional heterogeneity merely as confirmatory of previous data.

We would also like to make a minor note that we cannot find the criteria for a zipper to completely surround a LEC perimeter used in previous quantifications^{7,10-12}. For example, Jannaway et al. measured the length of VE-cadherin positive segments and determined any segment >7.85 μm in length as a zipper.

c. Now, the authors' propose a revised classification where some curvilinear and double junctions are discontinuous and others are continuous (lines 796-804, Supplement, Image quantification). This implies that zippers are not the only junctions that completely surround LECs.

Correct.

d. Apparently the authors' concept of discontinuous or continuous junctions has morphed from entire LECs – as used in previous publications - into individual lobes of LECs (Supplement, Image quantification). Therein lies the source of a problem.

We now clarify that quantification is indeed done at the level of individual lobes (lines 39, 117-118, 445, **Fig. 1g**). However, we are unsure why this poses a problem. To the best of our knowledge, previously published studies have also not considered the individual cell as the unit for junction classification. Instead, junctions have been analyzed as total number, area or length per a fixed area or field of view. For example, as described above in comment 7b, a criteria for a zipper to completely surround a LEC perimeter has not been used in previous quantifications^{7,10-12}.

e. A unique feature of initial lymphatic LECs is the presence of discontinuous junctions that permit entry of fluid, solutes, chylomicrons, etc. The presence of "intermediate" junctions indicates that the proportion of LEC perimeter with discontinuous junctions varies from cell to cell, but few LECs of initial lymphatics are completely surrounded by continuous junctions (zippers).

We fully agree with this. See the previous comments.

f. For the authors to solve their classification problem and meaningfully interpret their data for individual lobes (Fig. 1f, 1g), additional data are needed for the (i) % of individual LECs with one or more discontinuous junctions; (ii) % of the perimeter of individual LECs with discontinuous junctions; and (iii) frequency of discontinuous button, curvilinear, and double junctions expressed as % of lobes of individual LECs. Please use an LEC sample size to accommodate the cell to cell variability and give the mean \pm SE for each value for adult mice (n = 5 mice).

We quantified junctional frequencies at the level of the individual capillary vessel end, considering the lymphatic capillary as the functional unit, rather than the individual LEC.

Without mosaic cell labelling to identify the borders of individual LECs, the requested task is challenging and unprecise. Additionally, in our analysis, thin image stacks were acquired at high resolution, and therefore the full perimeter of a cell was rarely captured. To the best of our knowledge, these kind of data analysis has also never been provided in previous studies where

junctions were analyzed as total number, area or length per a fixed area or field of view (see comments 6d and 8b).

g. Classification of discontinuous junctions as “button”, “curvilinear”, or “double” is not a problem, as long as only discontinuous junctions are so classified. However, the authors should explain that they **subdivided junctions previously called “intermediate” into two groups, curvilinear and double**. The term **“zipper” should be restricted to continuous junctions that completely surround individual LECs, for consistency with the original definition**.

As we discuss above (comment 8b), describing our findings of new junction types as confirmatory and merely a subdivision of a previously defined category would be incorrect. Intermediate junctions have previously been defined as a transitional state present only during development and inflammation. Using the term “intermediate” for the curvilinear or double junctions that we show persist and are prevalent in normal adult tissues would be misleading in our opinion. Instead, we think it is more accurate to call them them by their morphological attributes.

As described above in comment 7b, we are puzzled by the request that “The term “zipper” should be restricted to continuous junctions that completely surround individual LECs, for consistency with the original definition”, since our understanding is that the criteria for a zipper to completely surround a LEC perimeter was not used in previous quantifications^{7,10-12}.

h. The corresponding text, figures, including summary Fig.6, and figure legends should be revised to accommodate the requested new data and revised descriptions and interpretations.

We have revised the text and figures as described in comments 8a-c above.

9. TEM studies of LEC junctions:

a. TEM values of 17% tight junctions and 83% adherens junctions (Extended Data Fig.2f, 2g) appear to reflect technical limitations of the TEM images (Extended Data Fig.2c) more than the actual % of junctions.

b. The authors’ TEM data appear to suffer from technical limitations where the identity of junctions cannot be determined with certainty in some panels of Extended Data Fig.2c A1-A5 (top row left to right) and B1-B5 (bottom row left to right) where plasma membranes were cut obliquely or sections had a thickness (60-70 nm, line 830) greater than optimal for resolving the trilaminar structure of unit membranes (<60 nm).

c. The authors do not acknowledge that some regions marked by red arrowheads in panels A1-A3, A5, B3 in Extended Data Fig.2c are uninterpretable for the presence of tight junctions and some regions marked by blue arrowheads in panels B2, B4 are oblique sections uninterpretable for any junctions.

d. In addition to the section plane and thickness issues, the trilaminar structure of unit membranes was difficult to visualized in the absence of en-block staining with uranyl acetate during tissue processing (lines 816-833). See Fig.11 in Leak, JCB 1971 and Fig.21 in Brightman & Reese, JCB 1969, where uranyl acetate en-block staining was used, for comparison.

We have removed some of the the TEM data and all statements about tight junctions in the results section, as suggested by the Reviewer (see comment 9h).

We agree that our TEM data was limited in value and depth. However, we note that previous TEM studies also report higher abundance of adherens junctions compared to tight junctions. For example:

Lauweryns and Cornillie 1984¹³:

At the intercellular junctions the lateral cell margins usually form simple overlappings (Fig. 6). No

specialized attachment devices such as desmosomes were seen. Along these overlappings the intercellular approximation may be very close, with an extracellular space of only 8-10 nm, although local gaps of about 60 nm are frequently present. Cytoplasmic densifications may occur at the luminal side of intercellular junctions, **although membrane fusions or tight junctions do not occur.**

Leak 1971¹:

In some of the junctions formed by the extensive overlapping of endothelial cells, there may be one point and, occasionally, several points along the length of the intercellular cleft where plasma membranes are closely approximated. Observation of these areas at higher magnifications reveals that the apposing membranes are held together by a macula adherens (Figs . 2-4, 6, 8). **Occasionally, areas of close apposition were observed where points of the intercellular cleft are indeed obliterated by the fusion of external leaflets,** as illustrated in Figs. 11 and 15 a.

Leak and Burke 1966¹⁴:

The zonula occludens (tight junction) was observed near the lumen (fig. 12). However, many junctions were observed in which this tight junctional complex was absent. The macula adhaerens (desmosomes) and the zonula adhaerens were encountered with some regularity between opposing membrane wall surfaces (figs.3, 6, 7, 12, 13). The zonula occludens (tight junction) is formed by a close apposition of the surface membranes of adjacent endothelial cells. The intercellular space contains a central membrane that is formed by the fusion of the two outer membranes of the two closely apposed unit membranes (fig. 12). The second element of the junctional complex, the zonula adherens appears as two dense leaflets that are separated by a distance of about 200 Å (figs. 7, 12, 13). **The most common adhesion device observed between apposing surface membranes of the lymphatic endothelial cells is the macula adhaerens (desmosomes).**

e. Confocal images (Fig.1b, Extended Data Fig.6b) show that VE-cadherin and claudin-5 have largely identical distributions, consistent with previous work (reference 4). Therefore, tight junctions would be expected to be at least as numerous as adherens junctions. Expression data for Cldn5 than Cdh5 mRNA (Suppl Fig.4a) are consistent with the IHC similarity.

While high mRNA expression does not necessarily correlate with high protein expression, we agree that the observed discrepancy between the immunofluorescence data and TEM data (see the previous comment) is surprising. Revised text is as follows:

Lines 494-498: Leak's studies also indicate that while adherens junctions are prevalent, ultrastructurally defined tight junctions are rarely observed in LECs^{21,37}. It is therefore unclear why immunofluorescence analysis revealed that the classical tight junction molecule CLDN5 co-localizes with VE-cadherin at the majority of LEC junctions, as reported previously^{4,6}, including in curvilinear and double junctions.

See also comment 6b.

f. The authors attempt to reconcile the discrepancies by speculating that adherens junctions contain tight junction proteins like claudin-5 (lines 486-487). They also argue that deletion of claudin-5, "the second major component of LEC junctions" (lines 472-473), did not disrupt LEC junctions (lines 287-288), but fail to acknowledge that button junctions contain multiple tight junctions proteins, including ESAM and JAM-A, in addition to claudin-5 (reference 4).

We previously stated 'It is possible that multiple cell-cell and cell-matrix adhesion receptors work cooperatively and compensate for each other's functions in maintaining lymphatic vessel integrity'.

We have now specified the tight junction proteins ESAM and JAM-A as candidate molecules citing ref 4 (line 488).

g. From their interpretations and claims, the authors seem to use infrequency of LEC tight junctions to support their lamellipodia hypothesis. However, the evidence argues otherwise and raises questions over the authors' interpretations and whether the TEM studies were technically suitable and sufficiently rigorous to determine the abundance of tight junctions.

As discussed, we have removed some of the the TEM data as per the Reviewer's suggestion (comment 9h). Our hypothesis is based on the dynamics of LEC lobes, reflected in the varied junctional morphologies, and not the infrequency of tight junctions.

h. Although the optimal solution is for the authors to obtain data from new TEM images that meet the standards of their predecessors cited above, an acceptable alternative would be to (i) delete Extended Data Figs.2e and 2g that show values for the number and % of tight junctions and adherens junctions; (ii) delete or relabel Extended Data Fig.2f to show the intercellular distance values do not apply to tight junctions. Unless better documented, values >20 nm are unlikely to fit with adherens junctions; (iii) replace all TEM images in Extended Data Fig.2c with three high mag images of LECs: a tight junction, an adherens junction, and no junction; (iv) delete from the text (lines 482-483 and elsewhere) all descriptions of TEM data for the number or % of junction type; and (v) acknowledge the technical limitations faced in visualizing LEC junctions by TEM.

We have followed the Reviewer's suggestion and removed Extended Data Fig. 2e (number of electron densities per overlap) and 2g (% of AJ/TJ). We have also removed panel 2f (intercellular distance) in the same Figure as it has limited value alone. We have kept a modified Extended Data Fig. 2c to simply illustrate the variability in cellular overlaps and indicate electron densities without specifying the junction type. In our opinion, these images meet the quality required to support the statements made.

10. Reasons for discrepancies with published data on lymphatic junctions: The authors' explanation for why their data for lymphatic buttons and other junctions differ from corresponding data in multiple previous papers is incomplete and unconvincing.

a. The Introduction (lines 57-76) should set out more background on what is known from previous work on junctions in initial lymphatics (cite references 4, 7, 8, 13-15, 32, 45).

As described in comment 4a, we have revised as follows:

Lines 63-65: Dynamic transitions between the two junction types occur in capillary lymphatic vessels during development and in disease states, **with several molecular players identified**⁶⁻⁹.

b. In the context of this background, the specific aims (lines 78-87) should more clearly define the intended purpose for analyzing and classifying LEC junctions in this study in the context of what is already known. See Specific Comment 3e.

The original aim of this study was not to reassess the composition or classification of capillary LEC junctions. Our focus was on understanding the mechanisms by which capillary LEC monolayer maintains its integrity. This initial research question prompted us to analyze cell-cell junctions in relation to cell shape and cytoskeletal organisation. We greatly extended our analysis of junctional configurations based on the valuable feedback from this Reviewer, which has strengthened our data. To incorporate the Reviewer's suggestions, but to also maintain the focus on the key aim and novel aspects of the study, we have revised the sentence as follows:

Lines 82-88: In the current study, we used novel mouse models and experimental approaches to characterize endothelial cells of lymphatic capillaries in mouse ear skin, elucidating the mechanisms and drivers underlying their attributes and significance for lymphatic vessel physiology. We identified new features of LEC cytoskeleton and their dynamic overlaps and uncovered a previously unappreciated spectrum of VE-cadherin-based junctional configurations in capillary LECs across juvenile and adult stages. Our results point to the role of dynamic cytoskeletal regulation of LEC shape in controlling the maintenance of vessel integrity and function.

c. The authors claim that their IHC and confocal imaging techniques had greater sensitivity and resolution than previous studies (lines 432-436). But this implies their data are more accurate and believable. However, this claim is not supported by evidence of greater sensitivity and resolution. It is unclear how sensitivity and resolution would be compared among studies.

As discussed above (comments 4d and 6d), we have removed reference to the question of resolution and sensitivity in the revised manuscript.

d. Differences in developmental age are mentioned but unlikely to be factors because the authors' own data show no difference between age 3 and 25 weeks (Fig. 1f, Extended Data Fig. 1b).

At 3 weeks of age, a significant proportion of dermal lymphatic vessels have active sprouts (**Fig. 1d**) that are always associated with an elongated LEC shape and zipper junctions (**Fig. 1c**). Inclusion of all capillary vessels including sprouting capillary ends, and not restricting the analysis to blunt ended vessels, may thus affect the results. We believe that this an important consideration for the lymphatic vascular field and we therefore decided to keep the statement (lines 442-443). However, during the revision, the discussion on discrepancies was deemed unnecessary and the issue of developmental age in this context was removed accordingly.

e. A factor not considered is the exposure of lymphatics to mechanical trauma by removing and cutting ear skin from cartilage BEFORE fixation, instead of fixing the tissue by vascular perfusion before removal. Manipulation of ear skin before fixation could contribute to morphological changes in lymphatics that would be prevented by vascular perfusion fixation (reference 4). Please address these issues in the Discussion.

As already addressed and described in our previous revision, we observed the same junction morphologies in ear tissue, which was fixed after dissection from the cartilage, or by vascular perfusion followed by dissection. We noticed that the description of vascular perfusion was omitted from the methods section, but has now been added (lines 1000-1006). Of note, we also observe the same junction morphologies in the diaphragm and trachea that are not exposed to mechanical stress during tissue preparation. See also specific comment 5c.

f. Differences in classification of lymphatic junctions in this and previous studies should also be considered in more detail, acknowledging the subjectivity and pros and cons of each approach.

Previous studies classify junctions as buttons and zippers, with some studies also classifying a transitional type of intermediate junctions. In our study, in addition to buttons and zippers, we define curvilinear and double junctions as additional entities, which are reminiscent of JBLs and persist in adult tissue.

We acknowledge and emphasise that previous studies have quantified junctions differently (lines 453-455), precluding direct comparison. As discussed in several comments above, we believe that the focus on the precise frequencies of junction types defined in different studies is therefore of

secondary importance. We have clarified that junctional heterogeneity has been observed in previous studies, in the developing tissues (lines 450-453). Without invalidating previous studies and classifications, we focus on our observed high heterogeneity in mature adult vessels with previously unappreciated presence of VE-cadherin⁺ junctions along the cellular overlap borders, reminiscent of JBLs (see comment 3h), and on the characterisation of this heterogeneity at the level of individual lobes, which we now illustrate more clearly in the new **Extended Data Fig. 1c**. Along with these changes, we have reduced the discussion on controversies and plausible explanations for differences in categorising LEC junctions (see comment 4f), as it detracts from the main scientific question. We believe that elaborating on this issue may be better suited for a review article.

g. Regardless of the terms used, the classification should distinguish discontinuous junctions (maculae) from continuous junctions (zonulae) in LECs.

We have revised the schematic drawings in the main figures to illustrate that curvilinear and double junctions can be discontinuous or continuous. See comment 3e, and specific comments 2a-c below.

11. LEC and HUVEC stretched *in vitro*: The *in vitro* studies of isotropic stretch described in the revised manuscript are strengthened by the addition of HUVEC for comparison to LECs. However, the comparison is limited by several problems:

a. The studies of stretch-induced shape changes in LEC overlap deserve more thorough consideration of the underlying assumptions. The presumption that studies of LEC stretch *in vitro* mimicked the process that initial lymphatics undergo during fluid entry is insufficient.

As discussed in comment 5b, we have now mentioned that the exposure of LECs to isotropic stretch *in vivo* is based on a hypothesis (lines 341-344), supported by compelling data from plant puzzle cells, which identify isotropic stretch as an upstream mechanism influencing their cell shape, that we have tested using a simplified *in vitro* system. We acknowledge that the parameters used for these experiments are not grounded in real measurements (lines 518-519), and that the model does not accurately replicate *in vivo* conditions and phenotypes (line 523-524). Despite the limitations, we consider the novel observations regarding the stretch effects on the curvature of cell-cell junctions and cellular overlaps highly interesting and of fundamental novelty, in particular given that attempts to recapitulate the key features of *in vivo* capillary LECs in culture conditions have not been successful so far. Indeed, despite the inherent differences between *in vitro* and *in vivo* studies (e.g. culturing ECs on unphysiologically stiff matrices (PDMS, plastic or glass)), results obtained from *in vitro* studies have been shown to be informative for studying fundamental cellular behaviour and for elucidating *in vivo* mechanisms.

b. What is known about the relative contributions of stretch and expansion without stretch as collapsed lymphatics fill under physiologic conditions? Is the process different from venules that collapse and expand moment by moment during positional changes? What process of lymphatic biology is mimicked by cyclic stretch for 100 seconds per cycle for 14 or 22 hours?

We acknowledge that the degree and frequency of exposure of LECs to isotropic stretch *in vivo* is not known, and parameters used are not grounded in real measurements (line 518-519). While few studies on the effect of isotropic stretch on mammalian cells are available, we considered parameters previously used for aortic ECs¹⁵ (0-5% and 0-10% cyclic stretch at 1Hz for 1, 3, 9 and 16 h), reasoning that a lower stretch frequency is applicable to LECs. The chosen value of 100 sec per stretch cycle equals 1.08 motor steps per second and 0.01 Hz on our device. Our findings are indicative of the relevance of isotropic stretch in LECs, which will open up new avenues for investigation.

We are unsure what the Reviewer means by stretch versus expansion without stretch. The expansion caused by the filling of a previously collapsed lymphatic vessel lumen would be expected to result in isotropic stretch in the endothelial monolayer, as the vessel lumen expands uniformly in all directions. Vessel collapse can be considered 'stretch release' to which cells would also be expected to respond.

See also comment 5b.

c. The subtlety of shape changes in LECs after 22 hours of stretch shown in the images (Fig.5g) should be acknowledged. The description of stretch-induced changes (lines 507-509) does not faithfully reflect the subtle changes shown by the images. Please revise.

Stretch applied consistently had a quantifiable effect on cellular overlaps and cell-cell contact linearity, but in order not to exaggerate these effects in the results description, we have removed the terms 'strikingly' and 'extensive':

Lines 519-521: ~~Strikingly, ... cyclic isotropic stretch for 0.01 Hz (100 seconds per cycle) over 14 hours induced extensive cellular overlaps reminiscent of the lobular overlaps observed in vivo; along with increased curvature of cell-cell junctions at 22 hours borders.~~

d. The conspicuous broad overlap of lobes in control LEC in vivo (Fig.2i) differs markedly from the narrow overlap of stretched LECs in vitro, except in focal regions (Fig.5g). However, the overlap measurements indicate the opposite (~2.4 μm in vivo, Fig.2j vs ~3.1 μm in vitro, Fig.5i). These differences are inconsistent with the statements on lines 507-509. Please revise.

We have revised the sentence as described above (comment 11c).

Although the overlaps observed *in vitro* do not fully recapitulate the morphology of those *in vivo*, we note similar heterogeneity with some overlaps lacking VE-cadherin while others showing weak VE-cadherin staining or 'double lines' of VE-cadherin⁺ junctions at the overlap borders, as shown in the new **Extended Data Fig. 7b**. See comment 11h below.

Of note, we observe narrowing of overlaps and increase in linear junctions with increased confluency, corresponding to a longer time in culture (compare overlaps in unstretched cells at 14 h (**Fig. 5e**) vs 22 h (**Fig. 5g**)), in agreement with previous findings¹⁶. While the absolute measure of overlap width is thus dependent on the culture conditions, and not directly comparable to the *in vivo* condition, importantly under both confluencies isotropic stretch increases overlap width compared to control unstretched conditions in cultured LECs, supporting the robustness of the observation.

e. Shape changes in the LEC monolayer shown in Supplementary Movie 7 after Cdc42 activation are more subtle than implied by the description (lines 359-362, 457-459). They are also not remotely reminiscent of oak leaf LECs in vivo. These discrepancies raise questions over whether the description and data in Fig.5k are truly representative of cells in the monolayer. Please align the text and Fig.5k more closely with what readers can see in the movie.

We have revised the text to provide clearer description of the results, including additional important experimental details. We have also provided a new movie of a zoomed region of an optogenetically activated LEC (**Supplementary Movie 8**), to better illustrate the rapid dynamic CDC42-driven cellular protrusion described in the results.

Revised text, lines 363-371: Conversely, activation of CDC42 induced the formation of cellular protrusions in LECs expressing an optogenetically recruitable RhoGEF for CDC42 activation, OptoISTN1³² (**Fig. 5k**). Time-lapse imaging of confluent monolayers showed dynamic protrusions, reminiscent of junction-based lamellipodia¹⁸, in OptoISTN1-expressing LECs (yellow) following local optogenetic activation (indicated by the blue boxed region) (**Supplementary Movies 7 and 8**). These protrusions extended beyond VE-cadherin⁺ junctions, visualised with the live labeling antibody (**Supplementary Movie 8**). Extraction of the outlines of individual OptoISTN1-expressing LECs showed a 20-30% increase in the cell area within 10 min of photo-activation (**Fig. 5k**). Control cells expressing the plasma membrane localised optogenetic recruitment tool, the improved light-induced dimer (iLID), alone (data not shown and ³²), as well as OptoISTN-expressing LECs not exposed to photoactivation (**Fig. 5k**) did not show a response.

We did not imply or state that this approach generates ‘oak leaf LECs’ as observed *in vivo*; the data merely shows the ability of active CDC42 to induce lamellipodia-like protrusions and increase in monolayer resistance.

f. The statement, “Together, these results show that junction-based lamellipodial protrusions promote LEC monolayer integrity (lines 371-372)” is not justified, as the reported correlation of shape change and monolayer resistance does not prove cause and effect. Please revise.

The statement is based on the finding that acute activation of CDC42 leads to formation of junction-based lamellipodia both when applied on individual cells, or on entire monolayers when it also induced an instantaneous increase in monolayer resistance (**Fig. 5k, l**). As shown by Mahland et al⁶, increase in electric resistance, measured via electric cell-substrate impedance sensing, can be directly correlated to endothelial cell-overlap extensions. Conversely, CDC42 inhibition prevents the formation of stretch-induced lamellipodia and leads to rupture of monolayer integrity upon stretch. We agree that the latter data on CDC42 inhibition does not explicitly prove cause and effect, and we have revised the sentence as follows:

Lines 380-382: ~~Together, t~~These results show that junction-based lamellipodial protrusions are associated with LEC monolayer integrity and barrier strength, similar to what was recently reported for HUVECs³².

g. The interpretation that stretch increases overlap of LECs but not HUVEC in vitro (lines 507-509, 515-517) is misleading because HUVEC already had more overlap before stretch (~3.6 μm , Extended Data Fig. 10b) than LECs had after stretch (~3.1 μm , Fig. 5i). The conspicuous difference in overlap of HUVEC compared to LECs, with or without stretch, is evident in the images (Fig. 5g vs Extended Data Fig. 10b). Please revise the descriptions to reflect these differences and address concerns over the meaningfulness of the LEC/HUVEC comparison.

We now emphasise that HUVECs possess large overlaps already at baseline conditions:

Lines 393-396: HUVECs exhibited **larger** irregular cellular overlaps, formed by a reticular network of VE-cadherin⁴³, and numerous stress fibers under baseline conditions **as compared to LECs**, but the total cellular overlap area remained unaltered upon stretching (**Extended Data Fig. 10a, b**).

As discussed in comment 5b, although we believe that the data, as presented, will be of value to the vascular biology community, we are willing to consider removing it.

h. Another difference that deserves comment is that most overlaps of “lobes” in stretched LECs lack

VE-cadherin staining (Fig.5f, 5g) and do not resemble LEC lobes *in vivo* (Fig.1a, 1c) or overlaps of HUVEC that have uniform VE-cadherin staining (Extended Data Fig.10a). These differences weigh against the relevance of shape changes of LECs *in vitro*. Please address.

To better illustrate the pattern of VE-cadherin distribution in stretch-induced LEC overlaps *in vitro*, we provide higher magnification images showing heterogeneity with some overlaps lacking VE-cadherin while others showing weak VE-cadherin staining or 'double lines' of VE-cadherin⁺ junctions at the overlap borders (**Extended Data Fig. 7b**). In HUVECs, VE-cadherin staining was not uniform, but a reticular pattern was observed, as stated (line 394).

See also comment 11c; we have revised the sentence describing the stretch-induced overlaps for added clarity.

i. Images and measurements for LECs and HUVEC stretched *in vitro* should be shown side-by-side in the same figure to make it easier for readers to compare the two cell types.

Given the density of the main figures and that data on HUVECs is not within the central focus of the paper, we consider the current presentation of data to be appropriate.

12. Statistics: Please address the following issues:

a. The authors should specify the statistical test used for graphs with three or more groups that show no P values (Figs.1e, 1f, 2d, 2f, 4k). Please confirm that these comparisons were made by ANOVA, as described in the Methods, and that Student's t-tests were used only when the dataset included two groups. Absence of statistical significance should be marked as such.

As stated in the methods and also specified in the respective figure legends, comparisons of three or more groups were made by ANOVA followed by multiple comparisons. We removed non-significant p-values at the request of another Reviewer, but have now included the statistics in the Source Data files.

b. Violin plots are confusing when added to graphs that show all individual values and should be removed unless clearly justified. As the individual values show the actual distribution, the empirical distribution shown by violin plots is unnecessary. Violin plots also overlap and obscure some data points and have the distraction of extending above and below the actual data. Prism statistical software recommends not using violin plots with plots of individual data points.

We have removed violin plots, and present individual data points as a scatter plot with mean \pm SD.

c. Showing medians (e.g., Fig.2d, 2e, 2f, 2j, 4f, 4h, 4k) instead of means implies the data are not normally distributed, yet parametric statistical tests were used. Please justify and change the statistical test or change medians to means.

We have removed violin plots, and present individual data points as a scatter plot with mean \pm SD.

d. In the analysis of junction types, it is unclear why Individual datapoints are presented as weighted averages per mouse instead of means. Please change to conventional means or justify the use of weighted averages and describe the weighting procedure.

As we have quantified the frequency of junctions per capillary end we chose to use a weighted average instead of a normal average to present this data at the level of individual animals. A weighted average is the average of values which are scaled by importance (number of observations in our case). As the number of accurately classifiable junctions differed greatly between different images, we opted to

normalize the data against the total number of junctions analyzed in that image (weight) as well as the total number of junctions analyzed per mice (sum of all the weights); this procedure is termed weighted average. If using conventional means the data would be skewed by images having low number of junctions analyzed. To illustrate the reason we provide one extreme fictional example below:

Mouse 1	Total junctions analyzed in picture	Zipper junctions	other types of junctions	Frequency of zippers (%)
Capillary end 1 (image 1)	2	1	1	50
Capillary end 2 (image 2)	30	1	29	3,333333333
Capillary end 3 (image 3)	30	2	28	6,666666667
			Average zipper frequency (%)	20
			weighted average zippers (%)	6,451612903

While a similar result would have been obtained by calculating the total frequencies per mouse, this approach allowed us to draw conclusions on junctional frequencies at the level of the individual capillary analyzed, while at the same time being able to accurately report on individual mice. This has now been clarified in the supplementary methods.

e. Where values are normalized, specify the reference used for normalization, e.g., Fig.4l.

The explanation is added in the respective figure legends:

Fig. 4l (tracer clearance): Intensity values were normalized to represent the percentage of the initial intensity at timepoint 0.

Fig. 5h (junction linearity): Junction linearity values were normalized to represent the percentage of the average of controls.

We noticed that the quantification of junction linearity was not described in the methods section in Supplementary Information, but this has now been added.

f. The number of mice per group should be increased for data reported for only 2 mice/group.

As requested, the number of mice have now been increased, see comment 5c.

13. Response to reviewers' comments: The current Responses to Reviewers' comments gives much greater attention to explaining the authors' views than to describing the changes made to address reviewers' comments. Please take to heart that the comments are intended to strengthen the manuscript by alerting the authors to issues likely to be problematic for readers. If given the opportunity to prepare another revision, the authors would benefit from focusing their responses on the changes made in the text and figures to address each comment and identifying where in the manuscript these changes were made (line numbers, figure numbers). Lengthy justifications of the authors' preferred views, instead of focusing on the changes made, are unnecessary and work against the opportunity to strengthen the manuscript for a broad readership.

We respect this view and hope that our responses are now satisfactory in this regard.

Specific Comments

1. Junctions on entire LECs versus individual lobes: Values in Fig.1f, 1g reflect the percent of each junction type determined by "numbering of individual lobes of...LECs and subsequent categorizing of lobe-associated junctions based on VE-cadherin signal (Supplement, page 2)."

We now clarify that our analysis of junctional heterogeneity was done at the level of individual lobes (lines 39, 117-118, 445, **Fig. 1g**).

a. Values for junctions on individual lobes do not inform the junctional composition of entire LECs on which barrier function is dependent.

As described in comment 8f, we calculated the junctional frequency per individual capillary vessel end, considering the lymphatic capillary as the functional unit, rather than the individual LEC.

b. The Methods (lines 796-804, Supplement: Image quantification) describe curvilinear and double junctions as continuous or discontinuous, but this distinction has little significance when applied to individual lobes instead of entire LECs.

We understand the Reviewer's comment referring to lack of significance if considering functionality with an individual LEC as a functional entity. However, as discussed in comment 8f, we believe that our definition is an accurate description of the heterogeneity of the junctions, which is now better illustrated in **Extended Data Fig. 1c**. We have also revised the schematic drawings in the main figures (**Fig. 1g**, **Fig. 6**) to illustrate that curvilinear and double junctions can be discontinuous or continuous.

c. When entire cells are considered, most LECs illustrated have regions of discontinuous VE-cadherin staining, which fits the original description of discontinuous LEC junctions, regardless of what they are called (maculae, buttons, curvilinear, etc.).

Indeed, we frequently observe a variable level of discontinuity in VE-cadherin staining, reflecting high heterogeneity in the organisation of adherens junctions in the mature vasculature, which is one of the findings of our study. The double junctions and curvilinear junctions described here do not fit the definition of buttons, characterised by VE-cadherin localisation parallel to the sides of junction-free flaps.

d. Other junctions are continuous (zonulae, zippers) around entire cells, but these were uncommon on oak leaf-shaped LECs.

We fully agree. As shown, at the lobe level, we observe on average <5% LYVE1⁺ zippers per capillary end.

e. In Fig.6, the junction labeled "Curvilinear" and "Zipper" is confusing because it is classified as both types of junction and is also misleading because it differs from the **conventional meaning of "zipper" as a continuous junction around an entire endothelial cell**. Please see "8. Classification of LEC junctions" and revise.

We have now clarified in the summary (line 39), results (line 117-118), discussion (line 445) and **Fig. 1g** that the junctions were assessed within individual LEC lobes. Of note, several previous publications have solely used the criterion of continuous lines of VE-cadherin⁺ junctions at cell borders^{7,11} or various thresholds based on area or length^{10,12}. However, none of these studies applied the criterion that a zipper junction should be present around an entire LEC.

f. Fig.6 should also be revised to match the authors' data, including evidence for no significant difference in junction types resulting from PBS injection (Fig.1g).

We have now revised **Fig. 6**. See comment 5c.

2. Categories of discontinuous junctions in ear skin: The idealized drawings of junctions (Fig.1f, Fig.6) are misleading because they do not faithfully match the corresponding images of these junctions.

We have provided additional images of capillary LEC junctions at the level of a single lobe, together with drawings closely depicting the real immunofluorescence signal, in **Extended Data Fig. 1c**. This is to better illustrate the spectrum of VE-cadherin⁺ junctional configurations without imposing subjective classification. See comment 8a.

a. The drawings show uniform bands of VE-cadherin, whereas the images show variable staining. Images throughout the figures show that most junctions around oak leaf shaped LECs are discontinuous, regardless of whether they are designated button, curvilinear, or double junctions (e.g., Fig.1b, 3b, 4b Cdc42flox/+, 4g Cdc42flox/+, 4g Ctrl, Extended Data Fig.1c). Please revise.

Schematic drawings in the main figures (**Fig. 1g, Fig. 6**) have been revised. See comment 8a, and specific comment above.

b. As described in “8. Classification of LEC junctions”, these junctions are distinctly different from continuous zonulae (zipper junctions) in collecting lymphatics (Fig. 1a right, 3g lower, etc.).

At the lobe level, as per the definition of a continuous VE-cadherin positive cell-cell contact without LYVE1, we defined these as zippers. The addition of the lack of LYVE1 as a criteria provides a stricter definition compared to that used in previous studies to describe ‘lymphatic zippering’.

c. The drawings in Fig.1f and Fig.6 and the corresponding descriptions should reflect the discontinuity of the junctions. Please see “8. Classification of LEC junctions” and revise.

The figures have been revised.

3. “Spectrum of junctional configurations”: Descriptions throughout the manuscript of initial lymphatics having a “spectrum of junctional configurations” are potentially misleading because they are insufficiently informative.

a. “Spectrum of junctional configurations” could be interpreted as a mixture of tight junctions, adherens junctions, gap junctions, desmosomes, etc.

b. This is of particular concern in the Summary and Introduction.

We acknowledge the potentially misleading statement. We have now specified ‘spectrum of VE-cadherin-based junctional configurations’.

c. More objective and informative alternatives are shown below in 3d and 3e.

d. Summary (lines 39-40): “LECs in dermal lymphatic capillaries had **a heterogeneous mixture of buttons and other discontinuous junctions** and unique...”

Since we also observe continuous VE-cadherin-based junctions in individual LEC lobes (e.g. **Extended Data Fig. 1c**), the suggested text edits would seem somewhat misleading to us. For added clarity, we have now specified that junctional heterogeneity was studied at the level of individual lobes.

Lines 37-39: The oak leaf shaped LECs in dermal lymphatic capillaries displayed a spectrum of VE-cadherin-based junctional configurations at the lobular intercellular interface, ...

e. Introduction (lines 82-87): “In the current study, we used novel mouse models and approaches to characterize endothelial cells of initial lymphatics in mouse ear skin and mechanisms responsible for the distinctive oak leaf shape of the LECs. We identified new features of LEC overlap in mosaic mice and **confirmed the heterogeneous mixture of buttons and other discontinuous junctions** between the LECs, as shown previously (references 4, 6-8, 13-15, 32, 45). We then determined the contributions of cortical actin, microtubules, and Cdc42 in regulating the oak leaf shape of LEC.”

As explained in multiple previous comments, summarising our findings as confirmatory of previous findings, showing a ‘heterogenous mixture of buttons and other discontinuous junctions’, would be misleading.

As a compromise, and explained also in comment 4e, we wish to incorporate the Reviewer’s suggestions while maintaining focus on the key aim and novel aspects of the study. We therefore revised the sentence as follows:

Lines 82-88: In the current study, we used novel mouse models and experimental approaches to characterize endothelial cells of lymphatic capillaries in mouse ear skin, elucidating the mechanisms and drivers underlying their attributes and significance for lymphatic vessel physiology. We identified new features of LEC cytoskeleton and their dynamic overlaps and uncovered a previously unappreciated spectrum of VE-cadherin-based junctional configurations in capillary LECs across juvenile and adult stages. Our results point to the role of dynamic cytoskeletal regulation of LEC shape in controlling the maintenance of vessel integrity and function.

4. “Dynamic” remodeling of LECs: The authors’ reference to “dynamic” remodeling of LECs (lines 406-411 and elsewhere) is confusing because they use the same term for diverse changes occurring at very different rates.

- a. Actin remodeling in LEC lobes occurred in seconds (Fig. 3h, Supplementary Movie 3 and 4).
- b. Changes in LEC junctions and lobe overlap after intradermal injection of PBS (lines 130-131, Fig. 1g, lines 165-167, Fig. 2i, 2j) occurred over 10 minutes.
- c. Changes in the shape of cultured LECs exposed to cyclic stretch (lines 346-351, Fig. 5e-h) occurred over 14 or 22 hours.
- d. Spontaneous changes in LEC lobes observed by intravital imaging (Fig. 2k) occurred over 6 to 14 weeks.
- e. Given the very different durations of these processes, the authors should distinguish changes occurring in seconds or minutes from those occurring in hours, days, or weeks. Use of the term “remodeling” for all these events conflates processes that could have different mechanisms and functional implications.

Indeed we show dynamics at the LEC lobes at different scales, which demonstrates that the changes are active and continuous (see the next comment). The term ‘remodelling’ is widely accepted and used to describe both short-term and long-term biological changes. We have checked all incidences in the text where the term remodelling was used to ensure that they also include a clarification of the scale concerned (actin, lobe, overlap, junction, cell-cell borders). If the Reviewer has a suggestion for a better term to use, we would be grateful for this advice.

f. As remodeling is, by definition, a dynamic process, the word “dynamic” in dynamic remodeling is redundant. Could remodeling be static? The same applies to “active” dynamics. Please justify or change.

‘Remodelling’ vs ‘dynamic remodelling’ have a different meaning, in our understanding. While remodeling is a more general term for any alteration or change, dynamic remodelling specifically highlights an active, continuous and responsive process. Specifically, the latter implies an adaptable process and ongoing adjustment in response to changing conditions, rather than a one-time change. With active, we refer to cell-intrinsic (in this case actin-driven) process rather than passive dynamics driven by external forces.

5. Initial lymphatics of trachea and diaphragm:

a. The claim that initial lymphatics of the trachea and diaphragm had junctions similar to those in the ear (lines 122, 527) is not convincing based on one image of one lymphatic in each organ (Extended Data Fig.1d) and no quantitative data.

Additional images of both diaphragm and tracheal lymphatic vessels available in the publicly available dataset. We recognize that detailed characterisation of different vascular beds would necessitate an extensive additional study, and we have revised the text to emphasize the limitation of our analysis arguing against generalization (lines 597-599). The provided images serve to show that the identified junction categories (i.e. double junctions and curvilinear junctions) were observed in other tissues.

b. The statement, “This finding is consistent with the original description of buttons in the terminal ends of adult tracheal lymphatic capillaries, where they were observed with a frequency of 25-30%4 (lines 431-432).” is incorrect. Instead, in reference 4, 24% of buttons in the initial 1500 µm of initial lymphatics in mouse tracheas were located within 125 µm of the tip, 50% were within 250 µm of the tip, and 75% were within 500 µm of the tip (Fig.1E of reference 4). Only 25% of buttons were located from 500-1500 µm from the tip, indicative of the heterogeneity of junctions and decreasing number of buttons with increasing distance from the tip. The proportions of buttons and other types of discontinuous junctions were not reported in reference 4.

We acknowledge that we had misinterpreted the graph presented in reference 4, see comment 6e. We thank the Reviewer for noting this mistake, and we have now removed this statement.

c. Furthermore, the authors’ claim applies to tracheas and diaphragms removed from dead mice before fixation, whereas published data were obtained from lymphatics fixed in situ by vascular perfusion. Please see “10. Reasons for discrepancies with published data...”.

As discussed above, we obtained similar findings from perfusion fixed tissues (see comment 10e). Of note, it appears, based on the information provided in the methods, that published studies by others have utilised immersion-fixed tissue, including Jannaway et al (diaphragm, ears)¹⁰, Zhang et al (ears, lacteals)¹¹, Churchill et al (ears)¹², Zheng et al (diaphragm)⁸.

d. It is unclear why the authors try to generalize their findings to apply to the trachea and diaphragm and then treat intestinal lacteals (not examined) as a special case (lines 581-585). Please see “6. Inconsistencies” and revise.

As explained in comment 6f, we have indeed not performed analysis of lacteal junctions ourselves, recognizing that detailed characterisation of different vascular beds would necessitate an extensive additional study. To further emphasize the limitation of our analysis, we have revised the sentence as follows:

Lines 597-599: Although we found **the presence of curvilinear and double junctions** similar junctional morphologies in lymphatic capillaries also **in the diaphragm and trachea in adult mice** in other adult organs, our analysis is not sufficient to generalize our findings from the skin to different

organs. it is important to acknowledge that the organization and remodelling of junctions may vary in different organs. **For example**, in the lacteals of the intestinal villi...’.

An aspect that we consider argues against generalisation in particular in the case of lacteals is that they, unlike dermal vessels, are in a state of continuous growth, with many exhibiting sprouting tips¹⁷. In addition, lacteals are specialized in the absorption of dietary lipids (chylomicrons) rather than interstitial fluid. However, we are willing to remove the reference to lacteals if requested.

e. The authors’ generalization of their findings in ear lymphatics to apply to lymphatics in other organs lacks adequate supportive evidence. These claims **should be validated by more convincing images and quantitative data obtained after perfusion fixation or revised**.

As explained in comment 6f and previous comment, we have revised the sentence to emphasize the limitation of our analysis of other organs. We are also willing to remove the reference to lacteals if requested (see the previous comment).

6. Additional issues:

a. Lines 49 and 550: “bellow-like” should be “bellows-like”: Please explain how the proposed “bellows-like” constriction of initial lymphatics, in the absence of a primary flap valve, would propel lymph toward the secondary valve without also driving it backward into the interstitium.

‘Bellows-like’ was corrected.

We have now added a brief clarification of the proposed mechanism of action:

Lines 563-566: In this scenario, passive shortening of LEC overlaps and lumen expansion during edema is countered by active actin-based lobe remodelling to increase cellular overlap and vessel constriction aiding fluid propulsion, a process reminiscent of a bellows-like mechanism. **The increase in cellular overlap would, in turn, promote tightening of the endothelial barrier to prevent fluid from re-entering the interstitium, while working in concert with the suction effect generated by the contraction of downstream collecting vessels to propel lymph forward.**

We find this question fascinating and clearly in need of further studies. Along these lines, we note that a two-valve pump mechanism, as proposed in previous studies, would only function if protruding cellular lobes (flaps) are present on the luminal side of the endothelial monolayer. Various studies and schematic illustrations place the flap valves at the abluminal side where they connect to anchoring filaments and are pulled open under interstitial tissue swelling. Such an abluminal location may be expected to push out lymph into the interstitium under conditions of external vessel compression by e.g. the action of skeletal muscle.

b. Lines 732 and 816: For animal welfare purposes, please specify the anesthetic, dose, route of administration and conditions for mice used for IHC whole mount staining or TEM. Also specify the method of euthanasia and describe in more detail the preparation of ear skin whole mounts, from anesthesia to immersion fixation.

We have added the requested details on tissue preparation in the methods section:

Lines 997-1007: Most tissues (juvenile and adult skin, adult trachea, diaphragm or embryonic back skin) were harvested from mice that were euthanised by cervical dislocation or CO₂ asphyxiation, and immediately dissected and placed for fixation in 4% paraformaldehyde for 2 h at RT or 4 h at 4°C. Tissue from *Cdh5-GFP* mice was harvested after transcardial perfusion following three different protocols yielding similar results: 1) perfusion with 10 ml of PBS, followed by 10 ml of 4% PFA (RT), ear

collection and post-fixation by immersion in 4% PFA overnight at 4°C, 2) perfusion with HBSS, followed by 4% PFA, ear collection and post-fixation by immersion in 4% PFA for 4 h at 4°C, 3) direct perfusion with 1% formaldehyde for 2-3 min, ear collection and post-fixation by immersion in 2% PFA for 4 h at 4°C. Skin on the dorsal side of the ear pinna was dissected from the underlying cartilage layer before (immersion) or after (perfusion) fixation.

Lines 1073-1075: Mice were euthanised by cervical dislocation or CO₂ asphyxiation. Skin on the dorsal side of the ear pinna was immediately dissected from the underlying cartilage layer, transferred to 2.5% Glutaraldehyde (Ted Pella) + 1 % Paraformaldehyde (Merck) in 0.1 M Phosphate buffer (PB) pH 7.4 and incubated in the fixative overnight at 4°C.

Details on the anesthetic, dose and route of administration for mice undergoing intravital imaging of dextran clearance assay were already provided (lines 1044-1045 and 1063-1064).

c. Line 756: “euthanized” is recommended in place of “sacrificed”.

We now use the term ‘euthanized’.

d. Lines 890-897: In “Author contributions”, please identify those responsible for preparing tissues for TEM, taking electron micrographs, and interpreting TEM images.

Only descriptive TEM images are now included without interpretation on junction types. Nina Daubel was responsible for harvesting tissues for TEM and taking electron micrographs, which is now specified in the author contributions. She was supported by the staff at our electron microscopy facility, Karin Staxäng and Monika Hodik, who processed the samples. We have added their names in the acknowledgements.

Referee #2

The revised Schoofs et al. manuscript addresses part of the raised concerns about novelty, molecular mechanism and technical controls in an insufficient manner. The question about human relevance of the reported findings was not even raised by any referee, so that a thorough rework of the manuscript with regard to molecular mechanism and technical controls was expected for publication in any kind of journal. Unfortunately, this expectation was not met by the authors. The specific points of critique about the authors' additions to the manuscript in response to this referee's suggestions are the following.

Major suggestion 1.

This referee criticized that *cdc42* does many things in a cell, so that the pathway involving *cdc42* must be better explored to obtain specificity and functional relevance. Just changing a major regulator of the F-actin cytoskeleton will induce many defects and alterations in a cell, without providing major insights into the physiologic mechanisms and functional significances of *cdc42*-regulated dynamic junctional overlaps in capillary LECs.

The experiments interfering with CDC42 function provide strong evidence that actin cytoskeletal dynamics play a critical role in regulating cellular overlaps and monolayer integrity in LECs, although these experiments were not designed to elucidate the specific molecular pathways and signals upstream or downstream of CDC42. Given the observed crucial requirement of CDC42 for the maintenance of lymphatic capillaries during homeostasis in adulthood, we expect that multiple pathways upstream of CDC42 contribute to LEC junction maintenance through actin cytoskeleton regulation, and therefore defining these would be beyond the scope of this work.

This referee asked (as major point 1) to integrate the role of integrins and ECM in their model, in particular since several publications showed that (i) integrins are mechanosensors and essential molecules in LECs (see e.g., Planas-Paz et al., EMBO J 2022; Kumaravel et al., Am J Physiol Cell Physiol 2020, 319: C1045) and (ii) integrin beta 1 and *cdc42* functionally interact in different cell types (see e.g., Keely et al., Nature 1997, 390: 632; Reymond et al., J Cell Biol 2012, 199: 653; Cerutti et al., Cell Rep 2024, 43: 113989), including LECs (see e.g., Valtcheva et al., JBC 2013, 288: 35736 and Liu et al., Development 2018, 145: dev165092).

We performed the suggested experiments to assess the role of integrin $\beta 1$ in the maintenance of capillary lymphatic endothelium. A requirement for lymphatic development does not necessarily imply a role in homeostatic maintenance; indeed, we did not observe a phenotype in *Itgb1* deleted mice recapitulating the phenotype we characterized in *Cdc42* deleted mice.

Further, Liu et al. have previously shown in their Development paper that *cdc42* affects VE-cadherin junctions in LECs, but these authors even integrated *cdc42* into a Rasip1- and integrin-containing molecular pathway, making their paper mechanistically superior over the work presented here.

Interestingly, the Rasip1-CDC42 pathway was found to be crucial during embryonic development of lymphatic vessels, with interference in Rasip1 impairing LEC junctions¹⁸. However, Rasip1 was not required postnatally, indicating a differential requirement between embryonic development and postnatal life¹⁸. These results indeed suggest that Rasip1-independent signaling mechanisms for maintaining capillary LEC junctions are at play in adult mice used in our study, where CDC42 deletion disrupted capillary LEC junctions.

Schoofs et al. conducted experiments on integrin beta 1 though, but **the experiments were technically insufficient for drawing strong conclusions. The negative results obtained are difficult to interpret, since they might be due to many possible technical issues.**

We confirmed successful deletion of integrin $\beta 1$ (n=3 mice per genotype from 2 different time points) in lymphatic capillaries after tamoxifen administration by whole-mount immunofluorescence and confocal microscopy of the ear tissue, which is a widely used method for confirming gene deletions. Thus, we disagree with the Reviewer's suggestion that experiments were technically insufficient and interpretation from the data provided is not possible due to technical issues.

They also **failed to expand on the underlying molecular mechanism of LEC overlaps and their functional relevance that likely involves not just a single molecule (that is, cdc42).**

Although we expect that there can be multiple receptors / signals upstream of CDC42 (see the comment above), our results clearly show that the maintenance of capillary LEC overlaps and junctions crucially depends on CDC42, and this cannot be compensated for by other factors. In addition, we provide direct functional relevance for the LEC overlaps by showing increase in barrier strength upon CDC42-mediated extension of cellular overlaps (**Fig. 5l**).

Firstly, the authors deleted *Itgb1* using *Vegfr3-CreERT2*, but they compared these mice with floxed mice harboring no Cre, even though this was specifically requested by this referee in Minor Suggestion 6, and this is important since Cre expression has been recently shown to have vascular defect on its own (see e.g., Rashbrook et al., *Nat Cardiovasc Res* 2022, 1: 806). So, **an important control is entirely missing in this revision.**

The Reviewer's concern about Cre-expressing mice showing a phenotype associated with Cre toxicity does not seem to be justified in this case since we did not observe a phenotype even in *Itgb1^{flox/flox};Vegfr3-CreER^{T2}* mice. As specifically requested by the Reviewer and as explained in our previous revision, a Cre-positive control was included for the analysis of the *Cdc42* mutant phenotype (**Fig. 4e-g**). In addition, we show several images of tissues from *Vegfr3-CreER^{T2}* positive mice in combination with a Cre reporter throughout the manuscript.

It should be noted that Cre toxicity becomes relevant when Cre is expressed constitutively or when CreERT is acutely translocated into nucleus upon tamoxifen administration. The latter scenario is of relevance especially in developmental contexts such as the developing retina¹⁹ but does not seem to be as significant in adult mice²⁰. Importantly, upon tamoxifen washout, CreERT is no longer present in the nucleus and potential acute effects subside²⁰. Given the extended time period (weeks) between tamoxifen induction and analysis in our experiments on adult mice, the concerns regarding Cre toxicity do not seem justified. In support of this, we did not observe any apoptotic events in LECs using longitudinal imaging of *R26-iMb2;Vegfr3-CreER^{T2}* mice.

Secondly, a quantification of junctional overlaps was not conducted (quantification of images is supposed to be a normal procedure published in contemporary papers, even in small journals).

An accurate quantification of junctional overlaps and comparison to the results obtained from the *Cdc42^{flox}* mice would require crossing the *R26-iMb2* allele into the *Itgb1^{flox}* background. Currently, the mice are in two different animal facilities and in different countries. Rederivation and breeding of the mice to homozygosity would require more than a year to establish this line. As explained in the next point, the existing data very clearly show that *Itgb1*-deleted mice do not recapitulate the phenotype observed in *Cdc42*-deleted mice, characterised by the loss of lymphatic capillary integrity and LEC shape, to justify pursuing this task.

Thirdly, the lymphatic vessels in the ear were not challenged by fluid injection as to investigate whether *Itgb1* changes the physiologic behavior of lymphatic vessels to an enhanced interstitial fluid

pressure. These kind of experiments are already part of this manuscript and not at all beyond the scope of this report.

The question raised by the Reviewer was whether integrin β 1 acts upstream of CDC42 in maintaining capillary LEC shape, overlaps and monolayer integrity. As shown in our revised manuscript, we did not find evidence for this from either *in vitro* or *in vivo* experiments; the CDC42 phenotype observed at 3 weeks after deletion under unchallenged homeostatic conditions was not recapitulated in *Itgb1*-deficient mice, even when the observation period in the latter was extended to 10 weeks post-deletion (**Extended Data Fig. 6e**). While the role of integrin β 1 in maintaining lymphatic capillaries under challenged conditions may be an interesting avenue for future research, it is certainly beyond the scope of the current study, and it is not ethically justified to subject additional mice to such experiments in the frame of the current study.

Fourthly, the authors tried to inhibit integrin beta 1 by adding 0.1 to 0.2 μ g/ml mAB13 to unstretched and stretched LECs, but they noticed only a slight decrease in CD31-positive cellular overlaps, even though it looks as if there might be a concentration-dependent effect of integrin blockade. Even though investigations on isolated integrin molecules are published with such low concentrations of antibodies and even though the authors observe an (again not quantified) reduction in integrin beta 1 activation, for cell culture experiments higher concentrations are often used, such 1 μ g/ml mAB13 or a 1:10-dilution of purified antibody (see e.g., Lee et al., *Circ Res* 1995, 76: 209; Aqino et al., *BBRC* 2024, 703: 149575). Thus, the authors could miss an effect of integrin beta 1 by using too low concentrations. Other inhibitors, such as the RGD peptides, as well as knockdown experiments for *Itgb1* are urgently needed as well in order to draw solid conclusions.

First, we would like to clarify that we did not observe a 'slight decrease' or 'concentration-dependent effect' of integrin inhibition on cellular overlaps as suggested by the Reviewer; no significant changes were observed and thus such conclusions cannot be drawn.

For the meaningful interpretation of the stretch experiments, it was critical to preserve EC monolayer integrity and adhesion, since the loss of adhesion would make the cells less responsive to the stretching of their substrate/membrane. The concentrations of mAb13 were based on Hakanpaa et al.²¹, who showed that in cultured ECs mAb13 concentrations > 1 μ g/ml decreased VE-cadherin at the cell-cell junctions already in static 2D cultures, while low mAb13 concentration (0.1 μ g/ml) preserved EC monolayer integrity. We confirmed by immunofluorescence staining that at the chosen concentrations (0.1-0.2 μ g/ml), the LEC monolayer integrity was not compromised and mAb13 treatment of LECs decreased the amount of active β 1 integrins in LECs (**Extended Data Fig. 9a**), thus validating our approach. We have now added the quantification of these validation data in **Extended Data Fig. 9a**.

Fifthly, if integrin appears to be not involved based on all experiments suggested above (and contrary to a large body of literature on its role in LEC and *cdc42*), the authors **must at least obtain the Rasip1 KO mice and conduct their studies in these mice with genetic rescue experiments to show that *cdc42*-dependent LEC overlaps are integrated within this pathway (that also involves integrins though)** to follow up on the paper by Liu et al. in *Development* 2018.

Since Liu et al showed that the postnatal deletion of *Rasip1* did not cause lymphatic vascular defects, thus concluding that the function of *Rasip1* was restricted to embryonic lymphatic vascular development¹⁸, it is not relevant to investigate *Rasip1* in our model system in adult mice. As discussed above, our data similarly shows a dispensable role of integrin β 1 in the maintenance of adult capillary LECs in homeostasis, which is not contradictory to its demonstrated role in lymphatic vascular morphogenesis during development where it is essential in controlling cell-matrix and cell-cell

interactions. Besides integrin $\beta 1$ and Rasip1, there are numerous upstream regulators of CDC42 that will be of interest for further research, but our primary goal in this study was to understand the broader requirement of CDC42-mediated cytoskeletal dynamics in the regulation of LEC shape and monolayer integrity during homeostasis.

Major suggestion 2:

The referee pointed out that a functional relevance of cdc42-dependent cellular overlaps of LECs could be lumen widening in order to be able to take up more fluid from the interstitial space. The authors now respond that they did not observe any differences in cdc42-deficient versus control lymphatic capillaries (data not shown). They argue that an increase of around 1 μm cannot be quantified due to high variability. This argument is surprising given that they quantified changes in LEC overlaps with an average width of 2 μm that decreases significantly to around 1.5 μm in their hands. So, the finding of no lumen increase is at odds to the authors' most relevant finding, i.e. the existence of overlaps that they suggest play a key role in lymphatic vessels. So, what is (expressed in 1-2 sentences) the key novel message that the authors wish to convey in terms of mechanistic insights into lymphatic biology that is of strong functional relevance? Why is the paper more relevant and why does it present a substantial increase in knowledge compared to the Liu et al., Development 2018 paper that shows cdc42 influencing LEC junctions by regulating VE-cadherin expression that involves Rasip1?

This appears to be a misunderstanding, and we would like to clarify. We observed a reduction in the cellular overlap width from 2 μm to 1 μm (Fig. 4f) upon *Cdc42* deletion, and with an average of three cells forming the vessel perimeter (*in vivo* parameters used for modelling, Fig. 5b), the expected increase in vessel circumference resulting from the overlap extension was calculated to be 3 μm , which corresponds to an increase of 0.95 μm in diameter. Considering that the width of a capillary was measured to be on average $42.8 \pm 2.4 \mu\text{m}$ (s.d.) (Fig. 2f), confidently detecting an increase of 1 μm (i.e., 2-2.5% increase that is smaller than s.d.) would not be feasible. Nevertheless, such a seemingly small increase would still result in a ca 5% increase in volume.

We believe the last two sentences of our abstract sufficiently summarize our key novel findings in relation to the newly discovery mechanism in lymphatic biology (lines 46-51):

'Collectively, our findings indicate that capillary LEC shape results from continuous remodelling of cell shape and overlaps that maintain vessel integrity while at the same time preserving dynamic and permeable cell-cell junctions compatible with vessel expansion and interstitial fluid uptake. We further propose a bellows-like fluid propulsion mechanism, whereby fluid-induced lumen expansion and shrinkage of LEC overlaps is countered by actin-based lamellipodia-like overlap extension to aid vessel constriction.'

Minor suggestion 5:

The referee asked the question why the authors switched from Vegfr3-CreERT2 to Prox1-CreERT2 to delete cdc42. The authors argued that Prox1 is stronger for gene deletion, but not specific for endothelium, while Vegfr3 is weaker, but specific for endothelium. Given this argument, couldn't it well be that deleting cdc42 via Prox1-Cre introduces cardiac defects that subsequently affect lymphatic vessels and interstitial fluid? Is there any possibility to inject tamoxifen into the ear skin to obtain more selective effects rather than bystander effects? How do the authors otherwise rule out that deletion of cdc42 in other tissues (including other vascular beds) affects the lymphatics indirectly rather than directly?

We agree with the Reviewer that the choice of the Cre line and tamoxifen administration regime are important considerations. While we have successfully utilized the topical application of 4-hydroxytamoxifen to induce mosaic Cre recombination of a single allele in the ear skin^{22,23}, ensuring efficient deletion of multiple alleles within a single cell while restricting Cre recombination locally

remains challenging with the currently available tools. For the *Cdc42* deletion experiment, which also required recombination of a reporter allele (*iMb2* or *Lifeact-GFP*), an efficient deleter line was necessary. We thus selected the *Prox1-CreER^{T2}* line, which has been widely used by our group and others, including in the study by Liu et al. referenced by the Reviewer¹⁸. In our experiments, we did not observe any apparent defects in the blood vasculature or in the collecting vessels, thereby excluding the presence of major vascular abnormalities in the *Cdc42*-deleted mice within the time frame of gene deletion (6 weeks) studied here.

Minor suggestion 6:

The referee asked for Cre lines as controls due to a recent publication indicating that Cre lines must be taken as controls (see Rashbrook et al., *Nat Cardiovasc Res* 2022, 1: 806). The authors answer that they did not observe any changes in TEM of LEC overlaps between Cre- and Cre+ mice, but this referee does not see any quantification of Cre- versus Cre+ LEC overlaps in Frye et al., *eLife* 2020. The TEM images shown in the Figure referred to did not have any Cre- control either. The referee wishes the authors **to include Cre+ Tx injected control mice for all comparisons to avoid artifacts induced by the presence of Cre recombinase**, as reported by Ruhrberg and colleagues in *Nature Cardiovasc Res* 2022

As discussed in the manuscript, and as also brought up by Reviewer 1, quantification of cellular overlaps from TEM images is not meaningful, because depending on the sectioning angle the overlap may represent either a perpendicular cut across or a longitudinal cut along the cell-cell borders. For a superior method of quantification of cellular overlaps we therefore included in the previous revision whole-mount immunofluorescent analysis of *Cdc42* mutant mice with the *iMb2* reporter, and in these experiments heterozygous mice carrying the *Prox1-CreER^{T2}* allele were included (**Fig. 4e-g**).

As discussed above (please see Major suggestion 1), Cre toxicity becomes relevant when Cre is expressed constitutively or when CreERT is acutely translocated into nucleus upon tamoxifen administration and analysis is done soon after Cre induction, which is not the case here.

Minor suggestion 8:

The referee asked for *cdc42* knockdown experiments that the authors provided. However, the results obtained are in contrast to Liu et al., *Development* 2018, who observed major alterations in F-actin intensity and lumen width in LECs after *cdc42* knockdown experiments. How do the authors explain and compare their results with these published and more mechanistic data?

Liu et al showed reduced phalloidin staining upon CDC42 knock-down in sparse LEC cultures and isolated cells, where control cells showed numerous stress fibers. Our studies were conducted in confluent monolayers to reflect a more physiological state of endothelial monolayers of the vessel wall, where LECs display mainly cortical actin, recapitulating the *in vivo* situation, both in control (e.g.²⁴ and our study) and *CDC42* knock-down experiments (**Extended Data Fig. 8d**). Direct comparison of the results between the two studies is therefore not feasible.

However, and as expected, considering the well-established role of CDC42, we also observed changes in the actin cytoskeleton in siCDC42-treated cells (**Extended Data Fig. 8d**). The timeframe of our experiment (**Extended Data Fig. 8b**) was optimized such that CDC42 silencing did not affect the integrity of unstretched monolayers under unstretched conditions (**Extended Data Fig. 8c**), allowing us to specifically assess the effect of loss of integrity upon isotropic stretching (**Extended Data Fig. 8d**).

References

1. Leak, L. V. Studies on the permeability of lymphatic capillaries. *J Cell Biol* **50**, 300–323 (1971).
2. Hirata, A., Baluk, P., Fujiwara, T. & McDonald, D. M. Location of focal silver staining at endothelial gaps in inflamed venules examined by scanning electron microscopy. *Am J Physiol* **269**, L403–418 (1995).
3. Zöltzer, H. Initial lymphatics--morphology and function of the endothelial cells. *Lymphology* **36**, 7–25 (2003).
4. Paatero, I. *et al.* Junction-based lamellipodia drive endothelial cell rearrangements in vivo via a VE-cadherin-F-actin based oscillatory cell-cell interaction. *Nat Commun* **9**, 3545 (2018).
5. Cao, J. & Schnittler, H. Putting VE-cadherin into JAIL for junction remodeling. *J Cell Sci* **132**, jcs222893 (2019).
6. Mahlandt, E. K. *et al.* Opto-RhoGEFs, an optimized optogenetic toolbox to reversibly control Rho GTPase activity on a global to subcellular scale, enabling precise control over vascular endothelial barrier strength. *Elife* **12**, RP84364 (2023).
7. Yao, L.-C., Baluk, P., Srinivasan, R. S., Oliver, G. & McDonald, D. M. Plasticity of button-like junctions in the endothelium of airway lymphatics in development and inflammation. *Am J Pathol* **180**, 2561–2575 (2012).
8. Zheng, W. *et al.* Angiopoietin 2 regulates the transformation and integrity of lymphatic endothelial cell junctions. *Genes Dev* **28**, 1592–1603 (2014).
9. Baluk, P. *et al.* Functionally specialized junctions between endothelial cells of lymphatic vessels. *J. Exp. Med.* **204**, 2349–2362 (2007).
10. Jannaway, M. *et al.* VEGFR3 is required for button junction formation in lymphatic vessels. *Cell Rep* **42**, 112777 (2023).
11. Zhang, F. *et al.* Lacteal junction zippering protects against diet-induced obesity. *Science* **361**, 599–603 (2018).
12. Churchill, M. J. *et al.* Infection-induced lymphatic zippering restricts fluid transport and viral dissemination from skin. *J Exp Med* **219**, e20211830 (2022).
13. Lauweryns, J. M. & Cornillie, F. J. Topography and ultrastructure of the uterine lymphatics in the rat. *Eur J Obstet Gynecol Reprod Biol* **18**, 309–327 (1984).
14. Leak, L. V. & Burke, J. F. Fine structure of the lymphatic capillary and the adjoining connective tissue area. *Am J Anat* **118**, 785–809 (1966).
15. Aguilera Suarez, S. *et al.* Studying the Mechanobiology of Aortic Endothelial Cells Under Cyclic Stretch Using a Modular 3D Printed System. *Front Bioeng Biotechnol* **9**, 791116 (2021).
16. Tatin, F. *et al.* Planar cell polarity protein Celsr1 regulates endothelial adherens junctions and directed cell rearrangements during valve morphogenesis. *Dev. Cell* **26**, 31–44 (2013).
17. Bernier-Latmani, J. & Petrova, T. V. Intestinal lymphatic vasculature: structure, mechanisms and functions. *Nat Rev Gastroenterol Hepatol* **14**, 510–526 (2017).
18. Liu, X. *et al.* Rasip1 controls lymphatic vessel lumen maintenance by regulating endothelial cell junctions. *Development* **145**, dev165092 (2018).
19. Brash, J. T. *et al.* Tamoxifen-Activated CreERT Impairs Retinal Angiogenesis Independently of Gene Deletion. *Circ Res* **127**, 849–850 (2020).
20. Garcia-Gonzalez, I. *et al.* iSuRe-HadCre is an essential tool for effective conditional genetics. *Nucleic Acids Res* gkae472 (2024) doi:10.1093/nar/gkae472.
21. Hakanpaa, L. *et al.* Targeting β 1-integrin inhibits vascular leakage in endotoxemia. *Proc Natl Acad Sci U S A* **115**, E6467–E6476 (2018).
22. Martinez-Corral, I. *et al.* Blockade of VEGF-C signaling inhibits lymphatic malformations driven by oncogenic PIK3CA mutation. *Nat Commun* **11**, 2869 (2020).
23. Petkova, M. *et al.* Immune-interacting lymphatic endothelial subtype at capillary terminals drives lymphatic malformation. *J Exp Med* **220**, e20220741 (2023).
24. Sabine, A. *et al.* FOXC2 and fluid shear stress stabilize postnatal lymphatic vasculature. *J. Clin. Invest.* **125**, 3861–3877 (2015).

Answers to the Reviewers' questions

#2023-08-13841B

We would like to thank the Reviewers for their continued thorough evaluation and valuable feedback.

We have incorporated the textual changes suggested by Reviewer 1, and demonstrated normal cardiac function in *Cdc42^{flox/flox};Prox1-CreER^{T2}* as requested by Reviewer 2. The changes in the text are highlighted in the manuscript in red.

Additionally, we have made minor textual changes to the abstract and figure legends to comply with Nature's guidelines for length.

Referee #1 (Remarks to the Author):

General Comments

1. Purpose of review: In the evaluation of Revision 2, the reviewer takes the authors at their word that they "are committed to transparency" in response to General Comment 4f of the previous review. Although the availability of all source data reflects this commitment, the purpose of this comment was to ensure that the text and figures in Revision 2 have the same transparency, accuracy, clarity, and balance as the source data. This was not the case in Revision 1. The reviewer is now charged with reassessing the authors' effort to achieve this transparency, accuracy, clarity, and balance by determining whether the changes made in the revised manuscript (not just in the Rebuttal) adequately address the issues raised previously by the reviewers.

2. Overall assessment of Revision 2:

a. Problem: In Revision 2 the authors have addressed some but not all of the reviewer's comments by making changes in the manuscript text and/or figures. General Comment 13 in the previous review recommended the authors to focus their Rebuttal on the changes they make in the text and figures to address each comment. The authors responded in their Rebuttal that they "hope that our responses are now satisfactory in this regard". However, again, much of their detailed 38-page response to the reviewers' comments describes the authors' views and explanations for why the requested changes were or were not made. By declining to make some of the recommended changes, the authors not only rejected this constructive feedback for strengthening the manuscript, but also ignored the likelihood that some readers would have the same concerns as the reviewers. And it prolonged the review process.

b. As an example of changes described in the Rebuttal but not in the manuscript, the authors were asked in the initial review to address concerns over whether the reported features of lymphatic junctions resulted from using immersion fixation instead of vascular perfusion fixation. The request was repeated in the second review because the comment was not addressed in Revision 1. Now the authors claim the issue was addressed, but the findings were reported only in the Rebuttal, not in the manuscript. In Revision 2, the issue is addressed in the Methods, but still the findings are not described in the Results or illustrated in the figures (see Specific Comment for lines 1000-1007).

c. Another example is reflected by the newly added data for silver nitrate staining, where the methods and findings are included in the manuscript, but the background and limitations of interpreting silver staining are not considered in the manuscript. Instead, the author argue in the Rebuttal – but not in the manuscript - that the findings are evidence against lobes being junction-free flap valves and in favor of the presence of double junctions (see General Comment 6).

d. **Solution:** The authors are asked to make changes in the text and figures that address: (1) the problems described below as not adequately resolved in the manuscript, and (2) issues considered in the Rebuttal but not in the manuscript. Parts of the Discussion can be trimmed if length constraints become an issue.

Each specific point has been addressed below in line with these comments after the Reviewer's suggested solution.

3. Junction classification:

a. Problem: The authors' description of junctions between lymphatic endothelial cells has been problematic from the beginning because their classification was confusing in the context of which findings were confirmatory and which were new and how the new findings fit with the literature. Changes made in Revision 1 and Revision 2 reduced but did not eliminate these problems.

b. **Solution:** As described in General Comments 4 and 5, the authors are asked to remedy the problems by: (1) changing the description of curvilinear and double junctions from “discontinuous or continuous” to “segmented or unsegmented” (or similar terms), and explicitly acknowledging that these junctions are, like buttons, discontinuous around endothelial cells, because they would otherwise be called zippers; and (2) using the term “zipper” only for continuous junctions that surround entire endothelial cells, as originally defined in Reference 4.

1) Description of curvilinear and double junctions have been changed from continuous/discontinuous to unsegmented/segmented as detailed in the next point. The definition of zippers as continuous junctions that surround entire endothelial cells is provided (lines 105, 124-125), differentiating them from curvilinear and double junctions.

2) The term 'zipper' is now used exclusively for continuous junctions present in collecting vessels and sprouting lymphatic capillaries. We did not observe them in lymphatic capillaries in adult mice, which is now stated in lines 124-125. Linear unsegmented LYVE1⁻ junctions observed along the borders of capillary LEC lobes have been renamed as LYVE1⁻ curvilinear junctions, instead of zippers.

We would like to kindly note that 'zipper' has been used in many previous studies to describe unsegmented junction fragments in capillary LECs even if not encircling the entire cell perimeter. For example, Jannaway et al's (PMID: 37454290) definition of zipper junction is any VE-cadherin⁺ fragment >7.85 μm in length. A new term, 'LYVE1⁻ curvilinear junction', might thus cause some confusion, but we hope that this new descriptive term will be acceptable.

4. Continuous vs. discontinuous junctions:

a. Problem: The authors' continued description of some curvilinear and double junctions as “continuous” is baffling. As described in previous reviews, the term “buttons” was introduced to distinguish discontinuous junctions in endothelial cells of initial lymphatics from continuous

junctions (“zippers”) around endothelial cells of collecting lymphatics and blood vessels: Reference 4, page 2350: “Endothelial cells of initial lymphatics were joined by discontinuous buttons (Fig. 1C), whereas endothelial cells of collecting lymphatics were joined by continuous zippers (Fig. 1D), similar to those in adjacent blood vessels”.

b. To describe some curvilinear and double junctions as “continuous” is inconsistent with the intended meaning of continuous junctions in endothelial cells, confusing, and in conflict with the authors’ own drawings in Fig. 1g and Fig. 6, showing red lines for curvilinear and double junctions that are discontinuous at the base of lobes.

c. If some curvilinear and double junctions are continuous on a lobe but not around the entire endothelial cell, then some button junctions would also be continuous. Describing discontinuous junctions as “continuous” is not only self-contradictory but is also inconsistent with a commitment to transparency, accuracy, clarity, and balance.

d. If some curvilinear and double junctions are unsegmented, then call them “unsegmented”, not “continuous”, and call the others “segmented” and reserve the term “continuous” for zipper junctions that surround entire endothelial cells, as originally defined in Reference 4.

e. **Solution:** The authors are asked to fix this problem by using the term “continuous” only for zipper junctions that surround entire endothelial cells and use the term “discontinuous” for all non-zipper junctions, regardless of whether they are called buttons, intermediate junctions, curvilinear junctions, or double junctions.

“Segmented” or similar term can be used for the subset of button, curvilinear, and double junctions that are subdivided or fragmented on a lobe. “Unsegmented” or similar term can be used for the subset of these junctions that are not subdivided on a lobe.

The solution will require four changes: (1) correcting the text (lines 99, 123-124, 448-449, 1057-1060, Supplement lines 81-83, etc.); (2) relabeling Fig.1g to distinguish discontinuous junctions (all but zippers) from continuous junctions (zippers); (3) reorganizing and relabeling the 8-part panel on the left side of Fig. 6 to distinguish discontinuous junctions (all but zippers) from continuous junctions (zippers); and (4) redefining and distinguishing button junctions, curvilinear junctions, and double junctions from one another in the main text (lines 120-125), Methods (lines 1057-1060), Supplemental Methods (lines 81-83), and legends for Fig. 1g (lines 743-750) and Fig. 6 (lines 873-881).

1) Description changed to segmented and unsegmented (lines 97, 128, 461, Fig 1 legend line 749, Methods lines 1041-1044, Supplementary page 5).

2) Fig. 1g has been relabelled. As explained above, LYVE1⁻ linear junctions (previously zippers) have been renamed as LYVE1⁻ curvilinear junctions.

3) Panels have been reorganised. Please note that no zippers (defined as proposed) were observed; unsegmented LYVE1⁻ junctions have been renamed as LYVE1⁻ curvilinear junctions.

4) We have redefined the 5 junction categories as: buttons (lines 125-126, methods 1040-1041, Supplementary page 5); curvilinear and double junctions (lines 128-130, 1041-1045,

Supplementary page 5), LYVE1⁻ curvilinear junctions (lines 130-131, methods 1044-1045, Supplementary page 5); and zipper junctions (lines 124-125, methods 1045-1046, Supplementary page 5).

Due to the word limit for figure legends, detailed definitions of the junctions could not be included in the Fig. 1g legend, but we believe that the revised panel together with the revised description of the results now make the definitions clear. Previous Fig. 6 is now included as a panel m in Fig. 5, limiting the length of the legend, but the figure and full revised legend is also included as Supplementary Fig. 8. Illustration of junction categories present in capillaries (all but zipper) are updated in the figure.

5. Zipper-like junctions:

a. Problem: As just described, “zipper” was introduced to describe continuous junctions that surround entire endothelial cells typical of collecting lymphatics AND blood vessels, neither of which has an oak leaf shape or lobes. Use of the term “zipper” for a junction on an individual lobe is inconsistent with the definition and usage in the literature, does not make sense, and confuses the authors’ message.

b. To remedy this issue, the authors were asked in the previous review to add data that express junctions per endothelial cell to distinguish endothelial cells with continuous junctions (zippers) around their entire perimeter from those with discontinuous junctions, regardless of what they are called. Instead, the authors’ rejected this recommendation and continued to use their redefined term “zipper” to apply to an individual lobe (Figs.1g, 1h, and 6, Supplement lines 75-90).

c. As a result, (1) endothelial cells with curvilinear or double junctions designated “continuous” are not distinguished from endothelial cells with continuous junctions around the entire cell, (2) readers are left to ponder the authors’ distinction between “zippers”, “continuous” curvilinear junctions, and “continuous” double junctions on lobes, and (3) readers must also puzzle over whether endothelial cells at the entry region of lymphatics have junctions that are discontinuous or continuous in the conventional sense, because the data apply to individual lobes instead of entire endothelial cells.

d. **Solution:** The authors are asked again to fix these problems by using the term “zipper” only for junctions that surround entire endothelial cells. This will require correcting the text (lines 118-120, 268-269, Supplement lines 83-84, etc.) and labels in Figs. 1g, 1h, 4h, and 6 to reflect the definition of zippers as junctions around entire endothelial cells and eliminating the use of “zippers” for junctions that are not continuous around the entire cell.

As described above, the term 'zipper' is now used exclusively for continuous junctions present in collecting vessels and sprouting lymphatic capillaries and the text (see above) and labels in Fig. 1g, 1h, 4h and 6 have been updated accordingly.

Description of the presence or absence of LYVE1 staining is informative, but LYVE1 staining should not be a criterion for the identification of zippers, which are simply defined as junctions that surround entire endothelial cells. LYVE1 staining and zipper junction designation are independent criteria and should be treated as such.

Values for zippers in graphs shown in Figs.1g and 1h should be adjusted to reflect the historical meaning of zippers.

No zippers according the proposed definition are now described in capillaries. LYVE1- linear junctions have instead been renamed as LYVE1- curvilinear junctions.

No changes are needed where “zippers” or “zipper-like” or “zippering” is used correctly for developing lymphatics, collecting lymphatics, and blood vessels (lines 63, 66, 68, 69, 81, 96, 106, 150, 432, 433, 440-443, etc.).

6. Silver nitrate staining:

a. Problem: The authors’ addition of new data for silver nitrate staining (lines 114-117, 126, 741-743, 1015-1027) to support their interpretation of junctions on lobes is puzzling. Why add data that raise more questions instead of using the space to strengthen existing data (e.g., expansion of data from staining for claudin-5 and other tight junction proteins) and discuss in more detail the strengths and limitations of assumptions and interpretations?

b. The authors confirm that silver nitrate staining outlines the approximate location of the border of lymphatic endothelial cells. However, silver nitrate also stains multiple other regions and does NOT faithfully mark intercellular junctions. As is well-documented in the literature, silver nitrate staining not only occurs in “intercellular cement” (yet to be identified) between overlapping endothelial cells but also in the basement membrane at gaps between endothelial cells where no junctions are present and some other regions (Majno et al. *Virchows Arch A* 408: 75-91, 1985; McDonald *Am J Physiol* 266:L61-83, 1994; Reference 16). The authors’ colocalization data illustrate this property by showing that silver staining coincides with PECAM (Fig. 1f, Extended Data Fig. 1b). PECAM, which is an adhesion molecule not a junctional protein, does not colocalize with VE-cadherin or claudin-5 at buttons, and deletion of PECAM does not change button structure (Reference 4).

c. Silver nitrate staining at the borders of lymphatic endothelial cell lobes cannot be equated to junctions and does not add unambiguous support for the authors’ view of junctions. In the absence of balanced consideration of what is known and not known about silver staining of endothelial cells accompanied by realistic interpretations of the observed staining in lymphatics, the authors’ new data are misleading and do not strengthen their story.

d. These new data in the Results are an example of the problem described in General Comment 2, where the authors do not explain in the Introduction their rationale for adding the findings and do not consider possible interpretations in the Discussion, leaving readers in the dark over the reason for adding silver nitrate staining. Instead, this is discussed in the Rebuttal seen only by the reviewers.

e. The authors’ prioritizing the addition of these new data to the manuscript also argues against their claims in their Rebuttal that space constraints prevented or abbreviated the changes they made in the manuscript in response to some reviewers’ comments.

f. **Solution:** The authors are asked to fix this problem by adding to the Introduction, Results, and Discussion the missing information that describes the (1) background and rationale for adding silver nitrate staining, (2) complex nature of silver staining that is not specific for tight junctions or

adherens junctions, and (3) assumptions underlying their interpretations of the findings. If space constraints become an issue, the authors could remove these new data and methods to give room for changes made in response to the reviewers' comments.

We believe the data is valuable, as it represent a methodologically distinct imaging approach to the immunofluorescence-based imaging, and have therefore kept it. We have now added information on the (lack of) specificity and assumptions underlying the interpretations of the findings (lines 116-121). We have also added additional information about silver staining, including references to previous work, in Supplementary Information.

7. Future responses to reviewers' comments:

a. **Problem:** As described in General Comment 2a, the authors have encumbered the review process by not simply making changes in the manuscript to address issues recommended by reviewers to improve the accuracy, clarity, and rigor of their story, but instead devoting unnecessarily large amounts of the Rebuttal to explaining their reasons for making or not making changes in the manuscript.

b. **Solution:** If given the opportunity to revise their manuscript, the authors are asked to streamline their Rebuttal by simply (1) confirming the changes they made in response to the reviewer's requests to correct inaccuracies and issues that could confuse readers, and (2) specifying the manuscript line numbers and figure panel numbers where the changes can be found.

Point taken and implemented here.

c. As the changes should speak for themselves, it is not necessary to explain them in the Rebuttal or to repeat the revised text or the authors' views described in the manuscript.

Specific Comments

1. Lines 111-112: "we focused solely on blunt-ended LYVE1+ vessels in all analyzed age groups..."

Problem: As the authors prefer to use "blunt-ended" instead of "initial lymphatics", readers should be reminded that "blunt-ended" lymphatics are functionally the beginning – not the end – of lymphatics.

Solution: Please change to: "...we focused on the entry region of initial lymphatics by examining only the blunt origin of LYVE1+ vessels, sometimes referred to as "blunt-ended" lymphatics, in all analyzed age groups..."

We have revised the sentence for improved clarity avoiding the term "-ended" (line 111). Of note, "blunt-ended" referred to the morphology/shape of the initial lymphatic vessel tip (opposite to 'spiky'), not to the initial lymphatic vessel segment.

2. Lines 129-130: Text describing lymphatics in the trachea and diaphragm (Extended Data Fig.1e).

Problem: The examples of lymphatic junctions in the trachea and diaphragm shown in Extended Data Fig.1e are not representative of those in the literature.

Solution: These examples should be: (1) described as selected to illustrate curvilinear and double junctions, and (2) acknowledged as not intended to be representative of initial lymphatics in these organs. Also see Comment for lines 597-599.

The images were acquired without pre-selection and are representative of the broader dataset, which we have chosen to illustrate in the diaphragm: we provide an overview of lymphatic vessels in the diaphragm with 10 randomly selected capillary ends at higher magnification, showing maximum intensity projections through the entire vessel (**Supplementary Fig. 2a**) as well as sub-stacks of a single layer of endothelium (**Supplementary Figure. 2b**). These images show similar junctional heterogeneity as observed in our earlier representative image and in skin samples. These images have been uploaded to Zenodo for transparency. We have decided to remove data on trachea, and have followed the Reviewer's suggestions on how to discuss findings from other organs (point 15).

3. Lines 149-150: "LYVE1-low sprouts located at the distal tip of lymphatic capillaries also showed elongated shape without lobes and the associated zipper junctions (Extended Data Fig. 2a)."

Problem: As written, the phrase "without lobes and the associated zipper junctions" could be misinterpreted as not having zipper junctions.

Solution: Please change to: "LYVE1-negative LEC of sprouts at the tip of lymphatic capillaries were elongated, lacked lobes, and had zipper junctions (Extended Data Fig. 2a)."

Done (lines 156-157). With the slight alteration that we prefer to keep LYVE1^{low} rather than LYVE1-negative, since as seen in Fig. 1c and Extended Data Fig. 2a, some LECs show weak LYVE1 expression.

4. Lines 161-167: Parallel lines of LYVE1 surface staining shown in Fig. 2h of lymphatics fixed and stained without permeabilization.

Problem: It is unclear how the LYVE1 antibody could access the interior edge of overlapping LEC flaps to create the second line of LYVE1 staining in the absence of permeabilization.

Solution: The authors should explain in the Results or Discussion their interpretation of how the antibody gained access to the interior edge - as well as the exterior edge - of LEC flaps to create two parallel lines of LYVE1 staining without permeabilization in the surface staining experiments. Explanation has been added (lines 173-174).

5. Lines 424-427: "This dynamic remodelling is required for maintaining the cellular overlaps that define the lobate cell shape, which in turn ensures integrity of the LEC monolayer under strain that is imposed by interstitial fluid pressure alterations."

Problem: This statement is the authors' interpretation, not an established fact.

Solution: Please rewrite this statement as an interpretation, e.g., "We interpret our evidence as indicating that the remodeling is required for maintaining the cellular overlaps..."

Changed (now lines 436-437).

6. Line 429: "Lymphatic vessels collect interstitial fluid from tissues back into the bloodstream..."

Problem: "...collect fluid...back to bloodstream..." does not make sense.

Solution: Please change to: "Lymphatic vessels collect and transport interstitial fluid from tissues back to the bloodstream..."

Changed (line 441).

7. Lines 452-453: "...intermediate junctions were described at developmental stages and were previously considered a transient state undergoing zipper-to-button transformation (Reference 6)."

Problem: This statement is inaccurate because it omits published descriptions of intermediate junctions under normal conditions, after infection, and with treatment.

Solution: Please replace with: "...intermediate junctions have been considered a stable intermediate between buttons and zippers in normal mice or a transient or transforming state during development, after genetic manipulation, or with untreated or treated infection when the junctions undergo button-to-zipper or zipper-to-button transformation (References 6, 8, 9)."

Changed and references added as suggested (lines 464-467).

8. Lines 453-455: "...variations in the criteria..."

Problem: This statement is inaccurate because it omits relevant published data.

Solution: Please change to: "Button junctions have been reported to constitute about 50% of junctions in initial lymphatics in mouse ear skin (Reference 8), but variations in the criteria..."

Changed (now lines 467-468).

9. Lines 484-485: "...CLDN5, the second major component of LEC junctions,..."

Problem: This statement is misleading.

Solution: Please change to: "...claudin-5, a tight junction protein in LEC,..."

Changed (lines 499).

10. Lines 487-488: "multiple cell-cell and cell-matrix adhesion receptors, such as the tight junction proteins ESAM or JAM-A present in capillary LEC junctions..."

Problem: "cell-matrix adhesion receptors, such as the tight junction proteins" is not correct.

Solution: Please change to: "multiple tight junction proteins, such as ESAM or JAM-A, and cell-matrix adhesion receptors, such as β 1 integrins, in capillary LEC..."

Changed (lines 502-503).

11. Lines 494-496: "Leak's studies also indicate that while adherens junctions are prevalent, ultrastructurally defined tight junctions are rarely observed in LECs (References 21,37)."

Problem: This sentence does not accurately represent Leak's findings reported in References 21 and 37, where no measurements were made of the relative abundance of tight junctions (maculae occludentes) and adherens junctions (called desmosomes in Reference 37). En block staining with uranyl acetate, which is required to convincingly distinguish the two types of junction, was used only in two figures (Fig.11 and 11a, Reference 21). There is no question of the historical value of Leak's beautiful work, but using his non-quantitative TEM images as a reference for comparison to junction frequencies in contemporary immunofluorescence imaging is unjustified and misleading.

Solution: Please replace this sentence with the following: "Leak's ultrastructural studies revealed both adherens junctions (maculae adherentes) and tight junctions (maculae occludentes) in overlapping regions of adjacent LECs (References 21,37)."

Changed, with the minor adjustment that the latin terms, which are not mentioned anywhere else in the text, were omitted (lines 509-510).

12. Lines 496-498: "It is therefore unclear why immunofluorescence analysis revealed that the classical tight junction molecule CLDN5 co-localizes with VE-cadherin at the majority of LEC junctions, as reported previously (References 4,6), including in curvilinear and double junctions."

Problem: This statement is not a valid or meaningful interpretation of coincident VE-cadherin and claudin-5 immunofluorescence in LEC. The obvious and most likely interpretation is that tight junctions and adherens junctions are both present in regions of coincident fluorescence. However, coincident immunofluorescence does not mean the junctions would be coincident when viewed at higher resolution by TEM, where they would be seen as adjacent.

Solution: Please replace this sentence with the following: "Consistent with Leak's findings, our immunofluorescence analysis revealed colocalization of VE-cadherin (adherens junctions) and claudin-5 (tight junctions) in overlapping regions of LECs, as reported previously (References 4,6). However, because confocal microscopy does not have the resolution of TEM, colocalization of immunofluorescence is described with the understanding that the two types of junction are adjacent, not superimposed."

Changed (lines 510-515).

13. Lines 563-566: Concept of tightening of the endothelial barrier to prevent fluid from re-entering the interstitium.

Problem: The mechanism described here resembles the primary/secondary lymphatic valve concept reported more than 20 years ago by Geert Schmid-Schönbein and colleagues.

Solution: Schmid-Schönbein's historically important concept should be acknowledged here as a precedent for the authors' proposal by citing: Trzewik J et al. Evidence for a second valve system in lymphatics: endothelial microvalves. FASEB J 15: 1711-7, 2001; Mendoza E, Schmid-Schönbein GW. A model for mechanics of primary lymphatic valves. J Biomech Eng 125: 407-4148, 2003.

References added (now line 581).

14. Lines 568-570: "...limited to a few electron microscopic studies (Reference 43)..."

Problem: This statement is inaccurate.

Solution: Please acknowledge at least one of the publications on the fibrillin composition of anchoring filaments, e.g., Gerli R, Solito R, Weber E, Aglianó M. Specific adhesion molecules bind anchoring filaments and endothelial cells in human skin initial lymphatics. *Lymphology*. 2000 Dec;33(4):148-57.

We have now removed the original statement on ‘few EM studies’ as it is not central to the discussion of our findings (lines 585-587).

Carefully reading the publication by Gerli et al, we did not find the evidence compelling, nor have we found other studies confirming this finding.

15. Lines 597-599: “Although we found the presence of curvilinear and double junctions in lymphatic capillaries also in the diaphragm and trachea in adult mice, our analysis is not sufficient to generalize our findings from the skin to different organs.”

Problem: The images illustrating lymphatic junctions in the diagram and trachea do not justify this statement.

Solution: Please change to: “Although we obtained evidence of curvilinear and double junctions in lymphatic capillaries in the diaphragm and trachea of adult mice, the proportions were not measured and our analysis was insufficient to determine whether our findings from skin apply to other organs.”

Changed, and reference to trachea removed (lines 614-617).

16. Line 602: “...large molecules such as chylomicrons...”

Problem: This statement is inaccurate because chylomicrons are multi-component lipoprotein protein particles, not large molecules.

Solution: (1) Please correct this error, and (2) correct the misleading implication that there is evidence documenting that lacteals permit the entry of molecules or particles larger than can enter other initial lymphatics.

We have removed this statement entirely and concluded the paragraph with the wording suggested by the Reviewer above, to emphasize the limitations of our findings in generalizing to other organs.

17. Lines 728-881: Please correct the typos in the figure legends.

We have corrected all identified typos.

18. Line 879: “bellow-like” should be “bellows-like”.

This was in the legend of Fig. 6, which has been moved to Supplementary Data and corrected.

19. Lines 1000-1007: Comparison in the Methods of lymphatic junctions after perfusion fixation versus immersion fixation.

Problem: The three words (“...yielding similar results”) in the Methods are not convincing evidence to justify the statement that LEC junction morphology was identical after immersion fixation and

perfusion fixation.

Solution: Please correct this problem by (1) describing in the Results (lines 90-140) the findings obtained in the comparison that support the authors' claim on line 1001 ("yielding similar results"); (2) identifying the figure panels that enable readers to review evidence underlying the claim; and (3) adding "fixation by immersion" or "fixation by vascular perfusion" to the legend of each example figure used for the comparison.

We have added **Supplementary Fig. 2c** for comparison of lymphatic capillaries from 25-week-old mice stained after immersion fixation or vascular perfusion, with the fixation method clearly stated in the figure and the legend. The figure is referred to in the results (lines 134-135). We have revised the sentence in methods (lines 985) to be more specific about the claim and to specify which tissues were fixed by vascular perfusion or by immersion fixation (lines 980-984).

20. Lines 1196-1206: Extended Data Fig.1 legend.

Problem: The legend for Extended Data Fig.1 does not match the figure.

Solution: Please (1) correct this figure legend, and (2) check all other figure legends to ensure the numbering and panel descriptions match the figures.

1) Corrected and 2) checked.

21. Problematic sequence of some figure panels: Please make the sequence easier for readers to understand in Fig. 1a, c, d, b, f, g, h and other figures that lack of the conventional a, b, c, d, left to right, top to bottom sequence of figure panels.

We regret that to optimally fit the multipanels into the available space, the conventional left-to-right and top-to-bottom sequence was not always feasible.

22. Figure numbers: Please add numbers to all figures to facilitate future reviews.

Added.

Referee #2 (Remarks to the Author):

Schoofs et al. resubmitted a revised manuscript for publication in Nature. This referee had asked to put *cdc42* into the context of what has been published on LEC and stretch-dependent mechanisms (already in her/his first review). The authors now argue that they expect multiple pathways to be upstream of *cdc42*, but without wishing to reveal one of them or even part of a pathway involved. They also argue that quantification of LEC junctional overlaps in *Itgb1* deficient mice requires crossing R26-iMb2 mice with *Itgb1* KO mice, which they argue requires too much time. The referee sees this point, even though a more thorough investigation of a well-characterized pathway upstream of *cdc42* would have been useful to put the data into any kind of molecular context (in particular given that the authors wish to publish in Nature). Further, the authors argue that injecting fluid into the ears of their *Itgb1* KO mice is beyond the scope of this manuscript, which irritates this referee even more, given that fluid injections into mouse ears take a day or two. Therefore, the role of integrin-beta1 as a key upstream receptor under experimental conditions that the authors describe and use in their manuscript (since the first submission) has not been investigated.

We would like to remind the Reviewer that we investigated, but did not find evidence for the involvement of integrin $\beta 1$ in the maintenance of capillary LEC integrity under the experimental conditions we used for studying CDC42 function. This included the requested analysis of the effect of LEC-specific deletion of *Itgb1* on lymphatic vessel maintenance in adult mice, and the effect of blockade of integrin $\beta 1$ function in human LECs exposed to isotropic stretch.

Since the lack of integrin $\beta 1$ function does not recapitulate the phenotype observed upon loss of CDC42 function (*in vitro* or *in vivo*), studies on integrin $\beta 1$ in edema-induced responses would be unrelated to the role of CDC42 in the maintenance of lymphatic capillary integrity through the regulation of capillary LEC shape and overlaps.

Again, the authors seem to not wish to explore this pathway or any pathway upstream of *cdc42*, even with tools that they have used in this manuscript. The authors also do not wish to investigate *Rasip1* and argue that it only plays a role in the embryo, but not adult, which is too strong a statement given that Liu et al. deleted *Rasip1* only postnatally and analyzed the neonatal mice within a week after birth, so that according to this referee a role cannot be excluded for the situation that Schoofs et al. investigate: the ears of 9-weeks-old adult mice after fluid challenge.

The authors were also asked to exclude the possibility that *Prox1*-Cre-driven deletion of *cdc42* affects cardiac tissue and thereby (as a secondary effect of a starting heart failure) affects the lymphatics, since it is well-known that fluid retention and lymphatic congestion can develop after cardiac issues (when the heart muscle is affected that expresses *Prox1*).

The authors state that they did not observe any major vascular defects, making this referee wonder (since no data are shown) what that actually means. Did the authors analyze heart pump function in their mice to exclude this issue that might entirely explain their *cdc42*-lymphatic phenotype?

We acknowledge the importance of considering the effect of gene deletion driven by *Prox1-CreER^{T2}* on cardiac function, given that this line also induces deletion in cardiomyocytes. We have now included echocardiography assessment of cardiac function 3 weeks after *Prox1-CreER^{T2}*-mediated *Cdc42* deletion (**Supplementary Fig. 7a**), when lymphatic defects were evident (Fig. 4). *Prox1-CreER^{T2}*-positive mice (n=2) and *Cdc42^{flox/flox}* mice (n=2) were used as controls. Our analysis

showed normal heart weight in the mutant (n=5) compared to control mice irrespective of sex (**Supplementary Fig. 7b**). Echocardiography showed no evidence of cardiac dysfunction in the mutant mice when comparing fractional shortening, ejection fraction and heart rate to the control mice (**Supplementary Fig. 7c**). In addition, left ventricular structure was not altered in the mutant compared to control mice (**Supplementary Fig. 7d**). Data was acquired and analyzed by experts in echocardiography, Amanda Marks and Marie Jeansson, who were added in the author list. We conclude that lymphatic phenotypes observed after inducible *Cdc42* deletion in *Prox1-CreER^{T2}* mice cannot be a secondary effect of heart failure (lines 279-283).

Notably, this is in agreement with previously published data showing that cardiomyocyte-specific deletion of *Cdc42* in the postnatal heart does not lead to observable effects on the baseline cardiac phenotype. Maillet et al¹ analyzed heart morphology and cardiac function in *Cdc42-flox;MHC-Cre* mice expressing a constitutively active form of Cre recombinase in cardiomyocytes. Efficient protein depletion was validated at the age of 2 months, and analyses of mutant mice of up to one year of age included heart weight measurement, echocardiography assessment of heart function, histological analysis of ventricular and cellular morphology in *Cdc42* mutant mice compared to control¹.

Finally, when looking at the raw data sheets, some of these do not align with the respective Figure panels and are not well-explained or sufficiently transparent. For example, in Figure 2j, the authors show the average width of collector vessels, but the last data in the excel table show as Area/length three values: 2.490, 1.246 and 1.269. This referee fails to see these values or their average in Figure 2j – what has happened? Were they or was their average taken out as outlier? In Ext. Data Figure 9c, the authors show 14 dots for stretch control (no-str.), but 23 values are shown in the corresponding excel sheet. So, where are the values? Same Figure: the authors show as “No stretch 0.1 mAb13” 8 or 9 dots in Ext. Data Figure 9c, whereas the corresponding excel sheet shows 13 values.

We thank the Reviewer for noting these mistakes in the source data.

Concerning Fig. 2j, the Reviewer is correct in noting that the last three values were not included in the graph. These measurements were mistakenly copied to the Source Data from a file that also contained an earlier preliminary quantification of overlaps in capillary LECs and that do not correspond to overlaps in collecting vessels. We have removed these erroneous data from the Source Data document.

Concerning Extended Data Fig. 9c, we had missed to update the graph in the figure after the last (third) experimental repetition that was performed during the second revision, although the Source Data file was updated. The graph has now been corrected. We have also reviewed all other Source Data files to confirm their accuracy.

References

1. Maillet, M. *et al.* *Cdc42* is an antihypertrophic molecular switch in the mouse heart. *J. Clin. Invest.* **119**, 3079–3088 (2009).

2023-08-13841C

Responses to referee's comments:

Please note that due to shortening the line numbers have changed. All changes have been therefore highlighted in the manuscript text in red.

Referee #1 (Remarks to the Author):

General Comments

1. The authors were very responsive and have adequately addressed the comments on the previous version of their manuscript. The remaining issues in the revised manuscript are minor.
2. The edits requested under Specific Comments are to correct minor errors and potentially misleading text and to improve clarity.
3. The authors are encouraged to make the title shorter and more eye-catching.

Specific Comments

1. Title: Current title "Resilience of lymphatic endothelium through isotropic stretch-induced cytoskeletal regulation of puzzle cell shape" describes in 13 words some of the findings but seems unnecessarily long, complex, and detailed to be sufficiently eye-catching to attract a broad readership.

Suggested alternatives are:

- Puzzle cell shape reinforces lymphatic vessel fluid uptake
- Puzzle cell shape increases resilience of initial lymphatics
- Lymphatic endothelial cell puzzle shape facilitates fluid uptake
- Lymphatic vessel resilience increased by puzzle cell shape
- Lymphatic vessel fluid uptake facilitated by puzzle cell shape
- Lymphatic endothelial cell puzzle shape increases resilience for fluid uptake

We have provided a shorter title that complies with Nature's guidelines:
Dynamic puzzle-like cell shape supports resilience of lymphatic endothelium

2. Line 38: "...spectrum of VE-cadherin-based junctional configurations at the lobular intercellular interface..."

•Needing attention: The discontinuous nature of these diverse junctions should be described in the Abstract.

•Recommended change: "...spectrum of shapes of discontinuous VE-cadherin-containing junctions at cell-cell interfaces..."

In the previous revision, the reviewer advised against using the term '(dis)continuous' to describe the new junction categories. As requested by the reviewer, we switched to use the terms 'segmented' and 'unsegmented' VE-cadherin-based junctions. Therefore, referring to them as 'discontinuous VE-cadherin-containing junctions at cell-cell interfaces' would be misleading as it conflicts with the terminology now used throughout the manuscript. In addition, it would not accurately describe that some of these junctions are unsegmented (i.e. not discontinuous) at the level of an individual lobe.

The description 'spectrum of VE-cadherin-based junctional configurations at the lobular interfaces' accurately describes the findings presented in Figure 1.

See also #9.

3. Line 58: "...intermittent junction-free regions..."

- Needing attention: "intermittent" is confusing (temporal versus spatial?) and unnecessary.
- Recommended change: Delete "intermittent".

Deleted (line 57).

4. Line 83: "...new features of LEC cytoskeleton and their dynamic overlaps..."

- Needing attention: Check grammar. In this phrase, "their" refers to "cytoskeleton", whereas it should refer to LEC.
- Recommended change: "...new features of the LEC cytoskeleton and dynamic properties of the overlapping cell borders..."

Revised as suggested (lines 80-81).

5. Line 100: "The classical tight junction protein Claudin 5 (CLDN5) was present in both buttons and in most, but not all, linear VE-cadherin+ adherens junctions (Fig. 1b)."

- Needing attention: This sentence is misleading and confusing. Claudin-5 is not "the classical" tight junction protein and is not present in adherens junctions. Occludin was the first tight junction protein identified in 1993 and has a broader distribution among cell types than claudin-5. Claudin-5, first described in 1999, is typically associated with endothelial cell tight junctions.
- Recommended change: "Staining for the endothelial cell tight junction protein claudin 5 (Cldn5) coincided with VE-cadherin staining of most buttons and other junctions in these LEC (Fig. 1b)."

Revised to include the requested definition 'endothelial cell tight junction protein claudin 5' (lines 97-98). The wording of the rest of the sentence has been slightly changed, due to the need to shorten the text.

VE-cadherin-based junctional configurations in capillary LECs (**Fig. 1b,c**), with the endothelial tight junction protein claudin 5 (CLDN5) often coinciding with VE-cadherin (**Fig. 1b**).

6. Lines 102-103: "...a significant proportion of lymphatic capillary ends displayed sprouting at three weeks of age (Fig. 1c, d)..."

- Needing attention: Significant in comparison to what? No statistics are shown in Fig. 1d.
- Recommended change: Delete "significant" or change to a percent.

Changed to ~20% (line 98). Exact value 23.6 ± 6.3 % (n=7 ears) is included in Source Data for Fig. 1d.

7. Lines 113-114: "Similar diversity was observed upon histological staining using silver nitrate..."

- Needing attention: This description of "similar diversity" is inconsistent with the images of silver staining in Fig. 1f and ED Fig. 1b, which show granular but largely continuous single or double silver lines at cell borders, unlike the heterogeneous discontinuous VE-cadherin staining shown in Fig. 1g and ED Fig. 1d.
- Recommended change: "Silver nitrate staining formed largely continuous single or double granular lines at the border of initial lymphatic LECs, unlike the heterogeneous, discontinuous VE-cadherin staining."

We have removed the statement 'similar diversity' as requested. The revised sentence additionally incorporates the reviewer's suggestion on the nature of the deposits, but also accurately describes the two important aspects revealed by our study: double lines of silver deposits and their presence at the tips of the lobes:

Silver nitrate staining¹⁶ additionally revealed predominantly single or double granular lines at capillary LEC borders, including the tips of the lobes (lines 107-108).

8. Lines 116-117: "Silver precipitation occurs due to an unspecified junctional component at the EC intercellular interface."

- Needing attention: The word "junctional" is misleading because it overstates what is known about the location of silver staining in the context of adherens junctions and tight junctions.
- Recommended change: "Silver nitrate marks endothelial cell borders by staining unknown components of the intercellular interface".

Revised to remove reference to 'junctional' component (line 109).

9. Lines 124-125: "This revealed the lack of zipper junctions, defined as linear LYVE1- junctions that surround the entire cell (Fig. 1g)."

- Needing attention: This statement is misleading because VE-cadherin is not mentioned and

therefore is inconsistent with the definition on Line 94: "...collecting vessels exhibited continuous VE-cadherin+ zipper junctions with no LYVE1 expression". Another issue is that the authors should explicitly state that the absence of zipper junctions in these vessels means that the junctions were discontinuous, albeit variable in morphology. A further issue is that the figure cited (Fig. 1g) does not show zipper junctions. However, zipper junctions are shown in Fig. 1a.

- Recommended changes: "This initial region of lymphatics lacked zipper junctions, defined as continuous VE-cadherin+ junctions around entire LECs (Fig. 1a). Initial lymphatics had discontinuous junctions with variable morphologies on individual lobes and were LYVE1+, unlike collecting lymphatics that had zipper junctions and lacked LYVE1 staining."

We have revised the sentence referring to the lack of zipper junctions in initial lymphatics as suggested (lines 114-116).

Regarding the terminology for the new junction categories, see #2 above. The terminology has been discussed with the reviewer through multiple rounds of revisions, and we have agreed to all changes that retain accurate description of our findings. For example, in the previous revision, the reviewer advised against using the term '(dis)continuous' to describe these junctions, but is now suggesting '**discontinuous unsegmented junction**', which we find confusing and may be perceived as contradictory. The suggested sentence ignores the key observation that some VE-cadherin-based junctions are unsegmented at the level of individual lobes, as shown in Figure 1 - data that the reviewer has previously accepted.

In our opinion, our description accurately describes the new junction types presented in Figure 1: *"...variations of unsegmented or segmented linear distribution of VE-cadherin lining one or both borders of LYVE1+ regions, which we termed curvilinear and double junctions, respectively"* (lines 119-121).

10. Lines 127-128: "...no significant increase in their frequency in older mice (Fig. 1g)."

- Needing attention: Significant in comparison to what? No statistics are shown in Fig. 1g.
- Recommended change: Delete "significant" or change to a percent.

Statistics is included in Source Data. During previous revision, we were asked to remove 'ns' indicating 'not significant' from the figures. We have included reference to Source data (line 118).

11. Lines 130-131: "Curvilinear junctions that were LYVE1- were rare observed (Fig. 1g)."

- Needing attention: Incorrect grammar.
- Recommended change: "Few curvilinear junctions lacked LYVE1 staining (Fig. 1g)."

Revised as suggested (lines 121-122).

12. Lines 137-138: "...Analysis of lymphatic vessels after 10 minutes revealed no significant changes in the relative frequency of junction types (Fig. 1h)."

- Needing attention: Significant in comparison to what? No statistics are shown in Fig. 1h.
- Recommended change: Delete “significant” or add results of statistical tests.

Statistics is included in Source Data. During previous revision, we were asked to remove ‘ns’ indicating ‘not significant’ from the figures. We have included reference to Source data (line 127).

13. Lines 173-174: “...which become inaccessible for staining after chemical crosslinking despite the antibody accessing the lumen at sites where the tissue was disrupted during preparation.”

- Needing attention: Rewording this sentence is recommended to reflect the speculative nature of the explanation of the staining of both lines.

•Recommended change: “...which become inaccessible for staining after chemical crosslinking. Staining of the intraluminal border presumably occurs where silver nitrate accesses the lumen through endothelial disruptions created during tissue processing.”

Revised as suggested, except that silver nitrate replaced by antibody (lines 148-150).

14. Lines 194-198: “remodelling”

- Needing attention: The use of the term “remodelling” for shape changes that occur over weeks or months (Lines 194-198) AND for shape changes that occur in minutes or hours (Lines 202-204) is confusing in the absence of clear descriptions of the different meanings of remodelling in these contexts.

- Recommended change: Please address this comment along with the comment for Lines 202-204.

Addressed below.

15. Lines 202-204: “...revealed continuous remodelling of cell cell borders”. This comment also applies to: Line 477: “Intravital imaging further revealed continuous remodelling of LEC overlaps...”, Line 574: “Continuous actin-driven remodelling of cellular overlaps...”, legend for Supplementary Movie 1: “...showing remodelling of cell-cell borders”, and other similar statements in the manuscript.

- Needing attention: The broad range of time-courses of “continuous” or “dynamic” remodelling reported throughout the manuscript and in figure legends as shape changes that occur in minutes to hours or over weeks to months is not adequately described or discussed. Shape changes occurring in minutes (Supplementary Movie 1) clearly differ mechanistically and functionally from the shape changes occurring over weeks or months (Fig. 2k). Another concern is that the changes shown in Movie 1 are subtle and occur in only focal regions of cells. The absence of the authors’ definition of the remodelling observed over minutes raises the question of whether “remodelling” in Supplementary Movie 1 refers to the tiny transient filapodial projections that rapidly form and retract on some lobes. Readers should be told the authors’ intended meanings of remodelling, where to look for remodelling in images and movies, and what they should expect to see.

- Recommended change: Please describe the types of remodelling in the Results and give more detailed descriptions of specific examples of remodelling in the corresponding figure legends, e.g., Supplementary Movie 1 (minutes to hours), Fig. 2k (weeks to months), to explain what is meant by “remodelling” in each case the term is used in the manuscript.

We have clarified previously that remodelling in the context of weeks-months refers to the observed changes in cellular lobes (e.g. lines 169, 210) and overlaps (e.g. line 175), minutes-hours to cellular borders (not only filopodial projections) (e.g. line 173), and sub-minute timeframe to actin cytoskeleton (e.g. lines 202). This is also clarified in legends of Movie 1 (SI data) and Fig. 2k (line 584). While we acknowledge the difficulty of directly extrapolating short-term observations to later events, the finding that actin cytoskeleton interference via *Cdc42* deletion disrupts lobate cell shape supports the notion that dynamic, actin-driven changes result in pronounced alterations in lobe structure. This provides functional evidence linking cytoskeletal dynamics to the maintenance of lobate shape.

16. Lines 220: “terminal capillary”; line 1190: “terminal capillaries”

- Needing attention: These terms, which are used only in these two locations, are confusing because the authors apparently mean the beginning of initial lymphatics rather than the end (terminal) of lymphatic capillaries where they join pre-collector or collecting lymphatics. The term “terminal capillary” was apparently adopted from studies of blood capillaries. The term “initial lymphatics” helps readers think functionally about these vessels, where flow is away from the tip, and reinforces the difference from blood capillaries, where flow is toward the tip of sprouts.

- Recommended change: “initial lymphatics” or “initial region of lymphatic capillaries”.

Revised as suggested (line 186).

17. Line 274: “...and loss of a uniform lobate shape (Fig. 4e, Extended Data Fig. 4a).”

- Needing attention: This is misleading. The cell shape change shown in these figures does not include a loss of lobate shape or conversion to the smooth contour of LEC in collecting venules. Instead, lobes are still present but the cell contour is more irregular.

- Recommended change: “...and change from the uniform oak leaf shape of normal LECs in initial lymphatics to more irregular shapes (Fig. 4e, Extended Data Fig. 4a). Importantly, the cells do not convert into the smooth shape of LEC in collecting venules.”

Revised as suggested (line 223-224). We believe that irregular shape now sufficiently describes the phenotype and addition of the second sentence is not needed.

18. Lines 340-341: “...the cells displayed significant bulging, leading to almost complete closure of the lumen (Extended Data Fig. 7a).”

- Needing attention: Significant in comparison to what? No statistics are shown in ED Fig. 7a.

- Recommended change: Delete “significant” or add results of statistical tests.

Significant replaced by notable (line 269).

19. Line 439: “...interstitial fluid pressure alterations (Supplementary Fig. 8).”

- Needing attention: Supplemental Fig. 8 appears to be identical to Fig. 5m.
- Recommended change: Move Supplementary Fig. 8 to become new main Fig. 6, delete Fig. 5m, and change line 439 to “...interstitial fluid pressure alterations (Fig. 6).”

We have implemented the suggested changes.

20. Lines 457-458: “...~20% of cellular lobes exhibited classical button junctions, with no significant increase in their frequency in older mice to suggest developmental maturation.”

- Needing attention: Does this mean no significant age-related changes were found in button junctions? If so, add the results of statistical tests to Fig. 1g. If not, delete “significant”.
- Recommended change: Delete “significant” or add results of statistical tests.

See #10. We have included reference to Source data in the results section.

21. Lines 467-470: “Button junctions have been reported to constitute about 50% of junctions in initial lymphatics...”

- Needing attention: Reference citations are missing.
- Recommended change: Add references that support this statement.

Reference has been added (line 354).

22. Line 545: “...exhibit significant cellular overlaps...”

- Needing attention: Where is the statistical significance reported?
- Recommended change: Delete “significant” or replace with a non-statistical term.

Replaced with ‘prominent’ (line 395).

23. Line 552: “...the absence of significant flow-induced mechanical forces...”

- Needing attention: Where is the statistical significance reported?
- Recommended change: Delete “significant” or replace with a non-statistical term.

Replaced with 'substantial' (line 399).

24. Lines 757-758: Fig. 1f legend: "...including the lobe tips (arrow) with discontinuities (arrowhead)."

- Needing attention: Please add that the tiny discontinuities in the granular silver staining are smaller than the discontinuities in VE-cadherin staining of initial lymphatic LECs. The legend should acknowledge that silver staining follows cell borders but does not faithfully match VE-cadherin staining in the images shown.

- Recommended change: "...including the lobe tips (arrow) with small discontinuities (arrowhead) accompanying the granular silver staining. These discontinuities are smaller than the discontinuities in VE-cadherin staining in Fig. 1a,b,g, and the silver staining pattern is largely continuous, unlike the staining of VE-cadherin at adherens junctions."

It would be clearly misleading to draw conclusions about the size or length of discontinuities by comparing the patterns of silver granules with those obtained from antibody-based labeling techniques, given the different size of silver granules (up to 1 μm) and the resolution of antibody-based detection (200 nm). Also, as pointed out by the reviewer (#7), the molecular component that initiates silver deposition is not known and may exhibit different distribution compared to VE-cadherin. The suggested change was therefore not implemented.

25. Line 797 Fig. 3 legend: "log₂ fold change".

- Needing attention: The comparison of capillary and collecting regions of lymphatics showed a difference in the two regions, not a change.

- Recommended change: "log₂-fold difference".

Revised as suggested (line 592).

26. Line 852 Fig. 5 legend: "(h-j) Quantification of junction linearity (n=38, 43 images; one experiment)..."

- Needing attention: The description in the Supplementary Methods (page 9), where normalized values are expressed as %, does not fit with normalized linearity values ranging from 1.0-2.0 in Fig.5h or with the statement (Line 856) "Junction linearity was normalized to the average of controls."

- Recommended change: Confirm the location in the manuscript where the method used to calculate linearity is explained (Supplementary Methods?). Also, briefly describe the meaning of "junction linearity" in Fig. 5h legend and add the units to the Y-axis label in Fig. 5h.

Thank you for noticing this mistake. We had missed to update the methods section when we changed the presentation of these results, to show the linearity index value without normalization, in line with previous publications. The method section has now been corrected. The linearity index is also described in the figure legend (line 636). As the value is ratio, it does not

have associated units.

27. Lines 864-865 Fig. 5m and Supplementary Fig. 8

- Needing attention: These figures appear to be identical.
- Recommended change: Move Supplementary Fig. 8 to become new main Fig. 6 and delete Fig. 5m.

The suggested changes have been implemented, see above.

28. Line 881: “4-hydroxytamoxifen”

- Needing attention: Please unify the abbreviations used for this agent throughout the text and figures.
- Recommended change: Use “tamoxifen” throughout and delete “Tam” and “4-OHT” in the text and figures.

4-hydroxytamoxifen is an active metabolite of tamoxifen that enables localized induction in the ear without requiring liver metabolism for systemic action, as is the case for tamoxifen. These are two different compounds used in different experiments, as described in the methods. We have replaced Tam by tamoxifen, and ensured that terminology is used correctly and consistently in the text and figures.

29. Lines 919-920: “Details of the MultiStretcher device will be presented elsewhere (Linsenmeier et al. 2024, in preparation).”

- Needing attention: Check the journal’s requirements regarding citation of manuscripts “in preparation”.
- Recommended change: If such citations are unacceptable, add the essential details of the device to the Methods or Supplementary Methods.

According to Nature’s guidelines, papers in preparation should be mentioned in the text with a list of authors (or initials if any of the authors are co-authors of the present contribution). We have updated the reference in the text to include a full list of authors (707-708).

30. Fig. 5m: This figure appears to be the same as Supplementary Fig. 8.

- Needing attention: Duplication of Fig. 5m and Supplemental Fig. 8
- Recommended change: Delete Fig. 5m and make Supplementary Fig. 8 the new main Fig. 6. See comments for Line 439 and Lines 864-865 regarding Fig. 5m and Supplementary Fig. 8.

The suggested changes have been implemented, see above.

31. Extended Data Fig. 3a, 3b

- Needing attention: Additional labels would be helpful.
- Recommended change: Please add labels to these figures to make it easier for readers to understand that 3a shows oak leaf endothelial cells of initial lymphatics, whereas 3b shows endothelial cells of collecting lymphatics. This distinction is evident in the legend but not in the figure.

Labels have been added.

32. Extended Data Figs. 9 and 10:

- Needing attention: Additional labels would be helpful.
- Recommended change: Please add labels to these figures to make it easier for readers to understand that Figure 9 shows cultured LEC and Figure 10 shows cultured HUVEC. This distinction is evident in the legends but not in the figures. There is plenty of room in the figures to add these labels. Also, in the legends, please comment on the width of VE-cadherin staining at junctions, where VE-cadherin forms a narrow line in LECs regardless of the width of overlap, but VE-cadherin staining is as wide as the overlap in HUVEC under unstretched and stretched conditions.

Labels have been added in the figures, and the description of junctions has been added in the legend, now Extended Data Fig. 11 (lines 1104-1106).

33. Supplementary Fig. 8 legend: "...a spectrum of junctional configurations organized as discontinuous buttons, as well as segmented or unsegmented double junctions and curvilinear junctions."

- Needing attention: Please correct the unintended but misleading implication that double and curvilinear junctions are not discontinuous.
- Recommended change: "...a spectrum of discontinuous junctions, including buttons, segmented or unsegmented double junctions, and curvilinear junctions."

To avoid confusing terminology 'discontinuos unsegmented junctions' (see #2, 9) we have revised as follows:

...a spectrum of junctional configurations organized as punctate buttons, as well as segmented or unsegmented double junctions and curvilinear junctions (line 650).